# Thermalization in Quenched Open Quantum Cosmology

Subhashish Banerjee[1], Sayantan Choudhury[2,3,4‡, §], Satyaki Chowdhury[3,4],
Johannes Knaute[5,6], Sudhakar Panda [3,4], K. Shirish[7]

[1] *Indian Institute of Technology Jodhpur, Jodhpur 342011, India.*
[2] *Centre For Cosmology and Science Popularization (CCSP), SGT University,
Gurugram, Delhi- NCR, Haryana- 122505, India.*
[3] *School of Physical Sciences, National Institute of Science Education and Research,
Bhubaneswar, Odisha - 752050, India.*
[4] *Homi Bhabha National Institute, Training School Complex, Anushakti Nagar, Mumbai -
400085, India.*
[5] *Max Planck Institute for Gravitational Physics (Albert Einstein Institute),
Am Mühlenberg 1, 14476 Potsdam-Golm, Germany.*
[6] *Department of Physics, Freie Universität Berlin, 14195 Berlin, Germany.*
[7] *Visvesvaraya National Institute of Technology, Nagpur, Maharashtra - 440010, India.*

## Abstract

In this article, we study the quantum field theoretic generalization of the Caldeira-Leggett model in general curved space-time considering interactions between two scalar fields in a classical gravitational background. The thermalization phenomena is then studied from the obtained de Sitter solution using quantum quench from one scalar field model obtained from path integrated effective action. We consider an instantaneous quench in the time-dependent mass protocol of the field of our interest. We find that the dynamics of the field post-quench can be described in terms of the state of the generalized Calabrese-Cardy (gCC) form and computed the different types of two-point correlation functions in this context. We explicitly found the conserved charges of $W_\infty$ algebra that represents the gCC state after a quench in de Sitter space and found it to be significantly different from the flat space-time results. We extend our study for the different two-point correlation functions not only considering the pre-quench state as the ground state, but also a squeezed state. We found that irrespective of the pre-quench state, the post quench state can be written in terms of the gCC state showing that the subsystem of our interest thermalizes in de Sitter space. Furthermore, we provide a general expression for the two-point correlators and explicitly show the thermalization process by considering a thermal Generalized Gibbs ensemble (GGE). Finally, from the equal time momentum dependent counterpart of the obtained results for the two-point correlators, we have studied the hidden features of the power spectra and studied its consequences for different choices of the quantum initial conditions.

**Keywords**: Quantum Quench, Thermalization, Quantum Field Theory in de Sitter Space.

‡ *Corresponding author, E-mail* : sayantan.choudhury@niser.ac.in, sayanphysicsisi@gmail.com
§ *NOTE: This project is the part of the non-profit virtual international research consortium "Quantum Aspects of Space-Time & Matter" (QASTM) .*

# 1 Introduction and summary

The study of Brownian motion [1–6] of a particle coupled to a thermal bath has assumed great significance owing to its relevance as a robust model for open quantum systems in the context of macroscopic properties of a particle in a general environment. This has been used to study quantum dissipation [1–5, 7, 8] and quantum decoherence due to the system's interaction with the environment [9]. This model of quantum brownian motion has proven to be useful not only in studies of open quantum systems but also in the field of quantum cosmology [10–19], quantum correlation problems [20–22], among others. It has also been extensively used in the context of AdS/CFT [23–25]. The usual approach of tackling this problem involves use of the influence functional technique developed by Feynman and Vernon [26]. The contribution of the environment degrees of freedom is quantified by the influence functional and one obtains the reduced subsystem of interest whose dynamics is of particular interest. A very well-known model in this direction was given by Caldeira and Leggett [2]. For the cosmological application look at ref. [27], where the authors have studied the origin of time dependent mass from the coupling to an inflaton field which is assumed to be in a coherent state leading to a time dependent mass and further the phenomena of particle production is studied in detail. For more details see also the other refs. [28–30].

The process of thermalization has grown to be an important area of research in the recent past. The advent of holography has provided a one-to-one correspondence of the subject of thermalization to the issue of gravitational collapse of a black hole. Quantum quench is one such technique where the process of thermalization can be realized in the system in the post-quench phase. In a quantum quench, some parameter of the Hamiltonian change over a finite duration of time, and the initial wave function in the pre-quench function evolves to a state after the quench that is not stationary. The evolution of the state after the quench is then guided by the post-quench Hamiltonian which is in general time-independent. This kind of study is crucial to find out if and when a closed system reaches equilibrium subject to any disturbances. Due to the growing interest in studying thermalization for integrable systems, there has been huge progress in the understanding of thermalization in scalar fields and extensive studies in the direction can be found in refs. [31]. Besides the theoretical motivation, in many experimental studies, the process of quantum quench has been realized using cold atoms and the post quench phase can be described in terms of free scalars or fermions [32]. Hence, the study of quantum quench involving scalar fields is of prime significance not only theoretically but also experimentally.

Quantum quench has been extensively studied in various contexts in recent times. Specifically, several studies have focused on the background of flat space-time, with the system undergoing a sudden change in its parameter under a well-defined quench protocol.

It has been seen that the system undergoing a quench tends to retain some memory of the sudden change at late times independent of its initial state condition. This quench protocol has also found its applications in the cosmology of the early universe. It has been used to study the characteristics of fast phase transitions, under the settings of early cosmology where temperature promptly decreases. An application of the quenching mechanism in the context of inflation provided many new results that were in contrast with the flat space-time results. During inflation, the post quench state in the background of de Sitter space-time doesn't retain the memory of the quench at late times which was in contrast with the flat space-time [33]. Quantum quench has been an effective model to study the undergoing transition to the broken phase, which is also used to study various physical processes such as baryogenesis due to electroweak phase transition [34]. The process of quantum quench has not only been studied for free fields but for the interacting fields as well. Late time thermal characteristics of interacting quantum fields have also been studied using the quenching mechanism in [35], especially for the $\phi^4$ model. Unlike free field theory which exhibits an exception in the $2d$ case due to a quantum quench of the energy gap or mass, interacting fields tend to thermalize even for the massless $2d$ case.

The quench approximation has also been studied in the context of conformal field theory [36, 37]. It was applied to study the properties of universal fast scaling of conformal operators undergoing fast quench, in the limit where the coupling suddenly changes its value from zero to $\delta\lambda$. The scale by which the holographic conformal operator changes has been found to be universal, i.e., the same scaling factor appears in the sudden quench limit of free scalar and fermionic field theories. One of the most interesting applications of quantum quench comes in the context of holographic thermalization, i.e., the thermalization of boundary operators, which has a direct correspondence with the collapse of gravitational matter in the bulk. Hence memory retention of quench protocol at late times by post quench state results in the retention of information of the collapsing matter by the final black hole [38]. In other words, a quantum quench could probe the inside geometry of a black hole. Besides all these applications, quantum quench could also be used to study general systems which don't involve phase transitions.

In this paper, we aim to study the thermalization phenomena at late times of two-point correlation functions from the solution obtained in the background of de Sitter space-time using quantum quench protocol. By making use of the well known *Caldeira Leggett Model*, we start with two interacting scalar fields in the background of de Sitter spactime. By doing the Euclidean path integration over one scalar field, we construct the reduced subsystem of our interest consisting of one scalar field described by an effective partition function. We then argue that our *Caldeira Leggett* (CL) Model in the context of cosmology, in the background of curved space-time which describes the particle production, could be translated in the language of Schrödinger quantum mechanics in one dimension where one studies the motion of electron in a wire in the presence of an impurity. We then identify the potential involved in the Schrödinger equation with the quench protocol and study the

thermalization properties of two-point correlators, their spatial derivatives and canonically conjugate momentum field in the ground state and generalized Calabrese-Cardy (gCC) states. We find that the dynamics of the post-quench state of the field of our interest can be described in terms of the state of the generalized Calabrese-Cardy (gCC) form and compute different types of two-point correlation functions in this context. We explicitly find that our post quench gCC state could be represented by the conserved $W_{\infty}$ algebra after the mechanism of quench protocol in de Sitter space and found the conserved charges to be significantly different from the flat space-time.

The underlying strong physical motivations and implications of this work are as follows:

1. The main motivation of the present work to provide a detailed framework of computing the cosmological correlation functions from a given open quantum mechanical system. Finding such correlations within the framework of cosmology is itself a very interesting problem itself. Recently, using the the same two field coupled model with a specific type of interaction (the QFT generalized version of CL model that we have used as our starting point of our paper) in some refs. [39, 40], the authors have tried to analyse this problem to address the phenomena of decoherence and recoherence from the evolution of the system reduced density matrix using the cosmological master equation perspective. However, in the mentioned references the authors have not addressed the structure, behaviour and the cosmological consequences of such correlations in the early time scale of the cosmological evolution. Though they have very clearly and in detail have established their findings and can be treated as the benchmark in a real sense in the context of cosmology with open quantum system. Two possibilities one can utilize to study the underlying framework of cosmological correlations from time dependent coupling parameter between the proposed two filed coupled CL model. Quantum mechanical quench naturally serves the purpose to provide the explicit form of the time dependent coupling parameter within the present framework. The mentioned two possibilities are slow and sudden time dependent profile for quench which helps us to trigger the thermalization process studied in the later half of this paper. We adopt the possibility of having fast or sudden quench which serves the purpose very smoothly in the present context [¶]. Sudden quench actually helps us to construct the accurate quantum states before quench, after quench and after sufficient enough time when the underlying physical system fully thermalizes. Once we fix the quantum initial condition, which is appearing in terms of the correct choice of the initial vacuum state, the structure of the pre-quench state is

---

[¶]In a more realistic cosmological set up, where the present methodology can be directly applicable, the scale of sudden quench can be fixed before reheating, more precisely before achieving thermalization. It might be fixed at end of inflation or just after the end of inflation. In the case of warm inflation since there is no reheating involved the framework and the thermaliziation is achieved completely in a different way, the choice of the quench scale should be chosen very appropriately.

automatically fixed. Next, using such pre-quench state one can immediately construct the post-quench state using Bogoliubov transformation and also using using the Dirichlet or Neumann type of boundary conditions. Finally, using this specific structure of the post-quench state one can able to fix the corresponding structure of the quantum state which can directly contribute and trigger the thermalization process in the present context of discussion. These constructed states helps us to explicitly compute the cosmological correlations before quench, just after quench and after a sufficient enough time when the thermalization is achieved. Before this particular work, these possibilities have not been explored in great detail for open quantum systems and we have tried our best to provide answers to the corresponding question.

2. Now it might be established in great detail, but a natural question comes in our mind that what the utility of the cosmological correlation computed from the present QFT generalized version of CL model in the de Sitter background? The specific answer to this question is as follows. We all know that the micro structure and the quantum mechanical origin of the reheating process is not well known and corresponding theoretical framework is not established yet. Till date this topic is completely untouched by the researcher due to having the lack of knowledge regarding the micro structure of reheating process. We strongly believe that, since we have now a proper understanding of the structure of the quantum states which helps us to thermalize an underlying quantum mechanical system and we also know how exactly to quantify the two-point cosmological correlation and its corresponding spectrum in the Fourier space, using the present methodology one need not to be foricibly assume thermalization of a theoretical set up written in the background of de Sitter space-time. We also believe that the developed methodology in this work can able to address many unexplored issues related to the phenomena of reheating in cosmology, which is treated completely from the phenomenological point of view before this work.

3. The prime motivation of using the CL model within the framework of cosmology is as follows. Actually this model automatically provide the theoretical origin of incorporating the phenomena of Quantum Brownian Motion in the present context. Now naturally another crucial question comes in our mind that why at all Quantum Brownian Motion is needed within the framework of cosmology? A correct answer to question when we try to incorporate the effects of anisotropy and inhomogeneity without introducing any concept of cosmological perturbations in the present framework. Quantum Brownian Motion within the framework of cosmology naturally helps us to incorporate the effects of anisotropy and inhomogeneity without introducing any perturbations. Such anisotropic and inhomogeneous effects helps us to construct the pre-quench state, post-quench state and the state responsible to achieve thermalization. In an effective framework where two fields are interacting via complicated

interactions, we really don't have any proper understanding of the path integration technique over an unwanted field in which at the end we are not interested in. This is because of the fact that, having complicated two field interactions within the framework of effective field theory we really don't know how to quantize these fields in a proper technical sense. CL model is the simplest framework where we really have understanding as well as control over the technical computational part that how to do the path integration over the fields which describes the thermal bath.

4. Also the present framework allows us to study the particle production process during and after reheating with the help of the constructed post-quench state and the state responsible to achieve thermalization in the present framework. Signatures of such particle productions can be directly found in the enhancement of the power spectra in Fourier space, which we have explicitly computed from the two-point cosmological correlations in this paper.

5. Last but not the least, additionally, the present framework can also be utilized to study the natural origin and outcomes of warm inflation where in absence of reheating one can thermalize an underlying theory.

The main results of the paper are as follows:

- Our prime motivation in this work, is to study the thermalization phenomenon in de Sitter space-time. It is important in the sense that if a system does not thermalize, we can't study its equilibrium properties for the system under consideration. This phenomenon was studied using free quantum field theories with massive scalar and fermion fields earlier in $1 + 1$ and $1 + 2$ dimensional flat space-time [41, 42], but not, to the best of our knowledge, in the context of de Sitter space, which has its own cosmological importance. In this paper, we have demonstrated how one can implement the same methodology to study the thermalization phenomena using free quantum field theory of a scalar field having an effective time-dependent mass term in $1 + 3$ dimensional de Sitter space written in planar coordinates.

- To implement this methodology we use the phenomena of quantum mechanical quench in our setup. This is a very successful technique providing a consistent theoretical way to equilibrate and hence thermalize a quantum mechanical system, initially out of equilibrium due to some response in the system. This technique provides a continuous description of the system in the associated time scale as it helps to express the quantum mechanical state of the system just before thermalization in terms of the state before applying quench. In this case, explicit solution of the time evolution of the quantum state from the time-dependent Hamiltonian of the system in $1 + 3$ dimensional de Sitter space is not needed.

- We do not use this methodology in our work in an ad hoc fashion. We provide a consistent theoretical framework from the beginning where one can naturally implement the above mentioned mechanism. In this work, we start with a theory of quantum Brownian motion in a general curved space-time background, described in terms of two scalar fields quadratically interacting with each other having minimal gravitational interaction, canonical kinetic terms as well as mass terms for both the fields. The model can be treated as a quantum field theoretic generalization of the well known *Caldeira Leggett model*, used to study the phenomena of quantum Brownian motion in the context of quantum mechanics. The original *Caldeira Leggett model* is approximated by a harmonic oscillator coupled to the environment consisting of $N$ oscillators, which are integrated out. However, in our case we have taken a simplified version where instead of $N$ scalar fields we have a single scalar field as our environment, which is technically identified with a noise field. On the other hand, the other scalar field in this context is identified to be the signal field, our main point of interest is to study the thermalization phenomena by implementing the methodology of quantum mechanical quench in $1 + 3$ dimensional de Sitter space. This hitherto unexplored possibility was not explored before in $1 + 3$ dimensional de Sitter space and has cosmological consequences.

- Since the quantum Brownian motion is studied here in $1 + 3$ dimensional de Sitter space, the signal and noise fields are dependent on both space and time. From the beginning, both the fields are considered to be inhomogeneous. See refs. [12, 43–50] where a similar approach has been followed earlier in various contexts. This approach is usually adapted to study outcomes of cosmological perturbation theory in the presence of a scalar field. There the field is taken be homogeneous in the $1 + 3$ dimensional de Sitter background and on top of that the inhomogeneous fluctuation of the field appears due to space-time-dependent perturbation with respect to the background. But in our computation we don't need to perform any perturbation on the background $1 + 3$ dimensional de Sitter space-time. The inhomogeneous effect in the signal and noise fields are considered from the beginning due to random movement in space-time in the presence of quantum Brownian motion.

- Since we are interested in the signal field, we path integrate the noise field using the Feynman path integral technique, treating the background $1+3$ dimensional de Sitter space classically. This is thus a semi-classical treatment allowing for the extraction of the information of the signal field.

- The quantum effective action of the signal field, in the Euclidean signature, is constructed using the saddle point technique, where the path integration is implemented at the local minimum of the noise field appearing in the model, described above. After carrying out the path integration, it is observed that the mass of signal field

gets modified in presence of the coupling parameter of the signal and noise field and the mass of the noise field. Here, during the implementation of the saddle point technique it is assured that at the local minimum of the noise field the gravitational back-reaction effect also gets minimized.

- As we are interested in understanding the large time behavior of the system in $1 + 3$ dimensional de Sitter space, the contribution from the quantum correction terms in the effective action goes to zero as in that limit the noise kernel appearing from two-point noise-noise field correlation function decays exponentially. As a result, the Klein Gordon equation of motion of the signal field appears to be similar to a damped parametric oscillator instead of a forced one in presence of the Hubble term in the d'Alembertian operator.

- Next, we Fourier transform the equation of motion in the momentum space. The sudden quench protocol in the effective mass profile of the signal field is implemented and the equations of motion for both the pre-quench and the post-quench phases of the evolution of the system under consideration are solved.

- Using the continuity condition for the solutions of the field and its conjugate momenta, we compute the Bogoliubov coefficients. This helps obtaining the solutions before the quench in terms of the solutions after quench and vice versa.

- After constructing the pre-quench, post-quench and the post thermalization state of the system, we study the signal-signal two-point correlation functions in the momentum space.

- Last but not least, instead of doing the exact computation of the two-point functions in the coordinate space, we study a much more observationally relevant quantity known as the power spectrum and observe various non-trivial features in the spectrum. We have also found that at a certain value of the co-moving wave number, the numerical amplitude of the spectrum exactly matches with the result obtained from the power spectrum using cosmological perturbation theory. This is quite interesting in the sense that it helps us to conclude that at very large time limit, when the effect of quantum corrections in the effective action for the signal field vanishes, the power spectrum evaluated from this computation and from cosmological perturbation theory exactly matches. On top of that our obtained results have the advantage that they naturally thermalize the system using quantum quench. This is not yet properly understood in the context of quantum fluctuations generated from cosmological perturbation theory.

The organization of the paper is as follows:

- In Sec. 2, we review the Caldeira-Leggett model in quantum mechanics and a quantum field theoretic generalized version of it in curved space-time consisting of scalar fields interacting with each other. We derive the effective action for the scalar field of our interest by path integrating out the contribution of the other field.

- In Sec. 3, we consider the solutions of the mode functions in spatially flat de Sitter space-time and by computing the Bogoliubov coefficients, derive the conserved charges of the $W_\infty$ algebra for the quench profile considered in this paper. We further provide a generalized expression of the correlation functions for different initial starting states of the pre-quench Hamiltonian. We choose the ground state as well as some squeezed state of the initial Hamiltonian as the starting wave functions and showed that the final state in the post-quench phase can be expressed in the gCC form. We also compute the thermal correlators to check whether the subsystem thermalizes or not.

- In Sec. 4, we provide the plots of the power spectrum obtained from the correlators for all different choices of the initial vacuum state and do a comparative analysis.

- In Sec. 5, we conclude and dicuss possible future prospects of the present work.

## 2 Quantum Field Theoretic generalization of Caldeira-Leggett model in curved space

In the Caldeira-Leggett (CL) model the phenomenon of quantum dissipation was discussed and closed equations for such a quantum system were obtained. For the purpose of studying such phenomenon, a particular model describing such system-bath interaction was chosen and the parameters of the model were fitted in such a way that the classical equations of Brownian motion were reproduced.

### 2.1 The two field interacting model

In this section, our prime objective is to provide the quantum field theoretic generalized version of *Caldeira-Leggett model* in a curved space-time. In general this framework is commonly used to describe *Quantum Brownian Motion* [51–53]. To describe this set up let us first start with the following two scalar field interacting theory, which is described

by the following action:

$$
S_{\mathbf{CL}}[\phi, \chi] = \int d^4 x \sqrt{-g} \left[ \underbrace{\left( -\frac{1}{2}(\partial\phi)^2 + \frac{m_\phi^2}{2}\phi^2 \right)}_{\textcolor{red}{\textbf{Free theory of } \phi}} + \underbrace{\left( -\frac{1}{2}(\partial\chi)^2 + \frac{m_\chi^2}{2}\chi^2 \right)}_{\textcolor{red}{\textbf{Free theory of } \chi}} + \underbrace{c\phi\chi}_{\textcolor{blue}{\textbf{Interaction}}} \right], \qquad (2.1)
$$

In this description both the fields are minimally coupled to the classical background gravity. In the above action, the first two underbrace terms represent two free massive scalar fields $\phi$ and $\chi$ and the last term represent the quadratic interaction term between them having interaction strength $c$ which is a function of space-time in general. We are identifying this action as the very simplest quantum field theory version of the *Caldeira-Leggett model* in curved space-time. In this description, the quantum harmonic oscillators are replaced by the scalar fields, which is quite justifiable. By following the same logical arguments applied in the *Caldeira-Leggett model*, in the present quantum field theoretic construction we path integrate over the field $\phi$ and construct an effective action for the field $\chi$. This is because of the fact that within the description of *Quantum Brownian Motion* we have identified $\phi$ as the noise field and $\chi$ is the field, of the system of interest.

To proceed further, let us write down the total contribution in the potential for the $\phi$ and $\chi(x)$ fields as appearing in the above action:

$$
V(\phi, \chi) = \left( \frac{m_\phi^2}{2}\phi^2 + \frac{m_\chi^2}{2}\chi^2 + c\phi\chi \right). \qquad (2.2)
$$

From this one can ask a question that for a given value of $\chi(x)$ what is the minimum of the above potential, which can be answered as:

$$
\left( \frac{\partial V(\phi, \chi)}{\partial \phi} \right)_{\phi=\phi_0} \sim m_\phi^2 \phi_0 + c\chi_0 = 0 \qquad \Longrightarrow \qquad \phi_0 \sim -\frac{c\chi_0}{m_\phi^2}, \qquad (2.3)
$$

$$
\left( \frac{\partial^2 V(\phi, \chi)}{\partial \phi^2} \right)_{\phi=\phi_0} \sim m_\phi^2 > 0 \quad \Longrightarrow \quad \text{minimum.} \qquad (2.4)
$$

Now at this point one can really think of the correctness of considering the minimization of the potential with respect to one field, where both the fields as well well as the coupling parameter is space-time dependent. The confusion arise because of the fact due to having space-time dependence in the coupling our general notion guided us to simplify the problem by solving the classical equations of motion of both fields, and due to having the coupling among both the fields we will have coupled equations in both the cases. Though this is the perfect approach to treat the underlying problem under consideration, but using this approach solving the coupled system of two fields is almost impossible in general. It may be done for very special type of restricted cases, which we obviously don't want to do in

this paper. The justification of performing minimization are as follows point-wise:

1. First of all, to solve this problem using the simplistic approach we have assumed that the coupling between the two fields vary with background space-time very slowly, so that one can approximately neglect the space and time derivatives of the coupling parameter from this present computation. For this reason we take, $c(x) \sim c(x_0) = c$. This approximation is completely justifiable with the quasi-de Sitter space-time where this methodology and the prescribed framework is further applied explicitly in the later half of this paper.

2. Now in between two fields $\chi$ is identified to be signal field and $\chi$ is identified to be the noise field in this framework, out of which we want to construct the effective theory of the signal field $\chi$ at the end. To technically perform this step we have considered a flat direction along the $\phi$ field and the position in the field space is implemented at the point of minimum, which is $\phi = \phi_0 \sim -\frac{c\chi_0}{m_\phi^2}$.

3. Our next job is to rewrite the action around the point $\phi = \phi_0$, which gives us:

$$S_{\mathbf{CL}}[\phi, \chi] \approx \int d^4x \sqrt{-g} \left[ -\frac{1}{2} \left( \partial \left( \phi - \phi_0 \right) \right)^2 + \frac{m_\phi^2}{2} \left( \phi - \phi_0 \right)^2 \right.$$
$$\left. -\frac{1}{2} \left( \partial \chi \right)^2 + \frac{m_\chi^2}{2} \chi^2 + c \left( \phi - \phi_0 \right) \chi \right]. \qquad (2.5)$$

After substituting $\phi_0 \sim -\frac{c\chi_0}{m_\phi^2}$ we get the following simplified form of the action:

$$S_{\mathbf{CL}}[\phi, \chi] \approx \int d^4x \sqrt{-g} \left[ -\frac{1}{2} \left( \partial \phi \right)^2 + \frac{m_\phi^2}{2} \phi^2 - \frac{1}{2} \left( \partial \left( \chi + \chi_0 \right) \right)^2 + \frac{m_\chi^2}{2} \chi^2 + c\phi \left( \chi + \chi_0 \right) \right.$$
$$\left. + \frac{c^2}{2} \chi_0^2 + \frac{c^2}{m_\phi^2} \chi \chi_0 \right]. \qquad (2.6)$$

Now we use the field redefinition, $\chi + \chi_0 = \tilde{\chi}$, which further gives:

$$S_{\mathbf{CL}}[\phi, \tilde{\chi}] \approx \int d^4x \sqrt{-g} \left[ -\frac{1}{2} \left( \partial \phi \right)^2 + \frac{m_\phi^2}{2} \phi^2 - \frac{1}{2} \left( \partial \tilde{\chi} \right)^2 + \frac{m_\chi^2}{2} \tilde{\chi}^2 + c\phi\tilde{\chi} \right.$$
$$\left. + \left( \frac{1}{2} \left( c^2 - m_\chi^2 \right) - \left( \frac{c^2}{m_\phi^2} - m_\chi^2 \right) \right) \chi_0^2 + \left( \frac{c^2}{m_\phi^2} - m_\chi^2 \right) \tilde{\chi}\chi_0 \right]. (2.7)$$

4. Further, we assume that $\tilde{\chi}$ is not very far from the field value $\chi_0$ then we finally get:

$$S_{\mathbf{CL}}[\phi, \tilde{\chi}] \approx \int d^4x \sqrt{-g} \left[ -\frac{1}{2} \left( \partial \phi \right)^2 + \frac{m_\phi^2}{2} \phi^2 - \frac{1}{2} \left( \partial \tilde{\chi} \right)^2 + \frac{m^2}{2} \tilde{\chi}^2 + c\phi\tilde{\chi} \right]. \qquad (2.8)$$

Here the following effective potential for the field $\tilde{\chi}$:

$$V_{\text{eff}}(\tilde{\chi}) = \frac{m^2}{2}\tilde{\chi}^2 \qquad \text{where} \qquad m = c, \tag{2.9}$$

where $m^2$ is the space-time-dependent effective mass term of the $\tilde{\chi}$ field. Here it is important to note that, henceforth we will not use the notation $\tilde{\chi}$ and instead of this for simplicity we will write it as $\chi$.

5. In terms of the above mentioned effective potential for the field $\chi$ (which is actually $\tilde{\chi}$) one can further recast the previously mentioned model action as:

$$S_{\mathbf{CL}}[\phi, \chi] = \int d^4x \sqrt{-g}\left[-\frac{1}{2}\left(\partial\phi\right)^2 + \frac{m_\phi^2}{2}\phi^2 - \frac{1}{2}\left(\partial\chi\right)^2 + V_{\text{eff}}(\chi) + c\phi\chi\right], \tag{2.10}$$

using which we now perform the path integration over the field $\phi$ in the next subsection. In this description we use a semi-classical treatment where we consider the background gravity classically and the fields quantum mechanically, which enables the determination of the partition function and path integration over the field $\phi$.

## 2.2   Quantum partition function and effective action

In this section our prime objective is to construct the quantum partition function and the effective action [54] for the field $\chi(x)$ by path integrating over the field $\phi(x)$. To perform this one needs to compute the following quantity:

$$\mathcal{Z}_{\text{eff}}[\chi] := \int \mathfrak{D}\phi \; \exp\left[iS_{\mathbf{CL}}[\phi, \chi]\right] = \exp\left[iS_{\text{eff}}[\chi]\right]. \tag{2.11}$$

However, instead of performing the above mention path integral in the Lorentzian signature we will do it in the Euclidean signature which can be obtained by replacing $S_{\mathbf{CL}}^{\text{eff}}[\phi, \chi]$ with the Euclidean action $iS_{E,\mathbf{CL}}^{\text{eff}}[\phi, \chi]$. In this new notation the above mentioned quantum partition function takes the following simplified form:

$$\mathcal{Z}_{\text{eff}}[\chi] := \int \mathfrak{D}\phi \; \exp\left[-S_{\mathbf{CL}}^E[\phi, \chi]\right] = \exp\left[-S_{\text{eff}}^E[\chi]\right]. \tag{2.12}$$

Here, $S_{\text{eff}}[\chi]$ and $S_{\text{eff}}^E[\chi]$ are the effective action for the field $\chi$ in the Lorentzian and Euclidean signatures, respectively.

In the Euclidean signature the quantum partition function can be further simplified to

the following form:

$$\mathcal{Z}_{\text{eff}}[\chi] = \mathcal{Z}_{\text{eff}}^{(0)}[\chi] \ \exp\left[\int d^4x \ \sqrt{-g(x)} \int d^4y \ \sqrt{-g(y)} \ c(x)\chi(x) \ G_\phi(x,y) \ c(y)\chi(y)\right], \quad (2.13)$$

where $G_\phi(x,y)$ is the Feynman Green's function (or the propagator) in this construction, which appears as a result of the two-point correlation of the $\phi$ field in a specific classical gravitational background. In the context of Quantum Brownian Motion this is commonly identified as the noise kernel. The explicit form of this Feynman Green's function is given by the following expression:

$$G_\phi(x,y) = \left(\frac{1}{\Box_x + m_\phi^2}\right)\left(\frac{\delta^4(x-y)}{\sqrt{-g(x)}}\right), \quad (2.14)$$

where the D'Alembertian operator in general gravitational background can be defined as:

$$\Box_x = \frac{1}{\sqrt{-g(x)}}\partial_\mu\left[\sqrt{-g(x)} \ g^{\mu\nu}(x)\partial_\nu\right] = g^{\mu\nu}(x)\nabla_\mu\nabla_\nu. \quad (2.15)$$

For a given gravitational classical background one can explicitly compute the mathematical structure of this Green's function. Additionally, the quantum partition function in the Euclidean signature without interaction ($c=0$) for the free massive theory of the $\chi$ field is given by the following expression:

$$\mathcal{Z}_{\text{eff}}^{(0)}[\chi] = \mathcal{Z}_{\text{eff}}^{(0)}[0] \ \exp\left[-\int d^4x\sqrt{-g}\left\{\left(-\frac{1}{2}\left(\partial\chi\right)^2 + V_{\text{eff}}(\chi)\right)\right\}\right]. \quad (2.16)$$

Here we define the contribution from the Euclidean quantum partition for the free massive scalar field $\phi$, after doing the path integration, as:

$$\begin{aligned}
\mathcal{Z}_{\text{eff}}^{(0)}[0] &= \int \mathfrak{D}\phi \ \exp\left[-\int d^4x\sqrt{-g}\left\{\left(-\frac{1}{2}\left(\partial\phi\right)^2 + \frac{m_\phi^2}{2}\phi^2\right)\right\}\right]\\
&= \frac{1}{\sqrt{\text{Det}\left(\Box_x + m_\phi^2\right)}}.
\end{aligned} \quad (2.17)$$

From this derived result the effective action for the field $\chi$ can be computed as:

$$\begin{aligned}
S_{\text{eff}}^E[\chi] &= -\ln\left[\mathcal{Z}_{\text{eff}}[\chi]\right]\\
&= \frac{1}{2}\ln\left[\text{Det}\left(\Box_x + m_\phi^2\right)\right] + \int d^4x\sqrt{-g}\left\{\left(-\frac{1}{2}\left(\partial\chi\right)^2 + V_{\text{eff}}(\chi)\right)\right\}\\
&\quad - \int d^4x \ \sqrt{-g(x)} \int d^4y \ \sqrt{-g(y)} \ c(x)\chi(x) \ G_\phi(x,y) \ c(y)\chi(y).
\end{aligned} \quad (2.18)$$

Up to this point the results are valid for any arbitrary general gravitational space-time. Now we derive the results with quasi de Sitter solution described by the following line element written in conformal time coordinate:

$$ds^2 = a^2(\tau)\left(-d\tau^2 + d\mathbf{x}^2\right) \quad \text{where} \quad a(\tau) = -\frac{1}{H\tau} \quad \text{and} \quad \sqrt{-g(\tau)} = a^4(\tau). \quad (2.19)$$

For de Sitter space-time one can explicitly show that:

$$\mathcal{Z}_{\text{eff}}^{(0)}[0] = \frac{1}{2}\text{cosech}\left(\frac{m_\phi T}{2}\right) \quad \text{where} \quad T = \frac{1}{H}\ln\left(-\frac{1}{H\tau_T}\right), \quad (2.20)$$

where $T$ represents the IR cut-off scale on the co-moving time. The physical origin comes from the fact that here during the integration over the co-moving conformal time instead of using $-\infty < \tau < 0$ (which is $0 < t < \infty$) we need to use $-\infty < \tau < \tau_T$ (which is $0 < t < T$) to avoid the IR divergence at the late time scale $T$ where CMB observations take place. This makes $\mathcal{Z}_{\text{eff}}^{(0)}[0]$ finite. However, the final outcomes, which is the correlation functions as well as the equation of motion for the field $\chi$ will be completely independent of such choice in this paper.

Then the corresponding quantum partition function in the quasi de Sitter space can be expressed as:

$$\mathcal{Z}_{\text{eff}}[\chi] = \frac{1}{2}\text{cosech}\left(\frac{m_\phi T}{2}\right)\exp\left[-\int d^4x\sqrt{-g}\left\{\left(-\frac{1}{2}(\partial\chi)^2 + V_{\text{eff}}(\chi)\right)\right\}\right]$$

$$\times \exp\left[\int d^3\mathbf{x}\int d^3\mathbf{y}\int_{-1/H}^{-\exp(-TH)/H}d\tau\,\sqrt{-g(\tau)}\int_{-1/H}^{-\exp(-TH)/H}d\tau'\sqrt{-g(\tau')}\right.$$

$$\left.\times c(\tau)\chi(\mathbf{x},\tau)\,G_\phi(\mathbf{x}-\mathbf{y},\tau,\tau')\,\chi(\mathbf{y},\tau')c(\tau')\right], \quad (2.21)$$

where the noise kernel or the propagator $G_\phi(\mathbf{x}-\mathbf{y},\tau,\tau')$ can be expressed as:

$$\langle\phi(\mathbf{x},\tau)\phi(\mathbf{y},\tau')\rangle = G_\phi(\mathbf{x}-\mathbf{y},\tau,\tau'). \quad (2.22)$$

Additionally, we have:

$$\langle\phi(\mathbf{x},\tau)\rangle = 0 = \langle\phi(\mathbf{y},\tau)\rangle. \quad (2.23)$$

Here we assume that the coupling parameter is only time-dependent in the de Sitter background for simplicity and there are no explicit or implicit dependencies on the space coordinates.

In this computation the temporal part of the propagator or the noise kernel can be

computed as:

$$G_\phi(\mathbf{x} - \mathbf{y}, \tau, \tau') = \frac{1}{4\pi^2} \left| \frac{\Gamma(\nu_\phi)}{\Gamma\left(\frac{3}{2}\right)} \right|^2 \frac{\cosh\left(m_\phi \left\{ |\tau - \tau'| - \frac{T}{2} \right\}\right)}{\sinh\left(\frac{m_\phi T}{2}\right)}$$

$$\times \frac{1}{|\mathbf{x} - \mathbf{y}|^{3-2\nu_\phi}} \times \frac{1}{\left[ |\mathbf{x} - \mathbf{y}|^2 - (|\tau - \tau'| - i\epsilon)^2 \right]}$$

$$= \frac{1}{4\pi^2} \left| \frac{\Gamma(\nu_\phi)}{\Gamma\left(\frac{3}{2}\right)} \right|^2 \exp\left(-m_\phi |\tau - \tau'|\right)$$

$$\times \left( \frac{1 + \exp\left(m_\phi |\tau - \tau'|\right) \exp(-m_\phi T)}{1 - \exp(-m_\phi T)} \right)$$

$$\times \frac{1}{|\mathbf{x} - \mathbf{y}|^{3-2\nu_\phi}} \times \frac{1}{\left[ |\mathbf{x} - \mathbf{y}|^2 - (|\tau - \tau'| - i\epsilon)^2 \right]}, \tag{2.24}$$

where the mass parameter $\nu_\phi$ for the field $\phi$ is given by the following expression:

$$\nu_\phi = \sqrt{\frac{9}{4} - \frac{m_\phi^2}{H^2}}. \tag{2.25}$$

Here, we have used the fact, in two different conformal times $\tau$ and $\tau'$ the Hubble parameters are exactly identical. Here we consider quasi de Sitter phase which is used throughout the paper. Only the tricky part is, instead of using the explicit structure of the interaction potential we are going to use a sudden quench profile in the effective mass of the required $\chi$ field which serves the same purpose in the present work effectively. Such choice actually helps us both theoretically as well from the observational perspective. We all know using just a quadratic potential of the $\chi$ field having constant mass one cannot satisfy strictly the observational constraints on inflation (from the amplitude, tilt of the spectrum and tensor-to-scalar ratio becomes large) using Planck 2018 data. Now if we insert a theoretically justifiable time dependent profile for the coupling parameter $c$ between the $\chi$ and $\phi$ field in the CL model action this will automatically fix the time dependent effective mass of the $\chi$ field. In this work, such time dependent profile is supplied by the quantum mechanical quench, which allows us take a sudden quench profile for the same purpose. Inserting a time dependent profile will going to directly effect the dynamic features before quench, just after quench and after long of the quench. This further implies that it is indirectly modifying the previously mentioned quadratic potential of the $\chi$ field having

constant mass in the present of a time dependent dynamical mass profile. It is important to note that, in the late time limit $\tau_T \to 0$ or $T \to \infty$, then we get the following simplified late time limiting result for the Green's function:

$$
\begin{aligned}
\mathcal{G}_\phi(\mathbf{x} - \mathbf{y}, \tau, \tau') &= \lim_{\tau_T \to 0} G_\phi(\mathbf{x} - \mathbf{y}, \tau, \tau') \\
&= \frac{1}{4\pi^2} \left| \frac{\Gamma(\nu_\phi)}{\Gamma\left(\frac{3}{2}\right)} \right|^2 \exp\left( -m_\phi |\tau - \tau'| \right) \\
&\quad \times \frac{1}{|\mathbf{x} - \mathbf{y}|^{3-2\nu_\phi}} \times \frac{1}{\left[ |\mathbf{x} - \mathbf{y}|^2 - (|\tau - \tau'| - i\epsilon)^2 \right]}.
\end{aligned} \tag{2.26}
$$

Further, varying this semi-classical effective action with respect to the field $\chi$ we get the following equation of motion in de Sitter space:

$$
\begin{aligned}
&\left[ \frac{1}{a^2(\tau)} \left( \frac{\partial^2}{\partial \tau^2} - \nabla^2 + 2\mathcal{H}(\tau) \frac{\partial}{\partial \tau} \right) + m^2(\tau) \right] \chi(\mathbf{x}, \tau) \\
&= \left( \int d^3\mathbf{x} \int d^3\mathbf{y} \int_{-1/H}^{-\exp(-TH)/H} d\tau \sqrt{-g(\tau)} \int_{-1/H}^{-\exp(-TH)/H} d\tau' \sqrt{-g(\tau')} \right. \\
&\quad \left. \times \mathcal{G}_\phi(\mathbf{x} - \mathbf{y}, \tau, \tau') \times c(\tau) c(\tau') \left( \chi(\mathbf{y}, \tau') + \chi(\mathbf{x}, \tau)\delta^3(\mathbf{x} - \mathbf{y})\delta(\tau - \tau') \right) \right), \tag{2.27}
\end{aligned}
$$

which describes the Brownian motion of the $\chi$ field in presence of the noise kernel $\mathcal{G}_\phi(\mathbf{x} - \mathbf{y}, \tau, \tau')$. Here one can consider the following two conditions to analyse the system:

1. One can consider that, $|\tau - \tau'| \to \infty$ which means that $\tau \ll \tau'$ i.e. large separation in time scale, then in this case we have:

$$
\lim_{|\tau - \tau'| \to \infty} \mathcal{G}_\phi(\mathbf{x} - \mathbf{y}, \tau, \tau') \approx 0. \tag{2.28}
$$

2. If we consider that the spatial separation between two points where the field $\chi$ is placed are separated by a large distance scale but in the time scale they are closely separated then we have:

$$
\lim_{|\tau - \tau'| \to 0} \lim_{|\mathbf{x} - \mathbf{y}| \to \infty} \mathcal{G}_\phi(\mathbf{x} - \mathbf{y}, \tau, \tau') \approx 0. \tag{2.29}
$$

In both the limiting cases we have the following simplified form of the equation of motion:

$$\left[\frac{1}{a^2(\tau)}\left(\frac{\partial^2}{\partial \tau^2} - \nabla^2 + 2\mathcal{H}(\tau)\frac{\partial}{\partial \tau}\right) + m^2(\tau)\right]\chi(\mathbf{x}, \tau) = 0, \tag{2.30}$$

For the rest of the analysis we will only concentrate on the free part of the effective action for the $\chi$ field as in both the limits no other terms contribute effectively. Hence we have:

$$S_{\text{eff}}[\chi] \approx \int d^4x \sqrt{-g}\left\{\left(-\frac{1}{2}(\partial\chi)^2 + V_{\text{eff}}(\chi)\right)\right\}. \tag{2.31}$$

Using the conformal coordinates the effective action for the $\chi$ field in the large time limit can be re-expressed as:

$$S_{\text{free}}^{\text{eff}}[\chi] = \frac{1}{2}\int d\tau\ d^3\mathbf{x}\ a^2(\tau)\left[(\partial_\tau\chi(\mathbf{x}, \tau))^2 - (\partial_i\chi(\mathbf{x}, \tau))^2 - m^2(\tau)a^2(\tau)\chi^2(\mathbf{x}, \tau)\right], \tag{2.32}$$

where the conformal time-dependent mass parameter for the field $\chi$ can be written in terms of the interaction strength $c(\tau)$ as $m^2(\tau) = c^2(\tau)$. Here the masses for the field $\chi$ are not initially conformal time-dependent. But since the coupling strength is time-dependent it turns out that the effective mass for the field $\chi$ eventually becomes time-dependent.

## 3    Mass quench in sudden limit in de Sitter space

Quantum quench has been proved to be very effective for probing the dynamics of a system undergoing a change in parameters over a short period of time [41, 42, 55, 56]. The initial wave function or in other words the state corresponding to the Hamiltonian before undergoing a change is called a *pre-quench state* while the state corresponding to the Hamiltonian after quench is called a *post-quench state*. The quench protocol that has been followed in recent times is to consider a mass function $m^2(\tau)$ such that in the sudden limit its value changes from $m_0^2$ in past to 0 in future, interpolating the behavior of correlators at late times. This method is known as sudden quenching of mass parameter from some constant value $m_0^2$ to 0 in the limit $-\tau \to \infty$. Now an important question to ask is do these late time correlators equilibrate and whether or not the post quench state remembers the quench protocol $m^2(\tau)$. In the context of the ADS/CFT correspondence these questions have direct relevance to the memory retention of the black hole of the collapsing matter and been studied in [11, 12, 41, 43, 56–63] by checking whether the post-quench state could be described by a thermal ensemble or not.

Let us start with the previously derived effective action for the dynamical scalar field $\chi$ to implement the phenomena of quantum mechanical quench in the present context:

$$S_{\text{free}}^{\text{eff}}[\chi] = \frac{1}{2} \int d\tau \ d^3\mathbf{x} \ a^2(\tau) \left[ (\partial_\tau \chi(\mathbf{x}, \tau))^2 - (\partial_i \chi(\mathbf{x}, \tau))^2 - m^2(\tau) a^2(\tau) \chi^2(\mathbf{x}, \tau) \right], \quad (3.1)$$

where we have used the de Sitter solution described by the following line element:

$$ds^2 = a^2(\tau) \left( -d\tau^2 + d\mathbf{x}^2 \right) \qquad \text{where} \qquad a(\tau) = -\frac{1}{H\tau}. \qquad (3.2)$$

Here conformal time-dependent quench protocol mass profile for the sudden quench phenomena is given by the following expression:

$$m^2(\tau) = c^2(\tau) = m_0^2 \Theta(-\tau) = \begin{cases} m_0^2 & \textbf{Before quench}: \ \tau < \eta; \\ \\ 0 & \textbf{After quench}: \ \tau \geq \eta, \end{cases} \qquad (3.3)$$

where $\eta$ is considered as the point of quench in the conformal time scale. Further for

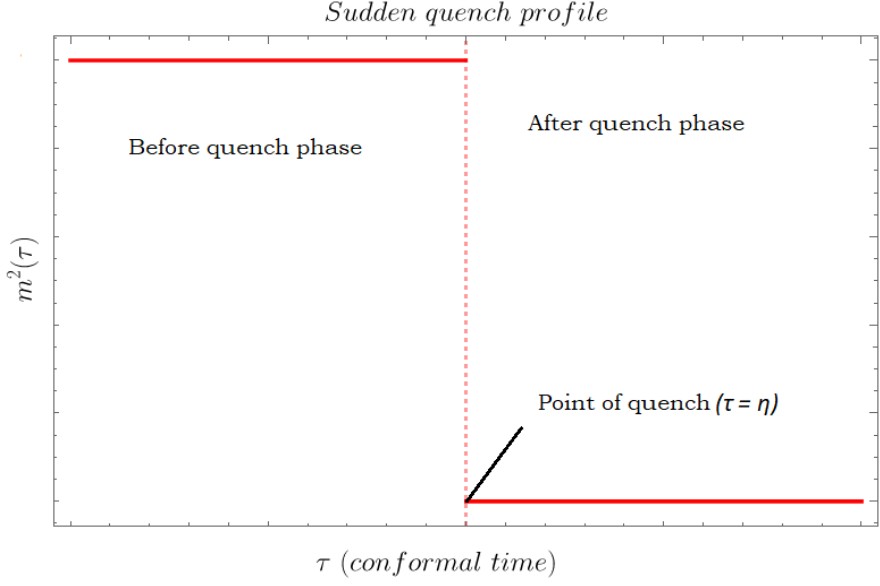

**Figure 3.1**: Mass profile in sudden quench limit.

computational simplicity we use the following redefinition:

$$v(\mathbf{x}, \tau) :\equiv a(\tau) \chi(\mathbf{x}, \tau). \qquad (3.4)$$

Now using this newly defined field $v(\mathbf{x}, \tau)$ one can further re-express the classical effective

action is defined as:

$$S_{\text{free}}^{\text{eff}}[\chi] = \frac{1}{2} \int d\tau \; d^3\mathbf{x} \; \left[ (\partial_\tau v(\mathbf{x}, \tau))^2 - (\partial_i v(\mathbf{x}, \tau))^2 - \left( m^2(\tau) a^2(\tau) - \frac{a''(\tau)}{a(\tau)} \right) v^2(\mathbf{x}, \tau) \right]. \quad (3.5)$$

Next, we choose the following ansatz for the Fourier transform to convert both the effective action and the Hamiltonian in the momentum space:

$$v(\mathbf{x}, \tau) := \int \frac{d^3\mathbf{k}}{(2\pi)^3} \; \exp(i\mathbf{k}.\mathbf{x}) \; v(\mathbf{k}, \tau). \quad (3.6)$$

Using this convention the effective action in Fourier space can be expressed as:

$$S_{\text{free}}^{\text{eff}}[\chi] = \int d\tau \; d^3\mathbf{k} \; \left[ |v'(\mathbf{k}, \tau)|^2 - \omega^2(\mathbf{k}, \tau) |v(\mathbf{k}, \tau)|^2 \right], \quad (3.7)$$

Here we have used the notation $\prime$ to represent the $\partial_\tau$ operation and will use this notation through out the paper.

After varying the action we found the following field equation for the redefined scalar field $v(\mathbf{k}, \tau)$ in Fourier space:

$$\left[ \frac{d^2}{d\tau^2} + \omega^2(\mathbf{k}, \tau) \right] v(\mathbf{k}, \tau) = 0. \quad (3.8)$$

The explicit solutions of the above equations before quench (incoming) and after quench (outgoing) solutions are explicitly derived and studied in the next subsection. This equation in general physically represents the particle production phenomena in de Sitter background [64]. In this work, our prime objective is to solve this classical field equation using the tools and techniques of quantum quench. On top of that, quench also provides us a theoretical framework of thermalization, which we implement in de Sitter space for the first time to study the thermalization process and its impact on quantum correlations in de Sitter space [46, 48]. Since the methodology is developed for conformally flat spacetime, classical solutions other than de Sitter can also be used to study the thermalization phenomena in other cosmologically relevant epochs of our universe.

Here in this construction the effective conformal time-dependent frequency in the Fourier space can be expressed as:

$$\omega^2(\mathbf{k}, \tau) = \left( k^2 + m_{\text{eff}}^2(\tau) \right), \quad (3.9)$$

and the conformal time-dependent effective mass can be expressed in terms of the sudden quench protocol as:

$$m_{\text{eff}}^2(\tau) = \left( m^2(\tau) a^2(\tau) - \frac{a''(\tau)}{a(\tau)} \right) = -\frac{1}{\tau^2} \left( \nu^2(\tau) - \frac{1}{4} \right). \quad (3.10)$$

Here we have used the fact that in the de Sitter space:

$$\frac{a''(\tau)}{a(\tau)} = \left(\mathcal{H}^2(\tau) + \mathcal{H}'(\tau)\right) = \frac{2}{\tau^2} \qquad \text{for} \qquad a(\tau) = -\frac{1}{H\tau}. \tag{3.11}$$

Here $\nu(\tau)$ is the conformal time mass parameter for the given quench protocol:

$$
\nu(\tau) = \sqrt{\frac{9}{4} - \frac{m^2(\tau)}{\mathcal{H}^2}}
$$

$$
= \begin{cases}
\nu_{in} = \sqrt{\dfrac{9}{4} - \dfrac{m_0^2}{\mathcal{H}^2}} & \textbf{Before quench}: \ \tau < \eta; \\[3mm]
\nu_{out} = \dfrac{3}{2} & \textbf{After quench}: \ \tau \geq \eta.
\end{cases} \tag{3.12}
$$

As mentioned above a mass quenching in the sudden limit is considered, i.e., we take a mass function $m^2(\tau)$ and change its value from $m_0^2$ to 0 in the future using which we compute the quantum correlators. Specifically in this paper we have computed the two-point correlators.

The present problem describing the particle production in de Sitter space can be translated in the language of Schrödinger quantum mechanics as a problem in 1 dimension, where one needs to study the movement of an electron inside an electrical wire in the presence of an impurity. This impurity is the quantum mechanical potential which is appearing in the corresponding Schrödinger equation:

$$\left[\frac{d^2}{dx^2} + (E - V(x))\right]\psi(x) = 0. \tag{3.13}$$

In this interpretation the following one-to-one map is set up between the particle production problem and the Schrödinger quantum mechanical problem:

$$\text{Distance } \ x \quad \longleftrightarrow \quad \text{Conformal time } \ \tau, \tag{3.14}$$

$$\text{Quantum impurity potential } \ V(x) \quad \longleftrightarrow \quad \text{Effective quench protocol } \ -m_{\text{eff}}^2(\tau), \tag{3.15}$$

$$\text{Quantum wave function } \ \psi(x) \quad \longleftrightarrow \quad \text{Rescaled mode function } \ v(\mathbf{k}, \tau). \tag{3.16}$$

In studying the behavior of wave functions in the quench protocol one of the main approximations we usually employ is solving the Klein-Gordon equation for constant masses instead for time-dependent parameter $m^2(\tau)$ which in turn is very difficult to interpolate. By doing the approximation $m^2(\tau) = m_0^2 = $ constant and repeating the procedure for each

recursion we get more and more precise results for the effective mass. However, in the context of sudden quenching we choose transition in masses close to zero and this diminishes our need for repeated iteration. In this we will also study quenches for masses close to zero because they correspond to half integer orders of the Hankel function which makes the wave-functions easy to interpolate. As mentioned above quantum quench, which corresponds to the change in the parameters of Hamiltonian for a short period of time has been employed in various areas. Starting from the study of various phenomenon under various regimes from studying the behavior of thermalization of correlators at late times in the de Sitter space-time, where the value of post-quench parameters doesn't depend on the quench protocol [10, 65–69]. In this paper, we are going to study the behavior of fields in terms of the correlators in intermediate time scales, we will encode the effects of the fields on the correlators through the quench profile followed by the mass parameter in the Hamiltonian of the field.

### 3.1 Solution of mode equation in de Sitter space

In this section, we study the solution of the equation of motion of the Fourier modes of the rescaled field in de Sitter background with scale factor $a(\tau) = -1/H\tau$, participating in the quantum quench driven Brownian motion [51, 70–72], which are given by:

$$\textbf{Before quench}: \qquad \left[\frac{d^2}{d\tau^2} + \omega_{\text{eff,in}}^2(k, \tau)\right] v_{in}(\mathbf{k}, \tau) = 0, \qquad (3.17)$$

$$\textbf{After quench}: \qquad \left[\frac{d^2}{d\tau'^2} + \omega_{\text{eff,out}}^2(k, \tau')\right] v_{out}(\mathbf{k}, \tau') = 0, \qquad (3.18)$$

where $v_{in}(\mathbf{k}, \tau)$ and $v_{out}(\mathbf{k}, \tau')$ signify the incoming and the outgoing solutions of the rescaled field, and particularly in the present context these play the role of the classical solution of the equation of motion before and after the quench mechanism. Due to having quantum quench in the time-dependent effective mass profile at a particular conformal time scale one can differentiate the solutions with respect to the mass parameters involved in the time-dependent effective frequencies, which are given by the following expressions:

$$\omega_{\text{eff,in}}^2(k, \tau) := \left(k^2 - \frac{\nu_{in}^2 - \frac{1}{4}}{\tau^2}\right) \qquad \text{with} \quad \nu_{in} = \sqrt{\frac{9}{4} - \frac{m_0^2}{H^2}}, \qquad (3.19)$$

$$\omega_{\text{eff,out}}^2(k, \tau') := \left(k^2 - \frac{\nu_{out}^2 - \frac{1}{4}}{\tau'^2}\right) \qquad \text{with} \quad \nu_{out} = \frac{3}{2}. \qquad (3.20)$$

Here it is important to note that, $\tau$ is the associated conformal time scale before the mass quench operation. Also $\tau' = \tau + \eta$ is the associated conformal time scale after the mass quench operation, where the quench is performed at the point $\eta$ in the forward direction in the conformal time scale.

Now, the solution of the mode equations in the Fourier space before and after quenched

mass profile can be written in spatially flat background de Sitter space as:

Before quench : $\qquad v_{in}(\mathbf{k}, \tau) = \sqrt{-\tau} \, [d_1 H^{(1)}_{\nu_{in}}(-k\tau) + d_2 H^{(2)}_{\nu_{in}}(-k\tau)],$ $\qquad$ (3.21)

After quench : $\qquad v_{out}(\mathbf{k}, \tau) = \sqrt{-\tau'} \, [d_3 H^{(1)}_{\nu_{out}}(-k\tau') + d_4 H^{(2)}_{\nu_{out}}(-k\tau')],$ $\quad$ (3.22)

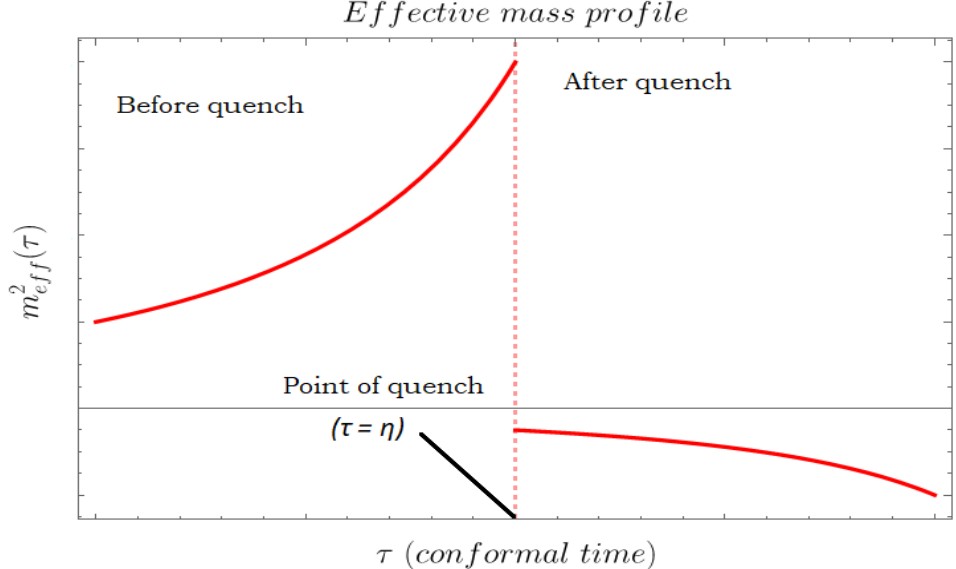

**Figure 3.2**: Effective mass profile.

where the solutions appear as linear combinations of the Hankel function of the first and second kind of order $\nu_{in}$ for the incoming and $\nu_{out}$ outgoing solutions.

It is important to note that here we have the following total effective mass for the sudden mass quench profile:

$$m^2_{\text{eff}}(\tau) = \frac{1}{\tau^2}\left(\frac{m^2(\tau)}{H^2} - 2\right) = \begin{cases} \dfrac{1}{\tau^2}\left(\dfrac{m_0^2}{H^2} - 2\right) & \textbf{Before quench}: \ \tau < \eta; \\[4mm] -\dfrac{2}{(\tau + \eta)^2} & \textbf{After quench}: \ \tau \geq \eta. \end{cases} \qquad (3.23)$$

This is plotted in Fig. (3.2). If we closely look into the obtained analytical solutions for the ingoing and the outgoing modes then we see that both the solutions are fixed with respect to choice of constants $d_1$, $d_2$ and $d_3$ , $d_4$. Due to having mass quench at a preferred conformal time scale $\eta$ in this particular set up the constants appearing in the outgoing after quench solution, $d_3$ and $d_4$ can be determined in terms of the constants appearing in the incoming before quench solution, $d_1$ and $d_2$. The specific choices for these constants

can be fixed by choosing the following set of quantum initial conditions [73]:

$$\textcolor{red}{\textbf{Bunch – Davies vacuum}} : \quad d_1 = 1, \qquad d_2 = 0, \qquad (3.24)$$

$$\textcolor{red}{\boldsymbol{\alpha} \textbf{ vacua}} : \qquad\qquad d_1 = \cosh\alpha, \qquad d_2 = \sinh\alpha, \qquad (3.25)$$

$$\textcolor{red}{\textbf{Motta – Allen vacua}} : \quad d_1 = \cosh\alpha, \qquad d_2 = \exp(i\gamma)\sinh\alpha. \quad (3.26)$$

For the Bunch-Davies case [74–76] we will get very simple expressions, though the expressions for the $\alpha$ or Motta-Allen case will become complicated. To avoid confusion during the computation we do not substitute these values of the constants for the three different choices of the quantum initial conditions. However, during the numerical computations from the obtained results we will use them explicitly to determine the differences in behavior. In the appendices we present some results pertaining to these initial conditions for completeness. Our result, presented here, are valid for the any arbitrary choice of the quantum initial conditions, out of which for numerical purpose we will only focus on the three above mentioned possibilities.

The Eqs.(3.21) represents the most general solution valid for all time scales. However, working with these general solutions is often cumbersome and the asymptotic limits of the above solutions are found convenient for analysis. The Hankel functions in these asymptotic limits can be expressed as:

$$\textcolor{red}{\textbf{Sub – horizon asymptotic expansion}} :$$

$$\lim_{-k\tau \to \infty} H_\nu^{(1)} = \sqrt{\frac{2}{\pi}} \frac{1}{\sqrt{-k\tau}} \exp(-i\{k\tau + \Delta_\nu\}), \quad (3.27)$$

$$\lim_{-k\tau \to \infty} H_\nu^{(2)} = -\sqrt{\frac{2}{\pi}} \frac{1}{\sqrt{-k\tau}} \exp(i\{k\tau + \Delta_\nu\}), \quad (3.28)$$

$$\textcolor{red}{\textbf{Super – horizon asymptotic expansion}} :$$

$$\lim_{-k\tau \to 0} H_\nu^{(1)} = \frac{i}{\pi}\Gamma(\nu)\left(\frac{-k\tau}{2}\right)^{(-\nu)}, \qquad (3.29)$$

$$\lim_{-k\tau \to 0} H_\nu^{(2)} = -\frac{i}{\pi}\Gamma(\nu)\left(\frac{-k\tau}{2}\right)^{(-\nu)}. \qquad (3.30)$$

where we define the factor $\Delta_\nu$ as:

$$\Delta_\nu = \frac{\pi}{2}\left(\nu + \frac{1}{2}\right) = \begin{cases} \dfrac{\pi}{2}\left(\nu_{in} + \dfrac{1}{2}\right) \quad \text{with} \quad \nu_{in} = \sqrt{\dfrac{9}{4} - \dfrac{m_0^2}{H^2}} \quad \textcolor{red}{\textbf{Before quench :}} \ \tau < \eta; \\[4mm] \dfrac{\pi}{2}\left(\nu_{out} + \dfrac{1}{2}\right) \quad \text{with} \quad \nu_{out} = \dfrac{3}{2} \qquad\qquad\qquad \textcolor{red}{\textbf{After quench :}} \ \tau \geq \eta. \end{cases}$$

$$(3.31)$$

Let us now discuss the solution of the above equation in the sub horizon limit where modes of quantum fluctuations are inside the cosmological horizon, it behaves like a quantum mechanical plane wave. In the limit $-k\tau \to \infty$ $(-k\tau \gg 1)$, using the above limiting solutions of the Hankel functions, the fluctuation solution reduces to:

**Sub-horizon asymptotic incoming solution:**

$$v_{in}(\mathbf{k},\tau)|_{-k\tau\to\infty} = \sqrt{\frac{2}{\pi k}}\left[d_1\exp\left\{-i\left(k\tau+\frac{\pi}{2}\left(\nu_{in}+\frac{1}{2}\right)\right)\right\} - d_2\exp\left\{-i\left(k\tau+\frac{\pi}{2}\left(\nu_{in}+\frac{1}{2}\right)\right)\right\}\right],$$
(3.32)

$$\Pi_{in}(\mathbf{k},\tau)|_{-k\tau\to\infty} = \frac{1}{i}\sqrt{\frac{2k}{\pi}}\left[d_1\exp\left\{-i\left(k\tau+\frac{\pi}{2}\left(\nu_{in}+\frac{1}{2}\right)\right)\right\} + d_2\exp\left\{-i\left(k\tau+\frac{\pi}{2}\left(\nu_{in}+\frac{1}{2}\right)\right)\right\}\right],$$
(3.33)

where $\Pi_{in}(\mathbf{k},\tau)$ is the canonically conjugate momentum of the field $v_{in}(\mathbf{k},\tau)$, which is defined as, $\Pi_{in}(\mathbf{k},\tau) = v'_{in}(\mathbf{k},\tau)$.

On the other hand, in the super-horizon limit when the fluctuating modes are goes outside the cosmological horizon it behaves classically. In the limit $-k\tau \to 0$ $(-k\tau \ll 1)$, using the above limiting solutions of the Hankel functions, the fluctuation solution reduces to:

**Super-horizon asymptotic incoming solution:**

$$v_{in}(\mathbf{k},\tau)|_{-k\tau\to 0} = \sqrt{\frac{2}{k}}\frac{i}{\pi}\Gamma(\nu_{in})\left(\frac{-k\tau}{2}\right)^{\frac{1}{2}-\nu_{in}}(d_1-d_2),$$
(3.34)

$$\Pi_{in}(\mathbf{k},\tau)|_{-k\tau\to 0} = \sqrt{\frac{2}{k}}\frac{i}{2\pi k}\left(\nu_{in}-\frac{1}{2}\right)\Gamma(\nu_{in})\left(\frac{-k\tau}{2}\right)^{-(\nu_{in}+\frac{1}{2})}(d_1-d_2).$$
(3.35)

**Sub-horizon asymptotic outgoing solution:**

$$v_{out}(\mathbf{k},\tau)|_{-k\tau\to\infty} = \sqrt{\frac{2}{\pi k}}\left[d_3\exp\left\{-i\left(k(\tau+\eta)+\frac{\pi}{2}\left(\nu_{out}+\frac{1}{2}\right)\right)\right\}\right.$$
$$\left. - d_4\exp\left\{-i\left(k(\tau+\eta)+\frac{\pi}{2}\left(\nu_{out}+\frac{1}{2}\right)\right)\right\}\right],$$
(3.36)

$$\Pi_{out}(\mathbf{k},\tau)|_{-k\tau\to\infty} = \frac{1}{i}\sqrt{\frac{2k}{\pi}}\left[d_3\exp\left\{-i\left(k\tau+\frac{\pi}{2}\left(\nu_{out}+\frac{1}{2}\right)\right)\right\}\right.$$
$$\left. + d_4\exp\left\{-i\left(k\tau+\frac{\pi}{2}\left(\nu_{out}+\frac{1}{2}\right)\right)\right\}\right].$$
(3.37)

**Super-horizon asymptotic outgoing solution:**

$$v_{out}(\mathbf{k},\tau)|_{-k\tau\to 0} = \sqrt{\frac{2}{k}}\frac{i}{\pi}\Gamma(\nu_{out})\left(\frac{-k(\tau+\eta)}{2}\right)^{\frac{1}{2}-\nu_{out}}(d_3-d_4),$$
(3.38)

$$\Pi_{out}(\mathbf{k}, \tau)|_{-k(\tau+\eta)\to 0} = \sqrt{\frac{2}{k}} \frac{i}{2\pi k}\left(\nu_{out} - \frac{1}{2}\right)\Gamma(\nu_{out})\left(\frac{-k(\tau+\eta)}{2}\right)^{-(\nu_{out}+\frac{1}{2})}(d_3 - d_4), \quad (3.39)$$

where $\Pi_{out}(\mathbf{k}, \tau)$ is the canonically conjugate momentum of the field $v_{out}(\mathbf{k}, \tau)$, which is defined as, $\Pi_{out}(\mathbf{k}, \tau) = v'_{out}(\mathbf{k}, \tau)$.

Combining the above two limiting solutions, the asymptotic solution of the mode equation can be written as:

**Asymptotic solution for the mode before quench:**

$$v_{in}(\mathbf{k}, \tau) = \frac{2^{\nu_{in}-\frac{3}{2}} \ i \ (-k\tau)^{\frac{3}{2}-\nu_{in}}}{\sqrt{2}k^{3/2}\tau}\left|\frac{\Gamma(\nu_{in})}{\Gamma(3/2)}\right| \times \left[d_1(1 + ik\tau)\exp\left(-i\left\{k\tau + \frac{\pi}{2}(\nu_{in} + \frac{1}{2})\right\}\right)\right.$$
$$\left. - d_2(1 - ik\tau)\exp\left(i\left\{k\tau + \frac{\pi}{2}(\nu_{in} + \frac{1}{2})\right\}\right)\right]. \quad (3.40)$$

The above equation basically represents the incoming solution before the point of quench. Similarly, the general expression for the canonically conjugate momentum variable for the incoming solutions (solution before the point of quench) in this asymptotic limit simplifies to the following expression:

**Asymptotic momentum before quench:**

$$\Pi_{in}(\mathbf{k}, \tau) = \frac{2^{\nu_{in}-\frac{3}{2}} \ i \ (-k\tau)^{\frac{3}{2}-\nu_{in}}}{\sqrt{2}k^{5/2}}\left|\frac{\Gamma(\nu_{in})}{\Gamma(3/2)}\right|\left[d_1\left\{\left(\frac{1}{2} - \nu_{in}\right)\frac{(1 + ik\tau)}{k^2\tau^2} + 1\right\}\right.$$
$$\exp\left(-i\left\{k\tau + \frac{\pi}{2}(\nu_{in} + \frac{1}{2})\right\}\right) - d_2\left\{\left(\frac{1}{2} - \nu_{in}\right)\frac{(1 + ik\tau)}{k^2\tau^2} + 1\right\}$$
$$\left.\exp\left(i\left\{k\tau + \frac{\pi}{2}\left(\nu_{in} + \frac{1}{2}\right)\right\}\right)\right]. \quad (3.41)$$

By following the same logical argument, the outgoing solutions can be calculated as:

**Asymptotic solution for the mode after quench:**

$$v_{out}(\mathbf{k}, \tau) = \frac{2^{\nu_{out}-\frac{3}{2}} \ i \ (-k(\tau+\eta))^{\frac{3}{2}-\nu_{out}}}{\sqrt{2}k^{3/2}(\tau+\eta)}\left|\frac{\Gamma(\nu_{out})}{\Gamma(3/2)}\right| \times \left[d_3(1 + ik(\tau+\eta))\exp\left(-i\left\{k(\tau+\eta)\right.\right.\right.$$
$$\left.\left.\left. + \frac{\pi}{2}\left(\nu_{out} + \frac{1}{2}\right)\right\}\right) - d_4(1 - ik(\tau+\eta))\exp\left(i\left\{k(\tau+\eta) + \frac{\pi}{2}\left(\nu_{out} + \frac{1}{2}\right)\right\}\right)\right].$$
$$(3.42)$$

The canonically conjugate momentum variable for the outgoing solution can also be described as:

**Asymptotic momentum after quench:**

$$\Pi_{out}(\mathbf{k}, \tau) = \frac{2^{\nu_{out}-\frac{3}{2}} \ i \ (-k(\tau+\eta))^{\frac{3}{2}-\nu_{out}}}{\sqrt{2}k^{5/2}}\left|\frac{\Gamma(\nu_{out})}{\Gamma(3/2)}\right|\left[d_3\left\{\left(\frac{1}{2} - \nu_{out}\right)\frac{(1 + ik(\tau+\eta))}{k^2(\tau+\eta)^2} + 1\right\}\right.$$

$$\exp\left(-i\left\{k(\tau+\eta)+\frac{\pi}{2}\left(\nu_{out}+\frac{1}{2}\right)\right\}\right)-d_4\left\{\left(\frac{1}{2}-\nu_{out}\right)\frac{(1+ik(\tau+\eta))}{k^2(\tau+\eta)^2}+1\right\}$$
$$\exp\left(i\left\{k(\tau+\eta)+\frac{\pi}{2}\left(\nu_{out}+\frac{1}{2}\right)\right\}\right)\Bigg)\Bigg].$$

If we closely look into the expressions for the field variables and their associated canonically conjugate momentum variables for the incoming and outgoing situations then we see that the solutions differ, (A). in terms of the mass parameters $\nu_{in}$ and $\nu_{out}$ and (B). in terms of the constants $d_i\forall i=1,\cdots,4$. As we have already mentioned, one can compute the expressions for the outgoing constants, $d_3$ and $d_4$ in terms of the incoming constants, $d_1$ and $d_2$, thereby expressing the incoming solution in terms of the outgoing solution or vice versa using the Bogoliubov transformation technique. This technique is particularly useful in the present context, not just for expressing one solution in terms of the other, but also for constructing the ground state as well as the excited generalized Calabresse Cardy (gCC) states, which are the key ingredients for computing the two-point functions for both the cases. The two-point functions also play another role here . They tell us that how the quantum correlations can be explicitly quantified when the system tending to thermalize. For the flat space-time, particularly in $1+1$ dimensional system this formalism is easily understandable and was explicitly studied in [41]. Later this work was generalized to $1+2$ dimensions in [42]. But there has been no such development in the presence of background classical gravitational solution. The presented technique in this paper will going to be an attempt for a very simplest case, where the space-time is described by de Sitter solution. The results that we have have obtained in this paper is an attempt to understand the underlying physical phenomena and its related physical explanation of the thermalization phenomena in de Sitter space-time in presence of sudden mass quench. We now develop the tools which would be needed for the mentioned purpose.

To determine the outgoing coeffcients $d_3$ and $d_4$ in terms of the ingoing coefficients $d_1$ and $d_2$ one needs to use the following two cruicial conditions:

1. **Continuity in the field variable:**

   First of all, the solution obtained before quench and after quench has to be continuous at the point of quench $\eta$, i.e.,

   $$v_{in}(\mathbf{k},\tau)|_{\tau=\eta}=v_{out}(\mathbf{k},\tau)|_{\tau=\eta}. \tag{3.43}$$

2. **Continuity in the momentum variable:**

Secondly, the canonically conjugate momenta obtained from both the solutions before quench and after quench has to be continuous at the point of quench $\eta$, i.e.,

$$\Pi_{in}(\mathbf{k}, \tau)|_{\tau=\eta} = \Pi_{out}(\mathbf{k}, \tau)|_{\tau=\eta}. \tag{3.44}$$

Again using the continuity condition of the solutions and its derivatives at the point of quench we can fix the constants $d_3$ and $d_4$ in terms of $d_1$ and $d_2$. It can be easily found that the constants $d_3$ and $d_4$ expressed in terms of $d_1$ and $d_2$ can be written as:

$$d_3 = \frac{2^{\nu_{in}-\frac{9}{2}} \exp(i\eta k)}{k\eta} \left[ d_1(6\eta k - 3i) + id_2(2\eta k + 3i) \exp(i(2\eta k + \pi\nu_{in})) \right], \tag{3.45}$$

$$d_4 = \frac{2^{\nu_{in}-\frac{9}{2}} \exp\{-i(3k\eta + \pi\nu_{in})\}}{k\eta} \left[ -d_1(3 + 2ik\eta) + 3d_2 \exp\{i(2k\eta + \pi\nu_{in})\}(i + 2k\eta) \right]. \tag{3.46}$$

Here it is important to note that, incoming and the outgoing mode functions before and after quench can be expressed in terms of each other via the following relations:

$$v_{in}(\mathbf{k}, \tau) = \alpha(k, \eta) \, v_{out}(\mathbf{k}, \tau) + \beta(k, \eta) \, v_{out}^*(-\mathbf{k}, \tau), \tag{3.47}$$

$$v_{out}(\mathbf{k}, \tau) = \alpha^*(k, \eta) \, v_{in}(\mathbf{k}, \tau) - \beta(k, \eta) \, v_{in}^*(-\mathbf{k}, \tau). \tag{3.48}$$

Consequently, the general solution for the field equation can be written as:

$$\begin{aligned} v(\mathbf{k}, \tau) &= a_{in}(\mathbf{k})v_{in}(\mathbf{k}, \tau) + a_{in}^\dagger(-\mathbf{k})v_{in}^*(-\mathbf{k}, \tau) \\ &= a_{out}(\mathbf{k})v_{out}(\mathbf{k}, \tau) + a_{out}^\dagger(-\mathbf{k})v_{out}^*(-\mathbf{k}, \tau), \end{aligned} \tag{3.49}$$

which satisfy the following reality constraint:

$$v^*(\mathbf{k}, \tau) = v(-\mathbf{k}, \tau). \tag{3.50}$$

Using these above mentioned equations one can explicitly show that:

$$a_{in}(\mathbf{k}) = \alpha^*(k, \eta)a_{out}(\mathbf{k}) - \beta^*(k, \eta)a_{out}^\dagger(-\mathbf{k}), \tag{3.51}$$

$$a_{out}(\mathbf{k}) = \alpha^*(k, \eta)a_{in}(\mathbf{k}) + \beta^*(k, \eta)a_{in}^\dagger(-\mathbf{k}). \tag{3.52}$$

Here the Bogolyubov coefficients at the point of quench $\eta$, are calculated using the following equations:

$$\alpha(k, \eta) = \left. \frac{v_{out}'(\mathbf{k}, \tau)v_{in}^*(\mathbf{k}, \tau) - v_{out}(\mathbf{k}, \tau)v_{in}'^*(\mathbf{k}, \tau)}{2i} \right|_\eta, \tag{3.53}$$

$$\beta^*(k, \eta) = \frac{v'_{out}(\mathbf{k}, \tau)v_{in}(\mathbf{k}, \tau) - v_{out}(\mathbf{k}, \tau)v'_{in}(\mathbf{k}, \tau)}{2i}\Bigg|_{\eta}. \qquad (3.54)$$

Using the above equation the Bogoliubov coefficients for our quench profile can be calculated as

$$\begin{aligned}
\alpha(k, \eta) = \frac{2^{2\nu_{in}-5}}{\pi} &\exp\{-i(2k\eta + \pi\nu_{in})\}(-k\eta)^{-2\nu_{in}}\Bigg[d_1 d_2^*(1 + ik\eta)(1 + k\eta(i + 2k\eta - 2i\nu_{in}) - 2\nu_{in}) \\
&+ d_1^* d_2 \exp\{2i(2k\eta + \pi\nu_{in})\}(i + k\eta)(i + k\eta(1 + 2ik\eta - 2\nu_{in}) - 2i\nu_{in}) \\
&- d_2 d_2^* \exp\{i(2k\eta + \pi\nu_{in})\}\ (i + k^2\eta^2(3i + 6k\eta - 2i\nu_{in}) - 2i\nu_{in}) \\
&+ d_1 d_1^* \exp\{i(2k\eta + \pi\nu_{in})\}\ (k^2\eta^2(-3i + 6k\eta + 2i\nu_{in}) + i(-1 + 2\nu_{in}))\Bigg]|\Gamma(\nu_{in})|^2,
\end{aligned}$$
$$(3.55)$$

$$\begin{aligned}
\beta(k, \eta) = \frac{2^{2\nu_{in}-5}}{\pi} &\exp\{i(2k\eta + \pi\nu_{in})\}(-k\eta)^{-2\nu_{in}}\Bigg[d_1(i + k\eta) - id_2 \exp\{-i(2k\eta + \pi\nu_{in})\}(-i + k\eta)\Bigg] \\
&\Bigg[d_2 \exp\{-i(2k\eta + \pi\nu_{in})\}(1 + k\eta(i + 2k\eta - 2i\nu_{in}) - 2\nu_{in}) \\
&+ d_1(-i + 2i\nu_{in} + k\eta(-1 - 2ik\eta + 2\nu_{in}))\Bigg]|\Gamma(\nu_{in})|^2. \qquad (3.56)
\end{aligned}$$

Once the Bogoliubov coefficients is found for a given quench profile, one defines a quantity $\gamma(k)$ which is defined as

$$\gamma(k) = \frac{\beta^*(k, \eta)}{\alpha^*(k, \eta)}, \qquad (3.57)$$

where in principle the coefficient $\gamma$ is functions of both $k$ and $\eta$, but for a given fixed value of the quench time scale, the coefficient $\gamma$ turns out to be a function of $k$ only.

Another quantity that will be of significance in the formulation of the in states is defined as

**For Dirichlet boundary state** : $\qquad \kappa(k) = -\frac{1}{2}\log(-\gamma(k)), \qquad (3.58)$

**For Neumann boundary state** : $\qquad \kappa(k) = -\frac{1}{2}\log(\gamma(k)). \qquad (3.59)$

A power series expansion of $\kappa$ and $\gamma$ around $k = 0$ gives us the conserved charges. In [41], the authors have explicitly found out the relationship between various coefficients of $\gamma(k)$ and $\kappa(k)$. For the quench profile considered above, it can be found that the series expansion of $\gamma(k)$ can be written as.

$$\gamma(k) = \gamma_0 + \gamma_2|k| + \gamma_3|k|^2 + \gamma_4|k|^3 + \gamma_5|k|^4 + \gamma_6|k|^5 + .... \qquad (3.60)$$

and the corresponding $\kappa(k)$ parameter for the Dirichlet and Neumann boundary states can be expressed in terms of the following series expansions around $k = 0$, as given by:

**For Dirichlet boundary state** :

$$\kappa(k) = \left( \kappa_{0,\mathbf{DB}} + \sum_{n=1}^{\infty} \kappa_{n+1,\mathbf{DB}} |k|^n \right), \qquad (3.61)$$

**For Neumann boundary state** :

$$\kappa(k) = \left( \kappa_{0,\mathbf{NB}} + \sum_{n=1}^{\infty} \kappa_{n+1,\mathbf{NB}} |k|^n \right), \qquad (3.62)$$

where it is important to note that:

$$\kappa_{0,\mathbf{DB}} = \left( \kappa_{0,\mathbf{NB}} + \frac{i\pi}{2} \right), \quad \text{and} \quad \kappa_{n+1,\mathbf{DB}} = \kappa_{n+1,\mathbf{NB}} \quad \forall \quad n = 1, 2, 3, \cdots, \infty \quad (3.63)$$

In this expansion the various non-vanishing coefficients of $\gamma(k)$ can be easily verified to be:

$$\gamma_0 = -\frac{id_1 + d_2 \exp(i\pi\nu_{in})}{id_2^* + d_1^* \exp(i\pi\nu_{in})}, \qquad (3.64)$$

$$\gamma_4 = -\frac{2(d_1 d_1^* - d_2 d_2^*) \exp(i\pi\nu_{in})\eta^3 (5 + 2\nu_{in})}{3((id_2^* + d_1^* \exp(i\pi\nu_{in}))^2(-1 + 2\nu_{in}))}, \qquad (3.65)$$

$$\gamma_6 = \frac{2(d_1 d_1^* - d_2 d_2^*) \exp(i\pi\nu_{in})\eta^5 (-29 + 4\nu_{in}(4 + \nu_{in}))}{5((id_2^* + d_1^* \exp(i\pi\nu_{in}))^2(1 - 2\nu_{in})^2)}. \qquad (3.66)$$

Similarly, the non-vanishing coefficients of the $\kappa(k)$ expansion can be calculated for Dirichlet and Neumann boundary state in the present context, which we have quoted explicitly in the Appendix A.

Thus for our quench profile, the relationship between the various coefficients of $\kappa(k)$ and $\gamma(k)$ can be found out. However, before doing that it can be seen that for the expansion contains an first constant term which is independent of $|k|$ and thus only acts as a phase for the states expressed in terms of them.

$$\kappa_{4,\mathbf{DB}} = \kappa_{4,\mathbf{NB}} = \frac{i}{2} \left( \frac{id_2^* + d_1^* \exp(i\pi\nu_{in})}{d_1 - id_2 \exp(i\pi\nu_{in})} \right) \gamma_4 = \frac{1}{2} \left( \frac{d_1 + id_2 \exp(i\pi\nu_{in})}{d_1 - id_2 \exp(i\pi\nu_{in})} \right) \frac{\gamma_4}{\gamma_0} \quad (3.67)$$

$$\kappa_{6,\mathbf{DB}} = \kappa_{6,\mathbf{NB}} = \frac{1}{2} \left( \frac{id_2^* + d_1^* \exp(i\pi\nu_{in})}{id_1 + d_2 \exp(i\pi\nu_{in})} \right) \gamma_6 = \frac{1}{2} \left( \frac{-id_1 + d_2 \exp(i\pi\nu_{in})}{id_1 + d_2 \exp(i\pi\nu_{in})} \right) \frac{\gamma_6}{\gamma_0} \quad (3.68)$$

The explicit expressions of the above coefficients for the three different choices of quantum initial conditions has been given in the Appendix.A

Additionally it is important to point that, the classical solution of the field $\chi$ can be

promoted further as a quantum operator by the following expression:

$$\hat{\chi}(\mathbf{k}, \tau) = \frac{a_{in}(\mathbf{k})v_{in}(\mathbf{k}, \tau) + a_{in}^\dagger(-\mathbf{k})v_{in}^*(-\mathbf{k}, \tau)}{a(\tau)} \tag{3.69}$$

$$= \frac{a_{out}(\mathbf{k})v_{out}(\mathbf{k}, \tau) + a_{out}^\dagger(-\mathbf{k})v_{out}^*(-\mathbf{k}, \tau)}{a(\tau)}, \tag{3.70}$$

where additionally the following reality condition in Fourier space has to be satisfied:

$$\hat{\chi}^*(\mathbf{k}, \tau) = \hat{\chi}(-\mathbf{k}, \tau). \tag{3.71}$$

By following this identification at the quantum level the canonically conjugate momentum operator corresponding to the field operator $\hat{\chi}(\mathbf{k}, \tau)$ can be expressed as:

$$\hat{\Pi}_\chi(\mathbf{k}, \tau) = \frac{a_{in}(\mathbf{k})v_{in}'(\mathbf{k}, \tau) + a_{in}^\dagger(-\mathbf{k})v_{in}^{*\prime}(-\mathbf{k}, \tau)}{a(\tau)} - \frac{a_{in}(\mathbf{k})v_{in}(\mathbf{k}, \tau) + a_{in}^\dagger(-\mathbf{k})v_{in}^*(-\mathbf{k}, \tau)}{a^2(\tau)}a'(\tau)$$

$$\tag{3.72}$$

$$= \frac{a_{out}(\mathbf{k})v_{out}'(\mathbf{k}, \tau) + a_{out}^\dagger(-\mathbf{k})v_{out}^{*\prime}(-\mathbf{k}, \tau)}{a(\tau)} - \frac{a_{out}(\mathbf{k})v_{out}(\mathbf{k}, \tau) + a_{out}^\dagger(-\mathbf{k})v_{out}^*(-\mathbf{k}, \tau)}{a^2(\tau)}a'(\tau).$$

$$\tag{3.73}$$

Further using Eq (3.69) and Eq (3.70) in Eq (3.74) and Eq (3.75) we finally get the following simplified form of the momentum operator:

$$\hat{\Pi}_\chi(\mathbf{k}, \tau) = \frac{a_{in}(\mathbf{k})\Pi_{in}(\mathbf{k}, \tau) + a_{in}^\dagger(-\mathbf{k})\Pi_{in}^*(-\mathbf{k}, \tau)}{a(\tau)} - \frac{\hat{\chi}(\mathbf{k}, \tau)}{a(\tau)}a'(\tau) \tag{3.74}$$

$$= \frac{a_{out}(\mathbf{k})\Pi_{out}(\mathbf{k}, \tau) + a_{out}^\dagger(-\mathbf{k})\Pi_{out}^*(-\mathbf{k}, \tau)}{a(\tau)} - \frac{\hat{\chi}(\mathbf{k}, \tau)}{a(\tau)}a'(\tau), \tag{3.75}$$

where we define the canonically conjugate momenta for the incoming and outgoing modes as:

$$\Pi_{in}(\mathbf{k}, \tau) = v_{in}'(\mathbf{k}, \tau), \tag{3.76}$$

$$\Pi_{out}(\mathbf{k}, \tau) = v_{out}'(\mathbf{k}, \tau). \tag{3.77}$$

Also it is important to note that the term $a(\tau) = -\dfrac{1}{H\tau}$ represents the scale factor in de Sitter space. All these expressions for the field and the momentum operators are very useful for computing the two-point correlation functions [77–79], explicitly computed in the next part of this paper.

## 3.2 Construction of in and out vacuum states

As discussed in the previous section, the solutions of the equation of motion before and after the point of quench is not exactly identical mainly because the mass profile changes. Physically it can be thought as two different oscillators with different masses. They define two distinct vacua $|0, in\rangle$ and $|0, out\rangle$, where the vacuum $|0, in\rangle$ represents the initial vacua of the oscillator before the point of quench and $|0, out\rangle$ represents the initial vacua of the oscillator after the point of quench. We begin with the assumption that we begin from the ground state of the initial massive theory, i.e., $|0, in\rangle$. Now since we are doing the computation in de Sitter background solution, the above mentioned in-vacuum state is not the usual Minkowski vacuum state used in the context of flat space-time. In this construction for any arbitrary choice of quantum initial vacuum state the in-vacuum and the out-vacuum state in general can be written in the following form:

$$|0, in\rangle = |d_1, d_2\rangle = \frac{1}{\sqrt{|d_1|}} \; |0, in\rangle_{\mathbf{vac}}, \tag{3.78}$$

where we define

$$|0, in\rangle_{\mathbf{vac}} = \exp\left(-\frac{i d_2^*}{2 d_1^*} \int \frac{d^3\mathbf{k}}{(2\pi)^3} \; a_{in}^\dagger(\mathbf{k}) a_{in}^\dagger(-\mathbf{k})\right) |0, in\rangle_{\mathbf{BD}}. \tag{3.79}$$

Here $|0, in\rangle_{\mathbf{BD}}$ is the Bunch Davies Euclidean vacuum state. In this construction the in-vacua state $|0, in\rangle_{\mathbf{vac}}$ can be expressed in terms of the out-vacua state using the above mentioned definition as:

$$|0, in\rangle_{\mathbf{vac}} = \exp\left[\frac{1}{2} \int \frac{d^3\mathbf{k}}{(2\pi)^3} \; \gamma(k) a_{out}^\dagger(\mathbf{k}) a_{out}^\dagger(-\mathbf{k})\right] |0, out\rangle. \tag{3.80}$$

In this context the in-vacuum can be recast in the following form:

$$|0, in\rangle_{\mathbf{vac}} = \exp\left[-\int \frac{d^3\mathbf{k}}{(2\pi)^3} \; \kappa(k) a_{out}^\dagger(\mathbf{k}) a_{out}(\mathbf{k})\right] |D\rangle, \tag{3.81}$$

$$|0, in\rangle_{\mathbf{vac}} = \exp\left[-\int \frac{d^3\mathbf{k}}{(2\pi)^3} \; \kappa(k) a_{out}^\dagger(\mathbf{k}) a_{out}(\mathbf{k})\right] |N\rangle, \tag{3.82}$$

where, $|D\rangle$ is the Dirichlet Boundary state and $|N\rangle$, represents the Neumann boundary state which are defined in terms of the out-vacuum $|0, out\rangle$ state as follows:

$$|D\rangle = \exp\left[-\frac{1}{2} \int \frac{d^3\mathbf{k}}{(2\pi)^3} \; a_{out}^\dagger(\mathbf{k}) a_{out}^\dagger(-\mathbf{k})\right] |0, out\rangle, \tag{3.83}$$

$$|N\rangle = \exp\left[\frac{1}{2} \int \frac{d^3\mathbf{k}}{(2\pi)^3} \; a_{out}^\dagger(\mathbf{k}) a_{out}^\dagger(-\mathbf{k})\right] |0, out\rangle. \tag{3.84}$$

Now using the power series expansion of $\kappa$ in Eqs (3.81) and (3.82), we find that our in vacuum-state can be expressed in the following simplified form [35, 68, 80, 81]:

$$|0, in\rangle = \frac{1}{\sqrt{|d_1|}} \; \exp\left[-\kappa_{0,\mathbf{DB}} W_0 - \sum_{n=2}^{\infty} \kappa_{2n,\mathbf{DB}} W_{2n,\mathbf{DB}}\right] |D\rangle, \tag{3.85}$$

$$|0, in\rangle = \frac{1}{\sqrt{|d_1|}} \; \exp\left[-\kappa_{0,\mathbf{NB}} W_0 - \sum_{n=2}^{\infty} \kappa_{2n,\mathbf{NB}} W_{2n,\mathbf{NB}}\right] |N\rangle. \tag{3.86}$$

Thus, for the instantaneous quench from non-zero to zero mass in de Sitter space the post quench wave function, starting from the ground state of the original Hamiltonian can be represented by the generalized Calabrese Cardy (gCC) form with the coefficients $\kappa'_n s$ given in (A.16), i.e.,

$$|0, in\rangle = |\psi\rangle_{gCC}. \tag{3.87}$$

Thus for the instantaneous quenched mass profile in de Sitter space-time, the in-state before quench takes the gCC form after the quench. Hence, one can represent the out-state in terms of the state $|\psi\rangle_{gCC}$ after the point of quench via the following relation:

**gCC in terms of Dirichlet boundary state** :

$$\begin{aligned}
|\psi_{gCC}\rangle_{\mathbf{DB}} &= \frac{1}{\sqrt{|d_1|}} \; \exp\left(-\kappa_{0,\mathbf{DB}} W_0 - \sum_{n=2}^{\infty} \kappa_{2n,\mathbf{DB}} W_{2n,\mathbf{DB}}\right) |D\rangle \\
&= \frac{1}{\sqrt{|d_1|}} \; \exp\left(-\kappa_{0,\mathbf{DB}} W_0 - \sum_{n=2}^{\infty} \kappa_{2n,\mathbf{DB}} W_{2n}\right) \\
&\qquad\qquad\qquad \exp\left(-\frac{1}{2} \int \frac{d^3 \mathbf{k}}{(2\pi)^3} a_{out}^\dagger(\mathbf{k}) a_{out}^\dagger(-\mathbf{k})\right) |0, out\rangle.
\end{aligned} \tag{3.88}$$

**gCC in terms of Neumann boundary state** :

$$\begin{aligned}
|\psi_{gCC}\rangle_{\mathbf{NB}} &= \frac{1}{\sqrt{|d_1|}} \; \exp\left(-\kappa_{0,\mathbf{NB}} W_0 - \sum_{n=2}^{\infty} \kappa_{2n,\mathbf{NB}} W_{2n}\right) |N\rangle \\
&= \frac{1}{\sqrt{|d_1|}} \; \exp\left(-\sum_{n=0}^{\infty} \kappa_{2n,\mathbf{NB}} W_{2n,\mathbf{NB}}\right) \\
&\qquad\qquad\qquad \exp\left(\frac{1}{2} \int \frac{d^3 \mathbf{k}}{(2\pi)^3} a_{out}^\dagger(\mathbf{k}) a_{out}^\dagger(-\mathbf{k})\right) |0, out\rangle.
\end{aligned} \tag{3.89}$$

One can also calculate the various conserved charges for the post quench phases from

the expansion of $\kappa$. In [41], the authors found that for the same quench profile in flat space-time, the in state after the point of quench can be expressed as

$$|0, in\rangle = \exp\left[-\frac{H}{m_0} + \frac{W_4}{6m_0^3} + ...\right]|D\rangle. \tag{3.90}$$

Thus, we find a significant difference in the nature of the gCC state after the point of quench for de Sitter space-time from the flat space results. The most striking difference being the absence of the coefficient $\kappa_2$ which implies the subsystem thermalization at a very large temperature. This claim can be made by understanding the fact that the coefficient $\kappa_2$ is related to the inverse temperature Also, another thing to note is the dependence of the coefficients on the choice of initial conditions. This is again a manifestation of the fact that the choice of initial vacuum is not unique in curved space-time. In our case, the expectation value of the number operator is given by:

$$\langle N \rangle = \frac{4^{-5+2\nu_{in}}}{\pi^2} \exp\{-2i(2k\eta + \pi\nu_{in})\}(-k\nu_{in})^{-4\nu_{in}} \bigg([d_2(-i + k\eta) + id_1^* \exp\{i(2k\eta + \pi\nu_{in})\}(i + k\eta)]$$

$$[(d_1(-i + k\eta) + id_2 \exp\{i(2k\eta + \pi\nu_{in})\})(i + k\eta)][d_2^*(1 + k\eta(i + 2k\eta - 2i\nu_{in}) - 2\nu_{in})$$
$$+ d_1^* \exp\{i(2k\eta + \pi\nu_{in})\}(i(-1 + 2\nu_{in}) + k\eta(-1 - 2ik\eta + 2\nu_{in}))]$$
$$[d_1(1 + k\eta(i + 2k\eta - 2i\nu_{in}) - 2\nu_{in})] + d_2 \exp\{i(2k\eta + \pi\nu_{in})\}$$
$$(i(-1 + 2\nu_{in}) + k\eta(-1 - 2i\eta k + 2\nu_{in}))\bigg)|\Gamma(\nu_{in})|^4, \tag{3.91}$$

which will finally appear in the following conserved charges of the $W_\infty$ algebra for gCC states:

$$\langle W_0 \rangle := \int \frac{d^3\mathbf{k}}{(2\pi)^3} \langle 0, in|a_{out}^\dagger(\mathbf{k})a_{out}(\mathbf{k})|0, in\rangle = \int \frac{d^3\mathbf{k}}{(2\pi)^3} \langle N(k) \rangle, \tag{3.92}$$

$$\langle W_{n+1} \rangle := \int \frac{d^3\mathbf{k}}{(2\pi)^3} |k|^n \langle 0, in|a_{out}^\dagger(\mathbf{k})a_{out}(\mathbf{k})|0, in\rangle = \int \frac{d^3\mathbf{k}}{(2\pi)^3} |k|^n \langle N(k) \rangle, \tag{3.93}$$

$$\forall \ n = 1, 2, \cdots, \infty \qquad \text{where} \qquad \langle N(k) \rangle = |\beta(k, \eta)|^2.$$

## 3.3 Quenched two-point correlation functions without squeezing

In this section, we will compute the two-point correlation function of the ground state, gCC in-vacuum in the post quench state by doing the mode expansion of fields in 3+1 dimensions. By changing the mass in the sudden limit from $m_0$ to $0$, which implies the changing mass parameter from $\nu_{in} = \sqrt{\frac{9}{4} - \frac{m_0^2}{H^2}}$ to $\nu_{out} = \frac{3}{2}$, the Hamiltonian of the system changes; the post-quench state is given by a gCC state as described in the previous section.

### 3.3.1 Two-point functions from ground state

Once we have constructed the in-states in terms of the out-states, we can calculate the following two-point correlation functions with respect to the ground state:

$$G^0_{\chi\chi}(\mathbf{x}_1, \mathbf{x}_2, \tau_1, \tau_2) = \langle 0, in| \, \chi(\mathbf{x}_1, \tau_1)\chi(\mathbf{x}_2, \tau_2) \, |0, in\rangle, \tag{3.94}$$

$$G^0_{\partial_i\chi\partial_i\chi}(\mathbf{x}_1, \mathbf{x}_2, \tau_1, \tau_2) = \langle 0, in| \, \partial_i\chi(\mathbf{x}_1, \tau_1)\partial_i\chi(\mathbf{x}_2, \tau_2) \, |0, in\rangle, \tag{3.95}$$

$$G^0_{\Pi_\chi\Pi_\chi}(\mathbf{x}_1, \mathbf{x}_2, \tau_1, \tau_2) = \langle 0, in| \, \Pi(\mathbf{x}_1, \tau_1)\Pi(\mathbf{x}_2, \tau_2) \, |0, in\rangle, \tag{3.96}$$

where, $G^0_{\chi\chi}(\mathbf{x}_1, \mathbf{x}_2, \tau_1, \tau_2)$, $G^0_{\partial_i\chi\partial_i\chi}(\mathbf{x}_1, \mathbf{x}_2, \tau_1, \tau_2)$ and $G^0_{\Pi_\chi\Pi_\chi}(\mathbf{x}_1, \mathbf{x}_2, \tau_1, \tau_2)$ represent the propagators in this computation. Additionally, we will define the spatial separation between the two points $\mathbf{x}_1$ and $\mathbf{x}_2$ as:

$$\mathbf{r} :\equiv \mathbf{x}_1 - \mathbf{x}_2, \tag{3.97}$$

which we willbe using in the subsequent computations.

It is important to note that, in this context, we are interested in the correlation function of the field $\chi$, its spatial derivative and canonically conjugate momenta. This field $\chi$ is redefined in terms of the classical mode function $\chi = v/a(\tau)$, which we use in the derivation of the two-point functions.

The two-point correlators can be expressed as:

$$G^0_{\chi\chi}(\mathbf{r}; \tau_1, \tau_2) = \int \frac{d^3\mathbf{k}}{(2\pi)^3} \, \langle 0, in| \, \chi_{in}(\mathbf{k}, \tau_1)\chi_{in}^*(\mathbf{k}, \tau_2) \, |0, in\rangle \, \exp(i\mathbf{k}.\mathbf{r})$$

$$= \int \frac{d^3\mathbf{k}}{(2\pi)^3} \mathcal{G}^0_{\chi\chi}(\mathbf{k}, \tau_1, \tau_2) \, \exp(i\mathbf{k}.\mathbf{r}), \tag{3.98}$$

$$G^0_{\partial_i\chi\partial_i\chi}(\mathbf{r}; \tau_1, \tau_2) = \int \frac{d^3\mathbf{k}}{(2\pi)^3} \, \langle 0, in| \, \partial_j\chi(\mathbf{k}, \tau_1)\partial_j\chi^*(\mathbf{k}, \tau_2) \, |0, in\rangle \, \exp(i\mathbf{k}.\mathbf{r})$$

$$= \int \frac{d^3\mathbf{k}}{(2\pi)^3} \mathcal{G}^0_{\partial_j\chi\partial_j\chi}(\mathbf{k}, \tau_1, \tau_2) \exp(i\mathbf{k}.\mathbf{r}), \tag{3.99}$$

$$G^0_{\Pi_\chi\Pi_\chi}(\mathbf{r}; \tau_1, \tau_2) = \int \frac{d^3\mathbf{k}}{(2\pi)^3} \, \langle 0, in| \, \Pi_\chi(\mathbf{k}, \tau_1)\Pi_\chi^*(\mathbf{k}, \tau_2) \, |0, in\rangle \exp(i\mathbf{k}.\mathbf{r})$$

$$= \int \frac{d^3\mathbf{k}}{(2\pi)^3} \mathcal{G}^0_{\Pi_\chi\Pi_\chi}(\mathbf{k}, \tau_1, \tau_2) \, \exp(i\mathbf{k}.\mathbf{r}), \tag{3.100}$$

where $\mathcal{G}^0_{\chi\chi}(\mathbf{k}, \tau_1, \tau_2)$, $\mathcal{G}^0_{\partial_j\chi\partial_j\chi}(\mathbf{k}, \tau_1, \tau_2)$ and $\mathcal{G}^0_{\Pi_\chi\Pi_\chi}(\mathbf{k}, \tau_1, \tau_2)$ representing the Fourier transform of the real space Green's functions. From the present computation we get the following

expressions for the Fourier transform of the real space Green's functions:

$$\mathcal{G}^0_{\chi\chi}(\mathbf{k}, \tau_1, \tau_2) = \frac{1}{a(\tau_1)a(\tau_2)} \frac{1}{|d_1|} \left[ \sum_{b=1}^{4} \Delta_b(\mathbf{k}, \tau_1, \tau_2) \right], \tag{3.101}$$

$$\mathcal{G}^0_{\partial_j\chi\partial_j\chi}(\mathbf{k}, \tau_1, \tau_2) = \frac{1}{a(\tau_1)a(\tau_2)} \frac{1}{|d_1|} \left[ -k^2 \sum_{b=1}^{4} \Delta_b(\mathbf{k}, \tau_1, \tau_2) \right], \tag{3.102}$$

$$\mathcal{G}^0_{\Pi_\chi\Pi_\chi}(\mathbf{k}, \tau_1, \tau_2) = \frac{1}{|d_1|} \left[ \frac{a'(\tau_1)a'(\tau_2)}{(a(\tau_1))^2(a(\tau_2))^2} \left( \sum_{b=1}^{4} \Delta_b(\mathbf{k}, \tau_1, \tau_2) \right) \right. \tag{3.103}$$

$$- \frac{a'(\tau_1)}{(a(\tau_1))^2(a(\tau_2))} \left( \sum_{b=5}^{8} \Delta_b(\mathbf{k}, \tau_1, \tau_2) \right)$$

$$- \frac{a'(\tau_2)}{(a(\tau_1))(a(\tau_2))^2} \left( \sum_{b=9}^{12} \Delta_b(\mathbf{k}, \tau_1, \tau_2) \right)$$

$$\left. + \frac{1}{a(\tau_1)a(\tau_2)} \left( \sum_{b=13}^{16} \Delta_b(\mathbf{k}, \tau_1, \tau_2) \right) \right]. \tag{3.104}$$

Here we have introduced new symbols $\Delta_i(\mathbf{k}, \tau_1, \tau_2) \ \forall \ i = 1, \cdots, 16$ which are used in the above mentioned expressions for propagators and are explicitly given in Appendix B.1.

Once we take the equal time case, $\tau_1 = \tau_2 = \tau$, it is easy to determine the expressions for the amplitude of the Power Spectrum of the field $\chi$, its spatial derivative and canonically conjugate momentum:

$$\mathcal{G}^0_{\chi\chi}(\mathbf{k}, \tau_1 = \tau, \tau_2 = \tau) := \mathcal{P}^0_{\chi\chi}(\mathbf{k}, \tau) = \frac{1}{a^2(\tau)} \frac{1}{|d_1|} \left[ \sum_{b=1}^{4} \Delta_b(\mathbf{k}, \tau) \right], \tag{3.105}$$

$$\mathcal{G}^0_{\partial_j\chi\partial_j\chi}(\mathbf{k}, \tau_1 = \tau, \tau_2 = \tau) := \mathcal{P}^0_{\partial_j\chi\partial_j\chi}(\mathbf{k}, \tau) = -k^2 \, \mathcal{P}^0_{\chi\chi}(\mathbf{k}, \tau), \tag{3.106}$$

$$\mathcal{G}^0_{\Pi_\chi\Pi_\chi}(\mathbf{k}, \tau_1 = \tau, \tau_2 = \tau) := \mathcal{P}^0_{\Pi_\chi\Pi_\chi}(\mathbf{k}, \tau) = \left[ \frac{(a'(\tau))^2}{a^2(\tau)} \mathcal{P}^0_{\chi\chi}(\mathbf{k}, \tau) \right.$$

$$\left. - \frac{a'(\tau)}{(a^3(\tau))} \frac{1}{|d_1|} \left( \sum_{b=5}^{12} \Delta_b(\mathbf{k}, \tau) \right) + \frac{1}{a^2(\tau)} \frac{1}{|d_1|} \left( \sum_{b=13}^{16} \Delta_b(\mathbf{k}, \tau) \right) \right], \tag{3.107}$$

which are all cosmologically significant quantities. This will finally give rise to the following cosmological two-point correlation function:

$$\langle 0, in| \, \chi(\mathbf{k}, \tau)\chi(\mathbf{k'}, \tau) \, |0, in\rangle = (2\pi)^3 \delta^3(\mathbf{k} + \mathbf{k'}) \mathcal{P}^0_{\chi\chi}(\mathbf{k}, \tau), \tag{3.108}$$

$$\langle 0, in| \, (ik\chi(\mathbf{k}, \tau))(ik\chi(\mathbf{k'}, \tau)) \, |0, in\rangle = (2\pi)^3 \delta^3(\mathbf{k} + \mathbf{k'}) \mathcal{P}^0_{\partial_j\chi\partial_j\chi}(\mathbf{k}, \tau)$$

$$= -(2\pi)^3 \delta^3(\mathbf{k} + \mathbf{k'}) \, k^2 \mathcal{P}^0_{\chi\chi}(\mathbf{k}, \tau), \tag{3.109}$$

$$\langle 0, in| \, \Pi(\mathbf{k}, \tau)\Pi(\mathbf{k}', \tau) \, |0, in\rangle = (2\pi)^3 \delta^3(\mathbf{k} + \mathbf{k}')\mathcal{P}^0_{\Pi_\chi \Pi_\chi}(\mathbf{k}, \tau). \qquad (3.110)$$

### 3.3.2  Two-point functions from gCC states

In this section, we focus on calculating the two-point correlation function for the gCC state:

$$G^{gCC}_{\chi\chi}(\mathbf{x}_1, \mathbf{x}_2, \tau_1, \tau_2) = \langle gCC| \, \hat{\chi}(\mathbf{x}_1, \tau_1)\hat{\chi}(\mathbf{x}_2, \tau_2) \, |gCC\rangle, \qquad (3.111)$$

$$G^{gCC}_{\partial_i\chi\partial_i\chi}(\mathbf{x}_1, \mathbf{x}_2, \tau_1, \tau_2) = \langle gCC| \, \hat{\partial_i\chi}(\mathbf{x}_1, \tau_1)\hat{\partial_i\chi}(\mathbf{x}_2, \tau_2) \, |gCC\rangle, \qquad (3.112)$$

$$G^{gCC}_{\Pi_\chi\Pi_\chi}(\mathbf{x}_1, \mathbf{x}_2, \tau_1, \tau_2) = \langle gCC| \, \hat{\Pi}(\mathbf{x}_1, \tau_1)\hat{\Pi}(\mathbf{x}_2, \tau_2) \, |gCC\rangle, \qquad (3.113)$$

where we use two types of gCC states, which are the $|\psi_{gCC}\rangle_{\mathbf{DB}}$ Dirichlet and $|\psi_{gCC}\rangle_{\mathbf{NB}}$ the Neumann boundary states, respectively.

The two-point correlators in terms of the Dirichlet boundary states can be expressed as:

$$G^{gCC\mathbf{DB}}_{\chi\chi}(\mathbf{r}; \tau_1, \tau_2) = \int \frac{d^3k}{(2\pi)^3} \, _{\mathbf{DB}}\langle gCC| \, \hat{\chi}_{in}(\mathbf{k}, \tau_1)\hat{\chi}^*_{in}(\mathbf{k}, \tau_2) \, |gCC\rangle_{\mathbf{DB}} \, \exp(i\mathbf{k.r})$$

$$= \frac{1}{|d_1|} \int \frac{d^3k}{(2\pi)^3} \exp\left( - (\kappa^*_{0,\mathbf{DB}} + \kappa_{0,\mathbf{DB}})W_0 - \sum_{n=2}^{\infty}(\kappa^*_{2n,\mathbf{DB}} + \kappa_{2n,\mathbf{DB}})W_{2n} \right)$$

$$\langle D| \, \hat{\chi}_{in}(\mathbf{k}, \tau_1)\hat{\chi}^*_{in}(\mathbf{k}, \tau_2) \, |D\rangle \, \exp(i\mathbf{k.r})$$

$$= \int \frac{d^3k}{(2\pi)^3} \, \mathcal{G}^{gCC\mathbf{DB}}_{\chi\chi}(\mathbf{k}, \tau_1, \tau_2) \, \exp(i\mathbf{k.r}), \qquad (3.114)$$

$$G^{gCC\mathbf{DB}}_{\partial_j\chi\partial_j\chi}(\mathbf{r}; \tau_1, \tau_2) = \int \frac{d^3\mathbf{k}}{(2\pi)^3} \, _{\mathbf{DB}}\langle gCC| \, \partial_j\hat{\chi}_{in}(\mathbf{k}, \tau_1)\partial_j\hat{\chi}^*_{in}(\mathbf{k}, \tau_2) \, |gCC\rangle_{\mathbf{DB}} \, \exp(i\mathbf{k.r})$$

$$= \frac{1}{|d_1|} \int \frac{d^3\mathbf{k}}{(2\pi)^3} \exp\left( - (\kappa^*_{0,\mathbf{DB}} + \kappa_{0,\mathbf{DB}})W_0 - \sum_{n=2}^{\infty}(\kappa^*_{2n,\mathbf{DB}} + \kappa_{2n,\mathbf{DB}})W_{2n} \right)$$

$$\langle D| \, \partial_j\hat{\chi}_{in}(\mathbf{k}, \tau_1)\hat{\partial_j}\chi^*_{in}(\mathbf{k}, \tau_2) \, |D\rangle \, \exp(i\mathbf{k.r})$$

$$= \int \frac{d^3\mathbf{k}}{(2\pi)^3} \, \mathcal{G}^{gCC\mathbf{DB}}_{\partial_j\chi\partial_j\chi}(\mathbf{k}, \tau_1, \tau_2) \, \exp(i\mathbf{k.r}), \qquad (3.115)$$

$$G^{gCC\mathbf{DB}}_{\Pi_\chi\Pi_\chi}(\mathbf{r}; \tau_1, \tau_2) = \int \frac{d^3\mathbf{k}}{(2\pi)^3} \, _{\mathbf{DB}}\langle gCC| \, \hat{\Pi}_\chi(\mathbf{k}, \tau_1)\hat{\Pi}^*_\chi(\mathbf{k}, \tau_2) \, |gCC\rangle_{\mathbf{DB}} \, \exp(i\mathbf{k.r})$$

$$= \frac{1}{|d_1|} \int \frac{d^3\mathbf{k}}{(2\pi)^3} \exp\left( - (\kappa^*_{0,\mathbf{DB}} + \kappa_{0,\mathbf{DB}})W_0 - \sum_{n=2}^{\infty}(\kappa^*_{2n,\mathbf{DB}} + \kappa_{2n,\mathbf{DB}})W_{2n} \right)$$

$$\langle D| \, \hat{\Pi}_\chi(\mathbf{k}, \tau_1)\hat{\Pi}^*_\chi(\mathbf{k}, \tau_2) \, |D\rangle \, \exp(i\mathbf{k.r})$$

$$= \int \frac{d^3\mathbf{k}}{(2\pi)^3} \, \mathcal{G}^{gCC\mathbf{DB}}_{\Pi_\chi\Pi_\chi}(\mathbf{k}, \tau_1, \tau_2) \, \exp(i\mathbf{k.r}), \qquad (3.116)$$

where $\mathcal{G}_{\chi\chi}^{gCC\text{DB}}(\mathbf{k}, \tau_1, \tau_2)$, $\mathcal{G}_{\partial_j\chi\partial_j\chi}^{gCC\text{DB}}(\mathbf{k}, \tau_1, \tau_2)$ and $\mathcal{G}_{\Pi_\chi\Pi_\chi}^{gCC\text{DB}}(\mathbf{k}, \tau_1, \tau_2)$ represent the Fourier transform of the real space Green's functions calculated between the Dirichlet boundary gCC states formed after quench. The state $|D\rangle$ is the Dirichlet boundary state which is defined in terms of the out-vacuum state by the following expression:

$$|D\rangle = \exp\left(-\frac{1}{2}\int \frac{d^3\mathbf{k}}{(2\pi)^3} a_{out}^\dagger(\mathbf{k}) a_{out}^\dagger(-\mathbf{k})\right) |0, out\rangle. \tag{3.117}$$

Now we can express the Fourier transform of the Green's functions $\mathcal{G}_{\chi\chi}^{gCC\text{DB}}(\mathbf{k}, \tau_1, \tau_2)$, $\mathcal{G}_{\partial_j\chi\partial_j\chi}^{gCC\text{DB}}(\mathbf{k}, \tau_1, \tau_2)$ and $\mathcal{G}_{\Pi_\chi\Pi_\chi}^{gCC\text{DB}}(\mathbf{k}, \tau_1, \tau_2)$ in terms of the out vacuum state. Hence, the outgoing solutions are represented by the following expressions:

$$\mathcal{G}_{\chi\chi}^{gCC\text{DB}}(\mathbf{k}, \tau_1, \tau_2) = \frac{1}{a(\tau_1)a(\tau_2)} \frac{1}{|d_1|}$$
$$\exp\left(-(\kappa_{0,\text{DB}}^* + \kappa_{0,\text{DB}})\langle N(k)\rangle - \sum_{n=2}^{\infty}(\kappa_{2n,\text{DB}}^* + \kappa_{2n,\text{DB}})|k|^{2n-1}\langle N(k)\rangle\right)$$
$$\sum_{c=1}^{4}\Theta_c(\mathbf{k}, \tau_1, \tau_2), \tag{3.118}$$

$$\mathcal{G}_{\partial_j\chi\partial_j\chi}^{gCC\text{DB}}(\mathbf{k}, \tau_1, \tau_2) = -k^2 \mathcal{G}_{\chi\chi}^{gCC\text{DB}}(\mathbf{k}, \tau_1, \tau_2), \tag{3.119}$$

$$\mathcal{G}_{\Pi_\chi\Pi_\chi}^{gCC\text{DB}}(\mathbf{k}, \tau_1, \tau_2) = \frac{1}{|d_1|} \exp\left(-(\kappa_{0,\text{DB}}^* + \kappa_{0,\text{DB}})\langle N(k)\rangle - \sum_{n=2}^{\infty}(\kappa_{2n,\text{DB}}^* + \kappa_{2n,\text{DB}})|k|^{2n-1}\langle N(k)\rangle\right)$$
$$\left\{\frac{a'(\tau_1)a'(\tau_2)}{(a(\tau_1)a(\tau_2))^2}\left[\sum_{c=1}^{4}\Theta_c(\mathbf{k}, \tau_1, \tau_2)\right] - \frac{a'(\tau_1)}{a^2(\tau_1)a(\tau_2)}\left[\sum_{c=5}^{8}\Theta_c(\mathbf{k}, \tau_1, \tau_2)\right]\right.$$
$$\left. - \frac{a'(\tau_2)}{a^2(\tau_2)a(\tau_1)}\left[\sum_{c=9}^{12}\Theta_c(\mathbf{k}, \tau_1, \tau_2)\right] + \frac{1}{a(\tau_1)a(\tau_2)}\left[\sum_{c=12}^{16}\Theta_c(\mathbf{k}, \tau_1, \tau_2)\right]\right\}, \tag{3.120}$$

where the functions $\Theta_c(\mathbf{k}, \tau_1, \tau_2)\forall\ c = 1, \cdots, 16$ are given in Appendix B.2.

Once we take the equal time case, which is $\tau_1 = \tau_2 = \tau$, then the expressions for the amplitude of the Power Spectrum of the field $\chi$, its spatial derivative and canonically conjugate momentum from the gCC Dirichlet boundary states can be easily obtained:

$$\mathcal{G}_{\chi\chi}^{gCC\text{DB}}(\mathbf{k}, \tau_1 = \tau, \tau_2 = \tau) := \mathcal{P}_{\chi\chi}^{gCC\text{DB}}(\mathbf{k}, \tau)$$
$$= \frac{1}{a^2(\tau)} \frac{1}{|d_1|}$$
$$\exp\left(-(\kappa_{0,\text{DB}}^* + \kappa_{0,\text{DB}})\langle N(k)\rangle - \sum_{n=2}^{\infty}(\kappa_{2n,\text{DB}}^* + \kappa_{2n,\text{DB}})|k|^{2n-1}\langle N(k)\rangle\right)$$
$$\left[\sum_{c=1}^{4}\Theta_c(\mathbf{k}, \tau)\right], \tag{3.121}$$

$$\mathcal{G}_{\partial_j\chi\partial_j\chi}^{gCC\mathbf{DB}}(\mathbf{k},\tau_1=\tau,\tau_2=\tau):=\mathcal{P}_{\partial_j\chi\partial_j\chi}^{gCC\mathbf{DB}}(\mathbf{k},\tau)=-k^2\,\mathcal{P}_{\chi\chi}^{gCC\mathbf{DB}}(\mathbf{k},\tau),\qquad\qquad(3.122)$$

$$\mathcal{G}_{\Pi_\chi\Pi_\chi}^{gCC\mathbf{DB}}(\mathbf{k},\tau_1=\tau,\tau_2=\tau):=\mathcal{P}_{\Pi_\chi\Pi_\chi}^{gCC\mathbf{DB}}(\mathbf{k},\tau)=\left[\frac{(a'(\tau))^2}{a^2(\tau)}\mathcal{P}_{\chi\chi}^{gCC\mathbf{DB}}(\mathbf{k},\tau)\right.$$

$$-\exp\left(-(\kappa_{0,\mathbf{DB}}^*+\kappa_{0,\mathbf{DB}})\langle N(k)\rangle-\sum_{n=2}^{\infty}(\kappa_{2n,\mathbf{DB}}^*+\kappa_{2n,\mathbf{DB}})|k|^{2n-1}\langle N(k)\rangle\right)$$

$$\left.\left\{\frac{a'(\tau)}{(a^3(\tau)}\frac{1}{|d_1|}\left(\sum_{c=5}^{12}\Theta_c(\mathbf{k},\tau)\right)-\frac{1}{a^2(\tau)}\frac{1}{|d_1|}\left(\sum_{b=13}^{16}\Theta_c(\mathbf{k},\tau)\right)\right\}\right].$$

$$(3.123)$$

These are cosmologically significant quantities. This will finally give rise to the following cosmological two-point correlation function for gCC Dirichlet boundary states:

$$_{\mathbf{DB}}\langle gCC|\chi(\mathbf{k},\tau)\chi(\mathbf{k}',\tau)|gCC\rangle_{\mathbf{DB}}=(2\pi)^3\delta^3(\mathbf{k}+\mathbf{k}')\mathcal{P}_{\chi\chi}^{gCC\mathbf{DB}}(\mathbf{k},\tau),\qquad(3.124)$$

$$_{\mathbf{DB}}\langle gCC|(ik\chi(\mathbf{k},\tau))(ik\chi(\mathbf{k}',\tau))|gCC\rangle_{\mathbf{DB}}=(2\pi)^3\delta^3(\mathbf{k}+\mathbf{k}')\mathcal{P}_{\partial_j\chi\partial_j\chi}^{gCC\mathbf{DB}}(\mathbf{k},\tau)$$

$$=-(2\pi)^3\delta^3(\mathbf{k}+\mathbf{k}')\,k^2\mathcal{P}_{\chi\chi}^{gCC\mathbf{DB}}(\mathbf{k},\tau),\quad(3.125)$$

$$_{\mathbf{DB}}\langle gCC|\Pi(\mathbf{k},\tau)\Pi(\mathbf{k}',\tau)|gCC\rangle_{\mathbf{DB}}=(2\pi)^3\delta^3(\mathbf{k}+\mathbf{k}')\mathcal{P}_{\Pi_\chi\Pi_\chi}^{gCC\mathbf{DB}}(\mathbf{k},\tau).\qquad(3.126)$$

Similarly, the two-point correlators in terms of the Neumann boundary states can be expressed as:

$$G_{\chi\chi}^{gCC\mathbf{NB}}(\mathbf{r};\tau_1,\tau_2)=\int\frac{d^3\mathbf{k}}{(2\pi)^3}\ _{\mathbf{NB}}\langle gCC|\hat{\chi}_{in}(\mathbf{k},\tau_1)\hat{\chi}_{in}^*(\mathbf{k},\tau_2)|gCC\rangle_{\mathbf{NB}}\ \exp(i\mathbf{k}.\mathbf{r})$$

$$=\frac{1}{|d_1|}\int\frac{d^3\mathbf{k}}{(2\pi)^3}\exp\left(-(\kappa_{0,\mathbf{NB}}^*+\kappa_{0,\mathbf{NB}})W_0-\sum_{n=2}^{\infty}(\kappa_{2n,\mathbf{NB}}^*+\kappa_{2n,\mathbf{NB}})W_{2n}\right)$$

$$\langle N|\hat{\chi}_{in}(\mathbf{k},\tau_1)\hat{\chi}_{in}^*(\mathbf{k},\tau_2)|N\rangle\ \exp(i\mathbf{k}.\mathbf{r})$$

$$=\int\frac{d^3\mathbf{k}}{(2\pi)^3}\ \mathcal{G}_{\chi\chi}^{gCC\mathbf{NB}}(\mathbf{k},\tau_1,\tau_2)\ \exp(i\mathbf{k}.\mathbf{r}),\qquad\qquad(3.127)$$

$$G_{\partial_j\chi\partial_j\chi}^{gCC\mathbf{NB}}(\mathbf{r};\tau_1,\tau_2)=\int\frac{d^3\mathbf{k}}{(2\pi)^3}\ _{\mathbf{NB}}\langle gCC|\partial_j\hat{\chi}_{in}(\mathbf{k},\tau_1)\partial_j\hat{\chi}_{in}^*(\mathbf{k},\tau_2)|gCC\rangle_{\mathbf{NB}}\ \exp(i\mathbf{k}.\mathbf{r})$$

$$=\frac{1}{|d_1|}\int\frac{d^3k}{(2\pi)^3}\exp\left(-(\kappa_{0,\mathbf{NB}}^*+\kappa_{0,\mathbf{NB}})W_0-\sum_{n=2}^{\infty}(\kappa_{2n,\mathbf{NB}}^*+\kappa_{2n,\mathbf{NB}})W_{2n}\right)$$

$$\langle N|\partial_j\hat{\chi}_{in}(\mathbf{k},\tau_1)\hat{\partial}_j\chi_{in}^*(\mathbf{k},\tau_2)|N\rangle\ \exp(i\mathbf{k}.\mathbf{r})$$

$$=\int\frac{d^3\mathbf{k}}{(2\pi)^3}\ \mathcal{G}_{\partial_j\chi\partial_j\chi}^{gCC\mathbf{NB}}(\mathbf{k},\tau_1,\tau_2)\exp(i\mathbf{k}.\mathbf{r}),$$

$$G_{\Pi_\chi\Pi_\chi}^{gCC\mathbf{NB}}(\mathbf{r};\tau_1,\tau_2)=\int\frac{d^3\mathbf{k}}{(2\pi)^3}\ _{\mathbf{NB}}\langle gCC|\hat{\Pi}_\chi(\mathbf{k},\tau_1)\hat{\Pi}_\chi^*(\mathbf{k},\tau_2)|gCC\rangle_{\mathbf{NB}}\ \exp(i\mathbf{k}.\mathbf{r})$$

$$= \frac{1}{|d_1|} \int \frac{d^3\mathbf{k}}{(2\pi)^3} \exp\left(-(\kappa^*_{0,\mathbf{NB}} + \kappa_{0,\mathbf{NB}})W_0 - \sum_{n=2}^{\infty}(\kappa^*_{2n,\mathbf{NB}} + \kappa_{2n,\mathbf{NB}})W_{2n}\right)$$

$$\langle N|\, \hat{\Pi}_\chi(\mathbf{k},\tau_1)\hat{\Pi}^*_\chi(\mathbf{k},\tau_2)\,|N\rangle \;\; \exp(i\mathbf{k}.\mathbf{r})$$

$$= \int \frac{d^3\mathbf{k}}{(2\pi)^3}\; \mathcal{G}^{gCC\mathbf{NB}}_{\Pi_\chi\Pi_\chi}(\mathbf{k},\tau_1,\tau_2)\;\exp(i\mathbf{k}.\mathbf{r}),$$

where $\mathcal{G}^{gCC\mathbf{NB}}_{\chi\chi}(\mathbf{k},\tau_1,\tau_2)$, $\mathcal{G}^{gCC\mathbf{NB}}_{\partial_j\chi\partial_j\chi}(\mathbf{k},\tau_1,\tau_2)$ and $\mathcal{G}^{gCC\mathbf{NB}}_{\Pi_\chi\Pi_\chi}(\mathbf{k},\tau_1,\tau_2)$ represents the Fourier transform of the real space Green's functions calculated between the gCC Neumann boundary state formed after quench. The state $|N\rangle$ is a Neumann boundary state which is defined as

$$|N\rangle = \exp\left(\frac{1}{2}\int \frac{d^3\mathbf{k}}{(2\pi)^3}\; a^\dagger_{out}(\mathbf{k})a^\dagger_{out}(-\mathbf{k})\right)|0,out.\rangle \tag{3.128}$$

Now we can express the Fourier transform of the Green's functions $\mathcal{G}^{gCC\mathbf{NB}}_{\chi\chi}(\mathbf{k},\tau_1,\tau_2)$, $\mathcal{G}^{gCC\mathbf{NB}}_{\partial_j\chi\partial_j\chi}(\mathbf{k},\tau_1,\tau_2)$ and $\mathcal{G}^{gCC\mathbf{NB}}_{\Pi_\chi\Pi_\chi}(\mathbf{k},\tau_1,\tau_2)$ in terms of the out vacuum state and hence the outgoing solutions represented by the following expressions:

$$\mathcal{G}^{gCC\mathbf{NB}}_{\chi\chi}(\mathbf{k},\tau_1,\tau_2) = \frac{1}{a(\tau_1)a(\tau_2)}\frac{1}{|d_1|}$$

$$\exp\left(-(\kappa^*_{0,\mathbf{NB}} + \kappa_{0,\mathbf{NB}}))\langle N(k)\rangle - \sum_{n=2}^{\infty}(\kappa^*_{2n,\mathbf{NB}} + \kappa_{2n,\mathbf{NB}}))|k|^{2n-1}\langle N(k)\rangle\right)$$

$$\sum_{c=1}^{4}\Theta_c(\mathbf{k},\tau_1,\tau_2), \tag{3.129}$$

$$\mathcal{G}^{gCC\mathbf{NB}}_{\partial_j\chi\partial_j\chi}(\mathbf{k},\tau_1,\tau_2) = -k^2\mathcal{G}^{gCC\mathbf{NB}}_{\chi\chi}(\mathbf{k},\tau_1,\tau_2), \tag{3.130}$$

$$\mathcal{G}^{gCC\mathbf{NB}}_{\Pi_\chi\Pi_\chi}(\mathbf{k},\tau_1,\tau_2) = \frac{1}{|d_1|}$$

$$\exp\left(-(\kappa^*_{0,\mathbf{NB}} + \kappa_{0,\mathbf{NB}}))\langle N(k)\rangle - \sum_{n=2}^{\infty}(\kappa^*_{2n,\mathbf{NB}} + \kappa_{2n,\mathbf{NB}}))|k|^{2n-1}\langle N(k)\rangle\right)$$

$$\left\{\frac{a'(\tau_1)a'(\tau_2)}{(a(\tau_1)a(\tau_2))^2}\left[\sum_{c=1}^{4}\Theta_c(\mathbf{k},\tau_1,\tau_2)\right] - \frac{a'(\tau_1)}{a^2(\tau_1)a(\tau_2)}\left[\sum_{c=5}^{8}\Theta_c(\mathbf{k},\tau_1,\tau_2)\right]\right.$$

$$\left. - \frac{a'(\tau_2)}{a^2(\tau_2)a(\tau_1)}\left[\sum_{c=9}^{12}\Theta_c(\mathbf{k},\tau_1,\tau_2)\right] + \frac{1}{a(\tau_1)a(\tau_2)}\left[\sum_{c=12}^{16}\Theta_c(\mathbf{k},\tau_1,\tau_2)\right]\right\}, \tag{3.131}$$

where the functions $\Theta_c(\mathbf{k},\tau_1,\tau_2)\forall\; c = 1,\cdots,16$ are defined earlier. Here one can further show that:

$$\frac{\mathcal{G}^{gCC\mathbf{NB}}_{\chi\chi}(\mathbf{k},\tau_1,\tau_2)}{\mathcal{G}^{gCC\mathbf{DB}}_{\chi\chi}(\mathbf{k},\tau_1,\tau_2)} = \frac{\mathcal{G}^{gCC\mathbf{NB}}_{\partial_j\chi\partial_j\chi}(\mathbf{k},\tau_1,\tau_2)}{\mathcal{G}^{gCC\mathbf{DB}}_{\partial_j\chi\partial_j\chi}(\mathbf{k},\tau_1,\tau_2)} = \frac{\mathcal{G}^{gCC\mathbf{NB}}_{\Pi_\chi\Pi_\chi}(\mathbf{k},\tau_1,\tau_2)}{\mathcal{G}^{gCC\mathbf{DB}}_{\Pi_\chi\Pi_\chi}(\mathbf{k},\tau_1,\tau_2)} = \exp\left(2\left(\kappa_{0,\mathbf{NB}} + \frac{i\pi}{2}\right)\langle N(k)\rangle\right)$$

$$= \exp(2\kappa_{0,\mathbf{DB}}\langle N(k)\rangle), \quad (3.132)$$

where we have used the fact that, all the forms of $W_{2n}$ $\forall$ $n = 0, 2, 3, \infty$ algebra for Dirichlet and Neumann boundary states are exactly same, but the coefficients for the $n = 0$ term is different and others are exactly same. Here particularly $n = 1$ is not allowed as for our set up the coefficient of $|k|$ term is trivially zero in the expansion of the $\kappa(k)$ parameter.

Once we take the equal time case, $\tau_1 = \tau_2 = \tau$, it is straighforward to determine the expressions for the amplitude of the Power Spectrum of the field $\chi$, its spatial derivative and canonically conjugate momentum from the gCC Neumann boundary states:

$$\mathcal{G}_{\chi\chi}^{gCC_{\mathbf{NB}}}(\mathbf{k}, \tau_1 = \tau, \tau_2 = \tau) := \mathcal{P}_{\chi\chi}^{gCC_{\mathbf{NB}}}(\mathbf{k}, \tau)$$
$$= \frac{1}{a^2(\tau)} \frac{1}{|d_1|} \exp\left(-(\kappa_{0,\mathbf{NB}}^* + \kappa_{0,\mathbf{NB}}))\langle N(k)\rangle\right.$$
$$\left.- \sum_{n=2}^{\infty}(\kappa_{2n,\mathbf{NB}}^* + \kappa_{2n,\mathbf{NB}}))|k|^{2n-1}\langle N(k)\rangle\right)\left[\sum_{c=1}^{4}\Theta_c(\mathbf{k},\tau)\right](3.133)$$

$$\mathcal{G}_{\partial_j\chi\partial_j\chi}^{gCC_{\mathbf{NB}}}(\mathbf{k}, \tau_1 = \tau, \tau_2 = \tau) := \mathcal{P}_{\partial_j\chi\partial_j\chi}^{gCC_{\mathbf{NB}}}(\mathbf{k}, \tau) = -k^2\ \mathcal{P}_{\chi\chi}^{gCC_{\mathbf{NB}}}(\mathbf{k}, \tau), \quad (3.134)$$

$$\mathcal{G}_{\Pi_\chi\Pi_\chi}^{gCC_{\mathbf{NB}}}(\mathbf{k}, \tau_1 = \tau, \tau_2 = \tau) := \mathcal{P}_{\Pi_\chi\Pi_\chi}^{gCC_{\mathbf{NB}}}(\mathbf{k}, \tau) =$$
$$\left[\frac{(a'(\tau))^2}{a^2(\tau)}\mathcal{P}_{\chi\chi}^{gCC_{\mathbf{NB}}}(\mathbf{k}, \tau) - \exp\left(-(\kappa_{0,\mathbf{NB}}^* + \kappa_{0,\mathbf{NB}}))\langle N(k)\rangle\right.\right.$$
$$\left.- \sum_{n=2}^{\infty}(\kappa_{2n,\mathbf{NB}}^* + \kappa_{2n,\mathbf{NB}}))|k|^{2n-1}\langle N(k)\rangle\right)$$
$$\left\{\frac{a'(\tau)}{(a^3(\tau)}\frac{1}{|d_1|}\left(\sum_{c=5}^{12}\Theta_c(\mathbf{k},\tau)\right) - \frac{1}{a^2(\tau)}\frac{1}{|d_1|}\left(\sum_{b=13}^{16}\Theta_c(\mathbf{k},\tau)\right)\right\}\right],$$
$$(3.135)$$

which all are cosmologically significant quantities. This will finally give rise to the following cosmological two-point correlation functions for gCC Neumann boundary states:

$$_{\mathbf{NB}}\langle gCC|\,\chi(\mathbf{k},\tau)\chi(\mathbf{k}',\tau)\,|gCC\rangle_{\mathbf{NB}} = (2\pi)^3\delta^3(\mathbf{k}+\mathbf{k}')\mathcal{P}_{\chi\chi}^{gCC_{\mathbf{NB}}}(\mathbf{k},\tau), \quad (3.136)$$

$$_{\mathbf{NB}}\langle gCC|\,(ik\chi(\mathbf{k},\tau))(ik\chi(\mathbf{k}',\tau))\,|gCC\rangle_{\mathbf{NB}} = (2\pi)^3\delta^3(\mathbf{k}+\mathbf{k}')\mathcal{P}_{\partial_j\chi\partial_j\chi}^{gCC_{\mathbf{NB}}}(\mathbf{k},\tau)$$
$$= -(2\pi)^3\delta^3(\mathbf{k}+\mathbf{k}')\ k^2\mathcal{P}_{\chi\chi}^{gCC_{\mathbf{NB}}}(\mathbf{k},\tau), \quad (3.137)$$

$$_{\mathbf{NB}}\langle gCC|\,\Pi(\mathbf{k},\tau)\Pi(\mathbf{k}',\tau)\,|gCC\rangle_{\mathbf{NB}} = (2\pi)^3\delta^3(\mathbf{k}+\mathbf{k}')\mathcal{P}_{\Pi_\chi\Pi_\chi}^{gCC_{\mathbf{NB}}}(\mathbf{k},\tau). \quad (3.138)$$

### 3.3.3 Two-point functions from Generalised Gibbs Ensemble without squeezing

In this section, we calculate the above two-point correlation functions for the Generalized

Gibbs Ensemble (GGE) [82, 83] after quench. They can be expressed as:

$$
G_{\chi\chi}^{GGE}(\beta, \mathbf{x}_1, \mathbf{x}_2, \tau_1, \tau_2) = \langle \hat{\chi}(\mathbf{x}_1, \tau_1)\hat{\chi}(\mathbf{x}_2, \tau_1)\rangle_\beta = \frac{1}{Z}\mathrm{Tr}\bigg( \exp\bigg( -\beta\hat{H}(\tau_1)
$$
$$
- \sum_{n=2}^{\infty} \kappa_{2n,\mathbf{DB/NB}} |k|^{2n-1}\hat{N}_k \bigg)\hat{\chi}(\mathbf{x}_1, \tau_1)\hat{\chi}(\mathbf{x}_2, \tau_2)\bigg),
$$

(3.139)

$$
G_{\partial_i\chi\partial_i\chi}^{GGE}(\beta, \mathbf{x}_1, \mathbf{x}_2, \tau_1, \tau_2) = \langle \partial_j\hat{\chi}(\mathbf{x}_1, \tau_1)\partial_j\hat{\chi}(\mathbf{x}_2, \tau_1)\rangle_\beta = \frac{1}{Z}\mathrm{Tr}\bigg( \exp\bigg( -\beta H
$$
$$
- \sum_{n=2}^{\infty} \kappa_{2n,\mathbf{DB/NB}} |k|^{2n-1}\hat{N}_k \bigg))\partial_j\chi(\mathbf{x}_1, \tau_1)\partial_j\chi(\mathbf{x}_2, \tau_2)\bigg),
$$

(3.140)

$$
G_{\Pi_\chi\Pi_\chi}^{GGE}(\beta, \mathbf{x}_1, \mathbf{x}_2, \tau_1, \tau_2) = \langle \hat{\Pi}_\chi(\mathbf{x}_1, \tau_1)\hat{\Pi}_\chi(\mathbf{x}_2, \tau_1)\rangle_\beta = \frac{1}{Z}\mathrm{Tr}\bigg( \exp(-\beta H
$$
$$
- \sum_{n=2}^{\infty} \kappa_{2n,\mathbf{DB/NB}} |k|^{2n-1}\hat{N}_k \bigg))\Pi_\chi(\mathbf{x}_1, \tau_1)\Pi_\chi(\mathbf{x}_2, \tau_2)\bigg),
$$

(3.141)

where, $Z$ is is the thermal partition function which in the present context is given by:

$$
Z = \mathrm{Tr}\bigg( \exp(-\beta\hat{H}(\tau_1) - \sum_{n=2}^{\infty} \kappa_{2n,\mathbf{DB/NB}} |k|^{2n-1}\hat{N}_k )) \bigg)
$$

which can be further represented in terms of the occupation number discrete representation of the Hamiltonian basis $|\{N_k\}\rangle \ \forall \ k$ as:

$$
Z = \frac{1}{|d_1|} \exp\bigg( -\frac{i}{2}\bigg\{ \frac{d_2^*}{d_1^*} - \frac{d_2}{d_1} \bigg\} \bigg) \times \sum_{\{N_k\}=0 \ \forall \ k}^{\infty} \langle\{N_k\}| \exp(-\beta \ \hat{H}_k(\tau_1))
$$
$$
- \sum_{n=2}^{\infty} \kappa_{2n,\mathbf{DB/NB}} |k|^{2n-1}\hat{N}_k )) |\{N_k\}\rangle
$$
$$
= \frac{1}{2|d_1|} \exp\bigg( -\frac{i}{2}\bigg\{ \frac{d_2^*}{d_1^*} - \frac{d_2}{d_1} \bigg\} \bigg) \exp\bigg( \frac{(\beta E_k(\tau_1))_{\mathrm{eff}}}{2} \bigg) \mathrm{cosech}\bigg( \frac{(\beta E_k(\tau_1))_{\mathrm{eff}}}{2} \bigg), (3.142)
$$

where $(\beta E_k(\tau_1))_{\mathrm{eff}}$ is given by:

$$
(\beta E_k(\tau_1))_{\mathrm{eff}} = \beta E_k(\tau_1) + \sum_{n=2}^{\infty} \kappa_{2n,\mathbf{DB/NB}} |k|^{2n-1},
$$

(3.143)

Thus, the expressions for the two-point for the GGE [84, 85] for the field $\chi$, its spatial

derivative and its canonically conjugate momentum as:

$$
G_{\chi\chi}^{GGE}(\beta, \mathbf{r}, \tau_1, \tau_2) = \int \frac{d^3\mathbf{k}}{(2\pi)^3} \left[ \mathcal{G}_{+,\chi\chi}^{GGE}(\beta, \mathbf{k}, \tau_1, \tau_2) \, \exp(i\mathbf{k.r}) \right.
$$
$$
\left. + \mathcal{G}_{-,\chi\chi}^{GGE}(\beta, \mathbf{k}, \tau_1, \tau_2) \, \exp(-i\mathbf{k.r}) \right], \qquad (3.144)
$$

$$
G_{\partial_i\chi\partial_i\chi}^{GGE}(\beta, \mathbf{k}, \tau_1, \tau_2) = \int \frac{d^3\mathbf{k}}{(2\pi)^3} \left[ \mathcal{G}_{+,\partial_i\chi\partial_i\chi}^{GGE}(\beta, \mathbf{k}, \tau_1, \tau_2) \, \exp(i\mathbf{k.r}) \right.
$$
$$
\left. + \mathcal{G}_{-,\partial_i\chi\partial_i\chi}^{GGE}(\beta, \mathbf{k}, \tau_1, \tau_2) \, \exp(-i\mathbf{k.r}) \right], \qquad (3.145)
$$

$$
G_{\Pi_\chi\Pi_\chi}^{GGE}(\beta, \mathbf{r}, \tau_1, \tau_2) = \int \frac{d^3\mathbf{k}}{(2\pi)^3} \left[ \mathcal{G}_{+,\Pi_\chi\Pi_\chi}^{GGE}(\beta, \mathbf{k}, \tau_1, \tau_2) \, \exp(i\mathbf{k.r}) \right.
$$
$$
\left. + \mathcal{G}_{-,\Pi_\chi\Pi_\chi}^{GGE}(\beta, \mathbf{k}, \tau_1, \tau_2) \, \exp(-i\mathbf{k.r}) \right], \qquad (3.146)
$$

where we have defined the spatial separation between the two points $\mathbf{x}_1$ and $\mathbf{x}_2$ as:

$$
\mathbf{r} := \equiv \mathbf{x}_1 - \mathbf{x}_2. \qquad (3.147)
$$

For each of the cases the corresponding thermal propagators in Fourier space are divided into two parts, one represents the advanced propagator appearing with $+$ symbol and the other one is the retarded propagator appearing with the $-$ symbol. To understand the mathematical structure of each of them let us first write their contributions independently in the following expressions:

$$
\mathcal{G}_{+,\chi\chi}^{GGE}(\beta, \mathbf{k}, \tau_1, \tau_2) = \frac{v_{out}(\mathbf{k}, \tau_1)v_{out}^*(-\mathbf{k}, \tau_2)}{2a(\tau_1)a(\tau_2)} \, \exp\left( \frac{(\beta E_k(\tau_1))_{\text{eff}}}{2} \right) \text{cosech}\left( \frac{(\beta E_k(\tau_1))_{\text{eff}}}{2} \right), \qquad (3.148)
$$

$$
\mathcal{G}_{-,\chi\chi}^{GGE}(\beta, \mathbf{k}, \tau_1, \tau_2) = \frac{v_{out}^*(-\mathbf{k}, \tau_1)v_{out}(\mathbf{k}, \tau_2)}{2a(\tau_1)a(\tau_2)} \, \exp\left( -\frac{(\beta E_k(\tau_1))_{\text{eff}}}{2} \right) \text{cosech}\left( \frac{(\beta E_k(\tau_1))_{\text{eff}}}{2} \right), \qquad (3.149)
$$

$$
\mathcal{G}_{+,\partial_i\chi\partial_i\chi}^{GGE}(\beta, \mathbf{k}, \tau_1, \tau_2) = -k^2 \, \mathcal{G}_{+,\chi\chi}^{GGE}(\beta, \mathbf{k}, \tau_1, \tau_2), \qquad (3.150)
$$

$$
\mathcal{G}_{-,\partial_i\chi\partial_i\chi}^{GGE}(\beta, \mathbf{k}, \tau_1, \tau_2) = -k^2 \, \mathcal{G}_{-,\chi\chi}^{GGE}(\beta, \mathbf{k}, \tau_1, \tau_2), \qquad (3.151)
$$

$$
\mathcal{G}_{+,\Pi_\chi\Pi_\chi}^{GGE}(\beta, \mathbf{k}, \tau_1, \tau_2) = \frac{v_{out}'(\mathbf{k}, \tau_1)v_{out}^{*\prime}(-\mathbf{k}, \tau_2)}{2a(\tau_1)a(\tau_2)} \, \exp\left( \frac{(\beta E_k(\tau_1))_{\text{eff}}}{2} \right) \text{cosech}\left( \frac{(\beta E_k(\tau_1))_{\text{eff}}}{2} \right)
$$
$$
- \frac{\mathcal{G}_{+,\chi\chi}^{GGE}(\beta, \mathbf{k}, \tau_1, \tau_2)}{a(\tau_1)a(\tau_2)} a'(\tau_1)a'(\tau_2), \qquad (3.152)
$$

$$
\mathcal{G}_{-,\Pi_\chi\Pi_\chi}^{GGE}(\beta, \mathbf{k}, \tau_1, \tau_2) = \frac{v_{out}^{*\prime}(-\mathbf{k}, \tau_1)v_{out}'(\mathbf{k}, \tau_2)}{2a(\tau_1)a(\tau_2)} \, \exp\left( -\frac{(\beta E_k(\tau_1))_{\text{eff}}}{2} \right) \text{cosech}\left( \frac{(\beta E_k(\tau_1))_{\text{eff}}}{2} \right)
$$
$$
- \frac{\mathcal{G}_{-,\chi\chi}^{GGE}(\beta, \mathbf{k}, \tau_1, \tau_2)}{a(\tau_1)a(\tau_2)} a'(\tau_1)a'(\tau_2). \qquad (3.153)
$$

All the technical details of the computations of the above mentioned expressions are explicitly presented in the Appendix.

Now we consider a special case, which is the equal time configuration $\tau_1 = \tau_2 = \tau$. In

that case we get the following expressions for the amplitude of the thermal power spectrum of the field $\chi$, its spatial derivative and its canonically conjugate momentum:

$$\mathcal{G}^{GGE}_{+,\chi\chi}(\beta,\mathbf{k},\tau,\tau) = \mathcal{P}^{GGE}_{+,\chi\chi}(\beta,\mathbf{k},\tau)$$

$$= \frac{v_{out}(\mathbf{k},\tau)v^*_{out}(-\mathbf{k},\tau)}{2a^2(\tau)} \exp\left(\frac{(\beta E_k(\tau_1))_{\text{eff}}}{2}\right) \text{cosech}\left(\frac{(\beta E_k(\tau_1))_{\text{eff}}}{2}\right), \quad (3.154)$$

$$\mathcal{G}^{GGE}_{-,\chi\chi}(\beta,\mathbf{k},\tau,\tau) = \mathcal{P}^{GGE}_{-,\chi\chi}(\beta,\mathbf{k},\tau)$$

$$= \frac{v^*_{out}(-\mathbf{k},\tau)v_{out}(\mathbf{k},\tau)}{2a^2(\tau)} \exp\left(-\frac{(\beta E_k(\tau_1))_{\text{eff}}}{2}\right) \text{cosech}\left(\frac{(\beta E_k(\tau_1))_{\text{eff}}}{2}\right), \quad (3.155)$$

$$\mathcal{G}^{GGE}_{+,\partial_i\chi\partial_i\chi}(\beta,\mathbf{k},\tau,\tau) = \mathcal{P}^{GGE}_{+,\partial_i\chi\partial_i\chi}(\beta,\mathbf{k},\tau) = -k^2\,\mathcal{P}^{GGE}_{+,\chi\chi}(\beta,\mathbf{k},\tau), \quad (3.156)$$

$$\mathcal{G}^{GGE}_{-,\partial_i\chi\partial_i\chi}(\beta,\mathbf{k},\tau,\tau) = \mathcal{P}^{GGE}_{-,\partial_i\chi\partial_i\chi}(\beta,\mathbf{k},\tau) = -k^2\,\mathcal{P}^{GGE}_{-,\chi\chi}(\beta,\mathbf{k},\tau), \quad (3.157)$$

$$\mathcal{G}^{GGE}_{+,\Pi_\chi\Pi_\chi}(\beta,\mathbf{k},\tau,\tau) = \mathcal{P}^{GGE}_{+,\Pi_\chi\Pi_\chi}(\beta,\mathbf{k},\tau)$$

$$= \frac{v'_{out}(\mathbf{k},\tau)v^{*'}_{out}(-\mathbf{k},\tau)}{2a^2(\tau)} \exp\left(\frac{(\beta E_k(\tau_1))_{\text{eff}}}{2}\right) \text{cosech}\left(\frac{(\beta E_k(\tau_1))_{\text{eff}}}{2}\right)$$

$$- \frac{\mathcal{P}^{GGE}_{+,\chi\chi}(\beta,\mathbf{k},\tau)}{a^2(\tau)}a'^2(\tau), \quad (3.158)$$

$$\mathcal{G}^{GGE}_{-,\Pi_\chi\Pi_\chi}(\beta,\mathbf{k},\tau,\tau) = \mathcal{P}^{GGE}_{-,\Pi_\chi\Pi_\chi}(\beta,\mathbf{k},\tau)$$

$$= \frac{v^{*'}_{out}(-\mathbf{k},\tau)v'_{out}(\mathbf{k},\tau)}{2a^2(\tau)} \exp\left(-\frac{(\beta E_k(\tau_1))_{\text{eff}}}{2}\right) \text{cosech}\left(\frac{(\beta E_k(\tau_1))_{\text{eff}}}{2}\right)$$

$$- \frac{\mathcal{P}^{GGE}_{-,\chi\chi}(\beta,\mathbf{k},\tau)}{a^2(\tau)}a'^2(\tau). \quad (3.159)$$

### 3.4   Quenched two-point correlation functions with squeezing

In this section, we will calculate the correlation functions for states which are not the ground state but excited states of the initial Hamiltonian. We will first show, that even if one starts from the excited state of the Hamiltonian before quench, the state after the quench can be expressed as gCC states. For this purpose, let's assume we start from a squeezed state [86–90] instead of the ground state of the pre-quench Hamiltonian. The language of squeezed states in the context of particle production in cosmology was also studied earlier in [91]. The two inter-related issues namely particle production and its relation in the dynamics of the early universe was established using the formalism of squeezed states. A squeezed state corresponding to the pre-quench Hamiltonian can be written as:

$$|\psi,in\rangle = |f\rangle = \exp\left(\frac{1}{2}\int \frac{d^3k}{(2\pi)^3}f(k)a^\dagger_{in}(\mathbf{k})a^\dagger_{in}(\text{-}\mathbf{k})\right)|0,in\rangle. \quad (3.160)$$

The above state can be written as:

$$|f\rangle = \exp\left(-\int \frac{d^3k}{(2\pi)^3}\kappa_{\text{eff}}(k)\hat{a}_{out}^\dagger(\mathbf{k})a_{out}(\mathbf{-k})\right)|Bd\rangle, \qquad (3.161)$$

where, $|Bd\rangle$ represents the boundary state and can be taken as two different possibilities $|D\rangle$(Dirichlet state) and $|N\rangle$(Neumann state) as already discussed in the previous subsection. The term $\kappa_{\text{eff}}$ is defined as

$$\textbf{For Dirichlet State}: \qquad \kappa_{\text{eff}}(k) = -\frac{1}{2}\log(-\gamma_{\text{eff}}(k)), \qquad (3.162)$$

$$\textbf{For Neumann State}: \qquad \kappa_{\text{eff}}(k) = -\frac{1}{2}\log(\gamma_{\text{eff}}(k)). \qquad (3.163)$$

In principle, the signature of $\gamma_{\text{eff}}(k)$ captures the effect of the boundary state and takes the negative signature for Dirichlet state and positive signature for the Neumann state. The quantity $\gamma_{\text{eff}}$ depends on the a particular combination of the ratio of the Bogoliubov coefficients and is given by:

$$\gamma_{\text{eff}}(k) = \left(\frac{\beta^*(k,\eta) + f(k)\alpha(k,\eta)}{\alpha^*(k,\eta) + f(k)\beta(k,\eta)}\right) = \exp(i\delta(k))\left(\frac{\gamma(k) + f(k)\exp(i\delta(k))}{1 + \exp(i\delta(k))f(k)\gamma^*(k)}\right), (3.164)$$

where we define the momentum dependent phase factor $\delta(k)$ as:

$$\exp(i\delta(k)) = \frac{\alpha(k)}{\alpha^*(k)}. \qquad (3.165)$$

For a fixed quench time scale $\eta$ it is expected to have only the momentum dependence in the $\gamma_{\text{eff}}$.

In this context the function $f(k)$ helps to create an arbitrary squeezed state from the initial Hamiltonian of the pre-quench phase. The role of $f(k)$ can be further understood by noting that a particular combination of $f(k)$ along with the operators $\hat{a}_{in}(\mathbf{k})$ and $\hat{a}_{in}^\dagger(\mathbf{k})$ annihilates the squeezed state:

$$\left(a_{in}(\mathbf{k}) - f(k)a_{in}^\dagger(-\mathbf{k})\right)|f\rangle$$
$$= \left(\left[\alpha^*(k,\eta) + f(k)\beta(k,\eta)\right]a_{out}(\mathbf{k}) - \left[\beta^*(k,\eta) + f(k)\alpha(k,\eta)\right]a_{out}^\dagger(-\mathbf{k})\right)|f\rangle$$
$$= 0. \qquad (3.166)$$

Particularly for a Gaussian squeeze state configuration the functional form of the squeezing function $f(k)$ is chosen have a Gaussian profile with standard deviation $\sigma = \sigma_0 m_0$, where $\sigma_0$ is the proportionality constant. In this case the squeezing function $f(k)$ can be written

as:

$$f(k) = \exp\left(-\frac{k^2}{2\sigma^2}\right). \tag{3.167}$$

Doing a series expansion of $\kappa_{\text{eff}}(k)$, for the specific choice of Gaussian profile of $f(k)$, it can be very easily verified that the non-vanishing expansion coefficients for the Dirichlet and Neumann boundary states can be written in a very simplified form, mentioned in Appendix B.3.

From the analysis the following additional relations between the non-vanishing expansion coefficients before and after squeezing operation are obtained:

$$\kappa_{0,\mathbf{DB}}^{\text{eff}} = \kappa_{0,\mathbf{DB}}, \tag{3.168}$$

$$\kappa_{0,\mathbf{NB}}^{\text{eff}} = \kappa_{0,\mathbf{NB}}, \tag{3.169}$$

$$\kappa_{4,\mathbf{DB}}^{\text{eff}} = \kappa_{4,\mathbf{NB}}^{\text{eff}} = \kappa_{4,\mathbf{DB}} = \kappa_{4,\mathbf{NB}}, \tag{3.170}$$

$$\kappa_{6,\mathbf{DB}}^{\text{eff}} = \kappa_{6,\mathbf{NB}}^{\text{eff}} = \kappa_{6,\mathbf{DB}} = \kappa_{6,\mathbf{NB}}, \tag{3.171}$$

$$\kappa_{7,\mathbf{DB}}^{\text{eff}} = \kappa_{7,\mathbf{NB}}^{\text{eff}} \neq \kappa_{7,\mathbf{DB}} = \kappa_{7,\mathbf{NB}}, \tag{3.172}$$

$$\kappa_{8,\mathbf{DB}}^{\text{eff}} = \kappa_{8,\mathbf{NB}}^{\text{eff}} = \kappa_{8,\mathbf{DB}} = \kappa_{8,\mathbf{NB}}, \tag{3.173}$$

$$\kappa_{9,\mathbf{DB}}^{\text{eff}} = \kappa_{9,\mathbf{NB}}^{\text{eff}} \neq \kappa_{9,\mathbf{DB}} = \kappa_{9,\mathbf{NB}}, \tag{3.174}$$

which implies that for some coefficients one can explicitly observe the deviation in the results before and after squeezing operation for the Gaussian squeezing profile function $f(k)$. One can explicitly check that the coefficients in which the effect of squeezing is noticeable, has two contributions, i.e.,

$$\kappa_{7,\mathbf{DB}}^{\text{eff}} = \kappa_{7,\mathbf{DB}} + M_{7,\mathbf{DB}}^{\text{sq}} = \kappa_{7,\mathbf{NB}} + M_{7,\mathbf{NB}}^{\text{sq}} = \kappa_{7,\mathbf{NB}}^{\text{eff}}, \tag{3.175}$$

$$\kappa_{9,\mathbf{DB}}^{\text{eff}} = \kappa_{9,\mathbf{DB}} + M_{9,\mathbf{DB}}^{\text{sq}} = \kappa_{9,\mathbf{NB}} + M_{9,\mathbf{NB}}^{\text{sq}} = \kappa_{9,\mathbf{NB}}^{\text{eff}}, \tag{3.176}$$

where $\kappa_{7,\mathbf{DB}}, \kappa_{7,\mathbf{NB}}$ and $\kappa_{9,\mathbf{DB}}, \kappa_{9,\mathbf{NB}}$ are appearing from the non-squeezing part and rest of the contributions $M_{7,\mathbf{DB}}^{\text{sq}}, M_{7,\mathbf{NB}}^{\text{sq}}$ and $M_{9,\mathbf{DB}}^{\text{sq}}, M_{9,\mathbf{NB}}^{\text{sq}}$ are appearing from the squeezing contributions, and are given by:

$$M_{7,\mathbf{DB}}^{\text{sq}} = M_{7,\mathbf{NB}}^{\text{sq}} =$$
$$\frac{16(d_1 d_1^* - d_2 d_2^*)^2 \eta^6 \exp(2i\pi\nu_{in})}{(1 - 2\nu_{in})^2 \left(id_1 + d_2 e^{i\pi\nu_{in}}\right)^2 \left(d_1^* e^{i\pi\nu_{in}} + id_2^*\right) \left(i(d_1 + d_1^* d_2^*) + e^{i\pi\nu_{in}}(d_1^* + d_2)\right)}, \tag{3.177}$$

$$M_{9,\mathbf{DB}}^{\text{sq}} = M_{9,\mathbf{NB}}^{\text{sq}} =$$
$$\frac{1}{(2\nu_{in} - 1)^3 \sigma^2 \left(d_1^* e^{i\pi\nu_{in}} + id_2^*\right) \left(ie^{i\pi\nu_{in}}(d_1(d_1^* + 2d_2) + d_2 d_2^*) - d_1(d_1 + d_2^*) + d_2 e^{2i\pi\nu_{in}}(d_1^* + d_2)\right)^2}$$

$$\times \left[ 8\eta^6 e^{2i\pi\nu_{in}} (d_1 d_1^* - d_2 d_2^*)^2 \left( -i \left( 4\eta^2 (2\nu_{in} - 3)\sigma^2 (d_1 + d_2^*) + d_1 (2\nu_{in} - 1) \right) \right. \right.$$

$$\left. \left. -e^{i\pi\nu_{in}} \left( 4\eta^2 (2\nu_{in} - 3)\sigma^2 (d_1^* + d_2) + d_2 (2\nu_{in} - 1) \right) \right) \right]., \qquad (3.178)$$

Similarly one can explicitly write down all the higher order odd contributions in the series which capture the effects of squeezing.

### 3.4.1 Two-point functions from squeezed state

Once we have constructed the in-states in terms of the out-states, we can calculate the following two-point correlation functions with respect to the ground state:

$$G_{\chi\chi}^{sq}(\mathbf{x}_1, \mathbf{x}_2, \tau_1, \tau_2) = \langle f| \chi(\mathbf{x}_1, \tau_1)\chi(\mathbf{x}_2, \tau_2) |f\rangle \qquad (3.179)$$

$$G_{\partial_i\chi\partial_i\chi}^{sq}(\mathbf{x}_1, \mathbf{x}_2, \tau_1, \tau_2) = \langle f| \partial_i\chi(\mathbf{x}_1, \tau_1)\partial_i\chi(\mathbf{x}_2, \tau_2) |f\rangle \qquad (3.180)$$

$$G_{\Pi_\chi\Pi_\chi}^{sq}(\mathbf{x}_1, \mathbf{x}_2, \tau_1, \tau_2) = \langle f| \Pi(\mathbf{x}_1, \tau_1)\Pi(\mathbf{x}_2, \tau_2) |f\rangle \qquad (3.181)$$

where, $G_{\chi\chi}^{sq}(\mathbf{x}_1, \mathbf{x}_2, \tau_1, \tau_2)$, $G_{\partial_i\chi\partial_i\chi}^{sq}(\mathbf{x}_1, \mathbf{x}_2, \tau_1, \tau_2)$ and $G_{\Pi_\chi\Pi_\chi}^{sq}(\mathbf{x}_1, \mathbf{x}_2, \tau_1, \tau_2)$ representing the propagators in this computation. Additionally, we will define the spatial separation between the two points $\mathbf{x}_1$ and $\mathbf{x}_2$ as:

$$\mathbf{r} :\equiv \mathbf{x}_1 - \mathbf{x}_2. \qquad (3.182)$$

We are also interested in the correlation functions of the field $\chi$, its spatial derivative and canonically conjugate momenta. This field $\chi$ is redefined in terms of classical mode function by $\chi = v/a(\tau)$, used during the derivation of the two-point functions.

The two-point correlators can be expressed as:

$$G_{\chi\chi}^{sq}(\mathbf{r}; \tau_1, \tau_2) = \int \frac{d^3\mathbf{k}}{(2\pi)^3} \langle f| \chi_{in}(\mathbf{k}, \tau_1)\chi_{in}^*(\mathbf{k}, \tau_2) |f\rangle \ \exp(i\mathbf{k}.\mathbf{r})$$

$$= \int \frac{d^3\mathbf{k}}{(2\pi)^3} \mathcal{G}_{\chi\chi}^{sq}(\mathbf{k}, \tau_1, \tau_2) \ \exp(i\mathbf{k}.\mathbf{r}), \qquad (3.183)$$

$$G_{\partial_i\chi\partial_i\chi}^{sq}(\mathbf{r}; \tau_1, \tau_2) = \int \frac{d^3\mathbf{k}}{(2\pi)^3} \langle f| \partial_j\chi(\mathbf{k}, \tau_1)\partial_j\chi^*(\mathbf{k}, \tau_2) |f\rangle \ \exp(i\mathbf{k}.\mathbf{r})$$

$$= \int \frac{d^3\mathbf{k}}{(2\pi)^3} \mathcal{G}_{\partial_j\chi\partial_j\chi}^{sq}(\mathbf{k}, \tau_1, \tau_2) \exp(i\mathbf{k}.\mathbf{r}), \qquad (3.184)$$

$$G_{\Pi_\chi\Pi_\chi}^{sq}(\mathbf{r}; \tau_1, \tau_2) = \int \frac{d^3\mathbf{k}}{(2\pi)^3} \langle f| \Pi_\chi(\mathbf{k}, \tau_1)\Pi_\chi^*(\mathbf{k}, \tau_2) |f\rangle \exp(i\mathbf{k}.\mathbf{r})$$

$$= \int \frac{d^3\mathbf{k}}{(2\pi)^3} \mathcal{G}_{\Pi_\chi\Pi_\chi}^{sq}(\mathbf{k}, \tau_1, \tau_2) \ \exp(i\mathbf{k}.\mathbf{r}), \qquad (3.185)$$

where $\mathcal{G}_{\chi\chi}^{sq}(\mathbf{k}, \tau_1, \tau_2)$, $\mathcal{G}_{\partial_j\chi\partial_j\chi}^{sq}(\mathbf{k}, \tau_1, \tau_2)$ and $\mathcal{G}_{\Pi_\chi\Pi_\chi}^{sq}(\mathbf{k}, \tau_1, \tau_2)$ representing the Fourier trans-

form of the real space Green's functions, as mentioned before. From the present computation we get the following expressions for the Fourier transform of the real space Green's functions:

$$\mathcal{G}_{\chi\chi}^{sq}(\mathbf{k},\tau_1,\tau_2) = \frac{1}{a(\tau_1)a(\tau_2)}\frac{1}{|d_1|}\left[\sum_{b=1}^{4}\Delta_b^{sq}(\mathbf{k},\tau_1,\tau_2)\right], \tag{3.186}$$

$$\mathcal{G}_{\partial_j\chi\partial_j\chi}^{sq}(\mathbf{k},\tau_1,\tau_2) = \frac{1}{a(\tau_1)a(\tau_2)}\frac{1}{|d_1|}\left[-k^2\sum_{b=1}^{4}\Delta_b^{sq}(\mathbf{k},\tau_1,\tau_2)\right], \tag{3.187}$$

$$\mathcal{G}_{\Pi_\chi\Pi_\chi}^{sq}(\mathbf{k},\tau_1,\tau_2) = \frac{1}{|d_1|}\left[\frac{a'(\tau_1)a'(\tau_2)}{(a(\tau_1))^2(a(\tau_2))^2}\left(\sum_{b=1}^{4}\Delta_b^{sq}(\mathbf{k},\tau_1,\tau_2)\right)\right. \tag{3.188}$$

$$-\frac{a'(\tau_1)}{(a(\tau_1))^2(a(\tau_2))}\left(\sum_{b=5}^{8}\Delta_b^{sq}(\mathbf{k},\tau_1,\tau_2)\right)$$

$$-\frac{a'(\tau_2)}{(a(\tau_1))(a(\tau_2))^2}\left(\sum_{b=9}^{12}\Delta_b^{sq}(\mathbf{k},\tau_1,\tau_2)\right)$$

$$\left.+\frac{1}{a(\tau_1)a(\tau_2)}\left(\sum_{b=13}^{16}\Delta_b^{sq}(\mathbf{k},\tau_1,\tau_2)\right)\right]. \tag{3.189}$$

Here we have introduced new symbols $\Delta_i^{sq}(\mathbf{k},\tau_1,\tau_2)\ \forall\ i=1,\cdots,16$ which are used in the above mentioned expressions for propagators, and are explicitly defined in the Appendix B.3.

Once we take the equal time case, $\tau_1 = \tau_2 = \tau$, the amplitude of the Power Spectrum of the field $\chi$, its spatial derivative and canonically conjugate momentum can be determined:

$$\mathcal{G}_{\chi\chi}^{sq}(\mathbf{k},\tau_1=\tau,\tau_2=\tau) := \mathcal{P}_{\chi\chi}^{sq}(\mathbf{k},\tau) = \frac{1}{a^2(\tau)}\frac{1}{|d_1|}\left[\sum_{b=1}^{4}\Delta_b^{sq}(\mathbf{k},\tau)\right], \tag{3.190}$$

$$\mathcal{G}_{\partial_j\chi\partial_j\chi}^{sq}(\mathbf{k},\tau_1=\tau,\tau_2=\tau) := \mathcal{P}_{\partial_j\chi\partial_j\chi}^{sq}(\mathbf{k},\tau) = -k^2\,\mathcal{P}_{\chi\chi}^{sq}(\mathbf{k},\tau), \tag{3.191}$$

$$\mathcal{G}_{\Pi_\chi\Pi_\chi}^{sq}(\mathbf{k},\tau_1=\tau,\tau_2=\tau) := \mathcal{P}_{\Pi_\chi\Pi_\chi}^{sq}(\mathbf{k},\tau) = \left[\frac{(a'(\tau))^2}{a^2(\tau)}\mathcal{P}_{\chi\chi}^{sq}(\mathbf{k},\tau)\right.$$

$$\left.-\frac{a'(\tau)}{(a^3(\tau))}\frac{1}{|d_1|}\left(\sum_{b=5}^{12}\Delta_b^{sq}(\mathbf{k},\tau)\right) + \frac{1}{a^2(\tau)}\frac{1}{|d_1|}\left(\sum_{b=13}^{16}\Delta_b^{sq}(\mathbf{k},\tau)\right)\right], \tag{3.192}$$

all cosmologically significant quantities. This will finally give rise to the following cosmological two-point correlation function:

$$\langle f|\,\chi(\mathbf{k},\tau)\chi(\mathbf{k}',\tau)\,|f\rangle = (2\pi)^3\delta^3(\mathbf{k}+\mathbf{k}')\mathcal{P}_{\chi\chi}^{sq}(\mathbf{k},\tau), \tag{3.193}$$

$$\langle f|)(ik\chi(\mathbf{k},\tau))(ik\chi(\mathbf{k}',\tau))\,|f\rangle = (2\pi)^3\delta^3(\mathbf{k}+\mathbf{k}')\mathcal{P}_{\partial_j\chi\partial_j\chi}^{sq}(\mathbf{k},\tau)$$

$$= -(2\pi)^3 \delta^3(\mathbf{k} + \mathbf{k}') \, k^2 \mathcal{P}_{\chi\chi}^{sq}(\mathbf{k}, \tau), \qquad (3.194)$$

$$\langle f | \, \Pi(\mathbf{k}, \tau) \Pi(\mathbf{k}', \tau) \, | f \rangle = (2\pi)^3 \delta^3(\mathbf{k} + \mathbf{k}') \mathcal{P}_{\Pi_\chi \Pi_\chi}^{sq}(\mathbf{k}, \tau). \qquad (3.195)$$

### 3.4.2 Two-point functions from squeezed gCC states

In this section, we focus on calculating the two-point correlation function for the squeezed gCC state:

$$G_{\chi\chi,sq}^{gCC}(\mathbf{x}_1, \mathbf{x}_2, \tau_1, \tau_2) = \langle gCC_{sq} | \, \hat{\chi}(\mathbf{x}_1, \tau_1) \hat{\chi}(\mathbf{x}_2, \tau_2) \, | gCC_{sq} \rangle, \qquad (3.196)$$

$$G_{\partial_i \chi \partial_i \chi, sq}^{gCC}(\mathbf{x}_1, \mathbf{x}_2, \tau_1, \tau_2) = \langle gCC_{sq} | \, \hat{\partial_i \chi}(\mathbf{x}_1, \tau_1) \hat{\partial_i \chi}(\mathbf{x}_2, \tau_2) \, | gCC_{sq} \rangle, \qquad (3.197)$$

$$G_{\Pi_\chi \Pi_\chi, sq}^{gCC}(\mathbf{x}_1, \mathbf{x}_2, \tau_1, \tau_2) = \langle gCC_{sq} | \, \hat{\Pi}(\mathbf{x}_1, \tau_1) \hat{\Pi}(\mathbf{x}_2, \tau_2) \, | gCC_{sq} \rangle, \qquad (3.198)$$

where we use two types of gCC states, the $|\psi_{gCC_{sq}}\rangle_{\mathbf{DB}}$ Dirichlet boundary state and $|\psi_{gCC_{sq}}\rangle_{\mathbf{NB}}$ Neumann boundary states, respectively.

The two-point correlators in terms of the Dirichlet boundary states can be expressed as:

$$G_{\chi\chi,sq}^{gCC\mathbf{DB}}(\mathbf{r}; \tau_1, \tau_2) = \int \frac{d^3k}{(2\pi)^3} \, {}_{\mathbf{DB}}\langle gCC_{sq} | \, \hat{\chi}_{in}(\mathbf{k}, \tau_1) \hat{\chi}_{in}^*(\mathbf{k}, \tau_2) \, | gCC_{sq} \rangle_{\mathbf{DB}} \, \exp(i\mathbf{k}.\mathbf{r})$$

$$= \frac{1}{|d_1|} \int \frac{d^3k}{(2\pi)^3} \exp\left( -(\kappa_{0,\mathbf{DB}}^{*\text{eff}} + \kappa_{0,\mathbf{DB}}^{\text{eff}})W_0 - \sum_{n=2}^{\infty} (\kappa_{n,\mathbf{DB}}^{*\text{eff}} + \kappa_{n,\mathbf{DB}}^{\text{eff}})W_n \right)$$

$$\langle D | \, \hat{\chi}_{in}(\mathbf{k}, \tau_1) \hat{\chi}_{in}^*(\mathbf{k}, \tau_2) \, | D \rangle \, \exp(i\mathbf{k}.\mathbf{r})$$

$$= \int \frac{d^3k}{(2\pi)^3} \, \mathcal{G}_{\chi\chi,sq}^{gCC\mathbf{DB}}(\mathbf{k}, \tau_1, \tau_2) \, \exp(i\mathbf{k}.\mathbf{r}), \qquad (3.199)$$

$$G_{\partial_j \chi \partial_j \chi, sq}^{gCC\mathbf{DB}}(\mathbf{r}; \tau_1, \tau_2) = \int \frac{d^3\mathbf{k}}{(2\pi)^3} \, {}_{\mathbf{DB}}\langle gCC | \, \partial_j \hat{\chi}_{in}(\mathbf{k}, \tau_1) \partial_j \hat{\chi}_{in}^*(\mathbf{k}, \tau_2) \, | gCC \rangle_{\mathbf{DB}} \, \exp(i\mathbf{k}.\mathbf{r})$$

$$= \frac{1}{|d_1|} \int \frac{d^3\mathbf{k}}{(2\pi)^3} \exp\left( -(\kappa_{0,\mathbf{DB}}^{*\text{eff}} + \kappa_{0,\mathbf{DB}}^{\text{eff}})W_0 - \sum_{n=2}^{\infty} (\kappa_{n,\mathbf{DB}}^{*\text{eff}} + \kappa_{n,\mathbf{DB}}^{\text{eff}})W_n \right)$$

$$\langle D | \, \partial_j \hat{\chi}_{in}(\mathbf{k}, \tau_1) \hat{\partial}_j \chi_{in}^*(\mathbf{k}, \tau_2) \, | D \rangle \, \exp(i\mathbf{k}.\mathbf{r})$$

$$= \int \frac{d^3\mathbf{k}}{(2\pi)^3} \, \mathcal{G}_{\partial_j \chi \partial_j \chi, sq}^{gCC\mathbf{DB}}(\mathbf{k}, \tau_1, \tau_2) \, \exp(i\mathbf{k}.\mathbf{r}), \qquad (3.200)$$

$$G_{\Pi_\chi \Pi_\chi, sq}^{gCC\mathbf{DB}}(\mathbf{r}; \tau_1, \tau_2) = \int \frac{d^3\mathbf{k}}{(2\pi)^3} \, {}_{\mathbf{DB}}\langle gCC | \, \hat{\Pi}_\chi(\mathbf{k}, \tau_1) \hat{\Pi}_\chi^*(\mathbf{k}, \tau_2) \, | gCC \rangle_{\mathbf{DB}} \, \exp(i\mathbf{k}.\mathbf{r})$$

$$= \frac{1}{|d_1|} \int \frac{d^3\mathbf{k}}{(2\pi)^3} \exp\left( -(\kappa_{0,\mathbf{DB}}^{*\text{eff}} + \kappa_{0,\mathbf{DB}}^{\text{eff}})W_0 - \sum_{n=2}^{\infty} (\kappa_{n,\mathbf{DB}}^{*\text{eff}} + \kappa_{n,\mathbf{DB}}^{\text{eff}})W_n \right)$$

$$\langle D | \, \hat{\Pi}_\chi(\mathbf{k}, \tau_1) \hat{\Pi}_\chi^*(\mathbf{k}, \tau_2) \, | D \rangle \, \exp(i\mathbf{k}.\mathbf{r})$$

$$= \int \frac{d^3\mathbf{k}}{(2\pi)^3} \, \mathcal{G}_{\Pi_\chi \Pi_\chi, sq}^{gCC\mathbf{DB}}(\mathbf{k}, \tau_1, \tau_2) \, \exp(i\mathbf{k}.\mathbf{r}), \qquad (3.201)$$

where $\mathcal{G}^{gCC\mathbf{DB}}_{\chi\chi,sq}(\mathbf{k},\tau_1,\tau_2)$, $\mathcal{G}^{gCC\mathbf{DB}}_{\partial_j\chi\partial_j\chi,sq}(\mathbf{k},\tau_1,\tau_2)$ and $\mathcal{G}^{gCC\mathbf{DB}}_{\Pi_\chi\Pi_\chi,sq}(\mathbf{k},\tau_1,\tau_2)$ represent the Fourier transform of the real space Green's functions calculated between the Dirichlet boundary squeezed gCC states formed after quench. The state $|D\rangle$ is the Dirichlet boundary state which was defined earlier.

Now we can express the Fourier transform of the Green's functions $\mathcal{G}^{gCC\mathbf{DB}}_{\chi\chi,sq}(\mathbf{k},\tau_1,\tau_2)$, $\mathcal{G}^{gCC\mathbf{DB}}_{\partial_j\chi\partial_j\chi,sq}(\mathbf{k},\tau_1,\tau_2)$ and $\mathcal{G}^{gCC\mathbf{DB}}_{\Pi_\chi\Pi_\chi,sq}(\mathbf{k},\tau_1,\tau_2)$ in terms of the out vacuum state and hence the outgoing solutions represented by the following expressions:

$$\mathcal{G}^{gCC\mathbf{DB}}_{\chi\chi,sq}(\mathbf{k},\tau_1,\tau_2) = \exp\left(-\sum_{n=7,9,11,\cdots}^{\infty}(M^{\mathrm{sq}*}_{n,\mathbf{DB}} + M^{\mathrm{sq}}_{n,\mathbf{DB}})|k|^{n-1}\langle N(k)\rangle\right)\mathcal{G}^{gCC\mathbf{DB}}_{\chi\chi}(\mathbf{k},\tau_1,\tau_2),\ (3.202)$$

$$\mathcal{G}^{gCC\mathbf{DB}}_{\partial_j\chi\partial_j\chi,sq}(\mathbf{k},\tau_1,\tau_2) = -k^2\mathcal{G}^{gCC\mathbf{DB}}_{\chi\chi,sq}(\mathbf{k},\tau_1,\tau_2), \tag{3.203}$$

$$\mathcal{G}^{gCC\mathbf{DB}}_{\Pi_\chi\Pi_\chi,sq}(\mathbf{k},\tau_1,\tau_2) = \exp\left(-\sum_{n=7,9,11,\cdots}^{\infty}(M^{\mathrm{sq}*}_{n,\mathbf{DB}} + M^{\mathrm{sq}}_{n,\mathbf{DB}})|k|^{n-1}\langle N(k)\rangle\right)\mathcal{G}^{gCC\mathbf{DB}}_{\Pi_\chi\Pi_\chi}(\mathbf{k},\tau_1,\tau_2),\ (3.204)$$

where the functions $M^{\mathrm{sq}}_{n,\mathbf{DB}}\ \forall\ n=7,9\cdots$ have been defined earlier.

Once we take the equal time case, $\tau_1=\tau_2=\tau$, it is easy to determine the expressions for the amplitude of the Power Spectrum of the field $\chi$, its spatial derivative and canonically conjugate momentum from the squeezed gCC Dirichlet boundary states:

$$\mathcal{G}^{gCC\mathbf{DB}}_{\chi\chi,sq}(\mathbf{k},\tau_1=\tau,\tau_2=\tau) := \mathcal{P}^{gCC\mathbf{DB}}_{\chi\chi,sq}(\mathbf{k},\tau)$$
$$= \exp\left(-\sum_{n=7,9,11,\cdots}^{\infty}(M^{\mathrm{sq}*}_{n,\mathbf{DB}} + M^{\mathrm{sq}}_{n,\mathbf{DB}})|k|^{n-1}\langle N(k)\rangle\right)\mathcal{P}^{gCC\mathbf{DB}}_{\chi\chi}(\mathbf{k},\tau),$$
$$(3.205)$$

$$\mathcal{G}^{gCC\mathbf{DB}}_{\partial_j\chi\partial_j\chi,sq}(\mathbf{k},\tau_1=\tau,\tau_2=\tau) := \mathcal{P}^{gCC\mathbf{DB}}_{\partial_j\chi\partial_j\chi,sq}(\mathbf{k},\tau) = -k^2\ \mathcal{P}^{gCC\mathbf{DB}}_{\chi\chi,sq}(\mathbf{k},\tau), \tag{3.206}$$

$$\mathcal{G}^{gCC\mathbf{DB}}_{\Pi_\chi\Pi_\chi,sq}(\mathbf{k},\tau_1=\tau,\tau_2=\tau) := \mathcal{P}^{gCC\mathbf{DB}}_{\Pi_\chi\Pi_\chi,sq}(\mathbf{k},\tau)$$
$$= \exp\left(-\sum_{n=7,9,11,\cdots}^{\infty}(M^{\mathrm{sq}*}_{n,\mathbf{DB}} + M^{\mathrm{sq}}_{n,\mathbf{DB}})|k|^{n-1}\langle N(k)\rangle\right)\mathcal{P}^{gCC\mathbf{DB}}_{\Pi_\chi\Pi_\chi}(\mathbf{k},\tau),$$
$$(3.207)$$

which are cosmologically significant quantities. This will finally give rise to the following cosmological two-point correlation function for the squeezed gCC Dirichlet boundary states:

$$_{\mathbf{DB}}\langle gCC_{sq}|\,\chi(\mathbf{k},\tau)\chi(\mathbf{k}',\tau)\,|gCC_{sq}\rangle_{\mathbf{DB}} = (2\pi)^3\delta^3(\mathbf{k}+\mathbf{k}')\mathcal{P}^{gCC\mathbf{DB}}_{\chi\chi,sq}(\mathbf{k},\tau), \tag{3.208}$$

$$_{\mathbf{DB}}\langle gCC_{sq}|\,(ik\chi(\mathbf{k},\tau))(ik\chi(\mathbf{k}',\tau))\,|gCC_{sq}\rangle_{\mathbf{DB}} = (2\pi)^3\delta^3(\mathbf{k}+\mathbf{k}')\mathcal{P}^{gCC\mathbf{DB}}_{\partial_j\chi\partial_j\chi,sq}(\mathbf{k},\tau)$$
$$= -(2\pi)^3\delta^3(\mathbf{k}+\mathbf{k}')\,k^2\mathcal{P}^{gCC\mathbf{DB}}_{\chi\chi,sq}(\mathbf{k},\tau), \tag{3.209}$$

$$_{\mathbf{DB}}\langle gCC_{sq}|\,\Pi(\mathbf{k},\tau)\Pi(\mathbf{k}',\tau)\,|gCC_{sq}\rangle_{\mathbf{DB}} = (2\pi)^3\delta^3(\mathbf{k}+\mathbf{k}')\mathcal{P}^{gCC\mathbf{DB}}_{\Pi_\chi\Pi_\chi,sq}(\mathbf{k},\tau). \tag{3.210}$$

Similarly, the two-point correlators in terms of the Neumann boundary states can be expressed as:

$$
\begin{aligned}
G_{\chi\chi,sq}^{gCC_{\mathbf{NB}}}(\mathbf{r};\tau_1,\tau_2) &= \int \frac{d^3\mathbf{k}}{(2\pi)^3} \ _{\mathbf{NB}}\langle gCC_{sq}|\,\hat{\chi}_{in}(\mathbf{k},\tau_1)\hat{\chi}_{in}^*(\mathbf{k},\tau_2)\,|gCC_{sq}\rangle_{\mathbf{NB}}\ \exp(i\mathbf{k}.\mathbf{r}) \\
&= \frac{1}{|d_1|}\int \frac{d^3\mathbf{k}}{(2\pi)^3} \exp\Bigg(-(\kappa_{0,\mathbf{NB}}^{\mathrm{eff}*}+\kappa_{0,\mathbf{NB}}^{\mathrm{eff}})W_0 - \sum_{n=2}^{\infty}(\kappa_{n,\mathbf{NB}}^{\mathrm{eff}*}+\kappa_{n,\mathbf{NB}}^{\mathrm{eff}})W_n\Bigg) \\
&\qquad\qquad \langle N|\,\hat{\chi}_{in}(\mathbf{k},\tau_1)\hat{\chi}_{in}^*(\mathbf{k},\tau_2)\,|N\rangle\ \exp(i\mathbf{k}.\mathbf{r}) \\
&= \int \frac{d^3\mathbf{k}}{(2\pi)^3}\ \mathcal{G}_{\chi\chi,sq}^{gCC_{\mathbf{NB}}}(\mathbf{k},\tau_1,\tau_2)\ \exp(i\mathbf{k}.\mathbf{r}), \qquad\qquad (3.211)
\end{aligned}
$$

$$
\begin{aligned}
G_{\partial_j\chi\partial_j\chi,sq}^{gCC_{\mathbf{NB}}}(\mathbf{r};\tau_1,\tau_2) &= \int \frac{d^3\mathbf{k}}{(2\pi)^3} \ _{\mathbf{NB}}\langle gCC_{sq}|\,\partial_j\hat{\chi}_{in}(\mathbf{k},\tau_1)\partial_j\hat{\chi}_{in}^*(\mathbf{k},\tau_2)\,|gCC_{sq}\rangle_{\mathbf{NB}}\ \exp(i\mathbf{k}.\mathbf{r}) \\
&= \frac{1}{|d_1|}\int \frac{d^3k}{(2\pi)^3} \exp\Bigg(-(\kappa_{0,\mathbf{NB}}^{\mathrm{eff}*}+\kappa_{0,\mathbf{NB}}^{\mathrm{eff}})W_0 - \sum_{n=2}^{\infty}(\kappa_{n,\mathbf{NB}}^{\mathrm{eff}*}+\kappa_{n,\mathbf{NB}}^{\mathrm{eff}})W_n\Bigg) \\
&\qquad\qquad \langle N|\,\partial_j\hat{\chi}_{in}(\mathbf{k},\tau_1)\hat{\partial}_j\chi_{in}^*(\mathbf{k},\tau_2)\,|N\rangle\ \exp(i\mathbf{k}.\mathbf{r}) \\
&= \int \frac{d^3\mathbf{k}}{(2\pi)^3}\ \mathcal{G}_{\partial_j\chi\partial_j\chi,sq}^{gCC_{\mathbf{NB}}}(\mathbf{k},\tau_1,\tau_2)\exp(i\mathbf{k}.\mathbf{r}),
\end{aligned}
$$

$$
\begin{aligned}
G_{\Pi_\chi\Pi_\chi,sq}^{gCC_{\mathbf{NB}}}(\mathbf{r};\tau_1,\tau_2) &= \int \frac{d^3\mathbf{k}}{(2\pi)^3} \ _{\mathbf{NB}}\langle gCC_{sq}|\,\hat{\Pi}_\chi(\mathbf{k},\tau_1)\hat{\Pi}_\chi^*(\mathbf{k},\tau_2)\,|gCC_{sq}\rangle_{\mathbf{NB}}\ \exp(i\mathbf{k}.\mathbf{r}) \\
&= \frac{1}{|d_1|}\int \frac{d^3\mathbf{k}}{(2\pi)^3} \exp\Bigg(-(\kappa_{0,\mathbf{NB}}^{\mathrm{eff}*}+\kappa_{0,\mathbf{NB}}^{\mathrm{eff}})W_0 - \sum_{n=2}^{\infty}(\kappa_{n,\mathbf{NB}}^{\mathrm{eff}*}+\kappa_{n,\mathbf{NB}}^{\mathrm{eff}})W_n\Bigg) \\
&\qquad\qquad \langle N|\,\hat{\Pi}_\chi(\mathbf{k},\tau_1)\hat{\Pi}_\chi^*(\mathbf{k},\tau_2)\,|N\rangle\ \exp(i\mathbf{k}.\mathbf{r}) \\
&= \int \frac{d^3\mathbf{k}}{(2\pi)^3}\ \mathcal{G}_{\Pi_\chi\Pi_\chi,sq}^{gCC_{\mathbf{NB}}}(\mathbf{k},\tau_1,\tau_2)\ \exp(i\mathbf{k}.\mathbf{r}),
\end{aligned}
$$

where $\mathcal{G}_{\chi\chi,sq}^{gCC_{\mathbf{NB}}}(\mathbf{k},\tau_1,\tau_2)$, $\mathcal{G}_{\partial_j\chi\partial_j\chi,sq}^{gCC_{\mathbf{NB}}}(\mathbf{k},\tau_1,\tau_2)$ and $\mathcal{G}_{\Pi_\chi\Pi_\chi,sq}^{gCC_{\mathbf{NB}}}(\mathbf{k},\tau_1,\tau_2)$ represent the Fourier transform of the real space Green's functions calculated between the squeezed gCC Neumann boundary state formed after quench. The state $|N\rangle$ is a Neumann boundary state, defined earlier.

Now we can express the Fourier transform of the Green's functions $\mathcal{G}_{\chi\chi,sq}^{gCC_{\mathbf{NB}}}(\mathbf{k},\tau_1,\tau_2)$, $\mathcal{G}_{\partial_j\chi\partial_j\chi,sq}^{gCC_{\mathbf{NB}}}(\mathbf{k},\tau_1,\tau_2)$ and $\mathcal{G}_{\Pi_\chi\Pi_\chi,sq}^{gCC_{\mathbf{NB}}}(\mathbf{k},\tau_1,\tau_2)$ in terms of the out vacuum state and hence the outgoing solutions represented by the following expressions:

$$
\mathcal{G}_{\chi\chi,sq}^{gCC_{\mathbf{NB}}}(\mathbf{k},\tau_1,\tau_2) = \exp\Bigg(-\sum_{n=7,9,11,\cdots}^{\infty}(M_{n,\mathbf{NB}}^{\mathrm{sq}*}+M_{n,\mathbf{NB}}^{\mathrm{sq}})|k|^{n-1}\langle N(k)\rangle\Bigg)\mathcal{G}_{\chi\chi}^{gCC_{\mathbf{NB}}}(\mathbf{k},\tau_1,\tau_2), \quad (3.212)
$$

$$
\mathcal{G}_{\partial_j\chi\partial_j\chi,sq}^{gCC_{\mathbf{NB}}}(\mathbf{k},\tau_1,\tau_2) = -k^2\mathcal{G}_{\chi\chi,sq}^{gCC_{\mathbf{NB}}}(\mathbf{k},\tau_1,\tau_2), \qquad\qquad (3.213)
$$

$$
\mathcal{G}_{\Pi_\chi\Pi_\chi}^{gCC_{\mathbf{NB}}}(\mathbf{k},\tau_1,\tau_2) = \exp\Bigg(-\sum_{n=7,9,11,\cdots}^{\infty}(M_{n,\mathbf{NB}}^{\mathrm{sq}*}+M_{n,\mathbf{NB}}^{\mathrm{sq}})|k|^{n-1}\langle N(k)\rangle\Bigg)\mathcal{G}_{\Pi_\chi\Pi_\chi}^{gCC_{\mathbf{NB}}}(\mathbf{k},\tau_1,\tau_2), \quad (3.214)
$$

where the functions $M_{n,\mathbf{DB}}^{\text{sq}} \forall\ n = 7, 9 \cdots$ have already been defined earlier.

Once again in the equal time case, $\tau_1 = \tau_2 = \tau$, it is strightforward to determine the expressions for the amplitude of the Power Spectrum of the field $\chi$, its spatial derivative and canonically conjugate momentum from the gCC Neumann boundary states:

$$
\begin{aligned}
\mathcal{G}_{\chi\chi,sq}^{gCC\mathbf{NB}}(\mathbf{k}, \tau_1 = \tau, \tau_2 = \tau) :&= \mathcal{P}_{\chi\chi,sq}^{gCC\mathbf{NB}}(\mathbf{k}, \tau) \\
&= \exp\left( -\sum_{n=7,9,11,\cdots}^{\infty} (M_{n,\mathbf{NB}}^{\text{sq}*} + M_{n,\mathbf{NB}}^{\text{sq}})|k|^{n-1}\langle N(k)\rangle \right) \mathcal{P}_{\chi\chi}^{gCC\mathbf{NB}}(\mathbf{k}, \tau),
\end{aligned}
$$
(3.215)

$$
\mathcal{G}_{\partial_j\chi\partial_j\chi,sq}^{gCC\mathbf{NB}}(\mathbf{k}, \tau_1 = \tau, \tau_2 = \tau) := \mathcal{P}_{\partial_j\chi\partial_j\chi,sq}^{gCC\mathbf{NB}}(\mathbf{k}, \tau) = -k^2\ \mathcal{P}_{\chi\chi,sq}^{gCC\mathbf{NB}}(\mathbf{k}, \tau),
$$
(3.216)

$$
\begin{aligned}
\mathcal{G}_{\Pi_\chi\Pi_\chi,sq}^{gCC\mathbf{NB}}(\mathbf{k}, \tau_1 = \tau, \tau_2 = \tau) :&= \mathcal{P}_{\Pi_\chi\Pi_\chi,sq}^{gCC\mathbf{NB}}(\mathbf{k}, \tau) \\
&= \exp\left( -\sum_{n=7,9,11,\cdots}^{\infty} (M_{n,\mathbf{NB}}^{\text{sq}*} + M_{n,\mathbf{NB}}^{\text{sq}})|k|^{n-1}\langle N(k)\rangle \right) \mathcal{P}_{\Pi_\chi\Pi_\chi}^{gCC\mathbf{NB}}(\mathbf{k}, \tau).
\end{aligned}
$$
(3.217)

These are cosmologically significant quantities. This will finally give rise to the following cosmological two-point correlation function for gCC Neumann boundary states:

$$
{}_{\mathbf{NB}}\langle gCC_{sq}|\,\chi(\mathbf{k}, \tau)\chi(\mathbf{k}', \tau)\,|gCC_{sq}\rangle_{\mathbf{NB}} = (2\pi)^3\delta^3(\mathbf{k}+\mathbf{k}')\mathcal{P}_{\chi\chi,sq}^{gCC\mathbf{NB}}(\mathbf{k}, \tau), \qquad (3.218)
$$

$$
\begin{aligned}
{}_{\mathbf{NB}}\langle gCC_{sq}|\,(ik\chi(\mathbf{k}, \tau))(ik\chi(\mathbf{k}', \tau))\,|gCC_{sq}\rangle_{\mathbf{NB}} &= (2\pi)^3\delta^3(\mathbf{k}+\mathbf{k}')\mathcal{P}_{\partial_j\chi\partial_j\chi,sq}^{gCC\mathbf{NB}}(\mathbf{k}, \tau) \\
&= -(2\pi)^3\delta^3(\mathbf{k}+\mathbf{k}')\ k^2\mathcal{P}_{\chi\chi,sq}^{gCC\mathbf{NB}}(\mathbf{k}, \tau), (3.219)
\end{aligned}
$$

$$
{}_{\mathbf{NB}}\langle gCC_{sq}|\,\Pi(\mathbf{k}, \tau)\Pi(\mathbf{k}', \tau)\,|gCC_{sq}\rangle_{\mathbf{NB}} = (2\pi)^3\delta^3(\mathbf{k}+\mathbf{k}')\mathcal{P}_{\Pi_\chi\Pi_\chi,sq}^{gCC\mathbf{NB}}(\mathbf{k}, \tau). \qquad (3.220)
$$

### 3.4.3 Two-point functions from Generalised Gibbs Ensemble with squeezing

In this section, we calculate the above two-point correlation functions for the Generalized Gibbs Ensemble (GGE) [82, 83] after quench. They can be expressed as:

$$
\begin{aligned}
G_{\chi\chi,sq}^{GGE}(\beta, \mathbf{x}_1, \mathbf{x}_2, \tau_1, \tau_2) = \langle\hat{\chi}(\mathbf{x}_1, \tau_1)\hat{\chi}(\mathbf{x}_2, \tau_1)\rangle_\beta = \frac{1}{Z}\text{Tr}\bigg( &\exp\bigg( -\beta\hat{H}(\tau_1) \\
&- \sum_{n=2}^{\infty} \kappa_{2n,\mathbf{DB/NB}}^{sq}\ |k|^{2n-1}\hat{N}_k \bigg)\hat{\chi}(\mathbf{x}_1, \tau_1)\hat{\chi}(\mathbf{x}_2, \tau_2) \bigg),
\end{aligned}
$$
(3.221)

$$
G_{\partial_i\chi\partial_i\chi,sq}^{GGE}(\beta, \mathbf{x}_1, \mathbf{x}_2, \tau_1, \tau_2) = \langle\partial_j\hat{\chi}(\mathbf{x}_1, \tau_1)\partial_j\hat{\chi}(\mathbf{x}_2, \tau_1)\rangle_\beta = \frac{1}{Z}\text{Tr}\bigg( \exp\bigg( -\beta H
$$

$$-\sum_{n=2}^{\infty} \kappa_{2n,\mathbf{DB/NB}}^{sq} |k|^{2n-1} \hat{N}_k \Big) \Big) \partial_j \chi(\mathbf{x}_1, \tau_1) \partial_j \chi(\mathbf{x}_2, \tau_2) \Big),$$

$$(3.222)$$

$$G_{\Pi_\chi \Pi_\chi, sq}^{GGE}(\beta, \mathbf{x}_1, \mathbf{x}_2, \tau_1, \tau_2) = \langle \hat{\Pi}_\chi(\mathbf{x}_1, \tau_1) \hat{\Pi}_\chi(\mathbf{x}_2, \tau_1) \rangle_\beta = \frac{1}{Z} \mathrm{Tr} \Big( \exp(-\beta H$$

$$-\sum_{n=2}^{\infty} \kappa_{2n,\mathbf{DB/NB}}^{sq} |k|^{2n-1} \hat{N}_k \Big) \Big) \Pi_\chi(\mathbf{x}_1, \tau_1) \Pi_\chi(\mathbf{x}_2, \tau_2) \Big),$$

$$(3.223)$$

where, $Z$ is is the thermal partition function which in the present context is given by:

$$Z = \mathrm{Tr} \Big( \exp(-\beta \hat{H}(\tau_1) - \sum_{n=2}^{\infty} \kappa_{2n,\mathbf{DB/NB}}^{sq} |k|^{2n-1} \hat{N}_k)) \Big)$$

which can be further represented in terms of the occupation number discrete representation of the Hamiltonian basis $|\{N_k\}\rangle \; \forall \; k$ as:

$$Z = \frac{1}{|d_1|} \exp \Big( -\frac{i}{2} \Big\{ \frac{d_2^*}{d_1^*} - \frac{d_2}{d_1} \Big\} \Big) \times \sum_{\{N_k\}=0 \; \forall \; k}^{\infty} \langle \{N_k\}| \exp(-\beta \; \hat{H}_k(\tau_1))$$

$$-\sum_{n=2}^{\infty} \kappa_{2n,\mathbf{DB/NB}}^{sq} |k|^{2n-1} \hat{N}_k \Big) ) |\{N_k\}\rangle$$

$$= \frac{1}{2|d_1|} \exp \Big( -\frac{i}{2} \Big\{ \frac{d_2^*}{d_1^*} - \frac{d_2}{d_1} \Big\} \Big) \; \exp \Big( \frac{(\beta E_k(\tau_1))_{\mathrm{eff},sq}}{2} \Big) \, \mathrm{cosech} \Big( \frac{(\beta E_k(\tau_1))_{\mathrm{eff},sq}}{2} \Big) (3.224)$$

where $(\beta E_k(\tau_1))_{\mathrm{eff}}$ is given by:

$$(\beta E_k(\tau_1))_{\mathrm{eff},sq} = \beta E_k(\tau_1) + \sum_{n=2}^{\infty} \kappa_{2n,\mathbf{DB/NB}}^{sq} |k|^{2n-1}, \qquad (3.225)$$

Thus, the expressions for the two-point for the GGE [84, 85] for the field $\chi$, its spatial derivative and its canonically conjugate momentum as:

$$G_{\chi\chi,sq}^{GGE}(\beta, \mathbf{r}, \tau_1, \tau_2) = \int \frac{d^3\mathbf{k}}{(2\pi)^3} \; \Big[ \mathcal{G}_{+,\chi\chi,sq}^{GGE}(\beta, \mathbf{k}, \tau_1, \tau_2) \; \exp(i\mathbf{k}.\mathbf{r})$$

$$+ \mathcal{G}_{-,\chi\chi,sq}^{GGE}(\beta, \mathbf{k}, \tau_1, \tau_2) \; \exp(-i\mathbf{k}.\mathbf{r}) \Big], \qquad (3.226)$$

$$G_{\partial_i\chi\partial_i\chi,sq}^{GGE}(\beta, \mathbf{k}, \tau_1, \tau_2) = \int \frac{d^3\mathbf{k}}{(2\pi)^3} \; \Big[ \mathcal{G}_{+,\partial_i\chi\partial_i\chi,sq}^{GGE}(\beta, \mathbf{k}, \tau_1, \tau_2) \; \exp(i\mathbf{k}.\mathbf{r})$$

$$+ \mathcal{G}_{-,\partial_i\chi\partial_i\chi,sq}^{GGE}(\beta, \mathbf{k}, \tau_1, \tau_2) \; \exp(-i\mathbf{k}.\mathbf{r}) \Big], \quad (3.227)$$

$$G_{\Pi_\chi\Pi_\chi,sq}^{GGE}(\beta, \mathbf{r}, \tau_1, \tau_2) = \int \frac{d^3\mathbf{k}}{(2\pi)^3} \; \Big[ \mathcal{G}_{+,\Pi_\chi\Pi_\chi,sq}^{GGE}(\beta, \mathbf{k}, \tau_1, \tau_2) \; \exp(i\mathbf{k}.\mathbf{r})$$

$$+ \mathcal{G}^{GGE}_{-,\Pi_\chi \Pi_\chi,sq} \left( \beta, \mathbf{k}, \tau_1, \tau_2 \right) \, \exp(-i\mathbf{k}.\mathbf{r}) \Bigg], \quad (3.228)$$

where we have defined the spatial separation between the two points $\mathbf{x}_1$ and $\mathbf{x}_2$ as:

$$\mathbf{r} :\equiv \mathbf{x}_1 - \mathbf{x}_2. \quad (3.229)$$

For each of the cases the corresponding thermal propagators in Fourier space are divided into two parts, one represents the advanced propagator appearing with $+$ symbol and the other one is the retarded propagator appearing with the $-$ symbol. To understand the mathematical structure of each of them let us first write their contributions independently in the following expressions:

$$\mathcal{G}^{GGE}_{+,\chi\chi,sq} \left( \beta, \mathbf{k}, \tau_1, \tau_2 \right) = \frac{v_{out}(\mathbf{k}, \tau_1) v^*_{out}(-\mathbf{k}, \tau_2)}{2a(\tau_1)a(\tau_2)} \, \exp\left( \frac{(\beta E_k(\tau_1))_{\text{eff,sq}}}{2} \right) \text{cosech}\left( \frac{(\beta E_k(\tau_1))_{\text{eff,sq}}}{2} \right),$$
$$(3.230)$$

$$\mathcal{G}^{GGE}_{-,\chi\chi,sq} \left( \beta, \mathbf{k}, \tau_1, \tau_2 \right) = \frac{v^*_{out}(-\mathbf{k}, \tau_1) v_{out}(\mathbf{k}, \tau_2)}{2a(\tau_1)a(\tau_2)} \, \exp\left( -\frac{(\beta E_k(\tau_1))_{\text{eff,sq}}}{2} \right) \text{cosech}\left( \frac{(\beta E_k(\tau_1))_{\text{eff,sq}}}{2} \right),$$
$$(3.231)$$

$$\mathcal{G}^{GGE}_{+,\partial_i\chi\partial_i\chi,sq} \left( \beta, \mathbf{k}, \tau_1, \tau_2 \right) = -k^2 \, \mathcal{G}^{GGE}_{+,\chi\chi,sq} \left( \beta, \mathbf{k}, \tau_1, \tau_2 \right), \quad (3.232)$$

$$\mathcal{G}^{GGE}_{-,\partial_i\chi\partial_i\chi,sq} \left( \beta, \mathbf{k}, \tau_1, \tau_2 \right) = -k^2 \, \mathcal{G}^{GGE}_{-,\chi\chi,sq} \left( \beta, \mathbf{k}, \tau_1, \tau_2 \right), \quad (3.233)$$

$$\mathcal{G}^{GGE}_{+,\Pi_\chi \Pi_\chi,sq} \left( \beta, \mathbf{k}, \tau_1, \tau_2 \right) = \frac{v'_{out}(\mathbf{k}, \tau_1) v^{*\prime}_{out}(-\mathbf{k}, \tau_2)}{2a(\tau_1)a(\tau_2)} \, \exp\left( \frac{(\beta E_k(\tau_1))_{\text{eff,sq}}}{2} \right) \text{cosech}\left( \frac{(\beta E_k(\tau_1))_{\text{eff,sq}}}{2} \right)$$
$$- \frac{\mathcal{G}^{GGE}_{+,\chi\chi} \left( \beta, \mathbf{k}, \tau_1, \tau_2 \right)}{a(\tau_1)a(\tau_2)} a'(\tau_1)a'(\tau_2),$$
$$(3.234)$$

$$\mathcal{G}^{GGE}_{-,\Pi_\chi \Pi_\chi,sq} \left( \beta, \mathbf{k}, \tau_1, \tau_2 \right) = \frac{v^{*\prime}_{out}(-\mathbf{k}, \tau_1) v'_{out}(\mathbf{k}, \tau_2)}{2a(\tau_1)a(\tau_2)} \, \exp\left( -\frac{(\beta E_k(\tau_1))_{\text{eff,sq}}}{2} \right) \text{cosech}\left( \frac{(\beta E_k(\tau_1))_{\text{eff,sq}}}{2} \right)$$
$$- \frac{\mathcal{G}^{GGE}_{-,\chi\chi,sq} \left( \beta, \mathbf{k}, \tau_1, \tau_2 \right)}{a(\tau_1)a(\tau_2)} a'(\tau_1)a'(\tau_2).$$
$$(3.235)$$

Now we consider a special case, which is the equal time configuration $\tau_1 = \tau_2 = \tau$. In that case we get the following expressions for the amplitude of the thermal power spectrum of the field $\chi$, its spatial derivative and its canonically conjugate momentum:

$$\mathcal{G}^{GGE}_{+,\chi\chi,sq} \left( \beta, \mathbf{k}, \tau_1 = \tau, \tau_2 = \tau \right) = \mathcal{P}^{GGE}_{+,\chi\chi,sq} \left( \beta, \mathbf{k}, \tau \right)$$
$$= \frac{v_{out}(\mathbf{k}, \tau) v^*_{out}(-\mathbf{k}, \tau)}{2a^2(\tau)} \, \exp\left( \frac{(\beta E_k(\tau_1))_{\text{eff,sq}}}{2} \right) \text{cosech}\left( \frac{(\beta E_k(\tau_1))_{\text{eff,sq}}}{2} \right),$$
$$(3.236)$$

$$\mathcal{G}^{GGE}_{-,\chi\chi,sq} \left( \beta, \mathbf{k}, \tau_1 = \tau, \tau_2 = \tau \right) = \mathcal{P}^{GGE}_{-,\chi\chi,sq} \left( \beta, \mathbf{k}, \tau \right)$$

$$= \frac{v_{out}^*(-\mathbf{k},\tau)v_{out}(\mathbf{k},\tau)}{2a^2(\tau)} \; \exp\left(-\frac{(\beta E_k(\tau_1))_{\text{eff,sq}}}{2}\right) \text{cosech}\left(\frac{(\beta E_k(\tau_1))_{\text{eff,sq}}}{2}\right)$$

$$(3.237)$$

$$\mathcal{G}_{+,\partial_i\chi\partial_i\chi,sq}^{GGE}\left(\beta,\mathbf{k},\tau_1=\tau,\tau_2=\tau\right) = \mathcal{P}_{+,\partial_i\chi\partial_i\chi,sq}^{GGE}\left(\beta,\mathbf{k},\tau\right) = -k^2\,\mathcal{P}_{+,\chi\chi,sq}^{GGE}\left(\beta,\mathbf{k},\tau\right),$$

$$(3.238)$$

$$\mathcal{G}_{-,\partial_i\chi\partial_i\chi,sq}^{GGE}\left(\beta,\mathbf{k},\tau_1=\tau,\tau_2=\tau\right) = \mathcal{P}_{-,\partial_i\chi\partial_i\chi,sq}^{GGE}\left(\beta,\mathbf{k},\tau\right) = -k^2\,\mathcal{P}_{-,\chi\chi,sq}^{GGE}\left(\beta,\mathbf{k},\tau\right),$$

$$(3.239)$$

$$\mathcal{G}_{+,\Pi_\chi\Pi_\chi,sq}^{GGE}\left(\beta,\mathbf{k},\tau_1=\tau,\tau_2=\tau\right) = \mathcal{P}_{+,\Pi_\chi\Pi_\chi,sq}^{GGE}\left(\beta,\mathbf{k},\tau\right)$$

$$= \frac{v_{out}'(\mathbf{k},\tau)v_{out}^{*\prime}(-\mathbf{k},\tau)}{2a^2(\tau)} \; \exp\left(\frac{(\beta E_k(\tau_1))_{\text{eff,sq}}}{2}\right) \text{cosech}\left(\frac{(\beta E_k(\tau_1))_{\text{eff,sq}}}{2}\right)$$

$$- \frac{\mathcal{P}_{+,\chi\chi,sq}^{GGE}\left(\beta,\mathbf{k},\tau\right)}{a^2(\tau)}a'^2(\tau),$$

$$(3.240)$$

$$\mathcal{G}_{-,\Pi_\chi\Pi_\chi,sq}^{GGE}\left(\beta,\mathbf{k},\tau_1=\tau,\tau_2=\tau\right) = \mathcal{P}_{-,\Pi_\chi\Pi_\chi,sq}^{GGE}\left(\beta,\mathbf{k},\tau\right)$$

$$= \frac{v_{out}^{*\prime}(-\mathbf{k},\tau)v_{out}'(\mathbf{k},\tau)}{2a^2(\tau)} \; \exp\left(-\frac{(\beta E_k(\tau_1))_{\text{eff,sq}}}{2}\right) \text{cosech}\left(\frac{(\beta E_k(\tau_1))_{\text{eff,sq}}}{2}\right)$$

$$- \frac{\mathcal{P}_{-,\chi\chi,sq}^{GGE}\left(\beta,\mathbf{k},\tau\right)}{a^2(\tau)}a'^2(\tau).$$

$$(3.241)$$

# 4   Numerical results

In this section, we study the behavior of the physically important power spectrum of the two-point correlators of different quantum states calculated in the Fourier transformed space. We plot the power spectrum with respect to the modes and it is expected that from our analysis these power spectrum and their associated signatures can be probed via various cosmological observational datasets. In each plot, we have incorporated the information regarding the three different choices of the initial conditions, which are appearing in terms of the Bunch Davies, $\alpha$ and Mota Allen vacua. We also have covered a large range of momentum modes to study the behavior of the obtained power spectra in small and large cosmological scales.

- In Fig. 4.1, the behavior of the power spectrum corresponding to the correlator $G_{\chi\chi}^0$ in the Fourier transformed space has been studied with respect to the mode functions. The difference in the effect of the choice of initial vacuum state can be very easily realized by seeing the behavior of the power spectrum for the lower modes. Three distinct lines are observed for the lower modes which suggest that the choice of initial vacuum has a non-trivial effect on the power spectrum. The amplitude of the correlator is the lowest for the Bunch Davies vacuum. However, the amplitude

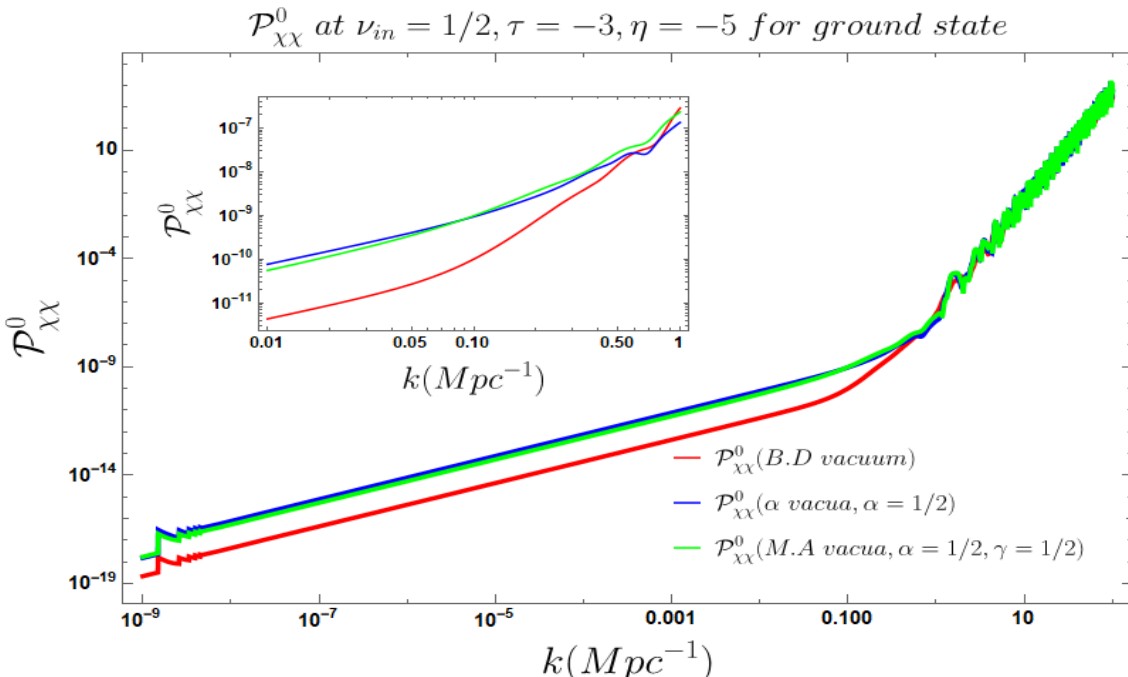

**Figure 4.1**: Behavior of the power spectrum of the correlator $G_{\chi\chi}$ for the ground state with respect to the comoving wave number/scale $k$.

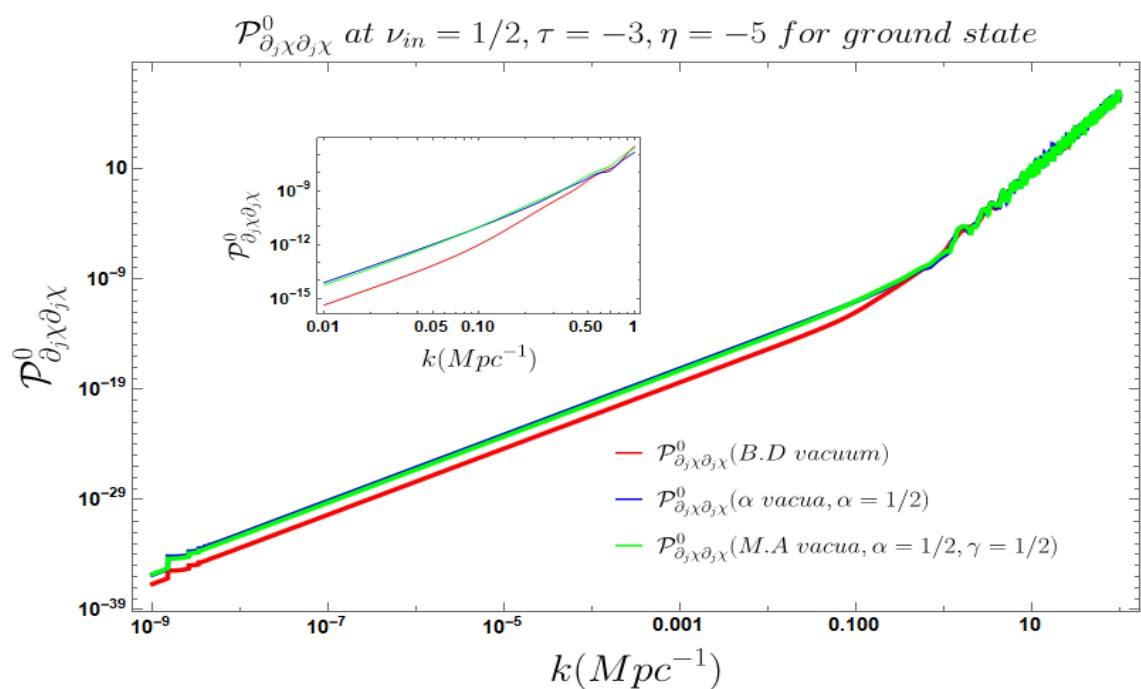

**Figure 4.2**: Behavior of the power spectrum of the correlator $G_{\partial_j\chi\partial_j\chi}$ for the ground state with respect to the comoving wave number/scale $k$.

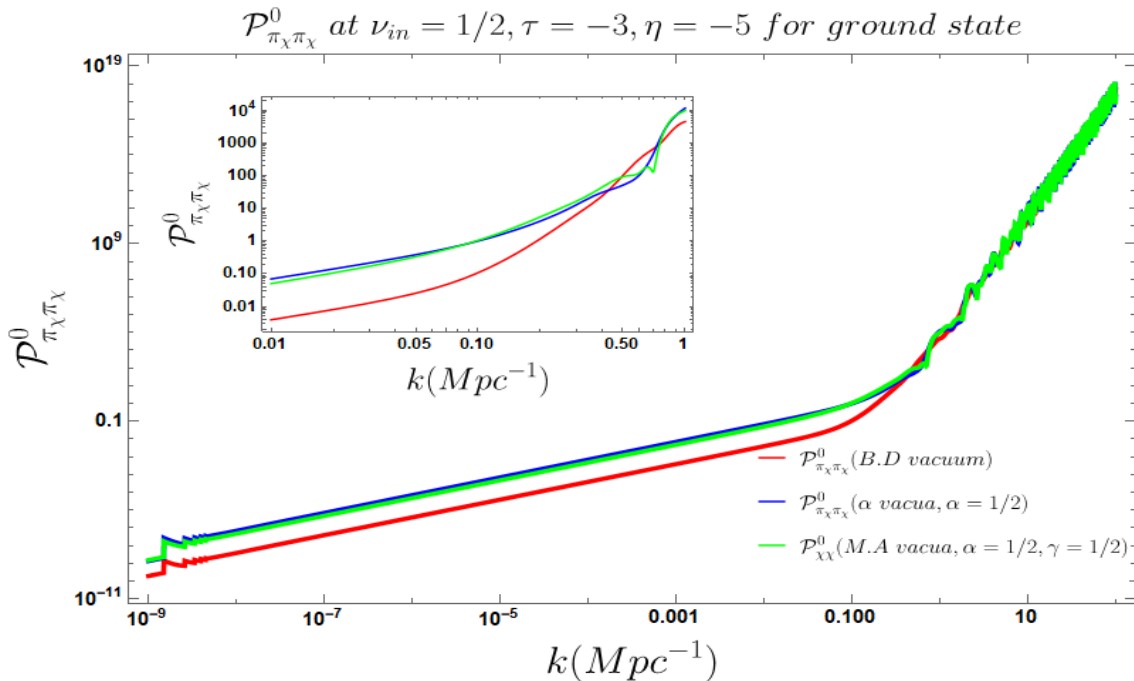

**Figure 4.3**: Behavior of the power spectrum of the correlator $G_{\Pi_\chi \Pi_\chi}$ for the ground state with respect to the comoving wave number/scale $k$.

for the alpha and the Mota Allen vacua cross over, as can be clearly seen from the inset of Fig. 4.1. From higher modes, it is extremely difficult to capture the role of the initial vacuum state in the power spectrum, due to the overlapping of the curves in that region. However, it should be noted that the overlap behavior of the power spectrum is independent of the choice of initial vacuum and more or less follows an identical pattern for all the vacuum states. From this plot, it is also observed that upto a certain range of the mode $k$ the obtained spectra grows almost linearly. After crossing the value $k \sim 1.20$ Mpc$^{-1}$ rapid oscillations with small amplitude can be observed, though the slope of the growth of the spectra in this region is higher than the previous one. From the present observational probes (Planck 2018 data [92]) the amplitude of the scalar modes from the power spectrum has to lie within the range $(2.975 \pm 0.056) \times 10^{-10}$ at 68% CL. From this plot, we have found that the amplitude of the spectrum exactly matches with the observed value within the range of the comoving scale $10^{-3}$ Mpc$^{-1} \le k \le 0.2$ Mpc$^{-1}$, which is a satisfactory finding of our analysis. Here it is important to note that for the observation purpose the pivot scale is chosen to be within the range of comoving scale $0.005$ Mpc$^{-1} \le k \le 0.2$ Mpc$^{-1}$, which again confronts well with our finding. Specific features appearing in the spectrum suggest that it should have spectral tilt ($n^0_{\chi\chi} = d\ln P^0_{\chi\chi}/d\ln k$), spectral running of the tilt ($\alpha^0_{\chi\chi} = dn^0_{\chi\chi}/d\ln k = d^2\ln P^0_{\chi\chi}/d\ln k^2$) and running of the running of tilt ($\beta^0_{\chi\chi} = d\alpha^0_{\chi\chi}/d\ln k = d^2 n^0_{\chi\chi}/d\ln k^2 = d^3\ln P^0_{\chi\chi}/d\ln k^3$) within the

observed range from the Planck 2018 data [92]. Due to the huge length of the paper we do not pursue these crucial possibilities explicitly. In the near future, we intend to explore these possibilities in detail.

- In Fig. 4.2, the behavior of the power spectrum corresponding to the correlator $G^0_{\partial_j \chi \partial_j \chi}$ in the Fourier transformed space has been studied with respect to the co-moving scale $k$. The overall behavior of the power spectrum is almost identical to the behavior of the correlator $G^0_{\chi\chi}$. However, the amplitude in the entire mode range is very small as compared to the power spectrum obtained for the $G^0_{\chi\chi}$ correlator. In the higher mode region, a difference in the behavior can also be observed. Though both the power spectrum exhibit a rising behavior in the higher mode region, the rate of increase for the $G^0_{\partial_j \chi \partial_j \chi}$ correlator is appreciably less than that of the $G^0_{\chi\chi}$ correlator which is again a new finding from our analysis. In the observational probes this type of two-point correlator and their associated power spectrum is not actually analyzed. But since we know the connection between this particular type of power spectrum with the previously derived one, it is expected to have smaller amplitude in this context. From the observational perspective it is expected that in near future, with the development of statistical accuracy in the CL, it may possible to directly probe this type of power spectrum.

- The behavior of the power spectrum corresponding to the correlator $G^0_{\Pi_\chi \Pi_\chi}$ in the Fourier transformed space has been plotted with respect to the mode functions in Fig. 4.3. We observe a behavior which is almost identical to the behavior shown by the power spectrum corresponding to the $G^0_{\chi\chi}$ correlator in the entire mode region. We have found that the corresponding amplitude of the power spectrum from the momentum two-point correlators are larger compared to the two types of spectra studied above. In the observational probes this type of two-point correlator and its associated power spectrum is not actually analyzed till date. However, it is expected to get signatures from two-point momentum correlator in future observational probes.

- In Fig. 4.4, we have plotted the behavior of the power spectrum corresponding to the correlator $G^{sq}_{\chi\chi}$ for the squeezed state. We observe three distinctive behavior in the three comoving scale regions. For the lower mode region, we observe rapid fluctuations in the power spectrum with the amplitude being the largest for the Bunch-Davies vacuum case. A decreasing behavior is also observed for the lower mode region. In the intermediate mode region, the decreasing behavior is continued with the rate of decrease being significantly larger than the lower mode region. However, the point worth mentioning is the fact that the amplitude for the Bunch-Davies case becomes lowest in this region. The higher mode region how-ever, shows a slowly rising behavior, with the contribution from the different vac-uum being almost identical, as evident from the overlapping curves. From the

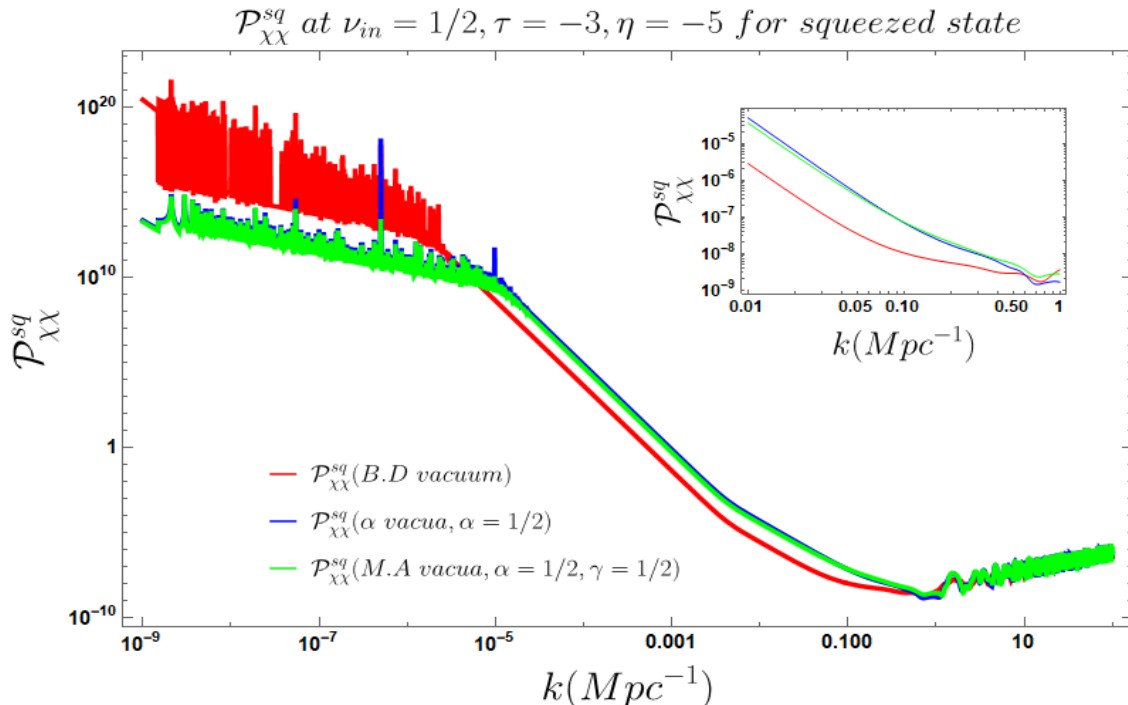

**Figure 4.4**: Behavior of the power spectrum of the correlator $G_{\chi\chi}$ for the squeezed state with respect to the comoving wave number/scale $k$.

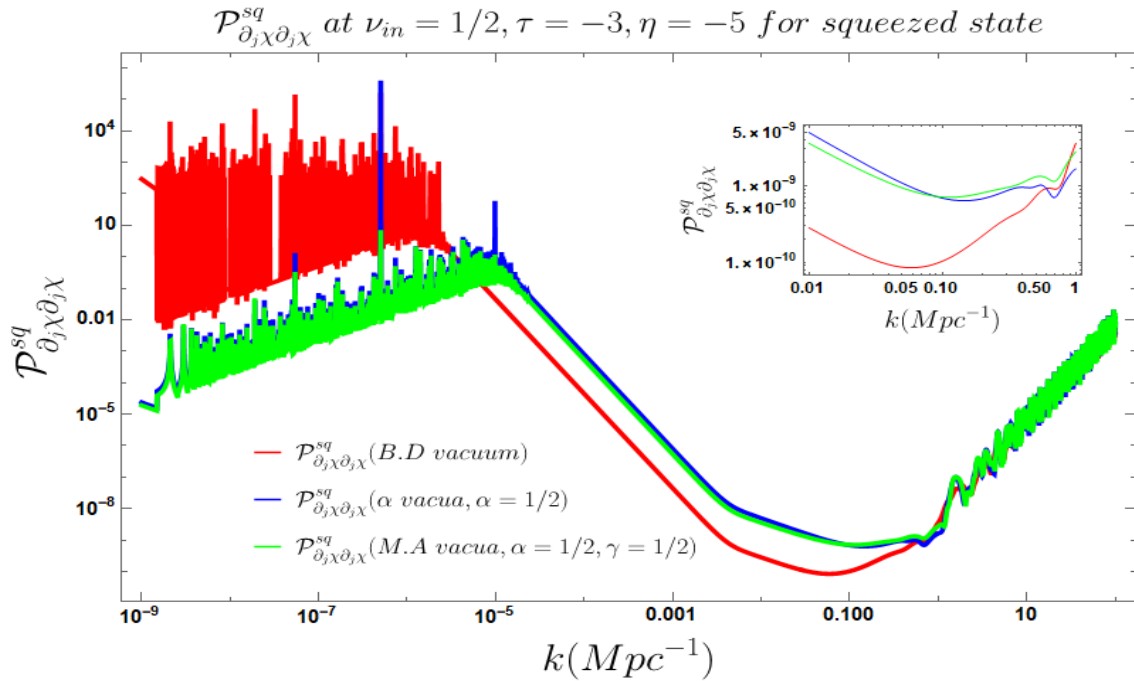

**Figure 4.5**: Behavior of the power spectrum of the correlator $G_{\partial_j\chi\partial_j\chi}$ for the squeezed state with respect to the comoving wave number/scale $k$.

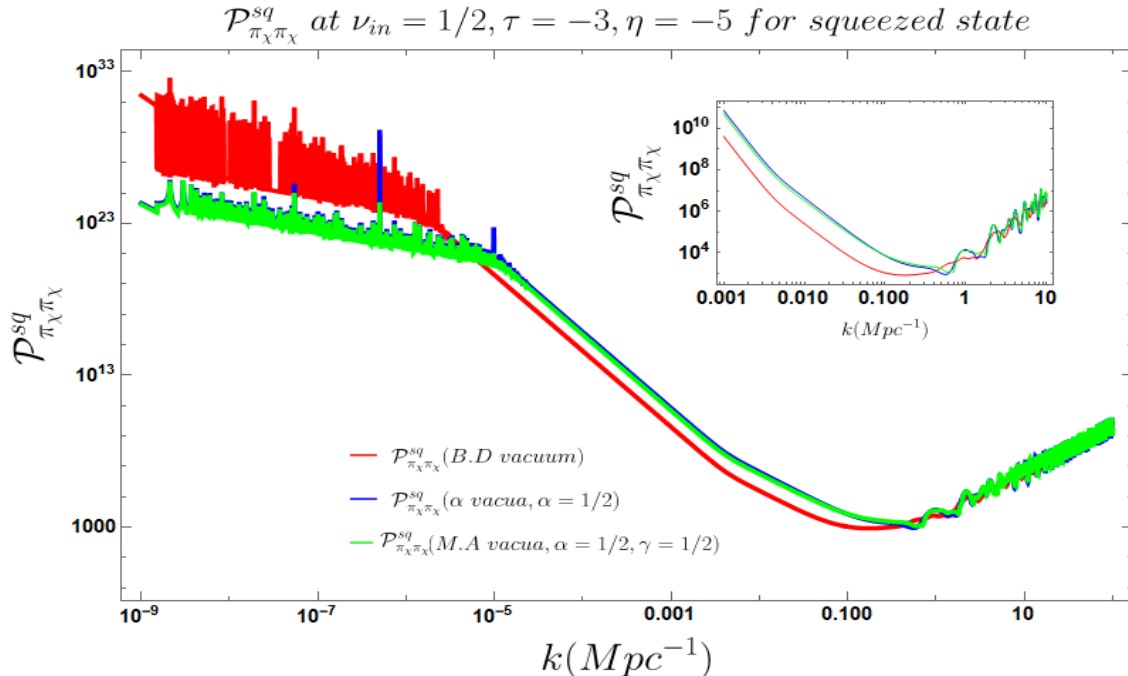

**Figure 4.6**: Behavior of the power spectrum of the correlator $G_{\Pi_\chi \Pi_\chi}$ for the ground state with respect to the comoving wave number/scale $k$.

present observational probes (Planck 2018 data [92]) the amplitude of the scalar modes from the power spectrum has to lie within the range $(2.975 \pm 0.056) \times 10^{-10}$ at 68% CL. From this plot, we have found that the amplitude of the spectrum exactly matches with the observed value within the range of the comoving scale $0.1 \text{ Mpc}^{-1} \leq k \leq 0.3 \text{ Mpc}^{-1}$, which is an interest observation from our analysis. Here it is important to note that for the observation purpose the pivot scale is chosen to be within the range of comoving scale $0.005 \text{ Mpc}^{-1} \leq k \leq 0.2 \text{ Mpc}^{-1}$, which again confronts well with our finding. Specific features appearing in the spectrum suggest that it should have spectral tilt $(n^{sq}_{\chi\chi} = d \ln P^{sq}_{\chi\chi}/d \ln k)$, spectral running of the tilt $(\alpha^{sq}_{\chi\chi} = dn^{sq}_{\chi\chi}/d \ln k = d^2 \ln P^{sq}_{\chi\chi}/d \ln k^2)$ and running of the running of tilt $(\beta^{sq}_{\chi\chi} = d\alpha^{sq}_{\chi\chi}/d \ln k = d^2 n^{sq}_{\chi\chi}/d \ln k^2 = d^3 \ln P^{sq}_{\chi\chi}/d \ln k^3)$ within the observed range from the Planck 2018 data [92]. In future we intend to explore these possibilities in detail.

- In Fig. 4.5, we have plotted the behavior of the power spectrum corresponding to the correlator $G^{sq}_{\partial_j\chi\partial_j\chi}$ for the squeezed state. The behavior of the power spectrum in the intermediate and the higher modes are nearly similar to the previous case. However, a difference exists in the lower mode region. Whereas in the previous case, we observed decreasing behavior for the lower modes, the power spectrum exhibits an increasing behavior in this case. The peculiar behavior for the Bunch-Davies case as was seen in the earlier case, also persists in this power spectrum. In the observational

probes this type of two-point correlator and their associated spectrum has not been analyzed yet. It is expected to have smaller amplitude in this context, which may be tested in near future with the development of statistical accuracy in the CL.

- In Fig. 4.6, we have plotted the behavior of the power spectrum corresponding to the correlator $G_{\Pi_\chi \Pi_\chi}$ for the squeezed state. We observe the behavior the power spectrum to be identical to that shown for the correlator $G_{\chi\chi}$.

- In Fig. 4.7(a) and Fig. 4.7(b) the behavior of the power spectrum corresponding to the correlator $G_{\chi\chi}$ for the gCC and the squeezed gCC states with respect to the comoving scale has been shown, respectively. We observe that the gCC state both with and without squeezing show a similar decreasing behavior in the lower and the intermediate regions, though the rate of decrease may not be identical in both the cases. However, strikingly different behavior can be observed for the higher modes. Whereas for the gCC state without squeezing, the power spectrum diverges to positive infinity for all the vacua, the same divergence to positive infinity is also observed for the case without squeezing but only for the $\alpha$ vacua case. The power spectrum for the Bunch-Davies and the Mota Allen vacua diverges to the negative infinity at a higher mode. From the present observational probes (Planck 2018 data [92]) the amplitude of the scalar modes from the power spectrum has to lie within the range $(2.975 \pm 0.056) \times 10^{-10}$ at 68% CL. From this plot, we find that the amplitude of the spectrum exactly matches with the observed value within the range of the comoving scale $0.2 \ \mathrm{Mpc}^{-1} \leq k \leq 0.3 \ \mathrm{Mpc}^{-1}$, which is an important consistency check for our analysis. Here it is important to note that for the observation purpose the pivot scale is chosen to be within the range of comoving scale $0.005 \ \mathrm{Mpc}^{-1} \leq k \leq 0.2 \ \mathrm{Mpc}^{-1}$, which again confronts well with our finding. Specific features appearing in the spectrum suggest that it should have spectral tilt ($n_{\chi\chi}^{gCC} = d \ln P_{\chi\chi}^{gCC}/d \ln k$ and $n_{\chi\chi}^{sq,gCC} = d \ln P_{\chi\chi}^{sq,gCC}/d \ln k$), spectral running of the tilt ($\alpha_{\chi\chi}^{gCC} = dn_{\chi\chi}^{gCC}/d \ln k = d^2 \ln P_{\chi\chi}^{gCC}/d \ln k^2$ and $\alpha_{\chi\chi}^{sq,gCC} = dn_{\chi\chi}^{sq,gCC}/d \ln k = d^2 \ln P_{\chi\chi}^{sq,gCC}/d \ln k^2$) and running of the running of tilt ($\beta_{\chi\chi}^{gCC} = d\alpha_{\chi\chi}^{gCC}/d \ln k = d^2 n_{\chi\chi}^{gCC}/d \ln k^2 = d^3 \ln P_{\chi\chi}^{gCC}/d \ln k^3$ and $\beta_{\chi\chi}^{sq,gCC} = d\alpha_{\chi\chi}^{sq,gCC}/d \ln k = d^2 n_{\chi\chi}^{sq,gCC}/d \ln k^2 = d^3 \ln P_{\chi\chi}^{sq,gCC}/d \ln k^3$) within the observed range from the Planck 2018 data [92]. In future we plan to explore these possibilities in detail.

- In Fig. 4.8(a) and Fig. 4.8(b), the behavior of the power spectrum corresponding to the correlator $G_{\partial_j \chi \partial_j \chi}$ for the gCC and the squeezed gCC states with respect to the modes has been shown, respectively. In contrast to the previous case, we observe that the gCC state both with and without squeezing shows a similar increasing behavior in the lower and the intermediate region, though the rate of increase may not be identical in both the cases. The divergence behavior at the higher modes however

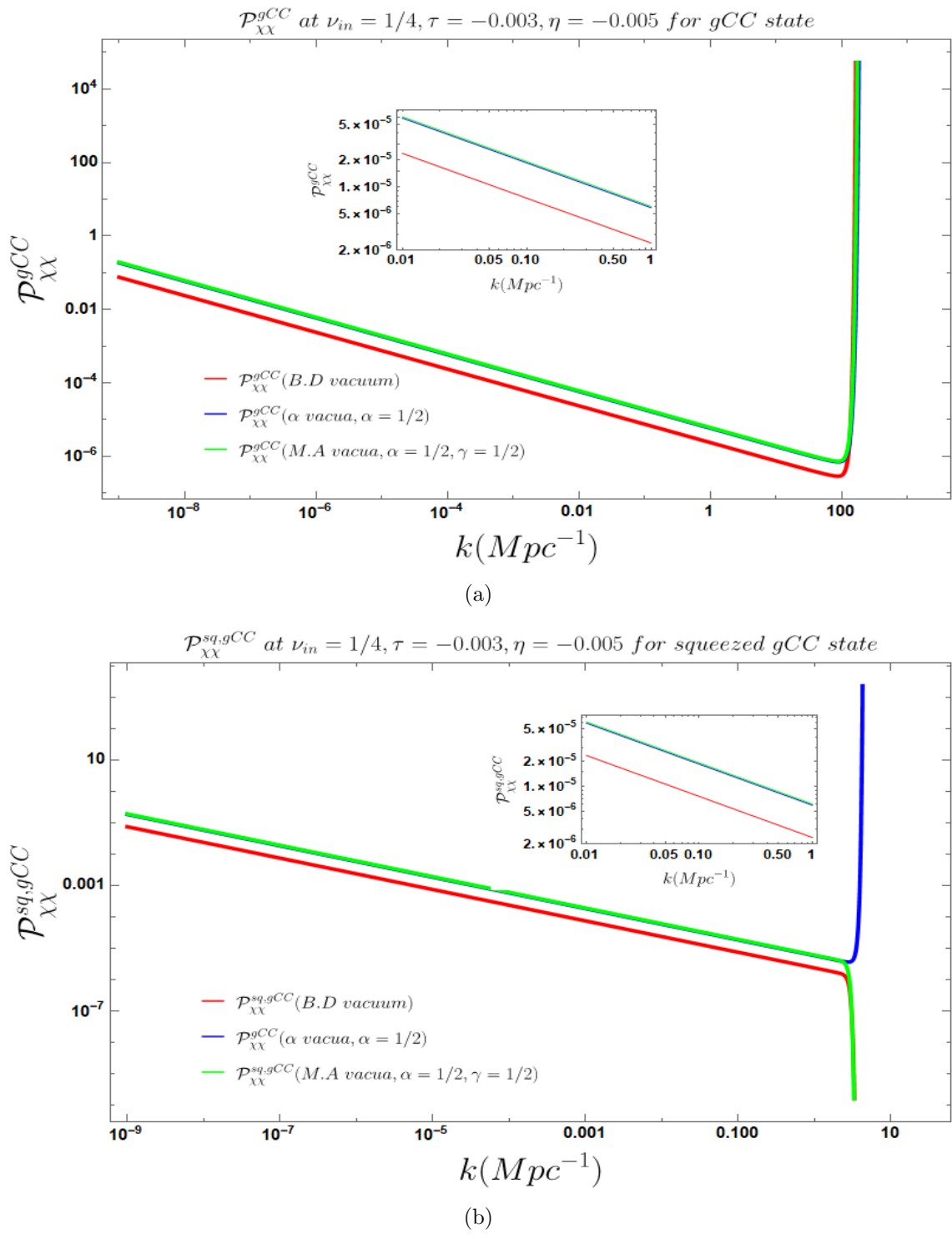

**Figure 4.7**: Behavior of the power spectrum of the correlator $G_{\chi\chi}$ for the gCC and the squeezed gCC state respectively obtained after quench with respect to the comoving wave number/scale $k$.

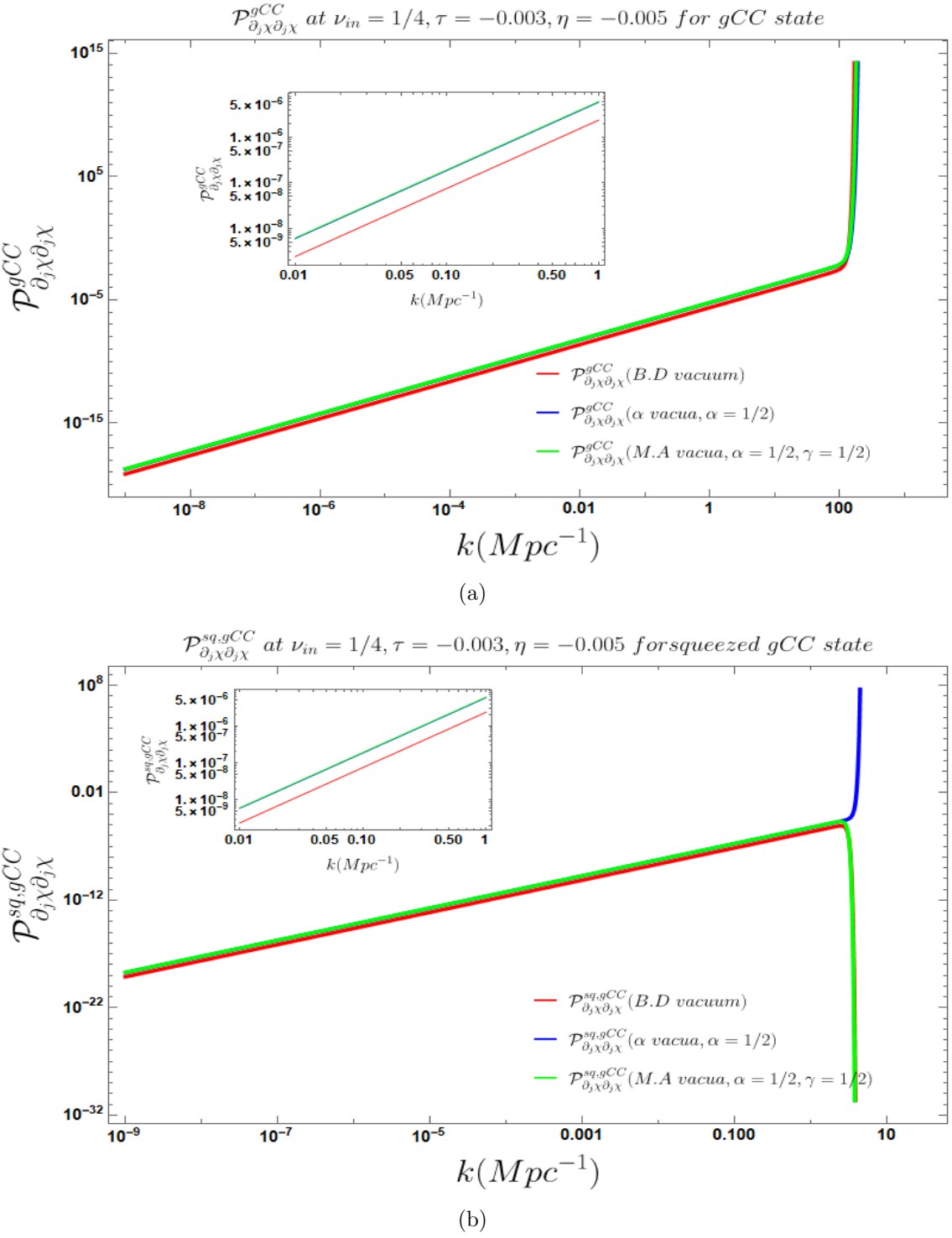

**Figure 4.8**: Behavior of the power spectrum of the correlator $G_{\partial_j\chi\partial_j\chi}$ for the gCC and the squeezed gCC state respectively obtained after quench with respect to the comoving wave number/scale $k$.

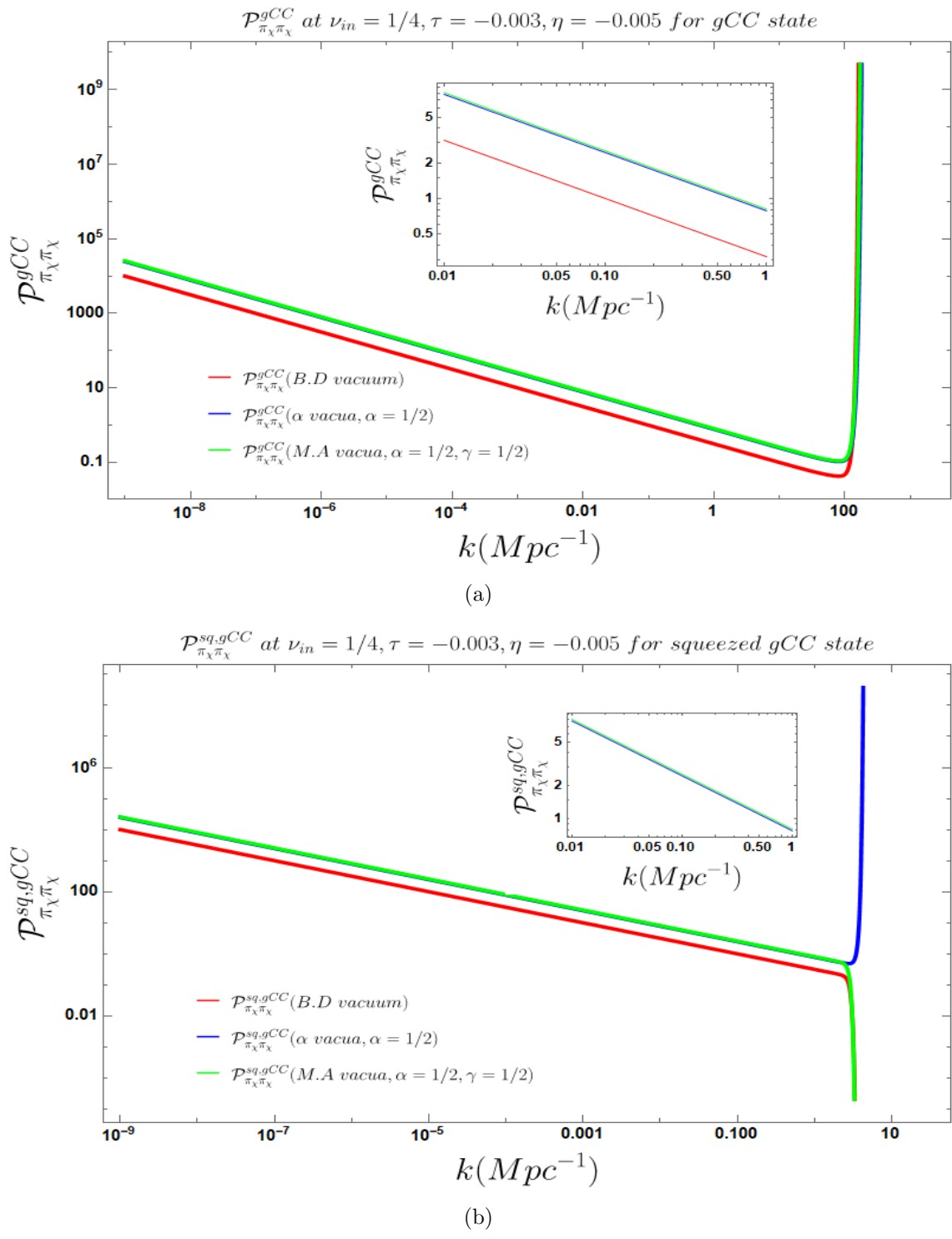

**Figure 4.9**: Behavior of the power spectrum of the correlator $G_{\Pi_\chi \Pi_\chi}$ for the gCC and the squeezed gCC state respectively obtained after quench with respect to the comoving wave number/scale $k$.

remains identical to the previous case. In the observational probes this type of two-point correlators and their associated spectra have not been actually analyzed yet. Though it is expected to have smaller amplitudes, it may be tested in the near future with the development of statistical accuracy in the CL.

- In Fig. 4.9(a) and Fig. 4.9(b), the behavior of the power spectrum corresponding to the correlator $G_{\Pi_\chi \Pi_\chi}$ for the gCC and the squeezed gCC states with respect to the modes has been shown, respectively. The behavior of the power spectrum in the entire mode region is identical to that for the correlator $G_{\chi\chi}$. The divergence pattern at the higher mode region is also similar in behavior.

- In Fig. 4.10(a), we have plotted the advanced part of the power spectrum corresponding to the Generalised Gibbs ensemble correlator $G_{\chi\chi}^{GGE}$ calculated for states without squeezing at a low value of $\beta$. We observe that for lower modes the amplitude of the power spectrum shows a gradual increasing behavior. In the intermediate mode range however the amplitude of the power spectrum saturates followed by a sharp decresing nature at the higher mode range.

- In Fig. 4.10(b), we have plotted the advanced part of the power spectrum corresponding to the Generalised Gibbs ensemble correlator $G_{\chi\chi}^{GGE}$ calculated for states without squeezing at a high value of $\beta$. The behavior of the power spectrum is almost identical to the one we observe for the low beta case. However, the most crucial difference is that the amplitude in this case is negligible as compared to the low beta case.

- In Fig. 4.11(a), we have plotted the retarded part of the power spectrum corresponding to the Generalised Gibbs ensemble correlator $G_{\chi\chi}^{GGE}$ calculated for post-quench state without squeezing at a low value of $\beta$. We observe that for lower modes, the amplitude of the power spectrum is constant. In the intermediate range the amplitude shows a gradual decreasing behavior followed by an overall increasing feature in the higher mode range. Also, the dependence of the amplitude on the choice of the intial vacuum condition is visible only in the intermediate mode range.

- In Fig. 4.11(b), we have plotted the retarded part of the power spectrum corresponding to the Generalised Gibbs ensemble correlator $G_{\chi\chi}^{GGE}$ calculated for post-quench state without squeezing at a high value of $\beta$. We observe that for lower and intermediate modes, the amplitude of the power spectrum shows a decreasing behavior which is widely different from what we observe in the low $\beta$ case. The behavior in the higher mode region is identical to what we observed in the low $\beta$ case. Also, the dependence of the amplitude on the choice of the intial vacuum condition is visible throughout the lower and intermediate mode region, which is also an interesting

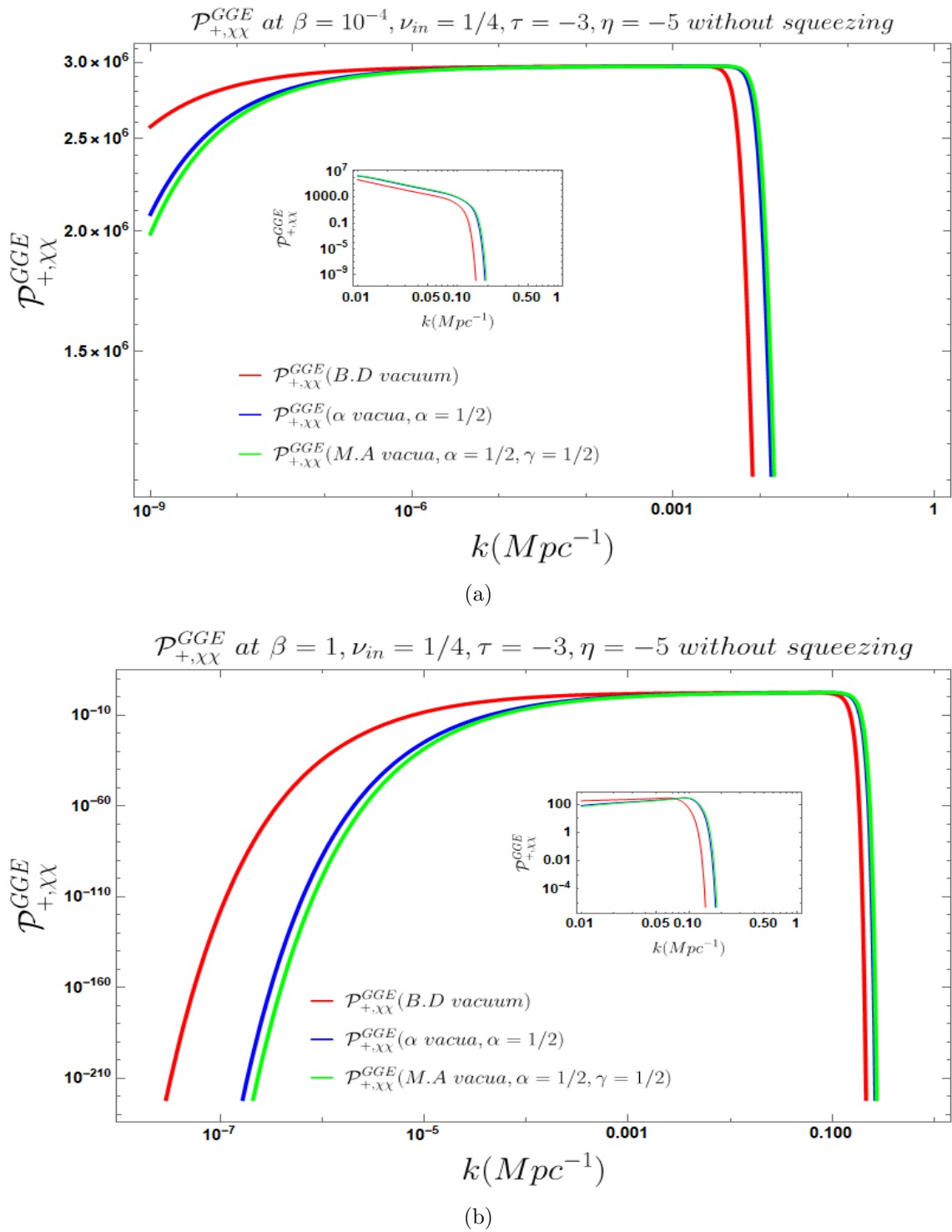

**Figure 4.10**: Behavior of the power spectrum corresponding to the advanced part of the correlator $G_{\chi\chi}^{GGE}$ with respect to the comoving wave number/scale $k$ at higher and lower temperatures.

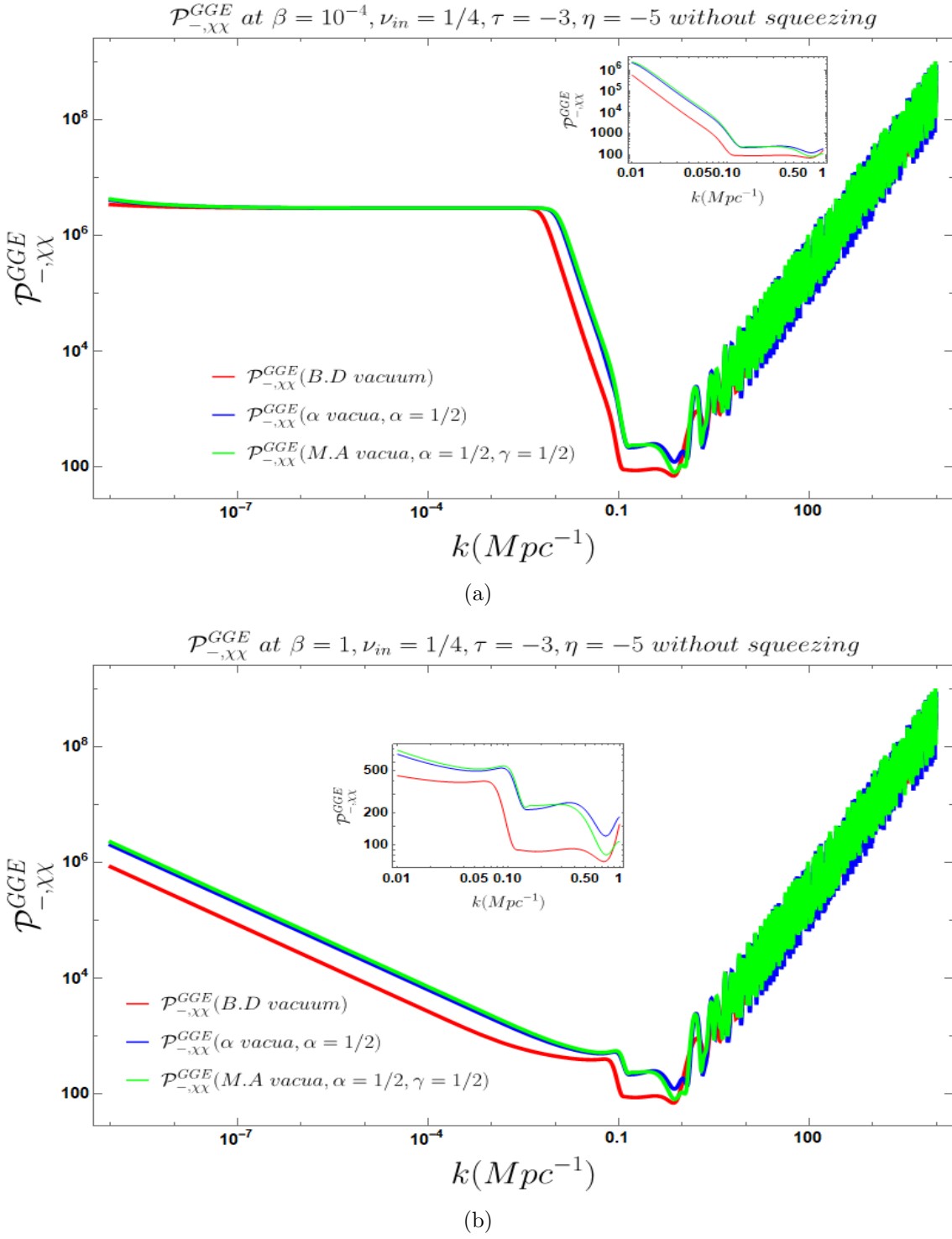

**Figure 4.11**: Behavior of the power spectrum corresponding to the retarded part of the correlator $G_{\chi\chi}^{GGE}$ with respect to the comoving wave number/scale $k$ at higher and lower temperatures.

difference from the low $\beta$ case. It is also worth mentioning that the amplitude of the retarded part takes almost similar values for high and low $\beta$, whereas for the advanced part the amplitude is almost negligible for the high $\beta$ as compared to low $\beta$.

- In Fig. 4.12(a), we have plotted the advanced part of the power spectrum corresponding to the Generalised Gibbs ensemble correlator $G^{GGE}_{\partial_i\chi\partial_i\chi}$ calculated for states without squeezing at a low value of $\beta$. We observe that for lower and intermediate modes, the power spectrum shows a strictly increasing behavior. However, it shows a sharp and abrupt decrease in the power spectrum for higher modes.

- In Fig. 4.12(b), we have plotted the advanced part of the power spectrum corresponding to the Generalised Gibbs ensemble correlator $G^{GGE}_{\partial_i\chi\partial_i\chi}$ calculated for states without squeezing at a high value of $\beta$. We observe that for lower modes, the behavior of the power spectrum shows a gradual increase. The rate of increase in the intermediate mode region is very less making the increasing nature very slow. However, the sharp fall in the higher mode region is observed in this case as well. Also, a key difference is observed in the lower modes region for the lower and higher $\beta$ case. The dependence of the power spectrum on the initial conditions is clearly visible in the higher $\beta$ case whereas the curves overlap in the lower $\beta$ case. Similar to the advanced part of the $G^{GGE}_{\chi\chi}$ correlator, the amplitude of the power spectrum for the low $\beta$ case is almost negligible compared to the high $\beta$ case.

- In Fig. 4.13(a), we have plotted the retarded part of the power spectrum corresponding to the Generalised Gibbs ensemble correlator $G^{GGE}_{\partial_i\chi\partial_i\chi}$ calculated for states without squeezing at a low value of $\beta$. We observe an overall increasing behavior for the entire lower mode region followed by a peculiar decreasing and then increasing nature of the spectrum in the intermediate mode region followed by an overall increasing behavior again at the higher mode region.

- In Fig. 4.13(b), we have plotted the retarded part of the power spectrum corresponding to the Generalised Gibbs ensemble correlator $G^{GGE}_{\partial_i\chi\partial_i\chi}$ calculated for states without squeezing at a high value of $\beta$. The overall behavior and the amplitude is almost identical to the low $\beta$ case. However, the inlets of the plots clearly show that the initial increase in the power spectrum occurs upto a large value of $k$ for high $\beta$.

- In Fig. 4.14(a), we have plotted the advanced part of the power spectrum corresponding to the Generalised Gibbs ensemble correlator $G^{GGE}_{\Pi_\chi\Pi_\chi}$ calculated for states without squeezing at a low value of $\beta$. We observe that for lower modes the power spectrum shows a strictly increasing behavior. The amplitude of the power spectrum in the

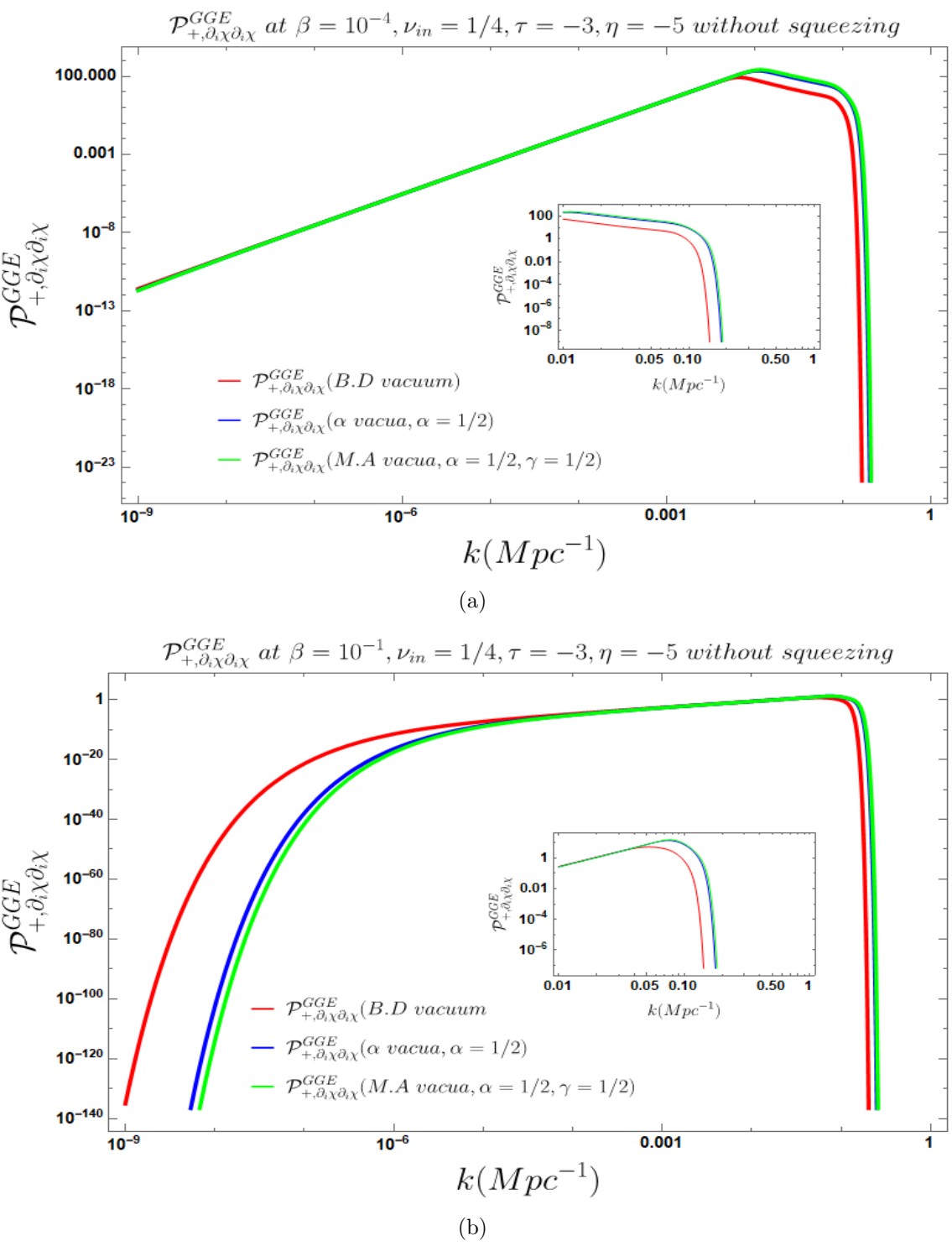

**Figure 4.12**: Behavior of the power spectrum corresponding to the advanced part of the correlator $G^{GGE}_{\partial_i \chi \partial_i \chi}$ with respect to the comoving wave number/scale $k$ at higher and lower temperatures.

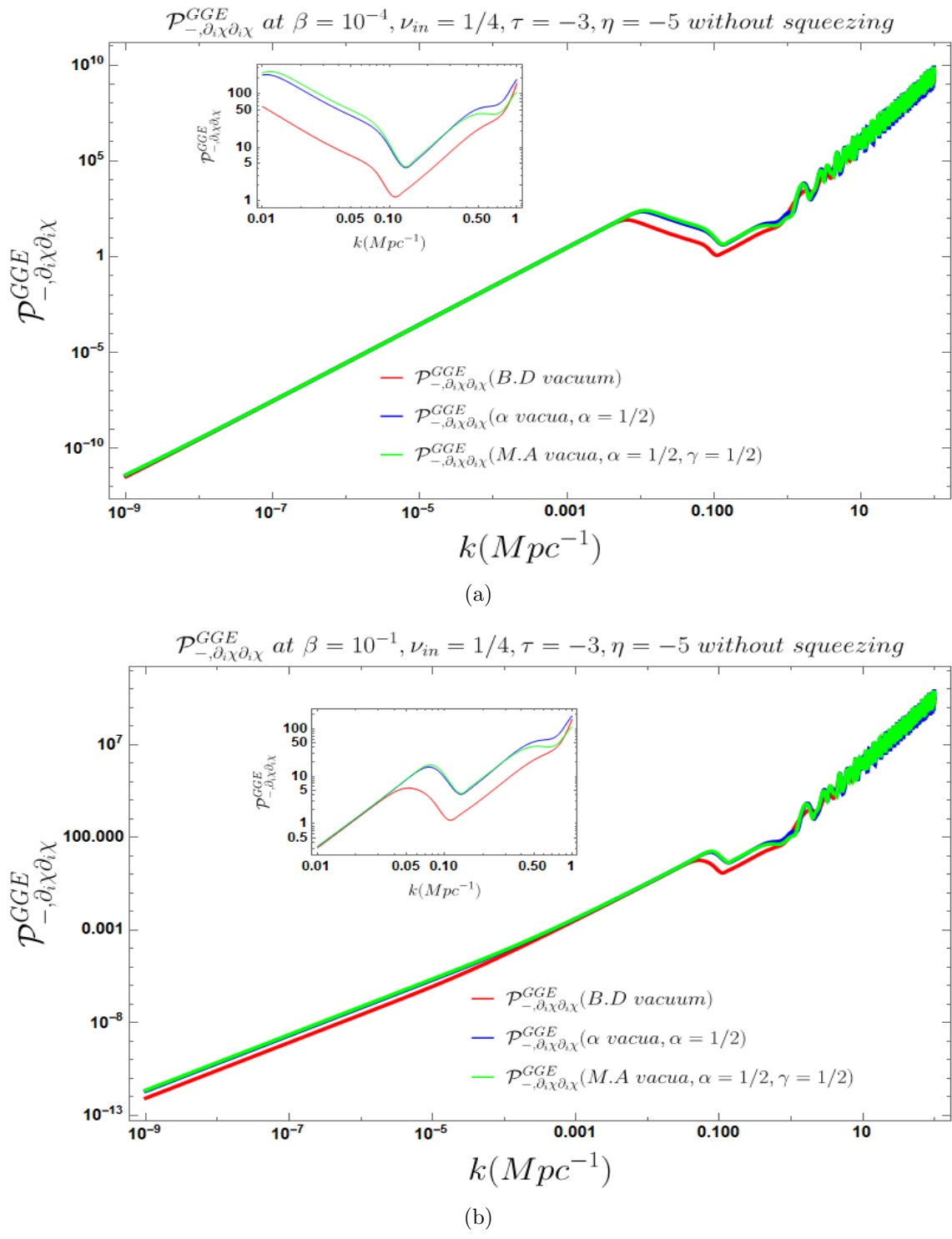

**Figure 4.13**: Behavior of the power spectrum corresponding to the retarded part of the correlator $G_{\partial_i\chi\partial_i\chi}^{GGE}$ with respect to the comoving wave number/scale $k$ at higher and lower temperatures.

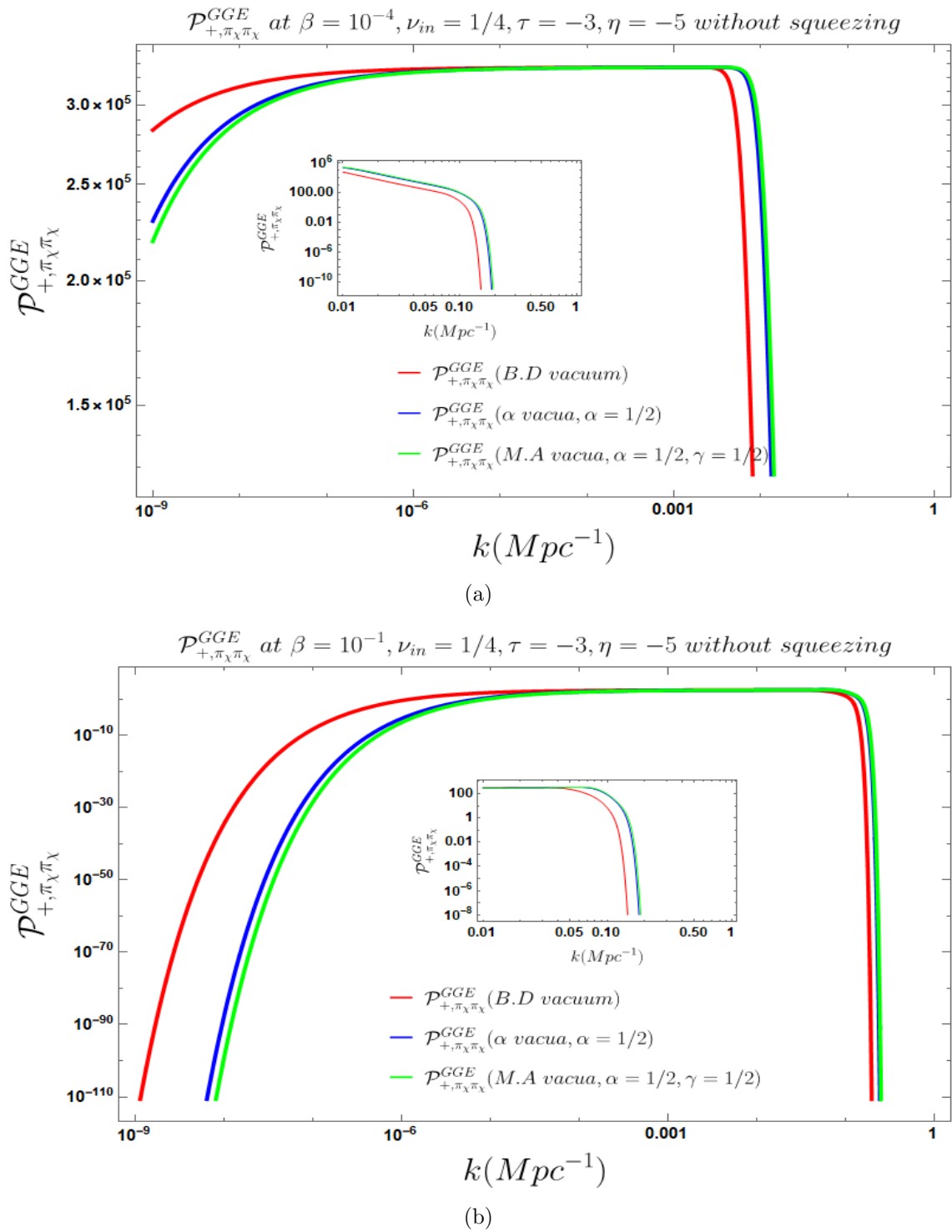

**Figure 4.14**: Behavior of the power spectrum corresponding to the advanced part of the correlator $G_{\Pi_\chi\Pi_\chi}^{GGE}$ with respect to the comoving wave number/scale $k$ at higher and lower temperatures.

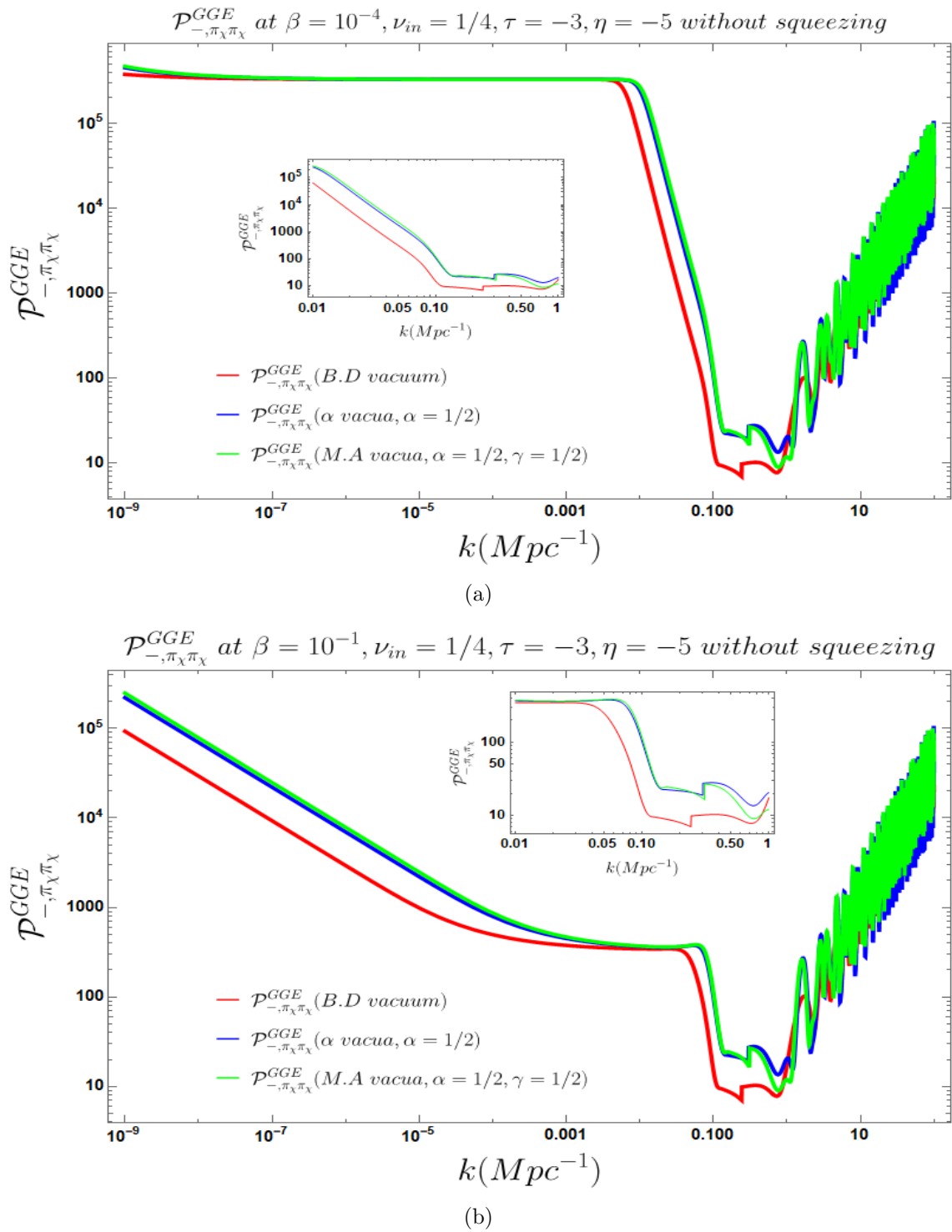

**Figure 4.15**: Behavior of the power spectrum corresponding to the retarded part of the correlator $G^{GGE}_{\Pi_\chi \Pi_\chi}$ with respect to the comoving wave number/scale $k$ at higher and lower temperatures.

intermediate modes is almost constant and the curves corresponding to the different initial conditions overlap. However, a sharp and abrupt fall in the power spectrum is observed for higher modes.

- In Fig. 4.14(b), we have plotted the advanced part of the power spectrum corresponding to the Generalised Gibbs ensemble correlator $G^{GGE}_{\Pi_\chi \Pi_\chi}$ calculated for states without squeezing at a high value of $\beta$. The overall behavior of the power spectrum is almost identical is identical to the low $\beta$ case apart from the fact that the amplitude of the power spectrum in this case is negligible compared to the low $\beta$ case.

- In Fig. 4.15(a), we have plotted the retarded part of the power spectrum corresponding to the Generalised Gibbs ensemble correlator $G^{GGE}_{\Pi_\chi \Pi_\chi}$ calculated for states without squeezing at a low value of $\beta$. It is observed that the power spectrum shows a saturation in its value for the lower mode region. This is followed by a decreasing behavior in the intermediate mode region. Clear distinction between the curves corresponding to different initial conditions is also visible in the intermediate mode region. In the higher mode region an overall increasing behavior of the power spectrum is observed.

- In Fig. 4.15(b), we have plotted the retarded part of the power spectrum corresponding to the Generalised Gibbs ensemble correlator $G^{GGE}_{\Pi_\chi \Pi_\chi}$ calculated for states without squeezing at a high value of $\beta$. The power spectrum in the lower mode region shows a smooth decreasing behavior. This is followed by a saturation for a small range of $k$ and then a sudden fall in the intermediate mode region. The behavior of the higher mode region is identical to the low $\beta$ case.

- In Fig. 4.16(a), we have plotted the advanced part of the power spectrum corresponding to the Generalised Gibbs ensemble correlator $G^{GGE}_{\chi\chi}$ calculated for states with squeezing at a low value of $\beta$. We observe that for lower modes the power spectrum slowly increases for a short range of $k$ after which it attains a saturation in its value. However, after a certain $k$, the power spectrum shows an abrupt fall in its value. The amplitude of the power spectrum is very high for the entire range of the $k$.

- In Fig. 4.16(b), we have plotted the advanced part of the power spectrum corresponding to the Generalised Gibbs ensemble correlator $G^{GGE}_{\chi\chi}$ calculated for states with squeezing at a high value of $\beta$. The behavior of the power spectrum in this case is widely different from the one observed for the low $\beta$ case. At lower values of $k$ the power spectrum increases at a very fast rate. The saturation value in the intermedimate mode region is observed in this case as well. However, the sharp fall in the higher $k$ values that was observed in the previous case does not happen in this scenario.

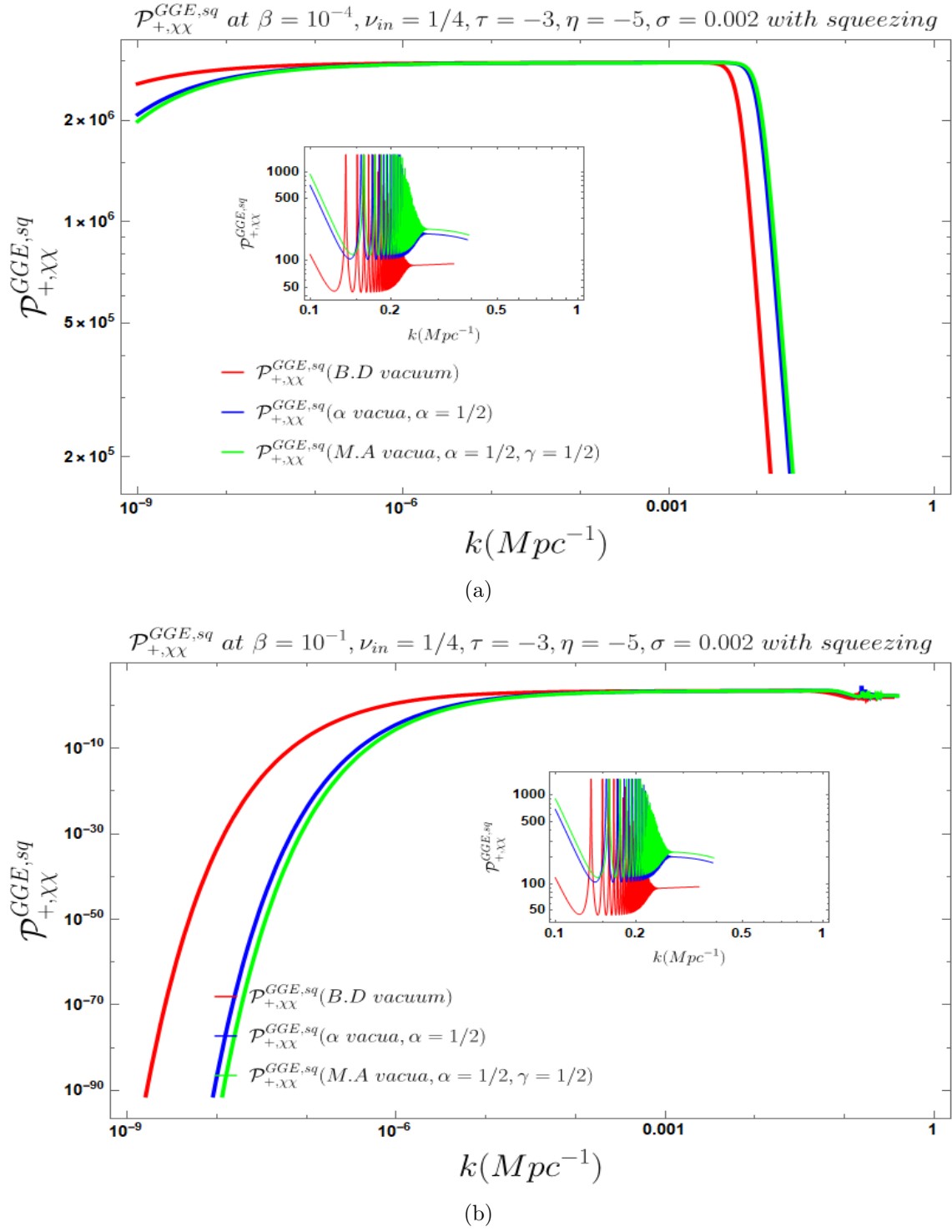

**Figure 4.16**: Behavior of the power spectrum corresponding to the advanced part of the correlator $G_{\chi\chi}^{GGE}$ with respect to the comoving wave number/scale $k$ at higher and lower temperatures in presence of squeezing.

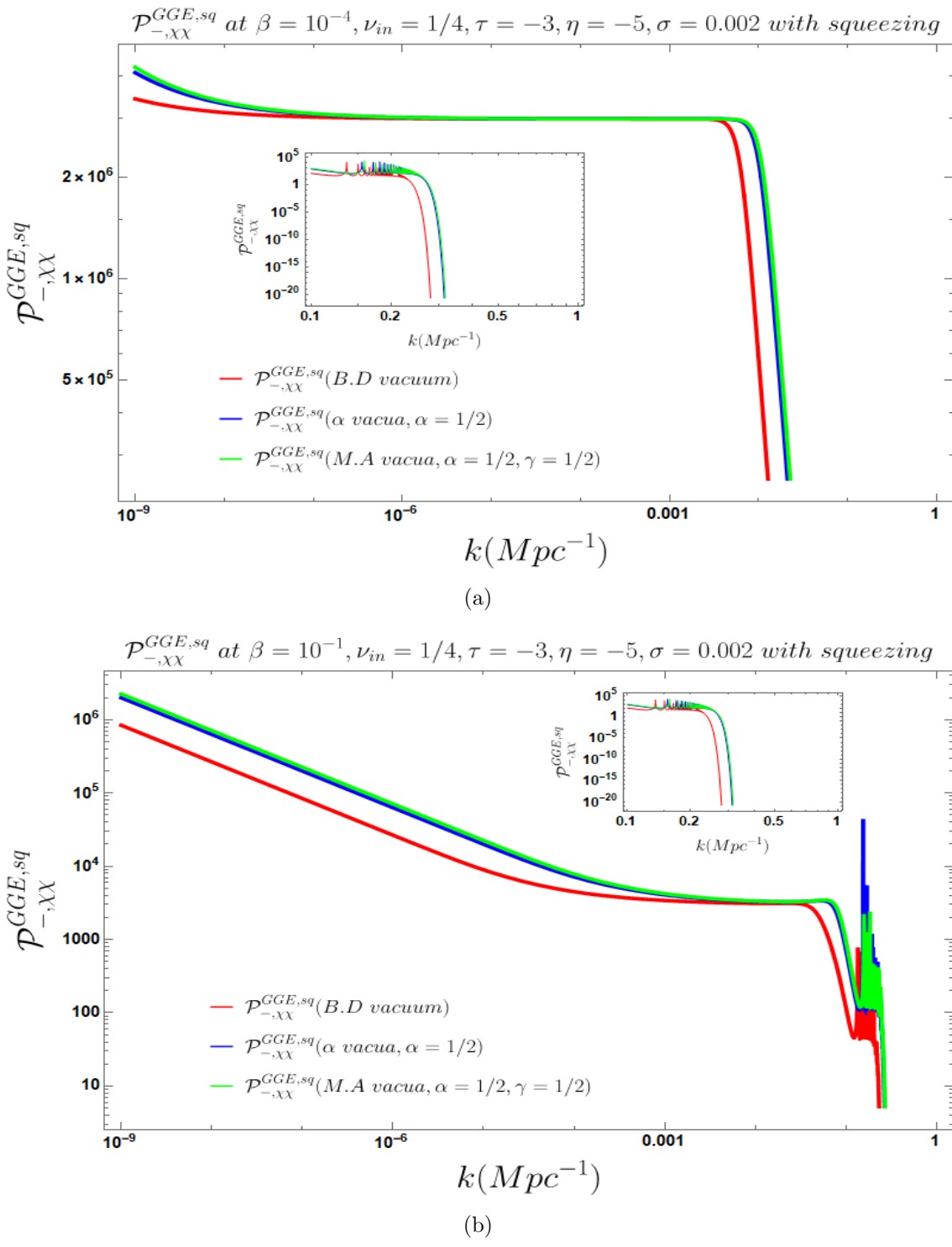

**Figure 4.17**: Behavior of the power spectrum corresponding to the retarded part of the correlator $G^{GGE}_{\Pi_\chi\Pi_\chi}$ with respect to the comoving wave number/scale $k$ at higher and lower temperatures in presence of squeezing.

- In Fig. 4.17(a), we have plotted the retarded part of the power spectrum corresponding to the Generalised Gibbs ensemble correlator $G^{GGE}_{\chi\chi}$ calculated for states with squeezing at a low value of $\beta$. The behavior of the power spectrum in this case is widely different from the advanced part. For very lower values of $k$ the spectrum shows a slightly decreasing behavior. It remains saturated for a very large range of $k$. However after a certain value of $k$ at a relatively large value, the power spectrum falls abruptly.

- In Fig. 4.17(b), we have plotted the retarded part of the power spectrum corresponding to the Generalised Gibbs ensemble correlator $G^{GGE}_{\chi\chi}$ calculated for states with squeezing at a high value of $\beta$. In this case the initial decreasing behavior occurs for a very large range of $k$. The intermediate saturation region is observed in this case as well. However, the saturation range of $k$ is much smaller in this case. At large values of $k$ some random fluctuations after a decreasing nature is observed.

- In Fig. 4.18(a), we have plotted the advanced part of the power spectrum corresponding to the Generalised Gibbs ensemble correlator $G^{GGE}_{\partial_i\chi\partial_i\chi}$ calculated for states with squeezing at a low value of $\beta$. The behavior of the correlator at low $\beta$ is widely different from the behavior of the corresponding part of the $G^{GGE}_{\chi\chi}$ correlator. We observe that the power spectrum monotonically increases for a very large range of $k$. The curves corresponding to different initial conditions also overlap in this region. However, after a certain characteristic value of $k$, the power spectrum falls a little and then exhibits an oscillatory feature with decreasing amplitude.

- In Fig. 4.18(b), we have plotted the advanced part of the power spectrum corresponding to the Generalised Gibbs ensemble correlator $G^{GGE}_{\partial_i\chi\partial_i\chi}$ calculated for states with squeezing at a high value of $\beta$. The behavior of the power spectrum in the low beta case is however identical to the corresponding part of the $G^{GGE}_{\chi\chi}$ correlator. Even the amplitude of the power spectrum in different ranges of $k$ is similar to the analogous part of the $G^{GGE}_{\chi\chi}$ correlator at low $\beta$.

- In Fig. 4.19(a), we have plotted the retarded part of the power spectrum corresponding to the Generalised Gibbs ensemble correlator $G^{GGE}_{\partial_i\chi\partial_i\chi}$ calculated for states with squeezing at a low value of $\beta$. It is observed that the behavior of the power spectrum is pretty similar to what we observe for the advanced part. The only difference that we observe is in the amplitude of the power spectrum. The amplitude in this case is one order less than the advanced part.

- In Fig. 4.19(b), we have plotted the retarded part of the power spectrum corresponding to the Generalised Gibbs ensemble correlator $G^{GGE}_{\partial_i\chi\partial_i\chi}$ calculated for states with squeezing at a high value of $\beta$. It is observed that the behavior of the power spectrum in this case is pretty similar to what we observe for the low $\beta$ case.

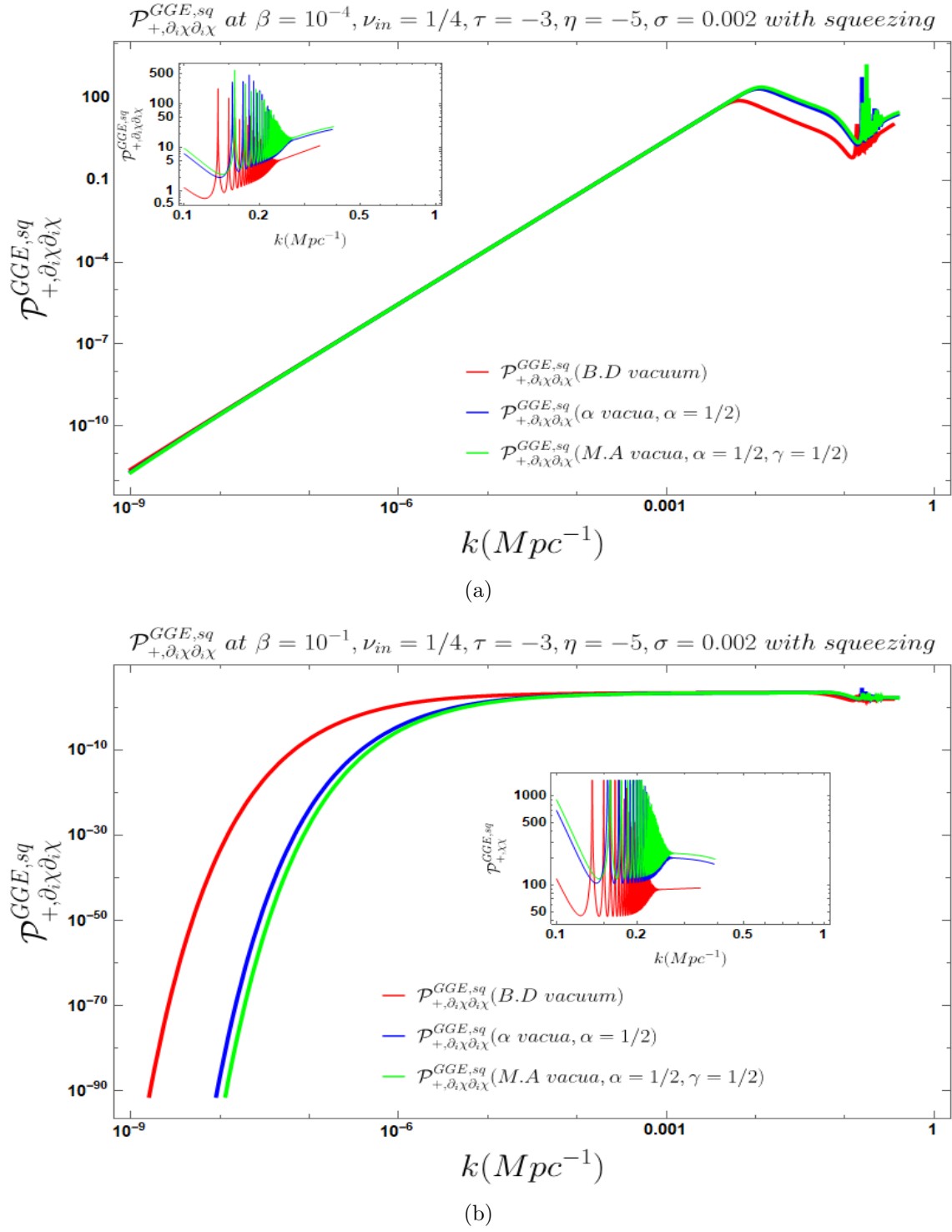

**Figure 4.18**: Behavior of the power spectrum corresponding to the advanced part of the correlator $G_{\partial_i \chi \partial_i \chi}^{GGE}$ with respect to the comoving wave number/scale $k$ at higher and lower temperatures in the presence of squeezing.

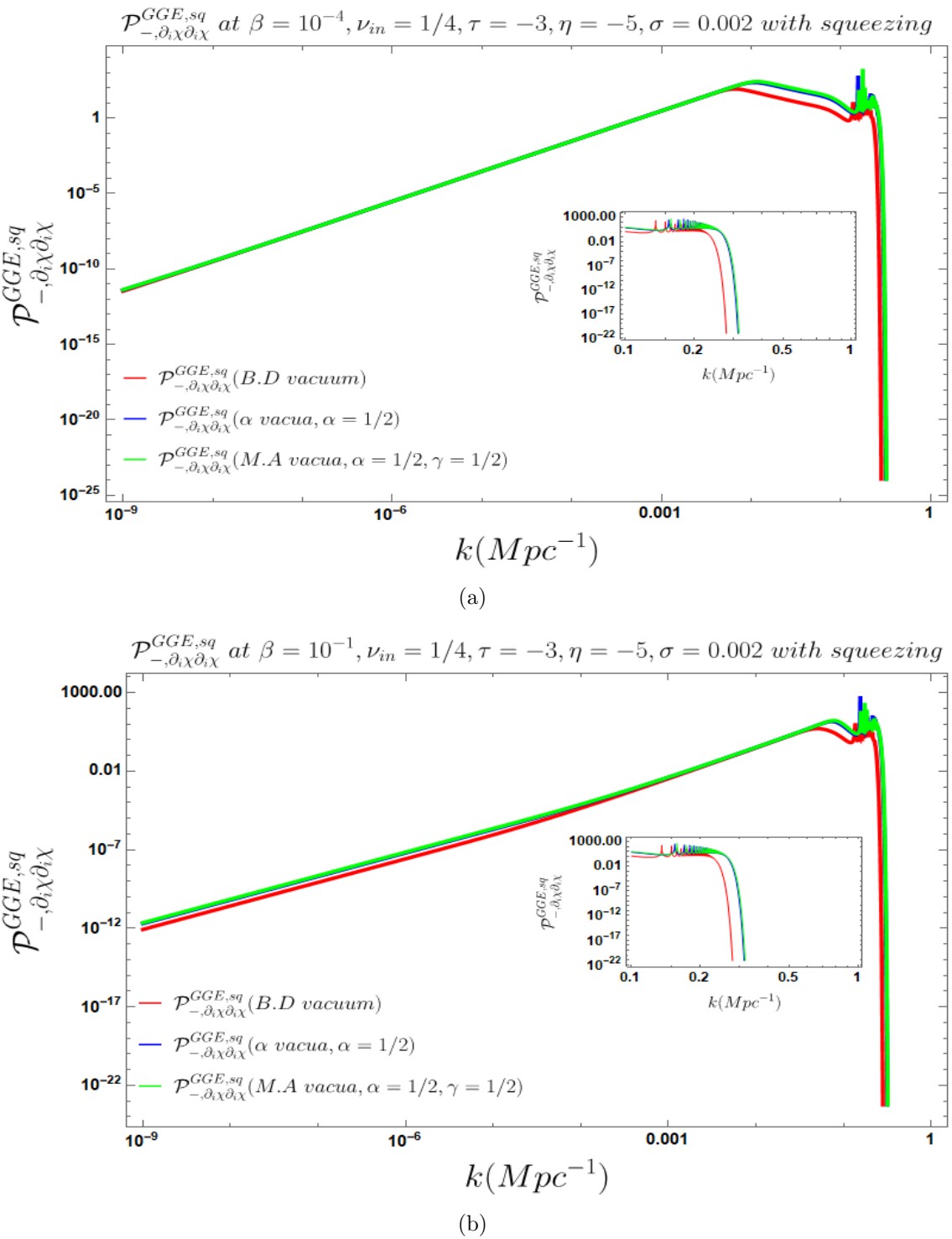

**Figure 4.19**: Behavior of the power spectrum corresponding to the retarded part of the correlator $G^{GGE}_{\partial_i\chi\partial_i\chi}$ with respect to the comoving wave number/scale $k$ at higher and lower temperatures in presence of squeezing.

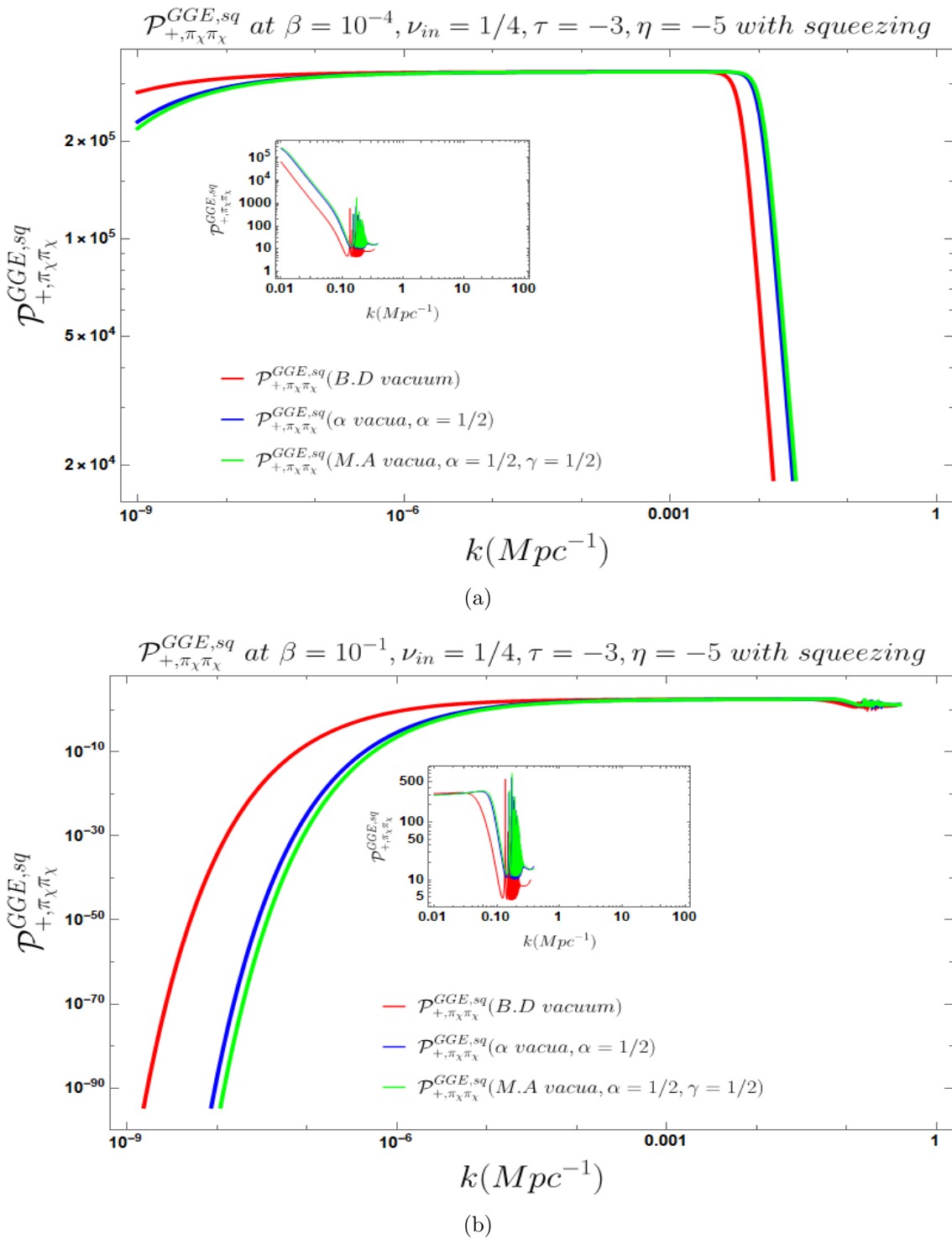

**Figure 4.20**: Behavior of the power spectrum corresponding to the advanced part of the correlator $G^{GGE}_{\Pi_\chi \Pi_\chi}$ with respect to the comoving wave number/scale $k$ at higher and lower temperatures in the presence of squeezing.

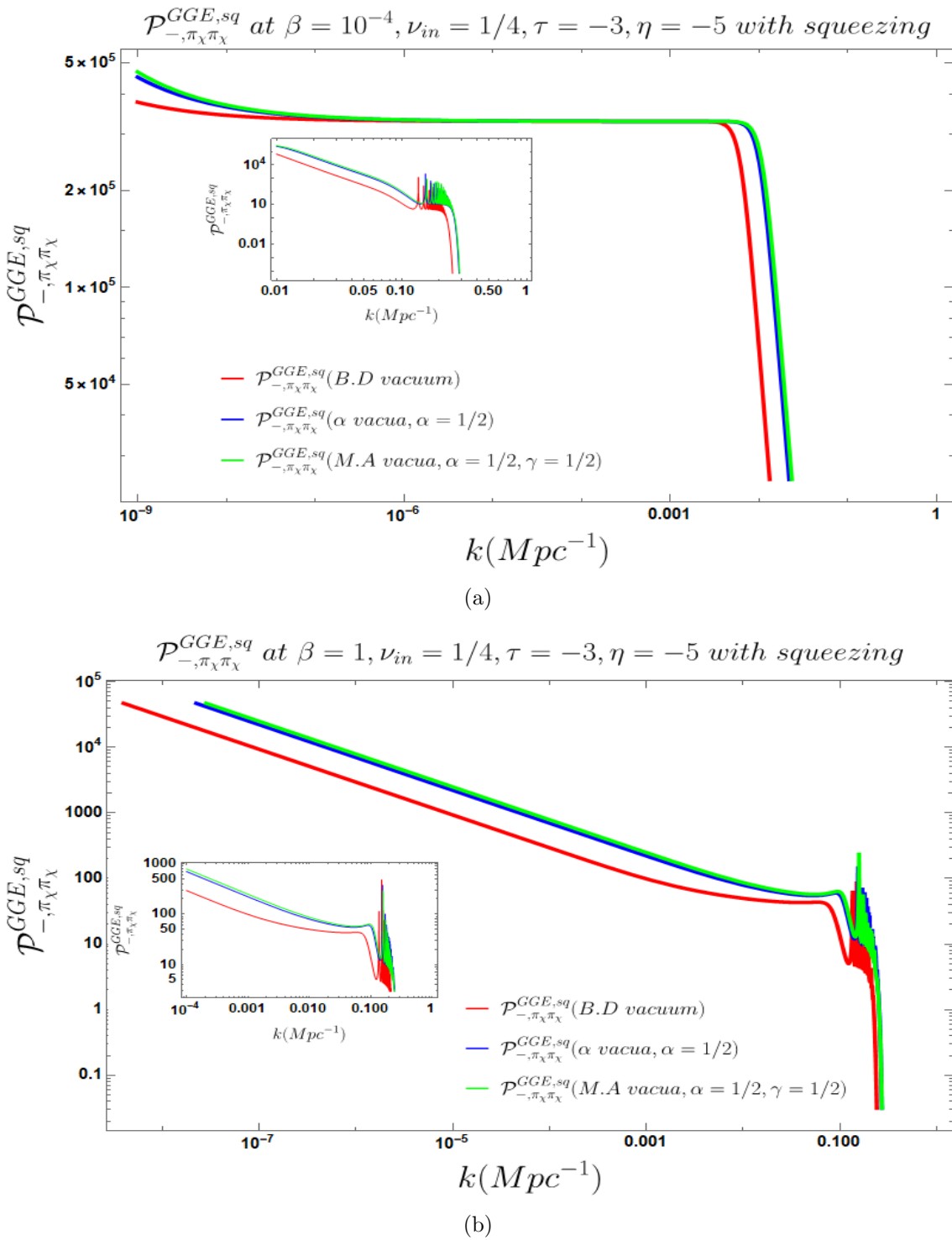

**Figure 4.21**: Behavior of the power spectrum corresponding to the retarded part of the correlator $G^{GGE}_{\Pi_\chi \Pi_\chi}$ with respect to the comoving wave number/scale $k$ at higher and lower temperatures in presence of squeezing.

- In Fig. 4.20(a), we have plotted the advanced part of the power spectrum corresponding to the Generalised Gibbs ensemble correlator $G_{\Pi_\chi \Pi_\chi}^{GGE}$ calculated for states with squeezing at a low value of $\beta$. The behavior of the power spectrum of this correlator is exactly identical to the $G_{\chi\chi}^{GGE}$ correlator. This identical nature is observed in the entire range of $k$.

- In Fig. 4.20(b), we have plotted the advanced part of the power spectrum corresponding to the Generalised Gibbs ensemble correlator $G_{\Pi_\chi \Pi_\chi}^{GGE}$ calculated for states with squeezing at a high value of $\beta$. The behavior at high $\beta$ is also identical to the corresponding part of the $G_{\chi\chi}^{GGE}$ correlator. The fast increase for lower values of $k$ followed by saturation in the power spectrum at intermediate and large values of $k$ occur in this case as well.

- In Fig. 4.21(a), we have plotted the retarded part of the power spectrum corresponding to the Generalised Gibbs ensemble correlator $G_{\Pi_\chi \Pi_\chi}^{GGE}$ calculated for states with squeezing at a low value of $\beta$. Again the behavior of the correlator is identical to the corresponding part of the $G_{\chi\chi}^{GGE}$ correlator. In this case too the initial decrease of the power spectrum for a small range of $k$ followed by the saturation for a large range of $k$ is observed. Moreover, the sudden fall that was observed after a certain value of $k$ for $G_{\chi\chi}^{GGE}$ correlator is also observed in this case. Though the behavior of the spectrum remains same the overall amplitude is one order small than the $G_{\chi\chi}^{GGE}$ correlator.

- In Fig. 4.21(b), we have retarded the advanced part of the power spectrum corresponding to the Generalised Gibbs ensemble correlator $G_{\Pi_\chi \Pi_\chi}^{GGE}$ calculated for states with squeezing at a high value of $\beta$. The feature shown by the correlator exactly matches the one showed by the corresponding part of the $G_{\chi\chi}^{GGE}$ correlator. The smooth decrease for a large range of $k$ followed by the fluctuations at large $k$'s is observed in this case too. Here also, the amplitude seems to be one order less in magnitude than its corresponding $G_{\chi\chi}^{GGE}$ correlator.

# 5 Conclusions

The concluding remarks of our analysis are as follows:

- We have developed the *curved space generalization* of quantum field theoretic version of the well known *Caldeira-Leggett model* consisting of two interacting scalar fields to describe the phenomena of *Quantum Brownian Motion*. In this construction, we have path integrated one scalar field from the two interacting scalar field theory and have

constructed the Euclidean partition function and the corresponding effective action for one scalar field. In this derivation, all the contributions from the interaction and the free part of the other field will be absorbed in the effective coupling parameter and consequently in the effective mass term of the scalar field in this effective description.

- In this construction, we have treated the gravitational sector classically and the interacting scalar fields quantum mechanically. For this reason during computing the effective action and partition function for one scalar field we have treated gravity as the background. Consequently, the result obtained in this construction is a semi-classical result. However, the path integral over the metric can also be done if we treat this quantum mechanically by following perturbative quantum gravity description. In this paper, we have restricted our analysis in the semi-classical regime and have not studied any quantum gravity description of the presented framework.

- Next we derive the results for conformally flat de Sitter space-time solution and used the phenomena of quantum quench as a special trick to study the two-point quantum correlation functions from the effective scalar field, its spatial derivative and its associated canonically conjugate effective field momentum. Particularly in this context, the phenomena of quantum quench is used to deal with the conformal time-dependent effective mass which we have obtained as an outcome of the previously mentioned semi-classical construction of partition function and the effective action for one scalar field in the de Sitter background geometry.

- We have chosen the sudden quench mass protocol using which we compute the classical solutions of the effective field in the Fourier transformed space, which is identified as the mode functions before and after the quench operation. In the technical description, the solutions obtained before and after quench are known as the incoming and outgoing modes. This further enabled us to compute the expressions for the two Bogoliubov coefficients which actually connect the solutions before and after the point of quench operation. However, it is important to note that the present computational methodology can be implemented for other time-dependent effective mass protocols and depending on the specific profile one can expect to get different types of solutions for the incoming modes, outgoing modes and for the two Bogoliubov coefficients which allow expressing one solution in terms of the other.

- From our study we found that irrespective of the initial starting state before the point of quench, the state of the system could be written in terms of some conserved charges of $W_\infty$ algebra, i.e., in the gCC form. This obtained result further implies that in the late time scale the subsystem that we are considering thermalizes. The above fact was true even if one doesn't take the ground state of the initial Hamiltonian as the starting state. Most significantly, the results that we have established for the thermalization within the framework of de Sitter background geometry was not

explicitly studied before. Also these obtained results can be further extended to study various early universe cosmological phenomena, particle production, reheating, etc., where it is needed to thermalize a system from out-of-equilibrium.

- We found that the conserved charges of $W_\infty$ algebra describing the gCC state post quench was dependent on the choice of the quantum initial conditions for de Sitter background.

- Additionally, we have studied the consequences within the context of a thermal GGE ensemble where we found that the results for the two-point quantum correlations are explicitly dependent on the factor $\beta$, which is the inverse equilibrium temperature of the GGE ensemble after thermalization. This is another evidence of the system attaining thermalization at the late time scale.

- We also extend the computation for finding the two-point quantum correlation functions from a Gaussian squeezed state and for a squeezed gCC state in this paper and found that the results are different from the results obtained without squeezing.

- We verify that an assumption of a non-Gaussian squeezed state as the starting wave function does not give any significant difference in the conserved charges of $W_\infty$ algebra and hence the structure of the gCC state describing the post-quench phase is almost identical with the gCC state obtained by assuming Gaussian squeezed state. This is nicely consistent with ref. [93], in which the author found that the non-Gaussian perturbations of the most dangerous type are practically absent.

The future prospects of the present work are as follows:

- The present work has been done by considering a specific instantaneous quench protocol. One can extend the present analysis by considering various other quench protocols in curved space-time.

- Another extension of the present work would be to try and consider non-quadratic interactions between the two scalar fields. Though an exact approach may not be possible in that case, but one can always resort to perturbative approaches while dealing with such non-quadratic interaction terms.

- A similar kind of study can be done by taking fermionic fields in the background of De-Sitter space instead of the scalar ones and we intend to do it in upcoming days.

- As already clear from the present analysis, the introduction of the curved background plays a tremendous role in constructing the gCC states for the post quench phase. One can extend the current work not only for different quench profiles but for different background space-times, probably in AdS space also.

- The system considered in this paper is a highly realistic one and can be a very useful model of many physical systems. One can thus think of studying chaos by computing OTOC's [79, 94–96] and circuit complexity [97–104] for such systems. These have attracted significant interest in recent times.

**Acknowledgements**

The research fellowship of SC is supported by the J. C. Bose National Fellowship of Sudhakar Panda. SC also would line to thank School of Physical Sciences, National Institute for Science Education and Research (NISER), Bhubaneswar for providing the work friendly environment. SC also thank all the members of our newly formed virtual international non-profit consortium Quantum Structures of the Space-Time & Matter (QASTM) for elaborative discussions. Satyaki Chowdhury and K. Shirish would like to thank NISER Bhubaneswar and VNIT Nagpur respectively, for providing fellowships. JK thanks Tung Tran for enlightening discussions. JK is a member of the Gravity, Quantum Fields and Information group at AEI, which is generously supported by the Alexander von Humboldt Foundation and the Federal Ministry for Education and Research through the Sofja Kovalevskaja Award. JK is partially supported by the International Max Planck Research School for Mathematical and Physical Aspects of Gravitation, Cosmology and Quantum Field Theory. The work of JK is supported in part by a fellowship from the Studienstiftung des deutschen Volkes (German Academic Scholarship Foundation). SP acknowledges the J. C. Bose National Fellowship for support of his research. Last but not least, we would like to acknowledge our debt to the people belonging to the various part of the world for their generous and steady support for research in natural sciences.

# A    Charges of $W_\infty$ algebra for different quantum initial conditions

## A.1    Expression for the coefficients of $\gamma(k)$

The specific choices for quantum initial conditions are fixed by choosing the following set of constants appearing in the incoming solution:

$$\textcolor{red}{\textbf{Bunch – Davies vacuum}}: \qquad d_1 = 1, \qquad\qquad d_2 = 0, \tag{A.1}$$
$$\textcolor{red}{\boldsymbol{\alpha}\ \textbf{vacua}}: \qquad\qquad\qquad d_1 = \cosh\alpha, \qquad d_2 = \sinh\alpha, \tag{A.2}$$
$$\textcolor{red}{\textbf{Motta – Allen vacua}}: \qquad\ d_1 = \cosh\alpha, \qquad d_2 = \exp(i\gamma)\sinh\alpha. \tag{A.3}$$

The various non-vanishing coefficients of $\gamma(k)$ can be computed as:

$$\gamma_0 = -\frac{id_1 + d_2 \exp(i\pi\nu_{in})}{id_2^* + d_1^* \exp(i\pi\nu_{in})}, \tag{A.4}$$

$$\gamma_4 = -\frac{2(d_1 d_1^* - d_2 d_2^*)\exp(i\pi\nu_{in})\eta^3(5 + 2\nu_{in})}{3((id_2^* + d_1^* \exp(i\pi\nu_{in}))^2(-1 + 2\nu_{in}))}, \tag{A.5}$$

$$\gamma_6 = \frac{2(d_1 d_1^* - d_2 d_2^*)\exp(i\pi\nu_{in})\eta^5(-29 + 4\nu_{in}(4 + \nu_{in}))}{5((id_2^* + d_1^* \exp(i\pi\nu_{in}))^2(1 - 2\nu_{in})^2)}. \tag{A.6}$$

In the next three subsections we mention the results for the above mentioned three different choices of the quantum initial conditions.

### A.1.1    Expressions for the Bunch Davies vacuum

For Bunch Davies vacuum we have the following results:

$$\gamma_0 = \exp\left(-i\pi\left(\nu_{in} + \frac{1}{2}\right)\right), \tag{A.7}$$

$$\gamma_4 = -\frac{2}{3}\exp(-i\pi\nu_{in})\eta^3\left(\frac{5 + 2\nu_{in}}{-1 + 2\nu_{in}}\right), \tag{A.8}$$

$$\gamma_6 = \frac{2}{5}\exp(-i\pi\nu_{in})\eta^5\left(\frac{-29 + 4\nu_{in}(4 + \nu_{in})}{(1 - 2\nu_{in})^2}\right). \tag{A.9}$$

### A.1.2    Expressions for the $\alpha$ vacua

For $\alpha$ vacua we have the following results:

$$\gamma_0 = \frac{\exp(i\pi\nu_{in})\sinh\alpha - i\cosh\alpha}{\exp(i\pi\nu_{in})\cosh\alpha + i\sinh\alpha}, \tag{A.10}$$

$$\gamma_4 = -\frac{2}{3}\frac{\exp(i\pi\nu_{in})\eta^3(5 + 2\nu_{in})}{(\cosh\alpha \exp(i\pi\nu_{in}) + i\sinh\alpha)^2(-1 + 2\nu_{in})}, \tag{A.11}$$

$$\gamma_6 = \frac{2}{5}\frac{\exp(i\pi\nu_{in})\eta^5(-29 + 4\nu_{in}(4 + \nu_{in}))}{(\cosh\alpha \exp(i\pi\nu_{in}) + i\sinh\alpha)^2(1 - 2\nu_{in})^2}. \tag{A.12}$$

### A.1.3 Expressions for the Mota-Allen vacua

For Mota-Allen vacua we have the following results:

$$\gamma_0 = \frac{\exp(i(\gamma + \pi\nu_{in}))\sinh\alpha - i\cosh\alpha}{\exp(i\pi\nu_{in})\cosh\alpha + i\exp(-i\gamma)\sinh\alpha}, \tag{A.13}$$

$$\gamma_4 = -\frac{2}{3}\frac{\exp(i\pi\nu_{in})\eta^3(5 + 2\nu_{in})}{(\exp(i\pi\nu_{in})\cosh\alpha + i\exp(-i\gamma)\sinh\alpha)^2(-1 + 2\nu_{in})}, \tag{A.14}$$

$$\gamma_6 = \frac{2}{5}\frac{\exp(i\pi\nu_{in})\eta^5(-29 + 4\nu_{in}(4 + \nu_{in}))}{(\exp(i\pi\nu_{in})\cosh\alpha + i\exp(-i\gamma)\sinh\alpha)^2(1 - 2\nu_{in})^2}. \tag{A.15}$$

### A.2 Expression for the coefficients of $\kappa(k)$ for ground state

The non-vanishing coefficients of the $\kappa(k)$ expansion for arbitrary quantum initial conditions, which are representing the non-vanishing charges of the $W_\infty$ algebra can be calculated for Dirichlet and Neumann boundary state as:

$$\kappa_{0,\mathbf{DB}} = -\frac{1}{2}\log\left[\frac{d_1 - id_2\exp(i\pi\nu_{in})}{d_2^* - id_1^*\exp(i\pi\nu_{in})}\right], \tag{A.16}$$

$$\kappa_{0,\mathbf{NB}} = -\frac{1}{2}\left\{\log\left[\frac{d_1 - id_2\exp(i\pi\nu_{in})}{d_2^* - id_1^*\exp(i\pi\nu_{in})}\right] + i\pi\right\}, \tag{A.17}$$

$$\kappa_{4,\mathbf{DB}} = \kappa_{4,\mathbf{NB}} = -\frac{(d_1 d_1^* - d_2 d_2^*)\exp(i\pi\nu_{in}))\eta^3(5 + 2\nu_{in})}{3(d_2^* - id_1^*\exp(i\pi\nu_{in}))(d_1 - id_2\exp(i\pi\nu_{in}))(-1 + 2\nu_{in}))}, \tag{A.18}$$

$$\kappa_{6,\mathbf{DB}} = \kappa_{6,\mathbf{NB}} = \frac{(d_1 d_1^* - d_2 d_2^*)\exp(i\pi\nu_{in})\eta^5(-29 + 4\nu_{in}(4 + \nu_{in}))}{5(id_2^* + d_1^*\exp(i\pi\nu_{in})(id_1 + d_2\exp(i\pi\nu_{in}))(1 - 2\nu_{in})^2)}. \tag{A.19}$$

$$\kappa_{7,\mathbf{DB}} = \kappa_{7,\mathbf{NB}} = \frac{1}{9(1 - 2\nu_{in})^2(d_1 - id_2\exp(i\pi\nu_{in}))^2(d_2^* - id_1^*\exp(i\pi\nu_{in}))^2}$$

$$\left[\eta^6\exp(i\pi\nu_{in})(d_1 d_1^* - d_2 d_2^*)\left(-72\exp(i\pi\nu_{in})(d_1 d_1^* - d_2 d_2^*)\right.\right.$$

$$\left.\left. + i(d_1 d_2^* + d_1^* d_2\exp(2i\pi\nu_{in})(4\nu_{in}(\nu_{in} + 5) - 47)\right)\right] \tag{A.20}$$

$$\kappa_{8,\mathbf{DB}} = \kappa_{8,\mathbf{NB}} = -\frac{(d_1 d_1^* - d_2 d_2^*)\eta^7\exp(i\pi\nu_{in})(2\nu_{in}(4\nu_{in}^2 + 22\nu_{in} - 81) + 125)}{7(2\nu_{in} - 1)^3(id_1 + d_2\exp(i\pi\nu_{in}))(d_1^*\exp(i\pi\nu_{in}) + id_2^*)} \tag{A.21}$$

$$\kappa_{9,\mathbf{DB}} = \kappa_{9,\mathbf{NB}} = \frac{1}{15(2\nu_{in} - 1)^3(id_1 + d_2\exp(i\pi\nu_{in}))^2(d_1^*\exp(i\pi\nu_{in}) + id_2^*)^2}$$

$$\left[2(d_1 d_1^* - d_2 d_2^*)\eta^8\exp(i\pi\nu_{in})\left(120\exp(i\pi\nu_{in})(2\nu_{in} - 3)(d_1 d_1^* - d_2 d_2^*) - id_1 d_2^*(8\nu_{in}^3\right.\right.$$

$$\left.\left. + 52\nu_{in}^2 - 218\nu_{in} + 215) - id_1^* d_2\exp(2i\pi\nu_{in})(8\nu_{in}^3 + 52\nu_{in}^2 - 218\nu_{in} + 215)\right)\right] \tag{A.22}$$

In the next three subsections we mention the results for the previously mentioned three different choices of the quantum initial conditions. Here we are computing the expressions

for the Dirichlet boundary states from which one can also derive the expressions for the Neumann boundary states using the above mentioned connecting relationships. For computational simplicity we will further drop the superscript **DB** in the further computations.

### A.2.1 Expressions for the Bunch Davies vacuum

For Bunch Davies vacuum we have the following results:

$$\kappa_0 = -\frac{i\pi}{2}\left(\frac{1}{2} - \nu_{in}\right) \tag{A.23}$$

$$\kappa_4 = -\frac{i}{3}\eta^3\left(\frac{5 + 2\nu_{in}}{-1 + 2\nu_{in}}\right) \tag{A.24}$$

$$\kappa_6 = -\frac{i}{5}\eta^5\left(\frac{-29 + 4\nu_{in}(4 + \nu_{in})}{(1 - 2\nu_{in})^2}\right) \tag{A.25}$$

$$\kappa_7 = \frac{8\eta^6}{(1 - 2\nu_{in})^2} \tag{A.26}$$

$$\kappa_8 = \frac{i}{7}\eta^7\left(\frac{(2\nu_{in}\left(4\nu_{in}^2 + 22\nu_{in} - 81\right) + 125)}{(2\nu_{in} - 1)^3}\right) \tag{A.27}$$

$$\kappa_9 = \frac{16\eta^8(3 - 2\nu_{in})}{(2\nu_{in} - 1)^3} \tag{A.28}$$

### A.2.2 Expressions for the $\alpha$ vacua

For $\alpha$ vacua we have the following results:

$$\kappa_0 = -\frac{1}{2}\log\left(\frac{\exp(i\pi\nu_{in})\sinh(\alpha) + i\cosh(\alpha)}{\exp(i\pi\nu_{in})\cosh(\alpha) + i\sinh(\alpha)}\right), \tag{A.29}$$

$$\kappa_4 = -\frac{1}{3}\frac{\exp(i\pi\nu_{in})\eta^3(5 + 2\nu_{in})}{(\sinh\alpha - i\cosh\alpha\exp(i\pi\nu_{in}))(\cosh\alpha - i\sinh\alpha\exp(i\pi\nu_{in}))(-1 + 2\nu_{in})}, \tag{A.30}$$

$$\kappa_6 = \frac{1}{5}\frac{\exp(i\pi\nu_{in})\eta^5(-29 + 4\nu_{in}(4 + \nu_{in}))}{(i\sinh\alpha + \cosh\alpha\exp(i\pi\nu_{in}))(i\cosh\alpha + \sinh\alpha\exp(i\pi\nu_{in}))(1 - 2\nu_{in})^2}, \tag{A.31}$$

$$\kappa_7 = \frac{\exp(i\pi\nu_{in})\eta^6\left(i\left(1 + \exp(2i\pi\nu_{in})\right)\left(4\nu_{in}(\nu_{in} + 5) - 47\right)\sinh(2\alpha) - 144\exp(i\pi\nu_{in})\right)}{18(1 - 2\nu_{in})^2\left(\exp(i\pi\nu_{in})\cosh(\alpha) + i\sinh(\alpha)\right)^2\left(\exp(i\pi\nu_{in})\sinh(\alpha) + i\cosh(\alpha)\right)^2} \tag{A.32}$$

$$\kappa_8 = -\frac{\eta^7\exp(i\pi\nu_{in})\left(8\nu_{in}^3 + 44\nu_{in}^2 - 162\nu_{in} + 125\right)}{7(2\nu_{in} - 1)^3\left(\exp(i\pi\nu_{in})\cosh(\alpha) + i\sinh(\alpha)\right)\left(\exp(i\pi\nu_{in})\sinh(\alpha) + i\cosh(\alpha)\right)}, \tag{A.33}$$

$$\kappa_9 = \frac{1}{15(2\nu_{in} - 1)^3(\sinh(2\alpha)\sin(\pi\nu_{in}) + \cosh(2\alpha))^2}$$
$$\phantom{'}\times\left[\eta^8\exp(-i\pi\nu_{in})\left(i\left(1 + \exp(2i\pi\nu_{in})\right)\left(8\nu_{in}^3 + 52\nu_{in}^2 - 218\nu_{in} + 215\right)\sinh(2\alpha)\right.\right.$$
$$\phantom{'}\left.\left. - 240\exp(i\pi\nu)(2\nu - 3)\right)\right]. \tag{A.34}$$

## A.2.3 Expressions for the Mota-Allen vacua

For Mota-Allen vacua we have the following results:

$$\kappa_0 = -\frac{1}{2}\log\left(\frac{\cosh\alpha - i\exp(i(\pi\nu_{in}+\gamma))\sinh\alpha}{\exp(-i\gamma)\sinh\alpha - i\exp(i\pi\nu_{in})\cosh\alpha}\right), \tag{A.35}$$

$$\kappa_4 = \frac{1}{3}\frac{\exp(i\pi\nu_{in})\eta^3(5+2\nu_{in})}{(\exp(-i\gamma)\sinh\alpha - i\cosh\alpha\exp(i\pi\nu_{in}))(\cosh\alpha - i\sinh\alpha\exp(i(\pi\nu_{in}+\gamma)))(-1+2\nu_{in})}, \tag{A.36}$$

$$\kappa_6 = \frac{1}{5}\frac{\exp(i\pi\nu_{in})\eta^5(-29+4\nu_{in}(4+\nu_{in}))}{(i\exp(-i\gamma))\sinh\alpha + \cosh\alpha\exp(i\pi\nu_{in}))(i\cosh\alpha + \sinh\alpha\exp(i(\pi\nu_{in}+\gamma)))(1-2\nu_{in})^2}, \tag{A.37}$$

$$\kappa_7 = \frac{1}{18(1-2\nu)^2\left(\cosh(\alpha)\exp(i(\gamma+\pi\nu_{in}))+i\sinh(\alpha)\right)^2\left(\sinh(\alpha)\exp(i(\gamma+\pi\nu_{in}))+i\cosh(\alpha)\right)^2},$$
$$\times\left[\exp(i(\gamma+\pi\nu_{in}))\eta^6\left(i(4\nu(\nu_{in}+5)-47)\sinh(2\alpha)\left(1+\exp(2i(\gamma+\pi\nu_{in}))\right)\right)\right.$$
$$\left.-144\exp(i(\gamma+\pi\nu_{in}))\right] \tag{A.38}$$

$$\kappa_8 = \frac{-\exp(i(\gamma+\pi\nu_{in}))\eta^7\left(2\nu_{in}(4\nu_{in}^2+22\nu_{in}-81)+125\right)}{7(2\nu_{in}-1)^3\left(\cosh(\alpha)\exp(i(\gamma+\pi\nu_{in}))+i\sinh(\alpha)\right)\left(\sinh(\alpha)\exp(i(\gamma+\pi\nu_{in}))+i\cosh(\alpha)\right)}, \tag{A.39}$$

$$\kappa_9 = \frac{1}{15(2\nu_{in}-1)^3\left(\cosh(\alpha)\exp(i(\gamma+\pi\nu_{in}))+i\sinh(\alpha)\right)^2\left(\sinh(\alpha)\exp(i(\gamma+\pi\nu_{in}))+i\cosh(\alpha)\right)^2}$$
$$\times\left[\eta^8\exp(i(\gamma+\pi\nu_{in}))(240(2\nu_{in}-3)\exp(i(\gamma+\pi\nu_{in}))\right.$$
$$\left.-i(8\nu_{in}^3+52\nu_{in}^2-218\nu_{in}+215)\sinh(2\alpha)(1+\exp(2i(\gamma+\pi\nu_{in}))))\right]. \tag{A.40}$$

## A.3 Expression for the coefficients of $\kappa(k)$ for squeezed states

Doing a series expansion of $\kappa_{eff}(k)$, for the specific choice of Gaussian $f(k)$, it can be very easily verified that the non-vanishing expansion coefficients for the Dirichlet and Neumann boundary states can be written in the following form:

$$\kappa_{0,\mathbf{DB}}^{\text{eff}} = -\frac{1}{2}\log\left[\frac{d_1 - id_2\exp(i\pi\nu_{in})}{d_2^* - id_1^*\exp(i\pi\nu_{in})}\right] \tag{A.41}$$

$$\kappa_{0,\mathbf{NB}}^{\text{eff}} = -\frac{1}{2}\left\{\log\left[\frac{d_1 - id_2\exp(i\pi\nu_{in})}{d_2^* - id_1^*\exp(i\pi\nu_{in})}\right]+i\pi\right\} \tag{A.42}$$

$$\kappa_{4,\mathbf{DB}}^{\text{eff}} = \kappa_{4,\mathbf{NB}}^{\text{eff}} = -\frac{(d_1d_1^* - d_2d_2^*)\exp(i\pi\nu_{in}))\eta^3(5+2\nu_{in})}{3(d_2^* - id_1^*\exp(i\pi\nu_{in}))(d_1 - id_2\exp(i\pi\nu_{in}))(-1+2\nu_{in}))} \tag{A.43}$$

$$\kappa_{6,\mathbf{DB}}^{\text{eff}} = \kappa_{6,\mathbf{NB}}^{\text{eff}} = \frac{(d_1d_1^* - d_2d_2^*)\exp(i\pi\nu_{in})\eta^5(-29+4\nu_{in}(4+\nu_{in}))}{5(id_2^* + d_1^*\exp(i\pi\nu_{in}))(id_1 + d_2\exp(i\pi\nu_{in}))(1-2\nu_{in})^2)} \tag{A.44}$$

$$\kappa_{7,\mathbf{DB}}^{\text{eff}} = \kappa_{7,\mathbf{NB}}^{\text{eff}}$$
$$= \frac{1}{9(d_2^* - id_1^*\exp(i\pi\nu_{in}))^2(id_1 + d_2\exp(i\pi\nu_{in}))^2(d_1 + d_2^* - i(d_1^* + d_2)\exp(i\pi\nu_{in}))(1-2\nu_{in})^2}$$

$$
\left[\exp(i\pi\nu_{in})\eta^6\left(id_1d_2^*(d_1+d_2^*)(-d_1d_1^*+d_2d_2^*)(-47+4\nu_{in}(5+\nu_{in}))+d_1^*d_2(d_1^*+d_2)\right.\right.
$$

$$
(-d_1d_1^*+d_2d_2^*))\exp(3i\pi\nu_{in})(-47+4\nu_{in}(5+\nu_{in}))+\left((d_1d_1^*-d_2d_2^*)(72d_1^2d_1^*-d_1d_2^*(2\nu_{in}+5)^2\right.
$$

$$
\left.(d_1^*+d_2)+72d_2d_2^{*2}\right)+d_1^*d_2\exp(3i\pi\nu_{in})(4\nu_{in}(\nu_{in}+5)-47)(d_1^*+d_2)(d_2d_2^*-d1d_1^*)
$$

$$
+id_1d_2^*(4\nu_{in}(\nu_{in}+5)-47)(d_1+d_2^*)(d_2d_2^*-d_1d_1^*)+ie^{2i\pi\nu_{in}}(d_1d_1^*-d_2d_2^*)
$$

$$
\left.\left.\left(d_1d_1^*(72d_1^*-d_2(2\nu_{in}+5)^2+d_2d_2^*(72d_2-d_1^*(2\nu_{in}+5)^2)\right)\right)\right]
\tag{A.45}
$$

$$
\kappa_{8,\mathbf{DB}}^{\mathrm{eff}}=\kappa_{8,\mathbf{NB}}^{\mathrm{eff}}-\frac{(d_1d_1^*-d_2d_2^*)\exp(i\pi\nu_{in})\eta^7(125+2\nu_{in}(-81+22\nu_{in}+4\nu_{in}^2))}{7(id_2^*+d_1^*\exp(i\pi\nu_{in}))(id_1+d_2\exp(i\pi\nu_{in}))(-1+2\nu_{in})^3}
$$

$$
\kappa_{9,\mathbf{DB}}^{\mathrm{eff}}=\kappa_{9,\mathbf{NB}}^{\mathrm{eff}}
$$

$$
=\frac{1}{15(2\nu_{in}-1)^3\sigma^2\left(id_1+d_2e^{i\pi\nu_{in}}\right)^2\left(d_2^*-id_1^*e^{i\pi\nu_{in}}\right)^2\left(d_1-ie^{i\pi\nu_{in}}(d_1^*+d_2)+d_2^*\right)^2}
$$

$$
\times\left[2\eta^6e^{i\pi\nu_{in}}(d_1d_1^*-d_2d_2^*)\left(d_1^3\eta^2\sigma^2\left(id_2^*\left(8\nu_{in}^3+52\nu_{in}^2-218\nu_{in}+215\right)-120d_1^*e^{i\pi\nu_{in}}(2\nu_{in}-3)\right)\right.\right.
$$

$$
+d_1^2\left(2d_2^*e^{i\pi\nu_{in}}\left(\eta^2\sigma^2\left(8\nu_{in}^3(d_1^*+d_2)+52\nu_{in}^2(d_1^*+d_2)-2\nu_{in}(109d_1^*+49d_2)+215d_1^*+35d_2\right)\right.\right.
$$

$$
\left.+30d_1^*(2\nu_{in}-1)\right)-id_1^*e^{2i\pi\nu_{in}}\left(60d_1^*(2\nu_{in}-1)-d_2\eta^2\left(8\nu_{in}^3+52\nu_{in}^2+262\nu_{in}-505\right)\sigma^2\right)
$$

$$
\left.+2id_2^{*2}\eta^2\left(8\nu_{in}^3+52\nu_{in}^2-218\nu_{in}+215\right)\sigma^2\right)+d_1\left(2d_2e^{i\pi\nu_{in}}\left(d_2^{*2}+d_1^{*2}e^{2i\pi\nu_{in}}\right)\right.
$$

$$
\left(\eta^2\left(8\nu_{in}^3+52\nu_{in}^2-218\nu_{in}+215\right)\sigma^2-60\nu_{in}+30\right)+d_2^2\eta^2e^{2i\pi\nu_{in}}\sigma^2\left(2d_1^*e^{i\pi\nu_{in}}\right.
$$

$$
\left.\left.\left(8\nu_{in}^3+52\nu_{in}^2-98\nu_{in}+35\right)-id_2^*\left(8\nu_{in}^3+52\nu_{in}^2+262\nu_{in}-505\right)\right)\right)+\eta^2\sigma^2\left(d_2^*-id_1^*e^{i\pi\nu_{in}}\right)^2
$$

$$
\left.\left(120d_1^*e^{i\pi\nu_{in}}(2\nu_{in}-3)+id_2^*\left(8\nu_{in}^3+52\nu_{in}^2-218\nu_{in}+215\right)\right)\right)
$$

$$
+d_2e^{i\pi\nu_{in}}\left(id_2^{*2}e^{i\pi\nu}\left(d_1^*\eta^2\left(8\nu_{in}^3+52\nu_{in}^2+262\nu_{in}-505\right)\sigma^2+60d_2(2\nu_{in}-1)\right)\right.
$$

$$
+2d_2^*e^{2i\pi\nu_{in}}\left(\eta^2\sigma^2(d_1^*+d_2)\left(d_1^*\left(8\nu_{in}^3+52\nu_{in}^2-98\nu_{in}+35\right)+60d_2(3-2\nu_{in})\right)\right.
$$

$$
\left.+30d_1^*d_2(2\nu_{in}-1)\right)-id_1^*\eta^2e^{3i\pi\nu_{in}}\left(8\nu_{in}^3+52\nu_{in}^2-218\nu_{in}+215\right)\sigma^2(d_1^*+d_2)^2
$$

$$
\left.\left.+120d_2^{*3}\eta^2(3-2\nu_{in})\sigma^2\right)\right)\right]
\tag{A.46}
$$

In the next three subsections we mention the results for the previously mentioned three different choices of the quantum initial conditions. Here we are computing the expressions for the Dirichlet boundary states from which one can also derive the expressions for the Neumann boundary states using the above mentioned connecting relationships. For computational simplicity we will further drop the superscript **DB** in the further computations.

### A.3.1    Expressions for the Bunch Davies vacuum

For Bunch Davies vacuum we have the following results:

$$\kappa_0^{\text{eff}} = \kappa_0 = -\frac{i\pi}{2}\left(\frac{1}{2} - \nu\right) \tag{A.47}$$

$$\kappa_4^{\text{eff}} = \kappa_4 = -\frac{i}{3}\eta^3\left(\frac{5 + 2\nu}{-1 + 2\nu}\right) \tag{A.48}$$

$$\kappa_6^{\text{eff}} = \kappa_6 = -\frac{i}{5}\eta^5\left(\frac{-29 + 4\nu(4 + \nu)}{(1 - 2\nu)^2}\right) \tag{A.49}$$

$$\kappa_7^{\text{eff}} \neq \kappa_7 = -\frac{8\eta^6\left(\exp(i\pi\nu) - i\right)}{\left(\exp(i\pi\nu) + i\right)(1 - 2\nu)^2} \tag{A.50}$$

$$\kappa_8^{\text{eff}} = \kappa_8 = \frac{i}{7}\eta^7\left(\frac{\left(2\nu\left(4\nu^2 + 22\nu - 81\right) + 125\right)}{(2\nu - 1)^3}\right) \tag{A.51}$$

$$\kappa_9^{\text{eff}} \neq \kappa_9 = \frac{8\eta^6\exp(i\pi\nu)\left(4\eta^2(2\nu - 3)\sigma^2\cos(\pi\nu) + i(2\nu - 1)\right)}{\left(\exp(i\pi\nu) + i\right)^2(2\nu - 1)^3\sigma^2} \tag{A.52}$$

### A.3.2    Expressions for the $\alpha$ vacua

For $\alpha$ vacua we have the following results:

$$\kappa_0^{\text{eff}} = \kappa_0 = -\frac{1}{2}\log\left(\frac{\exp(i\pi\nu_{in})\sinh(\alpha) + i\cosh(\alpha)}{\exp(i\pi\nu_{in})\cosh(\alpha) + i\sinh(\alpha)}\right), \tag{A.53}$$

$$\kappa_4^{\text{eff}} = \kappa_4 = -\frac{1}{3}\frac{\exp(i\pi\nu_{in})\eta^3(5 + 2\nu_{in})}{(\sinh\alpha - i\cosh\alpha\exp(i\pi\nu_{in}))(\cosh\alpha - i\sinh\alpha\exp(i\pi\nu_{in}))(-1 + 2\nu)}, \tag{A.54}$$

$$\kappa_6^{\text{eff}} = \kappa_6 = \frac{1}{5}\frac{\exp(i\pi\nu_{in})\eta^5(-29 + 4\nu_{in}(4 + \nu_{in}))}{(i\sinh\alpha + \cosh\alpha\exp(i\pi\nu_{in}))(i\cosh\alpha + \sinh\alpha\exp(i\pi\nu_{in}))(1 - 2\nu_{in})^2}, \tag{A.55}$$

$$\kappa_7^{\text{eff}} \neq \kappa_7 = \frac{1}{18(e^{i\pi\nu_{in}} + i)(1 - 2\nu_{in})^2(e^{i\pi\nu_{in}}\cosh(\alpha) + i\sinh(\alpha))^2(e^{i\pi\nu_{in}}\sinh(\alpha) + i\cosh(\alpha))^2}$$

$$\left[\eta^6 e^{i\pi\nu_{in}}(e^{i\pi\nu_{in}} - i)(i(-4\nu_{in}^2 + e^{2i\pi\nu_{in}}(4\nu_{in}^2 + 20\nu_{in} - 47) - 20\nu_{in} + 2ie^{i\pi\nu_{in}}(2\nu_{in} + 5)^2\right.$$

$$\left. + 47)\sinh(2\alpha) + 144e^{i\pi\nu_{in}}\cosh(2\alpha))\right], \tag{A.56}$$

$$\kappa_8^{\text{eff}} = \kappa_8 = -\frac{\eta^7\exp(i\pi\nu_{in})\left(8\nu_{in}^3 + 44\nu_{in}^2 - 162\nu_{in} + 125\right)}{7(2\nu_{in} - 1)^3\left(\exp(i\pi\nu_{in})\cosh(\alpha) + i\sinh(\alpha)\right)\left(\exp(i\pi\nu_{in})\sinh(\alpha) + i\cosh(\alpha)\right)}, \tag{A.57}$$

$$\kappa_9^{\text{eff}} \neq \kappa_9 = \frac{1}{15\left(e^{i\pi\nu}+i\right)^2\left(2\nu_{in}-1\right)^3\sigma^2\left(e^{i\pi\nu_{in}}\cosh(\alpha)+i\sinh(\alpha)\right)^2\left(e^{i\pi\nu_{in}}\sinh(\alpha)+i\cosh(\alpha)\right)^2}$$

$$\left[4\eta^6 e^{-2\alpha+3i\pi\nu_{in}}(\sinh(2\alpha)(e^{2\alpha}\eta^2\sigma^2((8\nu_{in}^3+52\nu_{in}^2-218\nu_{in}+215)\sin(\pi\nu_{in})+(2\nu_{in}+5)\right.$$

$$(4\nu_{in}(\nu_{in}+4)-29))\cos(\pi\nu_{in})-30i(2\nu_{in}-1)\sin(\pi\nu_{in}))+30\cosh(2\alpha)(-4e^{2\alpha}\eta^2(2\nu_{in}-3)$$

$$\left.\sigma^2\cos(\pi\nu_{in})-2i\nu_{in}+i))\right]. \tag{A.58}$$

### A.3.3 Expressions for the Mota-Allen vacua

For Mota-Allen vacua we have the following results:

$$\kappa_0^{\text{eff}} = \kappa_0 = -\frac{1}{2}\log\left(\frac{\cosh\alpha - i\exp(i\pi(\nu_{in}+\gamma))\sinh\alpha}{\exp(-i\gamma)\sinh\alpha - i\exp(i\pi\nu_{in})\cosh\alpha}\right), \tag{A.59}$$

$$\kappa_4^{\text{eff}} = \kappa_4 = \frac{1}{3}\exp(i\pi\nu_{in})\eta^3(5+2\nu_{in})$$

$$\times\left[(\exp(-i\gamma)\sinh\alpha - i\cosh\alpha\exp(i\pi\nu_{in}))(\cosh\alpha - i\sinh\alpha e^{i\pi(\nu_{in}+\gamma)}(-1+2\nu_{in}))\right]^{-1}, \tag{A.60}$$

$$\kappa_6^{\text{eff}} = \kappa_6 = \frac{1}{5}\exp(i\pi\nu_{in})\eta^5(-29+4\nu_{in}(4+\nu_{in}))$$

$$\times\left[i(\exp(-i\gamma)\sinh\alpha + \cosh\alpha\exp(i\pi\nu_{in}))(i\cosh\alpha + \sinh\alpha e^{i(\pi\nu_{in}+\gamma)}(1-2\nu_{in})^2\right]^{-1}, \tag{A.61}$$

$$\kappa_7^{\text{eff}} \neq \kappa_7 \tag{A.62}$$

$$= \frac{\left(-ie^{i\pi\nu_{in}}\left(e^{i\gamma}\sinh(\alpha)+\cosh(\alpha)\right)+e^{-i\gamma}\sinh(\alpha)+\cosh(\alpha)\right)^{-1}}{9(1-2\nu_{in})^2\left(e^{-i\gamma}\sinh(\alpha)-ie^{i\pi\nu_{in}}\cosh(\alpha)\right)^2\left(\sinh(\alpha)e^{i(\gamma+\pi\nu_{in})}+i\cosh(\alpha)\right)^2}$$

$$\left[\eta^6 e^{i\pi\nu_{in}}(e^{i\pi\nu_{in}}(-(2\nu_{in}+5)^2 e^{-i\gamma}\sinh\alpha\cosh\alpha(e^{i\gamma}\sinh(\alpha)+\cosh(\alpha))+72e^{-i\gamma}\sinh^3\alpha\right.$$

$$+72\cosh^3\alpha)+ie^{2i\pi\nu_{in}}(-(2\nu_{in}+5)^2 e^{i\gamma}\sinh\alpha\cosh^2\alpha - (2\nu_{in}+5)^2\sinh^2(\alpha)\cosh\alpha$$

$$+72e^{i\gamma}\sinh^3\alpha + 72\cosh^3\alpha)$$

$$+(-i)(4\nu_{in}(\nu_{in}+5)-47)e^{-2i\gamma}\sinh(\alpha)\cosh\alpha(e^{i\gamma}\cosh\alpha+\sinh\alpha)$$

$$\left.-(4\nu_{in}(\nu_{in}+5)-47)\sinh\alpha\cosh\alpha e^{i\gamma+3i\pi\nu_{in}}(e^{i\gamma}\sinh\alpha+\cosh\alpha))\right], \tag{A.63}$$

$$\kappa_8^{\text{eff}} = \kappa_8 = -\left(\exp(i(\gamma+\pi\nu_{in}))\eta^7\left(2\nu_{in}\left(4\nu_{in}^2+22\nu_{in}-81\right)+125\right)\right) \tag{A.64}$$

$$\times\left(7(2\nu_{in}-1)^3\left(\cosh(\alpha)e^{i(\gamma+\pi\nu_{in})}+i\sinh(\alpha)\right)\left(\sinh(\alpha)e^{i(\gamma+\pi\nu_{in})}+i\cosh(\alpha)\right)\right)^{-1},$$

$$\kappa_9^{\text{eff}} \neq \kappa_9$$

$$= \frac{1}{15(2\nu_{in}-1)^3\sigma^2\left(e^{-i\gamma}\sinh(\alpha)-ie^{i\pi\nu_{in}}\cosh(\alpha)\right)^2\left(\sinh(\alpha)e^{i\gamma+i\pi\nu_{in}}+i\cosh(\alpha)\right)^2}$$

$$
\times \frac{1}{\left(-ie^{i\pi\nu_{in}}\left(e^{i\gamma}\sinh(\alpha)+\cosh(\alpha)\right)+e^{-i\gamma}\sinh(\alpha)+\cosh(\alpha)\right)^2}
$$

$$
\Big[2\eta^6 e^{i\pi\nu_{in}}(\eta^2\sigma^2\cosh^3(\alpha)(i(8\nu_{in}^3+52\nu_{in}^2-218\nu_{in}+215)e^{-i\gamma}\sinh(\alpha)
$$

$$
-120e^{i\pi\nu_{in}}(2\nu_{in}-3)\cosh(\alpha))+\cosh(\alpha)(2\sinh(\alpha)e^{i\gamma+i\pi\nu_{in}}(\eta^2(8\nu_{in}^3+52\nu_{in}^2-218\nu_{in}+215)\sigma^2
$$

$$
-60\nu_{in}+30)(e^{-2i\gamma}\sinh^2(\alpha)+e^{2i\pi\nu_{in}}\cosh^2(\alpha))+\eta^2\sigma^2\sinh^2(\alpha)e^{2i\gamma+2i\pi\nu_{in}}(2e^{i\pi\nu_{in}}(8\nu_{in}^3
$$

$$
+52\nu_{in}^2-98\nu_{in}+35)\cosh(\alpha)-i(8\nu_{in}^3+52\nu_{in}^2+262\nu_{in}-505)e^{-i\gamma}\sinh(\alpha))
$$

$$
-\eta^2\sigma^2 e^{-3i\gamma}(\cosh(\alpha)e^{i\gamma+i\pi\nu_{in}}+i\sinh(\alpha))^2(i(8\nu_{in}^3+52\nu_{in}^2-218\nu_{in}+215)\sinh(\alpha)
$$

$$
+120(2\nu_{in}-3)\cosh(\alpha)e^{i\gamma+i\pi\nu_{in}}))
$$

$$
+\cosh^2(\alpha)(2i\eta^2(8\nu_{in}^3+52\nu_{in}^2-218\nu_{in}+215)\sigma^2 e^{-2i\gamma}\sinh^2(\alpha)
$$

$$
+2\sinh(\alpha)e^{-i\gamma+i\pi\nu_{in}}(\eta^2\sigma^2((8\nu_{in}^3+52\nu_{in}^2-98\nu_{in}+35)e^{i\gamma}\sinh(\alpha)
$$

$$
+(8\nu_{in}^3+52\nu_{in}^2-218\nu_{in}+215)\cosh(\alpha))+30(2\nu_{in}-1)\cosh(\alpha))
$$

$$
-ie^{2i\pi\nu_{in}}\cosh(\alpha)(60(2\nu_{in}-1)\cosh(\alpha)
$$

$$
-\eta^2(8\nu_{in}^3+52\nu_{in}^2+262\nu_{in}-505)\sigma^2 e^{i\gamma}\sinh(\alpha)))
$$

$$
+i\sinh^3(\alpha)e^{-i\gamma+2i\pi\nu_{in}}(\eta^2(8\nu_{in}^3+52\nu_{in}^2+262\nu_{in}-505)\sigma^2\cosh(\alpha)
$$

$$
+60(2\nu_{in}-1)e^{i\gamma}\sinh(\alpha))
$$

$$
+2e^{3i\pi\nu_{in}}\sinh^2(\alpha)(\eta^2\sigma^2(e^{i\gamma}\sinh(\alpha)+\cosh(\alpha))((8\nu^3+52\nu_{in}^2-98\nu_{in}+35)
$$

$$
\cosh(\alpha)+60(3-2\nu_{in})e^{i\gamma}\sinh(\alpha))+15(2\nu_{in}-1)e^{i\gamma}\sinh(2\alpha))
$$

$$
-i\eta^2(8\nu_{in}^3+52\nu_{in}^2-218\nu_{in}+215)
$$

$$
\sigma^2\sinh(\alpha)\cosh(\alpha)e^{i\gamma+4i\pi\nu_{in}}(e^{i\gamma}\sinh(\alpha)+\cosh(\alpha))^2
$$

$$
+120\eta^2(3-2\nu_{in})\sigma^2\sinh^4(\alpha)e^{-2i\gamma+i\pi\nu_{in}})\Big] \tag{A.65}
$$

## A.4 Consistency relations

In this appendix, we re-derive the relations between the various coefficients of $\gamma$ and $\kappa$ for the different choices of quantum initial conditions as discussed earlier in the text portion. For the sudden mass quench profile, the relationship between the various coefficients of $\kappa(k)$ and $\gamma(k)$ can be expressed as:

$$
\kappa_{4,\mathbf{DB}}=\kappa_{4,\mathbf{NB}}=\frac{i}{2}\left(\frac{id_2^*+d_1^*\exp(i\pi\nu_{in})}{d_1-id_2\exp(i\pi\nu_{in})}\right)\gamma_4 \quad=\frac{1}{2}\left(\frac{d_1+id_2\exp(i\pi\nu_{in})}{d_1-id_2\exp(i\pi\nu_{in})}\right)\frac{\gamma_4}{\gamma_0} \tag{A.66}
$$

$$
\kappa_{6,\mathbf{DB}}=\kappa_{6,\mathbf{NB}}=\frac{1}{2}\left(\frac{id_2^*+d_1^*\exp(i\pi\nu_{in})}{id_1+d_2\exp(i\pi\nu_{in})}\right)\gamma_6 \quad=\frac{1}{2}\left(\frac{-id_1+d_2\exp(i\pi\nu_{in})}{id_1+d_2\exp(i\pi\nu_{in})}\right)\frac{\gamma_6}{\gamma_0} \tag{A.67}
$$

In the next three subsections we mention the results for the previously mentioned three different choices of the quantum initial conditions. Here we are computing the expressions for the Dirichlet boundary states from which one can also derive the expressions for the Neumann boundary states using the above mentioned connecting relationships. For com-

putational simplicity we will further drop the superscript **DB** in the further computations.

### A.4.1 Expressions for the Bunch Davies vacuum

For Bunch Davies vacuum we have the following results:

$$\kappa_4 = \frac{1}{2}\left(\frac{\gamma_4}{\gamma_0}\right) \tag{A.68}$$

$$\kappa_6 = -\frac{1}{2}\left(\frac{\gamma_6}{\gamma_0}\right) \tag{A.69}$$

### A.4.2 Expressions for the $\alpha$ vacua

For $\alpha$ vacua we have the following results:

$$\kappa_4 = \frac{1}{2}\left(\frac{\cosh\alpha + i\sinh\alpha\exp(i\pi\nu_{in})}{i\cosh\alpha + \sinh\alpha\exp(i\pi\nu_{in})}\right)\left(\frac{\gamma_4}{\gamma_0}\right) \tag{A.70}$$

$$\kappa_6 = \frac{1}{2}\left(\frac{-i\cosh\alpha + \sinh\alpha\exp(i\pi\nu_{in})}{\cosh\alpha - i\sinh\alpha\exp(i\pi\nu_{in})}\right)\left(\frac{\gamma_6}{\gamma_0}\right) \tag{A.71}$$

### A.4.3 Expressions for the Mota-Allen vacua

For Mota-Allen vacua we have the following results:

$$\kappa_4 = \frac{1}{2}\left(\frac{\cosh\alpha + \sinh\alpha\exp(i\pi(\nu_{in} + 1/2) + \gamma)}{\cosh\alpha - \sinh\alpha\exp(i\pi(\nu_{in} + 1/2) + \gamma)}\right)\left(\frac{\gamma_4}{\gamma_0}\right) \tag{A.72}$$

$$\kappa_6 = \frac{1}{2}\left(\frac{\exp(-i\pi/2)\cosh\alpha + \sinh\alpha\exp(i(\pi\nu_{in} + \gamma))}{\exp(i\pi/2)\cosh\alpha + \sinh\alpha\exp(i(\pi\nu_{in} + \gamma))}\right)\left(\frac{\gamma_6}{\gamma_0}\right) \tag{A.73}$$

## B  Definition of the Symbols appearing in the two-point correlators

### B.1  Symbols appearing in the correlators of the ground state

Here we have defined the symbols $\Delta_i(\mathbf{k}, \tau_1, \tau_2) \; \forall \; i = 1, \cdots, 16$ that appeared in the correlators calculated for the ground state:

$$\Delta_1(\mathbf{k}, \tau_1, \tau_2) = |\alpha(\mathbf{k})|^2 v_{out}(\mathbf{k}, \tau_1) v_{out}^*(\mathbf{k}, \tau_2) \tag{B.1}$$

$$\Delta_2(\mathbf{k}, \tau_1, \tau_2) = \alpha(\mathbf{k})\beta^*(\mathbf{k}) v_{out}(\mathbf{k}, \tau_1) v_{out}(-\mathbf{k}, \tau_2) \tag{B.2}$$

$$\Delta_3(\mathbf{k}, \tau_1, \tau_2) = \alpha^*(\mathbf{k})\beta(\mathbf{k}) v_{out}^*(-\mathbf{k}, \tau_1) v_{out}^*(\mathbf{k}, \tau_2) \tag{B.3}$$

$$\Delta_4(\mathbf{k}, \tau_1, \tau_2) = |\beta(\mathbf{k})|^2 v_{out}^*(\mathbf{k}, \tau_1) v_{out}(-\mathbf{k}, \tau_2) \tag{B.4}$$

$$\Delta_5(\mathbf{k}, \tau_1, \tau_2) = |\alpha(\mathbf{k})|^2 v_{out}(\mathbf{k}, \tau_1) v_{out}'^*(\mathbf{k}, \tau_2) \tag{B.5}$$

$$\Delta_6(\mathbf{k}, \tau_1, \tau_2) = \alpha(\mathbf{k})\beta^*(\mathbf{k}) v_{out}(\mathbf{k}, \tau_1) v_{out}'(-\mathbf{k}, \tau_2) \tag{B.6}$$

$$\Delta_7(\mathbf{k}, \tau_1, \tau_2) = \alpha^*(\mathbf{k})\beta(\mathbf{k}) v_{out}^*(-\mathbf{k}, \tau_1) v_{out}'^*(\mathbf{k}, \tau_2) \tag{B.7}$$

$$\Delta_8(\mathbf{k}, \tau_1, \tau_2) = |\beta(\mathbf{k})|^2 v_{out}^*(\mathbf{k}, \tau_1) v_{out}'(-\mathbf{k}, \tau_2) \tag{B.8}$$

$$\Delta_9(\mathbf{k}, \tau_1, \tau_2) = |\alpha(\mathbf{k})|^2 v'_{out}(\mathbf{k}, \tau_1) v^*_{out}(\mathbf{k}, \tau_2) \tag{B.9}$$

$$\Delta_{10}(\mathbf{k}, \tau_1, \tau_2) = \alpha(\mathbf{k})\beta^*(\mathbf{k}) v'_{out}(\mathbf{k}, \tau_1) v_{out}(-\mathbf{k}, \tau_2) \tag{B.10}$$

$$\Delta_{11}(\mathbf{k}, \tau_1, \tau_2) = \alpha^*(\mathbf{k})\beta(\mathbf{k}) v'^*_{out}(-\mathbf{k}, \tau_1) v^*_{out}(\mathbf{k}, \tau_2) \tag{B.11}$$

$$\Delta_{12}(\mathbf{k}, \tau_1, \tau_2) = |\beta(\mathbf{k})|^2 v'^*_{out}(\mathbf{k}, \tau_1) v_{out}(-\mathbf{k}, \tau_2) \tag{B.12}$$

$$\Delta_{13}(\mathbf{k}, \tau_1, \tau_2) = |\alpha(\mathbf{k})|^2 v'_{out}(\mathbf{k}, \tau_1) v'^*_{out}(\mathbf{k}, \tau_2) \tag{B.13}$$

$$\Delta_{14}(\mathbf{k}, \tau_1, \tau_2) = \alpha(\mathbf{k})\beta^*(\mathbf{k}) v'_{out}(\mathbf{k}, \tau_1) v'_{out}(-\mathbf{k}, \tau_2) \tag{B.14}$$

$$\Delta_{15}(\mathbf{k}, \tau_1, \tau_2) = \alpha^*(\mathbf{k})\beta(\mathbf{k}) v'^*_{out}(-\mathbf{k}, \tau_1) v'^*_{out}(\mathbf{k}, \tau_2) \tag{B.15}$$

$$\Delta_{16}(\mathbf{k}, \tau_1, \tau_2) = |\beta(\mathbf{k})|^2 v'^*_{out}(\mathbf{k}, \tau_1) v'_{out}(-\mathbf{k}, \tau_2) \tag{B.16}$$

and $v_{in}$ and $v_{out}$ are the fluctuation solutions before and after the quench point respectively and $\alpha$ and $\beta$ are Bogoliubov coefficients which encodes the quench protocol in the form of the asymptotic expansion of the Hankel functions. The Bogoliubov coefficients could be written entirely in terms of $\gamma(k)$ as follows:

$$|\alpha(\mathbf{k})|^2 = \frac{1}{1 - |\gamma(\mathbf{k})|^2}, \tag{B.17}$$

$$|\beta(\mathbf{k})|^2 = \frac{|\gamma(\mathbf{k})|^2}{1 - |\gamma(\mathbf{k})|^2}, \tag{B.18}$$

$$\alpha(\mathbf{k})\beta^*(\mathbf{k}) = \frac{|\gamma(\mathbf{k})|}{1 - |\gamma(\mathbf{k})|^2}, \tag{B.19}$$

$$\alpha^*(\mathbf{k})\beta(\mathbf{k}) = \frac{|\gamma^*(\mathbf{k})|}{1 - |\gamma(\mathbf{k})|^2} \tag{B.20}$$

## B.2 Symbols appearing in the correlators of the gCC state

The symbols $\Theta_i(\mathbf{k}, \tau_1, \tau_2) \; \forall \; i = 1, \cdots, 16$ appearing in the correlators of the gCC states are given by:

$$\Theta_1(\mathbf{k}, \tau_1, \tau_2) = v_{out}(\mathbf{k}, \tau_1) v^*_{out}(\mathbf{k}, \tau_2) \tag{B.21}$$

$$\Theta_2(\mathbf{k}, \tau_1, \tau_2) = v_{out}(\mathbf{k}, \tau_1) v_{out}(-\mathbf{k}, \tau_2) \tag{B.22}$$

$$\Theta_3(\mathbf{k}, \tau_1, \tau_2) = v^*_{out}(-\mathbf{k}, \tau_1) v^*_{out}(\mathbf{k}, \tau_2) \tag{B.23}$$

$$\Theta_4(\mathbf{k}, \tau_1, \tau_2) = v^*_{out}(\mathbf{k}, \tau_1) v_{out}(-\mathbf{k}, \tau_2) \tag{B.24}$$

$$\Theta_5(\mathbf{k}, \tau_1, \tau_2) = v_{out}(\mathbf{k}, \tau_1) v'^*_{out}(\mathbf{k}, \tau_2) \tag{B.25}$$

$$\Theta_6(\mathbf{k}, \tau_1, \tau_2) = v_{out}(\mathbf{k}, \tau_1) v'_{out}(-\mathbf{k}, \tau_2) \tag{B.26}$$

$$\Theta_7(\mathbf{k}, \tau_1, \tau_2) = v^*_{out}(-\mathbf{k}, \tau_1) v'^*_{out}(\mathbf{k}, \tau_2) \tag{B.27}$$

$$\Theta_8(\mathbf{k}, \tau_1, \tau_2) = v^*_{out}(\mathbf{k}, \tau_1) v'_{out}(-\mathbf{k}, \tau_2) \tag{B.28}$$

$$\Theta_9(\mathbf{k}, \tau_1, \tau_2) = v'_{out}(\mathbf{k}, \tau_1) v^*_{out}(\mathbf{k}, \tau_2) \tag{B.29}$$

$$\Theta_{10}(\mathbf{k}, \tau_1, \tau_2) = v'_{out}(\mathbf{k}, \tau_1) v_{out}(-\mathbf{k}, \tau_2) \tag{B.30}$$

$$\Theta_{11}(\mathbf{k}, \tau_1, \tau_2) = v_{out}'^*(-\mathbf{k}, \tau_1)v_{out}^*(\mathbf{k}, \tau_2) \tag{B.31}$$

$$\Theta_{12}(\mathbf{k}, \tau_1, \tau_2) = v_{out}'^*(\mathbf{k}, \tau_1)v_{out}(-\mathbf{k}, \tau_2) \tag{B.32}$$

$$\Theta_{13}(\mathbf{k}, \tau_1, \tau_2) = v_{out}'(\mathbf{k}, \tau_1)v_{out}'^*(\mathbf{k}, \tau_2) \tag{B.33}$$

$$\Theta_{14}(\mathbf{k}, \tau_1, \tau_2) = v_{out}'(\mathbf{k}, \tau_1)v_{out}'(-\mathbf{k}, \tau_2) \tag{B.34}$$

$$\Theta_{15}(\mathbf{k}, \tau_1, \tau_2) = v_{out}'^*(-\mathbf{k}, \tau_1)v_{out}'^*(\mathbf{k}, \tau_2) \tag{B.35}$$

$$\Theta_{16}(\mathbf{k}, \tau_1, \tau_2) = v_{out}'^*(\mathbf{k}, \tau_1)v_{out}'(-\mathbf{k}, \tau_2) \tag{B.36}$$

and $v_{in}$ and $v_{out}$ are the fluctuation solutions before and after the quench point respectively.

## B.3 Symbols for squeezed state

Here we have introduced new symbols $\Delta_i^{sq}(\mathbf{k}, \tau_1, \tau_2) \ \forall \ i = 1, \cdots, 16$ which are used in the above mentioned expressions for propagators and given by:

$$\Delta_1^{sq}(\mathbf{k}, \tau_1, \tau_2) = \frac{|\alpha_{\text{eff}}(\mathbf{k})|^2}{|\alpha(\mathbf{k})|^2} \Delta_1(\mathbf{k}, \tau_1, \tau_2) \tag{B.37}$$

$$\Delta_2^{sq}(\mathbf{k}, \tau_1, \tau_2) = \frac{\alpha_{\text{eff}}(\mathbf{k})\beta_{\text{eff}}^*(\mathbf{k})}{\alpha(\mathbf{k})\beta^*(\mathbf{k})} \Delta_2(\mathbf{k}, \tau_1, \tau_2) \tag{B.38}$$

$$\Delta_3^{sq}(\mathbf{k}, \tau_1, \tau_2) = \frac{\alpha_{\text{eff}}^*(\mathbf{k})\beta_{\text{eff}}(\mathbf{k})}{\alpha^*(\mathbf{k})\beta(\mathbf{k})} \Delta_3(\mathbf{k}, \tau_1, \tau_2) \tag{B.39}$$

$$\Delta_4^{sq}(\mathbf{k}, \tau_1, \tau_2) = \frac{|\beta_{\text{eff}}(\mathbf{k})|^2}{|\beta(\mathbf{k})|^2} \Delta_4(\mathbf{k}, \tau_1, \tau_2) \tag{B.40}$$

$$\Delta_5^{sq}(\mathbf{k}, \tau_1, \tau_2) = \frac{|\alpha_{\text{eff}}(\mathbf{k})|^2}{|\alpha(\mathbf{k})|^2} \Delta_5(\mathbf{k}, \tau_1, \tau_2) \tag{B.41}$$

$$\Delta_6^{sq}(\mathbf{k}, \tau_1, \tau_2) = \frac{\alpha_{\text{eff}}(\mathbf{k})\beta_{\text{eff}}^*(\mathbf{k})}{\alpha(\mathbf{k})\beta^*(\mathbf{k})} \Delta_6(\mathbf{k}, \tau_1, \tau_2) \tag{B.42}$$

$$\Delta_7^{sq}(\mathbf{k}, \tau_1, \tau_2) = \frac{\alpha_{\text{eff}}^*(\mathbf{k})\beta_{\text{eff}}(\mathbf{k})}{\alpha^*(\mathbf{k})\beta(\mathbf{k})} \Delta_7(\mathbf{k}, \tau_1, \tau_2) \tag{B.43}$$

$$\Delta_8^{sq}(\mathbf{k}, \tau_1, \tau_2) = \frac{|\beta_{\text{eff}}(\mathbf{k})|^2}{|\beta(\mathbf{k})|^2} \Delta_8(\mathbf{k}, \tau_1, \tau_2) \tag{B.44}$$

$$\Delta_9^{sq}(\mathbf{k}, \tau_1, \tau_2) = \frac{|\alpha_{\text{eff}}(\mathbf{k})|^2}{|\alpha(\mathbf{k})|^2} \Delta_9(\mathbf{k}, \tau_1, \tau_2) \tag{B.45}$$

$$\Delta_{10}^{sq}(\mathbf{k}, \tau_1, \tau_2) = \frac{\alpha_{\text{eff}}(\mathbf{k})\beta_{\text{eff}}^*(\mathbf{k})}{\alpha(\mathbf{k})\beta^*(\mathbf{k})} \Delta_{10}(\mathbf{k}, \tau_1, \tau_2) \tag{B.46}$$

$$\Delta_{11}^{sq}(\mathbf{k}, \tau_1, \tau_2) = \frac{\alpha_{\text{eff}}^*(\mathbf{k})\beta_{\text{eff}}(\mathbf{k})}{\alpha^*(\mathbf{k})\beta(\mathbf{k})} \Delta_{11}(\mathbf{k}, \tau_1, \tau_2) \tag{B.47}$$

$$\Delta_{12}^{sq}(\mathbf{k}, \tau_1, \tau_2) = \frac{|\beta_{\text{eff}}(\mathbf{k})|^2}{|\beta(\mathbf{k})|^2} \Delta_{12}(\mathbf{k}, \tau_1, \tau_2) \tag{B.48}$$

$$\Delta_{13}^{sq}(\mathbf{k}, \tau_1, \tau_2) = \frac{|\alpha_{\text{eff}}(\mathbf{k})|^2}{|\alpha(\mathbf{k})|^2} \Delta_{13}(\mathbf{k}, \tau_1, \tau_2) \tag{B.49}$$

$$\Delta_{14}^{sq}(\mathbf{k}, \tau_1, \tau_2) = \frac{\alpha_{\text{eff}}(\mathbf{k})\beta_{\text{eff}}^*(\mathbf{k})}{\alpha(\mathbf{k})\beta^*(\mathbf{k})} \Delta_{14}(\mathbf{k}, \tau_1, \tau_2) \tag{B.50}$$

$$\Delta_{15}^{sq}(\mathbf{k}, \tau_1, \tau_2) = \frac{\alpha_{\text{eff}}^*(\mathbf{k})\beta_{\text{eff}}(\mathbf{k})}{\alpha^*(\mathbf{k})\beta(\mathbf{k})} \Delta_{15}(\mathbf{k}, \tau_1, \tau_2) \tag{B.51}$$

$$\Delta_{16}^{sq}(\mathbf{k}, \tau_1, \tau_2) = \frac{|\beta_{\text{eff}}(\mathbf{k})|^2}{|\beta(\mathbf{k})|^2} \Delta_{16}(\mathbf{k}, \tau_1, \tau_2) \tag{B.52}$$

and $v_{in}$ and $v_{out}$ are the fluctuation solutions before and after the quench point respectively and $\alpha$ and $\beta$ are Bogoliubov coefficients which encodes the quench protocol in the form of the asymptotic expansion of the Hankel functions. These Bogoliubov coefficients could be written entirely in terms of $\gamma_{\text{eff}}(k)$ as follows:

$$|\alpha_{\text{eff}}(\mathbf{k})|^2 = \frac{1}{1 - |\gamma_{\text{eff}}(\mathbf{k})|^2}, \tag{B.53}$$

$$|\beta_{\text{eff}}(\mathbf{k})|^2 = \frac{|\gamma_{\text{eff}}(\mathbf{k})|^2}{1 - |\gamma_{\text{eff}}(\mathbf{k})|^2}, \tag{B.54}$$

$$\alpha_{\text{eff}}(\mathbf{k})\beta_{\text{eff}}^*(\mathbf{k}) = \frac{|\gamma_{\text{eff}}(\mathbf{k})|}{1 - |\gamma_{\text{eff}}(\mathbf{k})|^2}, \tag{B.55}$$

$$\alpha_{\text{eff}}^*(\mathbf{k})\beta_{\text{eff}}(\mathbf{k}) = \frac{|\gamma_{\text{eff}}^*(\mathbf{k})|}{1 - |\gamma_{\text{eff}}(\mathbf{k})|^2} \tag{B.56}$$

## C  Quantization of Hamiltonian in occupation number representation

Now in the quantum description the corresponding quantized normal ordered Hamiltonian operator can be written as:

$$\hat{H}(\tau) = \sum_{\{N_k\}=0 \ \forall \ k}^{\infty} \hat{H}_k(\tau), \tag{C.1}$$

where in the occupation number representation of the Hamiltonian one can write:

$$\hat{H}_k(\tau) = \hat{N}_k \ E_k(\tau) \quad \text{where} \quad \hat{N}_k = a_{out}^\dagger(-\mathbf{k})a_{out}(\mathbf{k}). \tag{C.2}$$

Here $E_k(\tau)$ is the dispersion relation which is defined in the present context as:

$$E_k(\tau) = \left[ |\Pi_{out}(\mathbf{k}, \tau)|^2 + \omega_{out}^2(k, \tau)|v_{out}(\mathbf{k}, \tau)|^2 \right]. \tag{C.3}$$

Hence in the occupation number representation we have:

$$\langle \{N_k\}| \ \hat{H}_k(\tau) \ |\{N_k\}\rangle = N_k E_k(\tau). \tag{C.4}$$

# D  Derivation for thermal partition function for GGE ensemble

First of all we derive the expression for the thermal partition function $Z$ for GGE ensemble. For this purpose we start with the following definition:

$$Z(\tau_1) = \mathrm{Tr}\left( \exp(-\beta\hat{H}(\tau_1) - \sum_{n=2}^{\infty} \kappa_{2n,\mathbf{DB/NB}}|k|^{2n-1}\hat{N}(k)) \right)$$

$$= \int d\Psi_{out} \; {}_{out}\langle\Psi| \exp(-\beta\hat{H}(\tau_1)) - \sum_{n=2}^{\infty} \kappa_{2n,\mathbf{DB/NB}}|k|^{2n-1}\hat{N}(k) |\Psi\rangle_{out}, \quad \text{(D.1)}$$

where we have translated the trace operation in terms of an outgoing quantum state after quench in continuous representation of wave function. But technically computation of this result is very cumbersome in terms of a thermal state. For this reason the above mentioned expression can be further represented in terms of the occupation number discrete representation of the Hamiltonian basis $|\{N_k\}\rangle \; \forall \; k$ as:

$$Z(\tau_1) = \underbrace{\frac{1}{|d_1|} \exp\left( -\frac{i}{2}\left\{ \frac{d_2^*}{d_1^*} - \frac{d_2}{d_1} \right\} \right)}_{\textcolor{red}{\text{This factor is the outcome of arbitrary quantum vacuum}}}$$

$$\times \sum_{\{N_k\}=0 \; \forall \; k}^{\infty} \langle\{N_k\}| \exp(-\beta\,\hat{H}_k(\tau_1)) - \sum_{n=2}^{\infty} \kappa_{2n,\mathbf{DB/NB}}|k|^{2n-1}\hat{N}(k)) |\{N_k\}\rangle,$$

$$= \frac{1}{|d_1|} \exp\left( -\frac{i}{2}\left\{ \frac{d_2^*}{d_1^*} - \frac{d_2}{d_1} \right\} \right) \times \sum_{\{N_k\}=0 \; \forall \; k}^{\infty} \exp(-\beta E_k(\tau_1))N_k - \sum_{n=2}^{\infty} \kappa_{2n,\mathbf{DB/NB}}|k|^{2n-1}N(k)),$$

$$= \frac{1}{|d_1|} \exp\left( -\frac{i}{2}\left\{ \frac{d_2^*}{d_1^*} - \frac{d_2}{d_1} \right\} \right) \left( \frac{\exp(\beta E_k(\tau_1))_{\mathrm{eff}}}{\exp(\beta E_k(\tau_1))_{\mathrm{eff}} - 1} \right),$$

$$= \frac{1}{2|d_1|} \exp\left( -\frac{i}{2}\left\{ \frac{d_2^*}{d_1^*} - \frac{d_2}{d_1} \right\} \right) \exp\left( \frac{(\beta E_k(\tau_1))_{\mathrm{eff}}}{2} \right) \mathrm{cosech}\left( \frac{(\beta E_k(\tau_1))_{\mathrm{eff}}}{2} \right) \quad \text{(D.2)}$$

where $E_k(\tau_1)$ is the cosmological dispersion relation, which is given by:

$$E_k(\tau_1) = \left[ |\Pi_{out}(\mathbf{k}, \tau_1)|^2 + \omega_{out}^2(k, \tau_1)|v_{out}(\mathbf{k}, \tau_1)|^2 \right], \quad \text{(D.3)}$$

having the frequency $\omega_{out}$ of the outgoing modes after the quench operation is given by the following expression:

$$\omega_{out}^2(k, \tau_1) = \left( k^2 - \frac{2}{\tau_1^2} \right) \quad \text{where} \quad \tau_1 = \tau + \eta \quad \text{(D.4)}$$

where, in the above mentioned notation $\eta$ represents the time scale where the quantum quench operation have been performed. Further translating the dispersion relation in terms

of the $\chi$ field we get the following expression:

$$E_k(\tau_1) = a^2(\tau_1)\left[E_k^\chi(\tau_1) + \mathcal{H}(\tau_1)\,\mathcal{O}_k^\chi(\tau_1)\right] \qquad \text{where} \qquad \mathcal{H}(\tau_1) = \left(\frac{a'(\tau_1)}{a(\tau_1)}\right), \quad \text{(D.5)}$$

where the energy dispersion relation in terms of the field $\chi$ and the new contribution $\mathcal{O}_k^\chi(\tau_1)$ can be expressed as:

$$E_k^\chi(\tau_1) = \left[|\Pi_\chi(\mathbf{k},\tau_1)|^2 + \omega_\chi^2(k,\tau_1)|\chi(\mathbf{k},\tau_1)|^2\right], \qquad\qquad \text{(D.6)}$$

$$\mathcal{O}_k^\chi(\tau_1) = \left[\Pi_\chi(-\mathbf{k},\tau_1)\chi(\mathbf{k},\tau_1) + \Pi_\chi(\mathbf{k},\tau_1)\chi(-\mathbf{k},\tau_1)\right]. \qquad \text{(D.7)}$$

Here the new effective frequency $\omega_\chi$ after the quench operation for the outgoing field can be written as:

$$\omega_\chi^2(k,\tau_1) = \omega_{out}^2(k,\tau_1) + \mathcal{H}^2(\tau_1) \qquad \text{where} \qquad \mathcal{H}(\tau_1) = \left(\frac{a'(\tau_1)}{a(\tau_1)}\right). \qquad \text{(D.8)}$$

# E  Subsystem thermalization from gCC to GGE

Now our aim is to explicitly establish the statement of subsystem thermalization from a gCC state to thermal GGE ensemble. and the equivalence between them. The derived results in this section is new in the sense that we have done the computation for the $1+3$ dimension de Sitter curved space-time and can be used these results further to interpret various unknown physical concepts including the thermalization phenomena in the context of early universe cosmology.

For the post-quench gCC type of quantum states constructed in this paper using the Dirichlet and Neumann boundary states within the perturbative regime of the expansion coefficients of the $W_\infty$ conserved charges, the reduced density matrix of a region $\mathcal{A}$, which can be obtained by performing a partial trace operation on a region $\mathcal{B}$ and treated to be the complement of the region $\mathcal{A}$ can be asymptotically approaches to a GGE, which is technically demonstrated as:

**For Dirichlet boundary state :**

$$\text{Tr}_\mathcal{B}\left[\exp(-iH\tau)\,|\psi(\kappa_n)\rangle\,\langle\psi(\kappa_n)|\,\exp(iH\tau)\right]$$

$$= \text{Tr}_\mathcal{B}\left[\exp(-iH\tau)\exp\left(-\int\frac{d^3\mathbf{k}}{(2\pi)^3}\,\kappa(k)\hat{N}(k)\right)|D\rangle\,\langle D|\exp\left(-\int\frac{d^3\mathbf{k}}{(2\pi)^3}\,\kappa(k)\hat{N}(k)\right)\exp(iH\tau)\right]$$

$$\xrightarrow{\tau\to 0}$$

$$\text{Tr}_\mathcal{B}\left[\frac{1}{Z(\tau)}\exp\left(-\int\frac{d^3\mathbf{k}}{(2\pi)^3}\,4\kappa(k)\hat{N}(k)\right)\right]$$

$$= \text{Tr}_{\mathcal{B}} \Big[ \rho_{\text{GGE}}(\beta, 4\kappa_{n,\textbf{DB}}) \Big] \qquad \text{where} \qquad \rho_{\text{GGE}}(\beta, 4\kappa_{n,\textbf{DB}}) = \frac{1}{Z(\tau)} \exp \left( -\beta H - 4 \sum_n \kappa_{n,\textbf{DB}} W_n \right),$$

$$\text{(E.1)}$$

and

**For Neumann boundary state :**

$$\text{Tr}_{\mathcal{B}} \Big[ \exp(-iH\tau) |\psi(\kappa_n)\rangle \langle \psi(\kappa_n)| \exp(iH\tau) \Big]$$

$$= \text{Tr}_{\mathcal{B}} \left[ \exp(-iH\tau) \exp \left( \int \frac{d^3\textbf{k}}{(2\pi)^3} \kappa(k) \hat{N}(k) \right) |N\rangle \langle N| \exp \left( \int \frac{d^3\textbf{k}}{(2\pi)^3} \kappa(k) \hat{N}(k) \right) \exp(iH\tau) \right]$$

$$\xrightarrow{\tau \to 0}$$

$$\text{Tr}_{\mathcal{B}} \left[ \frac{1}{Z(\tau)} \exp \left( \int \frac{d^3\textbf{k}}{(2\pi)^3} 4\kappa(k) \hat{N}(k) \right) \right]$$

$$= \text{Tr}_{\mathcal{B}} \Big[ \rho_{\text{GGE}}(\beta, 4\kappa_{n,\textbf{NB}}) \Big] \qquad \text{where} \qquad \rho_{\text{GGE}}(\beta, 4\kappa_{n,\textbf{NB}}) = \frac{1}{Z(\tau)} \exp \left( -\beta H - 4 \sum_n \kappa_{n,\textbf{NB}} W_n \right),$$

$$\text{(E.2)}$$

Here it is important to note that all the quantum operators of the $W_\infty$ algebra in the present context can be expressed as:

$$W_n = |k|^{n-1} \hat{N(k)} \quad \text{where} \quad \hat{N(k)} = a_{out}^\dagger(\textbf{k}) a_{out}(\textbf{k}). \tag{E.3}$$

This further implies the ensemble average of the conserved charges of $W_\infty$ algebra for gCC and GGE turn out to be exactly same because of subsystem thermalization, i.e.

$$\langle W_n \rangle_{\text{gCC}} = \langle W_n \rangle_{\text{GGE}}. \tag{E.4}$$

It can be explicitly verified that in the present prescription the following statement is true:

$$\langle N(k) \rangle_{\text{gCC}} = |\beta(k)|^2 = \frac{|\gamma(k)|^2}{1 - |\gamma(k)|^2}, \tag{E.5}$$

$$\langle N(k) \rangle_{\text{GGE}} = \frac{1}{\exp(4\kappa(k)) - 1}, \tag{E.6}$$

$$\langle N(k) \rangle_{\text{gCC}} = \langle N(k) \rangle_{\text{GGE}}, \tag{E.7}$$

where all the quantities are evaluated at a fixed value of conformal time $\eta$ where the quench operation is performed. For simplicity we have dropped the $\eta$ dependence in the above expressions. But remind ourself it is important to note that all functions of $k$ would be actually representing functions of $(k, \eta)$ in this context.

## F  Derivation for thermal Green's functions for GGE ensemble without squeezing in Fourier space

The thermal Green's functions for the GGE ensemble for the field $\chi$, its spatial derivative and its canonically conjugate momentum can be expressed as:

$$
G_{\chi\chi}^{GGE}(\beta, \mathbf{r}, \tau_1, \tau_2) = \int \frac{d^3\mathbf{k}}{(2\pi)^3} \left[ \mathcal{G}_{+,\chi\chi}^{GGE}(\beta, \mathbf{k}, \tau_1, \tau_2) \exp(i\mathbf{k}.\mathbf{r}) \right.
$$
$$
\left. + \mathcal{G}_{-,\chi\chi}^{GGE}(\beta, \mathbf{k}, \tau_1, \tau_2) \exp(-i\mathbf{k}.\mathbf{r}) \right], \qquad \text{(F.1)}
$$

$$
G_{\partial_i\chi\partial_i\chi}^{GGE}(\beta, \mathbf{k}, \tau_1, \tau_2) = \int \frac{d^3\mathbf{k}}{(2\pi)^3} \left[ \mathcal{G}_{+,\partial_i\chi\partial_i\chi}^{GGE}(\beta, \mathbf{k}, \tau_1, \tau_2) \exp(i\mathbf{k}.\mathbf{r}) \right.
$$
$$
\left. + \mathcal{G}_{-,\partial_i\chi\partial_i\chi}^{GGE}(\beta, \mathbf{k}, \tau_1, \tau_2) \exp(-i\mathbf{k}.\mathbf{r}) \right], \qquad \text{(F.2)}
$$

$$
G_{\Pi_\chi\Pi_\chi}^{GGE}(\beta, \mathbf{r}, \tau_1, \tau_2) = \int \frac{d^3\mathbf{k}}{(2\pi)^3} \left[ \mathcal{G}_{+,\Pi_\chi\Pi_\chi}^{GGE}(\beta, \mathbf{k}, \tau_1, \tau_2) \exp(i\mathbf{k}.\mathbf{r}) \right.
$$
$$
\left. + \mathcal{G}_{-,\Pi_\chi\Pi_\chi}^{GGE}(\beta, \mathbf{k}, \tau_1, \tau_2) \exp(-i\mathbf{k}.\mathbf{r}) \right], \qquad \text{(F.3)}
$$

where we define, $\mathbf{r} :\equiv \mathbf{x}_1 - \mathbf{x}_2$.

For each of the cases the corresponding thermal propagators in Fourier space are divided into two parts, one of them represents the advanced propagator which are appearing with $+$ symbol and the other one is the retarded propagator which are appearing with the $-$ symbol. In the occupation number representation for the Hamiltonian we get:

$$
\mathcal{G}_{+,\chi\chi}^{GGE}(\beta, \mathbf{k}, \tau_1, \tau_2) = \frac{1}{Z(\tau_1)} \frac{v_{out}(\mathbf{k}, \tau_1) v_{out}^*(-\mathbf{k}, \tau_2)}{a(\tau_1)a(\tau_2)} \frac{1}{|d_1|} \exp\left( -\frac{i}{2} \left\{ \frac{d_2^*}{d_1^*} - \frac{d_2}{d_1} \right\} \right)
$$
$$
\times \sum_{\{N_k\}=0 \ \forall \ k}^{\infty} \langle \{N_k\}| \exp(-\beta \ \hat{H}_k(\tau_1) - \sum_{n=2}^{\infty} \kappa_{2n,\mathbf{DB/NB}}|k|^{2n-1}))
$$
$$
a_{out}(\mathbf{k}) a_{out}^\dagger(-\mathbf{k}) |\{N_k\}\rangle
$$
$$
= \frac{1}{Z(\tau_1)} \frac{v_{out}(\mathbf{k}, \tau_1) v_{out}^*(-\mathbf{k}, \tau_2)}{a(\tau_1)a(\tau_2)} \frac{1}{|d_1|} \exp\left( -\frac{i}{2} \left\{ \frac{d_2^*}{d_1^*} - \frac{d_2}{d_1} \right\} \right)
$$
$$
\times \sum_{\{N_k\}=0 \ \forall \ k}^{\infty} (N_k + 1) \exp(-(\beta E_k(\tau_1))_{\text{eff}} N_k)
$$
$$
= \frac{1}{Z(\tau_1)} \frac{v_{out}(\mathbf{k}, \tau_1) v_{out}^*(-\mathbf{k}, \tau_2)}{a(\tau_1)a(\tau_2)} \frac{1}{|d_1|} \exp\left( -\frac{i}{2} \left\{ \frac{d_2^*}{d_1^*} - \frac{d_2}{d_1} \right\} \right)
$$
$$
\times \frac{\exp\left(2(\beta E_k(\tau_1))_{\text{eff}}\right)}{\left(\exp\left((\beta E_k(\tau_1))_{\text{eff}}\right) - 1\right)^2}
$$
$$
= \frac{v_{out}(\mathbf{k}, \tau_1) v_{out}^*(-\mathbf{k}, \tau_2)}{a(\tau_1)a(\tau_2)} \times \frac{\exp\left(2(\beta E_k(\tau_1))_{\text{eff}}\right)}{\left(\exp\left((\beta E_k(\tau_1))_{\text{eff}}\right) - 1\right)^2} \times \left( \frac{\exp\left((\beta E_k(\tau_1))_{\text{eff}}\right)}{\left(\exp\left((\beta E_k(\tau_1))_{\text{eff}}\right) - 1\right)} \right)^{-1}
$$

$$= \frac{v_{out}(\mathbf{k}, \tau_1) v_{out}^*(-\mathbf{k}, \tau_2)}{2a(\tau_1)a(\tau_2)} \, \exp\left(\frac{(\beta E_k(\tau_1))_{\text{eff}}}{2}\right) \text{cosech}\left(\frac{(\beta E_k(\tau_1))_{\text{eff}}}{2}\right), \qquad \text{(F.4)}$$

and

$$\mathcal{G}_{-,\chi\chi}^{GGE}(\beta, \mathbf{k}, \tau_1, \tau_2) = \frac{1}{Z(\tau_1)} \frac{v_{out}^*(-\mathbf{k}, \tau_1) v_{out}(\mathbf{k}, \tau_2)}{a(\tau_1)a(\tau_2)} \frac{1}{|d_1|} \exp\left(-\frac{i}{2}\left\{\frac{d_2^*}{d_1^*} - \frac{d_2}{d_1}\right\}\right)$$

$$\times \sum_{\{N_k\}=0 \ \forall \ k}^{\infty} \langle\{N_k\}| \exp(-\beta \ \hat{H}_k(\tau_1) - \sum_{n=2}^{\infty} \kappa_{2n,\mathbf{DB/NB}}|k|^{2n-1})) \qquad \text{(F.5)}$$

$$a_{out}^\dagger(-\mathbf{k})a_{out}(\mathbf{k})\,|\{N_k\}\rangle$$

$$= \frac{1}{Z(\tau_1)} \frac{v_{out}^*(-\mathbf{k}, \tau_1) v_{out}(\mathbf{k}, \tau_2)}{a(\tau_1)a(\tau_2)} \frac{1}{|d_1|} \exp\left(-\frac{i}{2}\left\{\frac{d_2^*}{d_1^*} - \frac{d_2}{d_1}\right\}\right)$$

$$\times \sum_{\{N_k\}=0 \ \forall \ k}^{\infty} N_k \ \exp(-(\beta E_k(\tau_1))_{\text{eff}} N_k)$$

$$= \frac{1}{Z(\tau_1)} \frac{v_{out}^*(-\mathbf{k}, \tau_1) v_{out}(\mathbf{k}, \tau_2)}{a(\tau_1)a(\tau_2)} \frac{1}{|d_1|} \exp\left(-\frac{i}{2}\left\{\frac{d_2^*}{d_1^*} - \frac{d_2}{d_1}\right\}\right)$$

$$\times \frac{\exp(\beta E_k(\tau_1))_{\text{eff}}}{(\exp(\beta E_k(\tau_1))_{\text{eff}} - 1)^2}$$

$$= \frac{v_{out}^*(-\mathbf{k}, \tau_1) v_{out}(\mathbf{k}, \tau_2)}{a(\tau_1)a(\tau_2)}$$

$$\times \frac{\exp(\beta E_k(\tau_1))_{\text{eff}}}{(\exp(\beta E_k(\tau_1))_{\text{eff}} - 1)^2} \times \left(\frac{\exp((\beta E_k(\tau_1))_{\text{eff}})}{(\exp((\beta E_k(\tau_1))_{\text{eff}}) - 1)}\right)^{-1}$$

$$= \frac{v_{out}^*(-\mathbf{k}, \tau_1) v_{out}(\mathbf{k}, \tau_2)}{2a(\tau_1)a(\tau_2)} \, \exp\left(-\frac{(\beta E_k(\tau_1))_{\text{eff}}}{2}\right) \text{cosech}\left(\frac{(\beta E_k(\tau_1))_{\text{eff}}}{2}\right), \quad \text{(F.6)}$$

By following the same steps one can further show the following results in the present context:

$$\mathcal{G}_{+,\partial_i\chi\partial_i\chi}^{GGE}(\beta, \mathbf{k}, \tau_1, \tau_2) = -k^2 \ \mathcal{G}_{+,\chi\chi}^{GGE}(\beta, \mathbf{k}, \tau_1, \tau_2), \qquad \text{(F.7)}$$

$$\mathcal{G}_{-,\partial_i\chi\partial_i\chi}^{GGE}(\beta, \mathbf{k}, \tau_1, \tau_2) = -k^2 \ \mathcal{G}_{-,\chi\chi}^{GGE}(\beta, \mathbf{k}, \tau_1, \tau_2), \qquad \text{(F.8)}$$

$$\mathcal{G}_{+,\Pi_\chi\Pi_\chi}^{GGE}(\beta, \mathbf{k}, \tau_1, \tau_2) = \frac{v_{out}'(\mathbf{k}, \tau_1) v_{out}^{*\prime}(-\mathbf{k}, \tau_2)}{2a(\tau_1)a(\tau_2)} \, \exp\left(\frac{(\beta E_k(\tau_1))_{\text{eff}}}{2}\right) \text{cosech}\left(\frac{(\beta E_k(\tau_1))_{\text{eff}}}{2}\right)$$

$$-\frac{\mathcal{G}_{+,\chi\chi}^{GGE}(\beta, \mathbf{k}, \tau_1, \tau_2)}{a(\tau_1)a(\tau_2)} a'(\tau_1)a'(\tau_2), \qquad \text{(F.9)}$$

$$\mathcal{G}_{-,\Pi_\chi\Pi_\chi}^{GGE}(\beta, \mathbf{k}, \tau_1, \tau_2) = \frac{v_{out}^{*\prime}(-\mathbf{k}, \tau_1) v_{out}'(\mathbf{k}, \tau_2)}{2a(\tau_1)a(\tau_2)} \, \exp\left(-\frac{(\beta E_k(\tau_1))_{\text{eff}}}{2}\right) \text{cosech}\left(\frac{(\beta E_k(\tau_1))_{\text{eff}}}{2}\right)$$

$$-\frac{\mathcal{G}_{-,\chi\chi}^{GGE}(\beta, \mathbf{k}, \tau_1, \tau_2)}{a(\tau_1)a(\tau_2)} a'(\tau_1)a'(\tau_2). \quad \text{(F.10)}$$

# G   Derivation for thermal Green's functions for GGE ensemble with squeezing in Fourier space

The thermal Green's functions for the GGE ensemble for the field $\chi$, its spatial derivative and its canonically conjugate momentum can be expressed as:

$$
G^{GGE}_{\chi\chi,sq}(\beta, \mathbf{r}, \tau_1, \tau_2) = \int \frac{d^3\mathbf{k}}{(2\pi)^3} \left[ \mathcal{G}^{GGE}_{+,\chi\chi,sq}(\beta, \mathbf{k}, \tau_1, \tau_2) \, \exp(i\mathbf{k}.\mathbf{r}) \right.
$$
$$
\left. + \mathcal{G}^{GGE}_{-,\chi\chi,sq}(\beta, \mathbf{k}, \tau_1, \tau_2) \, \exp(-i\mathbf{k}.\mathbf{r}) \right], \qquad \text{(G.1)}
$$

$$
G^{GGE}_{\partial_i\chi\partial_i\chi,sq}(\beta, \mathbf{k}, \tau_1, \tau_2) = \int \frac{d^3\mathbf{k}}{(2\pi)^3} \left[ \mathcal{G}^{GGE}_{+,\partial_i\chi\partial_i\chi,sq}(\beta, \mathbf{k}, \tau_1, \tau_2) \, \exp(i\mathbf{k}.\mathbf{r}) \right.
$$
$$
\left. + \mathcal{G}^{GGE}_{-,\partial_i\chi\partial_i\chi,sq}(\beta, \mathbf{k}, \tau_1, \tau_2) \, \exp(-i\mathbf{k}.\mathbf{r}) \right], \quad \text{(G.2)}
$$

$$
G^{GGE}_{\Pi_\chi\Pi_\chi,sq}(\beta, \mathbf{r}, \tau_1, \tau_2) = \int \frac{d^3\mathbf{k}}{(2\pi)^3} \left[ \mathcal{G}^{GGE}_{+,\Pi_\chi\Pi_\chi,sq}(\beta, \mathbf{k}, \tau_1, \tau_2) \, \exp(i\mathbf{k}.\mathbf{r}) \right.
$$
$$
\left. + \mathcal{G}^{GGE}_{-,\Pi_\chi\Pi_\chi,sq}(\beta, \mathbf{k}, \tau_1, \tau_2) \, \exp(-i\mathbf{k}.\mathbf{r}) \right], \quad \text{(G.3)}
$$

where we define, $\mathbf{r} :\equiv \mathbf{x}_1 - \mathbf{x}_2$.

For each of the cases the corresponding thermal propagators in Fourier space are divided into two parts, one of them represents the advanced propagator which are appearing with $+$ symbol and the other one is the retarded propagator which are appearing with the $-$ symbol. In the occupation number representation for the Hamiltonian we get:

$$
\mathcal{G}^{GGE}_{+,\chi\chi,sq}(\beta, \mathbf{k}, \tau_1, \tau_2) = \frac{1}{Z(\tau_1)} \frac{v_{out}(\mathbf{k}, \tau_1)v_{out}^*(-\mathbf{k}, \tau_2)}{a(\tau_1)a(\tau_2)} \frac{1}{|d_1|} \exp\left( -\frac{i}{2}\left\{ \frac{d_2^*}{d_1^*} - \frac{d_2}{d_1} \right\} \right)
$$
$$
\times \sum_{\{N_k\}=0 \; \forall \; k}^{\infty} \langle\{N_k\}| \exp(-\beta \, \hat{H}_k(\tau_1) - \sum_{n=2}^{\infty} \kappa^{sq}_{2n,\mathbf{DB/NB}}|k|^{2n-1}))
$$
$$
a_{out}(\mathbf{k})a_{out}^\dagger(-\mathbf{k}) \, |\{N_k\}\rangle
$$
$$
= \frac{1}{Z(\tau_1)} \frac{v_{out}(\mathbf{k}, \tau_1)v_{out}^*(-\mathbf{k}, \tau_2)}{a(\tau_1)a(\tau_2)} \frac{1}{|d_1|} \exp\left( -\frac{i}{2}\left\{ \frac{d_2^*}{d_1^*} - \frac{d_2}{d_1} \right\} \right)
$$
$$
\times \sum_{\{N_k\}=0 \; \forall \; k}^{\infty} (N_k + 1) \, \exp(-(\beta E_k(\tau_1))_{\text{eff,sq}} N_k)
$$
$$
= \frac{1}{Z(\tau_1)} \frac{v_{out}(\mathbf{k}, \tau_1)v_{out}^*(-\mathbf{k}, \tau_2)}{a(\tau_1)a(\tau_2)} \frac{1}{|d_1|} \exp\left( -\frac{i}{2}\left\{ \frac{d_2^*}{d_1^*} - \frac{d_2}{d_1} \right\} \right)
$$
$$
\times \frac{\exp\left(2(\beta E_k(\tau_1))_{\text{eff,sq}}\right)}{\left(\exp\left((\beta E_k(\tau_1))_{\text{eff,sq}}\right) - 1\right)^2}
$$
$$
= \frac{v_{out}(\mathbf{k}, \tau_1)v_{out}^*(-\mathbf{k}, \tau_2)}{a(\tau_1)a(\tau_2)} \times \frac{\exp\left(2(\beta E_k(\tau_1))_{\text{eff,sq}}\right)}{\left(\exp\left((\beta E_k(\tau_1))_{\text{eff,sq}}\right) - 1\right)^2}
$$

$$\times \left( \frac{\exp\left((\beta E_k(\tau_1))_{\text{eff,sq}}\right)}{(\exp\left((\beta E_k(\tau_1))_{\text{eff,sq}}\right) - 1)} \right)^{-1}$$

$$= \frac{v_{out}(\mathbf{k}, \tau_1) v_{out}^*(-\mathbf{k}, \tau_2)}{2a(\tau_1)a(\tau_2)} \exp\left( \frac{(\beta E_k(\tau_1))_{\text{eff,sq}}}{2} \right) \operatorname{cosech}\left( \frac{(\beta E_k(\tau_1))_{\text{eff,sq}}}{2} \right), \quad \text{(G.4)}$$

and

$$\mathcal{G}^{GGE}_{-,\chi\chi,sq}(\beta, \mathbf{k}, \tau_1, \tau_2) = \frac{1}{Z(\tau_1)} \frac{v_{out}^*(-\mathbf{k}, \tau_1) v_{out}(\mathbf{k}, \tau_2)}{a(\tau_1)a(\tau_2)} \frac{1}{|d_1|} \exp\left( -\frac{i}{2}\left\{ \frac{d_2^*}{d_1^*} - \frac{d_2}{d_1} \right\} \right)$$

$$\times \sum_{\{N_k\}=0 \ \forall \ k}^{\infty} \langle \{N_k\}| \exp(-\beta \ \hat{H}_k(\tau_1) - \sum_{n=2}^{\infty} \kappa_{2n,\mathbf{DB/NB}} |k|^{2n-1})) \quad \text{(G.5)}$$

$$a_{out}^\dagger(-\mathbf{k}) a_{out}(\mathbf{k}) \, |\{N_k\}\rangle$$

$$= \frac{1}{Z(\tau_1)} \frac{v_{out}^*(-\mathbf{k}, \tau_1) v_{out}(\mathbf{k}, \tau_2)}{a(\tau_1)a(\tau_2)} \frac{1}{|d_1|} \exp\left( -\frac{i}{2}\left\{ \frac{d_2^*}{d_1^*} - \frac{d_2}{d_1} \right\} \right)$$

$$\times \sum_{\{N_k\}=0 \ \forall \ k}^{\infty} N_k \ \exp(-(\beta E_k(\tau_1))_{\text{eff,sq}} N_k)$$

$$= \frac{1}{Z(\tau_1)} \frac{v_{out}^*(-\mathbf{k}, \tau_1) v_{out}(\mathbf{k}, \tau_2)}{a(\tau_1)a(\tau_2)} \frac{1}{|d_1|} \exp\left( -\frac{i}{2}\left\{ \frac{d_2^*}{d_1^*} - \frac{d_2}{d_1} \right\} \right)$$

$$\times \frac{\exp(\beta E_k(\tau_1))_{\text{eff,sq}}}{\left( \exp\left(\beta E_k(\tau_1)\right)_{\text{eff,sq}} - 1 \right)^2}$$

$$= \frac{v_{out}^*(-\mathbf{k}, \tau_1) v_{out}(\mathbf{k}, \tau_2)}{a(\tau_1)a(\tau_2)}$$

$$\times \frac{\exp(\beta E_k(\tau_1))_{\text{eff}}}{\left( \exp\left(\beta E_k(\tau_1)\right)_{\text{eff}} - 1 \right)^2} \times \left( \frac{\exp\left((\beta E_k(\tau_1))_{\text{eff}}\right)}{(\exp\left((\beta E_k(\tau_1))_{\text{eff}}\right) - 1)} \right)^{-1}$$

$$= \frac{v_{out}^*(-\mathbf{k}, \tau_1) v_{out}(\mathbf{k}, \tau_2)}{2a(\tau_1)a(\tau_2)} \exp\left( -\frac{(\beta E_k(\tau_1))_{\text{eff}}}{2} \right) \operatorname{cosech}\left( \frac{(\beta E_k(\tau_1))_{\text{eff}}}{2} \right), \quad \text{(G.6)}$$

By following the same steps one can further show the following results in the present context:

$$\mathcal{G}^{GGE}_{+,\partial_i\chi\partial_i\chi,sq}(\beta, \mathbf{k}, \tau_1, \tau_2) = -k^2 \ \mathcal{G}^{GGE}_{+,\chi\chi,sq}(\beta, \mathbf{k}, \tau_1, \tau_2), \quad \text{(G.7)}$$

$$\mathcal{G}^{GGE}_{-,\partial_i\chi\partial_i\chi,sq}(\beta, \mathbf{k}, \tau_1, \tau_2) = -k^2 \ \mathcal{G}^{GGE}_{-,\chi\chi,sq}(\beta, \mathbf{k}, \tau_1, \tau_2), \quad \text{(G.8)}$$

$$\mathcal{G}^{GGE}_{+,\Pi_\chi\Pi_\chi,sq}(\beta, \mathbf{k}, \tau_1, \tau_2) = \frac{v_{out}'(\mathbf{k}, \tau_1) v_{out}^{*\prime}(-\mathbf{k}, \tau_2)}{2a(\tau_1)a(\tau_2)} \exp\left( \frac{(\beta E_k(\tau_1))_{\text{eff,sq}}}{2} \right) \operatorname{cosech}\left( \frac{(\beta E_k(\tau_1))_{\text{eff,sq}}}{2} \right)$$

$$- \frac{\mathcal{G}^{GGE}_{+,\chi\chi,sq}(\beta, \mathbf{k}, \tau_1, \tau_2)}{a(\tau_1)a(\tau_2)} a'(\tau_1)a'(\tau_2), \quad \text{(G.9)}$$

$$\mathcal{G}^{GGE}_{-,\Pi_\chi\Pi_\chi,sq}(\beta, \mathbf{k}, \tau_1, \tau_2) = \frac{v_{out}^{*\prime}(-\mathbf{k}, \tau_1) v_{out}'(\mathbf{k}, \tau_2)}{2a(\tau_1)a(\tau_2)} \exp\left( -\frac{(\beta E_k(\tau_1))_{\text{eff,sq}}}{2} \right) \operatorname{cosech}\left( \frac{(\beta E_k(\tau_1))_{\text{eff,sq}}}{2} \right)$$

$$-\frac{\mathcal{G}^{GGE}_{-,\chi\chi,sq}(\beta,\mathbf{k},\tau_1,\tau_2)}{a(\tau_1)a(\tau_2)}a'(\tau_1)a'(\tau_2). \quad \text{(G.10)}$$

# H  From Schrödinger scattering problem in Quantum Mechanics to Particle Production in de Sitter Space

Initially, we have stated with a two interacting scalar field theory describing Quantum Brownian motion by following the quantum field theoretic generalization of the *Caldeira-Leggett Model*. Further performing the Euclidean path integration over one scalar field we have derived an effective theory of the other scalar field. Now for the conformally flat de Sitter background we have shown that in the Fourier space the Klein Gordon field equation for the modes of survived field after path integration can be written as:

$$\left[\frac{d^2}{d\tau^2} + \left(k^2 + m_{\text{eff}}^2(\tau)\right)\right]v(\mathbf{k},\tau) = 0 \qquad \text{where} \quad m_{\text{eff}}^2(\tau) = \frac{1}{\tau^2}\left(\frac{m^2(\tau)}{\mathcal{H}^2} - 2\right). \quad \text{(H.1)}$$

The analogous problem in quantum mechanics is to solve a Schrödinger scattering problem in one dimension inside an electrical conduction wire in presence of an impurity potential, which is described by [‖]:

$$\left[\frac{d^2}{dx^2} + (E - V(x))\right]\psi\left(\sqrt{E},x\right) = 0. \quad \text{(H.2)}$$

Here $V(x)$ is the impurity potential which mimics the role of negative of the effective conformal time-dependent mass protocol used in the quenched Quantum Brownian Motion problem. By replacing the time coordinate $\tau$ with $x$ one can write down the following form of the impurity potential in the one dimensional Schrödinger problem:

$$V(x) = \frac{1}{x^2}\left(2 - U(x)\right), \quad \text{(H.3)}$$

where the quantum mechanical quench protocol in one dimension quantum mechanical problem in the present context is described by:

$$U(x) = U_0\Theta(-x) = \begin{cases} U_0 & \textbf{\textcolor{red}{Before quench}}: \ x < x_0; \\ \\ 0 & \textbf{\textcolor{red}{After quench}}: \ x \geq x_0. \end{cases} \quad \text{(H.4)}$$

Here $x_0$ is identified to be point where the quench operation is performed.

Also it is important to note that, the wave function $\psi\left(\sqrt{E},x\right)$ in one dimensional Schrödinger problem mimics the role of the mode function as appearing in the particle

---

[‖] Here we have assumed $\hbar = 1$ and $2m = 1$ in the Schrödinger equation.

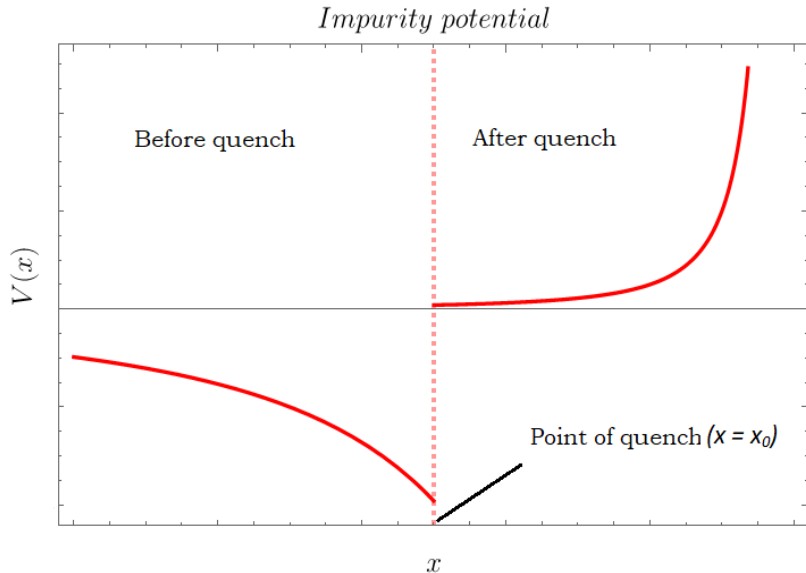

**Figure H.1**: The impurity potential profile.

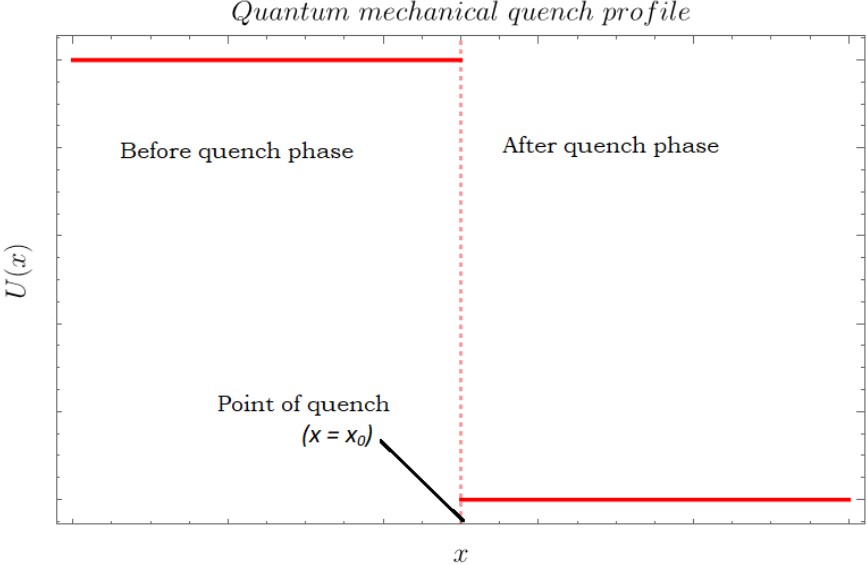

**Figure H.2**: Quantum mechanical quench profile.

production problem in de Sitter space. Finally the energy $E$ in the Schrödinger problem mimics the role of $k^2$ in Fourier space in the particle production problem in de Sitter space.

In this description the solutions for the Schrödinger equation before and after quench can be written as:

$$\textbf{Before quench}: \quad \psi_{in}\left(\sqrt{E},x\right) = \sqrt{x}\left[C_1\ H^{(1)}_{\frac{1}{2}\sqrt{9-4U_0}}\left(\sqrt{E}x\right) + C_2\ H^{(2)}_{\frac{1}{2}\sqrt{9-4U_0}}\left(\sqrt{E}x\right)\right], \text{(H.5)}$$

**After quench :** $\quad \psi_{out}\left(\sqrt{E},x\right) = \sqrt{\dfrac{2}{\pi\sqrt{E}}}\Bigg[C_3\left(\dfrac{\sin\left(\sqrt{E}x\right)}{\sqrt{E}x} - \cos\left(\sqrt{E}x\right)\right)$

$$-C_4\left(\dfrac{\cos\left(\sqrt{E}x\right)}{\sqrt{E}x} + \sin\left(\sqrt{E}x\right)\right)\Bigg]. \qquad (H.6)$$

Here $\psi_{in}(x)$ and $\psi_{out}(x)$ are the representative solutions of the Schrödinger equation before and after quench respectively. Also, $C_1$, $C_2$ and $C_3$, $C_4$ are the arbitrary integration constants which are fixed by the appropriate choice of the boundary conditions, which are the continuity of the in and out solutions and it derivatives at the point of quench $x_0$. This helps us to write $C_3$, $C_4$ in terms of $C_1$, $C_2$. Additionally it is important to note that, to serve this purpose instead of using the actual solution one need to use the asymptotic solutions of the Schrödinger equation before and after quench at $x \to -\infty$ and $x \to \infty$ respectively.

In this construction one can actually write down the total asymptotic solution ($x \to \pm\infty$) of the Schrödinger equation by the following expression:

$$\psi\left(\sqrt{E},x\right) = C_1\ f_{in}\left(\sqrt{E},x\right) + C_2\ f_{in}^*\left(\sqrt{E},x\right) = C_3\ f_{out}\left(\sqrt{E},x\right) + C_4\ f_{out}^*\left(\sqrt{E},x\right).$$
$$(H.7)$$

Here $f_{in}\left(\sqrt{E},x\right)$ and $f_{out}\left(\sqrt{E},x\right)$ are the combined asymptotic solutions at $x \to \pm\infty$ for the actual solutions obtained in the previous page.

Here it is important to note that, incoming and the outgoing solutions before and after quench can be expressed in terms of each other via the following relations:

$$f_{in}\left(\sqrt{E},x\right) = \alpha\left(\sqrt{E},x_0\right)\ f_{out}\left(\sqrt{E},x\right) + \beta\left(\sqrt{E},x_0\right)\ f_{out}^*\left(\sqrt{E},x\right), \qquad (H.8)$$
$$f_{out}\left(\sqrt{E},x\right) = \alpha^*\left(\sqrt{E},x_0\right)\ f_{in}\left(\sqrt{E},x\right) - \beta\left(\sqrt{E},x_0\right)\ f_{in}^*\left(\sqrt{E},x\right). \qquad (H.9)$$

Consequently, the general solution for the field equation can be written as:

$$\psi\left(\sqrt{E},x\right) = a_{in}\left(\sqrt{E}\right) f_{in}\left(\sqrt{E},x\right) + a_{in}^\dagger\left(\sqrt{E}\right) f_{in}^*\left(\sqrt{E},x\right)$$
$$= a_{out}\left(\sqrt{E}\right) f_{out}\left(\sqrt{E},x\right) + a_{out}^\dagger\left(\sqrt{E}\right) f_{out}^*\left(\sqrt{E},x\right), \qquad (H.10)$$

which satisfy the following reality constraint:

$$\psi^*\left(\sqrt{E},x\right) = \psi\left(\sqrt{E},x\right). \qquad (H.11)$$

Using these above mentioned equations one can explicitly show that:

$$a_{in}\left(\sqrt{E}\right) = \alpha^*\left(\sqrt{E}, x_0\right) a_{out}\left(\sqrt{E}\right) - \beta^*\left(\sqrt{E}, x_0\right) a_{out}^\dagger\left(\sqrt{E}\right),$$ (H.12)

$$a_{out}\left(\sqrt{E}\right) = \alpha^*\left(\sqrt{E}, x_0\right) a_{in}\left(\sqrt{E}\right) + \beta^*\left(\sqrt{E}, x_0\right) a_{in}^\dagger\left(\sqrt{E}\right).$$ (H.13)

Here the Bogolyubov coefficients at the point of quench $x_0$, are calculated using the following equations:

$$\alpha\left(\sqrt{E}, x_0\right) = \frac{1}{2i}\left[\frac{df_{out}\left(\sqrt{E}, x\right)}{dx} f_{in}^*\left(\sqrt{E}, x\right) - f_{out}\left(\sqrt{E}, x\right) \frac{df_{in}^*\left(\sqrt{E}, x\right)}{dx}\right]_{x_0},$$ (H.14)

$$\beta^*\left(\sqrt{E}, x_0\right) = \frac{1}{2i}\left[\frac{df_{out}\left(\sqrt{E}, x\right)}{dx} f_{in}\left(\sqrt{E}, x\right) - f_{out}\left(\sqrt{E}, x\right) \frac{df_{in}\left(\sqrt{E}, x\right)}{dx}\right]_{x_0}.$$ (H.15)

In this context, one can explicitly show that the incoming coefficients $C_1, C_2$ and the outgoing coefficients $C_3, C_4$ are related via the following matrix equation:

$$\begin{pmatrix} C_3 \\ C_4 \end{pmatrix} = \underbrace{\begin{pmatrix} \alpha\left(\sqrt{E}, x_0\right) & \beta\left(\sqrt{E}, x_0\right) \\ \beta^*\left(\sqrt{E}, x_0\right) & \alpha^*\left(\sqrt{E}, x_0\right) \end{pmatrix}}_{\textcolor{red}{\textbf{Transfer Matrix}}} \begin{pmatrix} C_1 \\ C_2 \end{pmatrix}$$ (H.16)

which finally leads to the following constraint:

$$\left|\alpha\left(\sqrt{E}, x_0\right)\right|^2 - \left|\beta\left(\sqrt{E}, x_0\right)\right|^2 = 1.$$ (H.17)

Now, for the scattering problem one can define the reflection and transmission coefficients for the wave travelling from left to right as:

$$r = \frac{C_2}{C_1} = -\frac{\beta^*\left(\sqrt{E}, x_0\right)}{\alpha^*\left(\sqrt{E}, x_0\right)},$$ (H.18)

$$t = \frac{C_3}{C_1} = \alpha\left(\sqrt{E}, x_0\right) + \beta\left(\sqrt{E}, x_0\right) \; r = \left(\alpha\left(\sqrt{E}, x_0\right) - \frac{\left|\beta\left(\sqrt{E}, x_0\right)\right|^2}{\alpha^*\left(\sqrt{E}, x_0\right)}\right),$$ (H.19)

which finally implies the following conservation equation:

$$|r|^2 + |t|^2 = 1.$$ (H.20)

Similarly, for the scattering problem one can define the reflection and transmission coefficients for the wave travelling from right to left as:

$$r' = \frac{C_3}{C_4} = \frac{\beta\left(\sqrt{E}, x_0\right)}{\alpha^*\left(\sqrt{E}, x_0\right)}, \tag{H.21}$$

$$t' = \frac{C_2}{C_4} = \frac{1}{\alpha^*\left(\sqrt{E}, x_0\right)}, \tag{H.22}$$

which further implies:

$$|r| = |r'|, \quad \frac{t}{t'} = \left(\frac{1}{|t'|^2} + \frac{rr'}{t'^2}\right). \tag{H.23}$$

Finally, for this scattering problem the transfer matrix can be written in terms of the reflection and transmission coefficients as:

$$\underbrace{\begin{pmatrix} \alpha\left(\sqrt{E}, x_0\right) & \beta\left(\sqrt{E}, x_0\right) \\ \beta^*\left(\sqrt{E}, x_0\right) & \alpha^*\left(\sqrt{E}, x_0\right) \end{pmatrix}}_{\textcolor{red}{\textbf{Transfer Matrix}}} = \begin{pmatrix} t - \dfrac{rr'}{t'} & \dfrac{r'}{t'} \\[2ex] -\dfrac{r}{t'} & \dfrac{1}{t'} \end{pmatrix}. \tag{H.24}$$

After getting the expression for the reflection coefficient after quench one can further expand it around $\sqrt{E} = 0$, which gives:

$$r' = \sum_{n=0}^{\infty} r_n \, E^{\frac{n}{2}}, \tag{H.25}$$

which is exactly analogous to the expansion of the factor $\gamma$, which we have computed in the main subject content of the paper.

## I    Determining coefficients for outgoing modes in terms of full solutions

We first consider the case of instantaneous quench where the mass of the field suddenly falls of to 0 at a particular conformal time denoted by $\eta$ in this case. The incoming solutions before the point of quench is denoted by:

$$v_{in}(\tau) = \sqrt{-k\tau} \, [d_1 H^{(1)}_{\nu_{in}}(-k\tau) + d_2 H^{(2)}_{\nu_{in}}(-k\tau)]. \tag{I.1}$$

The derivatives of the above solution can be calculated as:

$$v'_{in}(\tau) = \frac{1}{2\sqrt{-k\tau}}\left[2d_1 k\tau\ H^{(1)}_{\nu_{in}-1}(-k\tau) + d_1(-1+2\nu_{in})\ H^{(1)}_{\nu_{in}}(-k\tau)\right.$$
$$\left.+2d_2 k\tau\ H^{(2)}_{\nu_{in}-1}(-k\tau) + d_2(-1+2\nu_{in})H^{(2)}_{\nu_{in}}(-k\tau)\right]. \tag{I.2}$$

The outgoing solution after the quench point is given by

$$v_{out}(\tau) = \sqrt{-k(\tau+\eta)}\left[d_3\ H^{(1)}_{\frac{3}{2}}(-k(\tau+\eta)) + d_4\ H^{(2)}_{\frac{3}{2}}(-k(\tau+\eta))\right]. \tag{I.3}$$

The derivatives of the outgoing solution is calculated as:

$$v'_{out}(\tau) = \frac{1}{\sqrt{-k\tau+\eta}}\left(d_3 k(\tau+\eta)H^{(1)}_{\frac{1}{2}}(-k(\tau+\eta)) + d_3 H^{(1)}_{\frac{3}{2}}(-k(\tau+\eta))\right.$$
$$\left.+d_4(k(\tau+\eta))H_{\frac{1}{2}}(-k(\tau+\eta)) + d_4 H^{(2)}_{\frac{3}{2}}(-k(\tau+\eta))\right).$$

Generally out of the four arbitrary constants, two can be fixed by the initial choice of vacuum state. Hence, expressing any two arbitrary constants in terms of the other two is quite natural. We proceed by expressing the constants appearing in the outgoing solutions in terms of the constants of the incoming solution. These fixing is carried out by using the continuity of the solutions and its first derivatives at the point of quench. Thus the arbitrary constants $d_3$ and $d_4$ expressed in terms of $d_1$ and $d_2$ can be written as

$$d_3 = \frac{i\pi}{8\sqrt{2}}\left(d_1 H^{(1)}_{\nu_{in}}(-k\eta)\ \{-4k\eta H^{(2)}_{\frac{1}{2}}(-2k\eta) + (-3+2\nu_{in})H^{(2)}_{\frac{3}{2}}(-2k\eta)\}\right.$$
$$+2k\tau H^{(2)}_{\frac{3}{2}}(-2k\eta)\{d_1 H^{(1)}_{\nu_{in}-1}(-k\eta) + d_2 H^{(2)}_{\nu_{in}-1}(-k\eta)\}$$
$$\left.+d_2\{-4k\tau H^{(2)}_{\frac{1}{2}}(-2k\eta) + (-3+2\nu_{in})H^{(2)}_{\frac{3}{2}}(-2k\eta)\}H^{(2)}_{\nu}(-k\eta)\right) \tag{I.4}$$

$$d_4 = \frac{i\pi}{8\sqrt{2}}\left(4k\eta H^{(2)}_{\frac{1}{2}}(-2k\eta)\{d_1 H^{(1)}_{\nu_{in}}(-k\eta) + d_2 H^{(2)}_{\nu_{in}}(-k\eta)\} + H^{(1)}_{\frac{3}{2}}(-2k\eta)\right.$$
$$\{-2d_1 k\eta H^{(1)}_{\nu_{in}-1}(-k\eta) + d_1(3-2\nu_{in})H^{(1)}_{\nu_{in}}(-k\eta) - 2d_2 k\eta H^{(2)}_{\nu_{in}-1}(-k\eta)$$
$$\left.+(3-2\nu_{in})H^{(2)}_{\nu_{in}}(-k\eta)\}\right) \tag{I.5}$$

Though in this article we have not used the analytical computations from the full solution of the mode equation as computing the two-point correlators and preparing the post quench states are extremely time consuming and sometimes impossible to simplify. For this reason we have used the asymptotic solution which combines the effect at $\tau \to -\infty$ and $\tau \to 0$ to serve the purpose.

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
