# Peer review of "Thermalization in Quenched De Sitter Space"

_SciPost Physics_

## Round 4 · Referee Report · Anonymous · 2022-6-25

The authors considered a quantum quench of a scalar field on dS space, and investigated the correlation functions of the scalar field. Although the results are new and interesting, I found several problems on the manuscript and I would like to request the authors to make revisions.

1. The authors studied the sudden quench (3.3) in the model (3.1). But the motivation is unclear. Is this just a mathematical model? Is there any cosmological reason to study the quench (3.3)? If so, what is $\chi$ and $\phi$, and how does the time dependent $m(\tau)$ (or $c(\tau)$) arise? (In condensed matter systems, experimentalists might be able to control the mass or the coupling, but, in cosmology, who can do it?)

2. Related to the first problem, the importance of the Caldeira-Leggett model (2.1) is unclear for me. The authors considered the sudden quench (3.3), and it means that certain dynamics involving other fields must causes it. In my opinion, it is better to simply start from the model (3.1) from the beginning rather than start from the Caldeira-Leggett model (2.1). Whether starting from (3.1) or starting from (2.1), both must assume the unknown assumption (3.3), and starting from (3.1) is better, since it is simpler.

3. This manuscript is not well explained and is difficult to understand in several places. I could not understand the following arguments:

   (a) After (2.8), the authors employed the Euclidean action. However, several equations seem Lorentzian. For example $\sqrt{-g}$ and the action in (2.14). (2.15) also looks Lorentzian but it is unclear whether the authors really studied Lorentzian dS or it is a typo.

   (b) Around (2.15), the definition of $H$, $H(\tau)$, $H'$ and their related quantities such as $\tau_T$ are unclear.

   (c) In (3.11), $\mathcal{H}$ and $\mathcal{H}'$ are not defined.

   (d) In section 3, the authors studied various vacua and states such as Bunch-Davies vacuum. Their definitions are needed to make the manuscript self-contained. Relatedly in (3.58) and (3.59), Dirichlet and Neumann boundary states are not defined.

4. I found several typos:

   (a) On page 21, "we have have".

   (b) In (3.221), (3.222) and (3.223), $\tau_1$ in the second equation.

   (c) On page 47, there are several typos on the parentheses.

---

## Round 4 · Referee Report · Anonymous · 2022-6-27

Strengths
Writing is clear at a technical level
Weaknesses
The motivation is poor
Report
This paper studies quantum quenches for fields in de Sitter spacetime. While the calculations appear correct, the big problem with the paper is that there is little explanation for why one should study quantum quenches in de Sitter spacetime. There is some mention at the beginning about this relating to fast phase transitions in the early universe, but that's all. It's very slim motivation given that there is then well over 100 pages devoted to the subject.
Moreover, it's quite difficult to think of a genuine motivation for these calculations. The usual phase transitions that take place in the early universe arise because of the expansion of the universe and the related cooling. There is nothing fast about this. The time scale of the phase transition is set by the inverse Hubble scale. Here, the idea is that there is out-of-equilibrium dynamics taking place in the early universe in a smooth de Sitter background and there seems little reason to think that this is an interesting thing to study.
My impression is that the real reason the author's explored this is because there were calculations that they could do. But this is not enough to warrant publication in a top journal like SciPost. For that reason, I recommend that this paper is rejected.

---

## Editorial Decision

submission_&_refereeing_history